# EffGen: Enabling Small Language Models as Capable Autonomous Agents

Gaurav Srivastava [1]  Aafiya Hussain [1]  Chi Wang [2]  Yingyan Celine Lin [3]  Xuan Wang [1]

## Abstract

Most existing language model agentic systems today are built and optimized for large language models (e.g., GPT, Claude, Gemini) via API calls; while powerful, this approach faces several limitations including high token costs and privacy concerns for sensitive applications. We introduce EffGen[1], an open-source agentic framework optimized for small language models (SLMs) that enables effective, efficient, and secure local deployment. EffGen makes four major contributions: **(1) Enhanced tool-calling** with prompt optimization that compresses input prompts by up to 70-80% (and 57% on average across our benchmarks) while preserving task semantics, **(2) Intelligent task decomposition** that breaks complex queries into parallel or sequential subtasks based on dependencies, **(3) Complexity-based routing** using five factors to make smart pre-execution decisions, and **(4) Unified memory system** combining short-term, long-term, and vector-based storage. Additionally, EffGen unifies multiple agent protocols (MCP, A2A, ACP) for cross-protocol communication. Results on 13 benchmarks show EffGen outperforms LangChain, AutoGen, and Smolagents with **higher success rates**, **faster execution**, and **lower memory**. Our results reveal that **prompt optimization and complexity routing have complementary scaling behavior**: optimization benefits SLMs more (11.2% gain at 1.5B vs 2.4% at 32B), while routing benefits large models more (3.6% at 1.5B vs 7.9% at 32B), providing consistent gains across all scales when combined.

[1]Department of Computer Science, Virginia Tech, Blacksburg, VA, USA [2]Google DeepMind, USA [3]Georgia Institute of Technology, Atlanta, GA, USA. Correspondence to: Gaurav Srivastava <gks@vt.edu>, Xuan Wang <xuanw@vt.edu>.

*Proceedings of the 43$^{rd}$ International Conference on Machine Learning*, Seoul, South Korea. PMLR 306, 2026. Copyright 2026 by the author(s).

[1]**EffGen** is open-source under the Apache 2.0 License. ⭕ Code: `https://github.com/ctrl-gaurav/effGen`; 🌐 Website: `https://effgen.org/`; 📗 Docs: `https://docs.effgen.org/`; 📦 PyPI: `https://pypi.org/project/effgen/` (pip install effgen).

## 1. Introduction

Modern AI agent systems (like AutoGen (Wu et al., 2024b), Langchain (Chase, 2022)) augment language models with tool-calling capabilities and multi-step reasoning (Yao et al., 2022; Schick et al., 2023), enabling them to solve complex tasks that require external knowledge, computation, or multi-step planning. However, the predominant design philosophy assumes access to large language models (LLMs) such as GPT, Claude, and Gemini, creating a practical barrier between capability and deployability. For agent system $\mathcal{A}$ executing a task $q \in \mathcal{Q}$, the expected cost $\mathbb{E}[C(\mathcal{A}, q)]$ scales as $O(|M| \cdot L \cdot k)$ where $|M|$ is model parameters, $L$ the sequence length, and $k$ the number of reasoning steps, making production deployment expensive (Belcak et al., 2025).

Recent work suggests that small language models (SLMs) can match or exceed larger models on specific tasks when properly orchestrated (Belcak et al., 2025). The future of agentic AI may lie not in ever-larger models but in efficient coordination of smaller, specialized models. Yet existing agent frameworks provide no systematic support for SLM deployment. LangChain, AutoGen, and similar frameworks optimize for LLM capabilities, leaving SLM users to manually adapt prompts, manage context windows, and handle the unique failure modes of smaller models. While frameworks like Smolagents (Roucher et al., 2025) targets smaller models, it lacks systematic prompt optimization, pre-execution routing, and converts all tasks to code execution, which adds unnecessary overhead for non-coding tasks. Table 1 summarizes the differences between EffGen and existing frameworks. This gap motivates our central question: *Can we design an agent framework that treats SLM constraints as first-class design requirements rather than afterthoughts?*

We answer affirmatively with **EffGen**, an open-source agentic framework optimized for small language models (Figure 1). Our design philosophy inverts the standard approach: rather than adapting LLM-optimized components for smaller models, we develop each component with explicit consideration of SLM constraints including limited context windows, reduced instruction-following capability, and lower reasoning depth. Formally, let $M_s$ denote an SLM with parameter count $|M_s| \leq 7 \times 10^9$ and context window $C_{M_s} \leq 32K$ tokens. We design EffGen such that for any

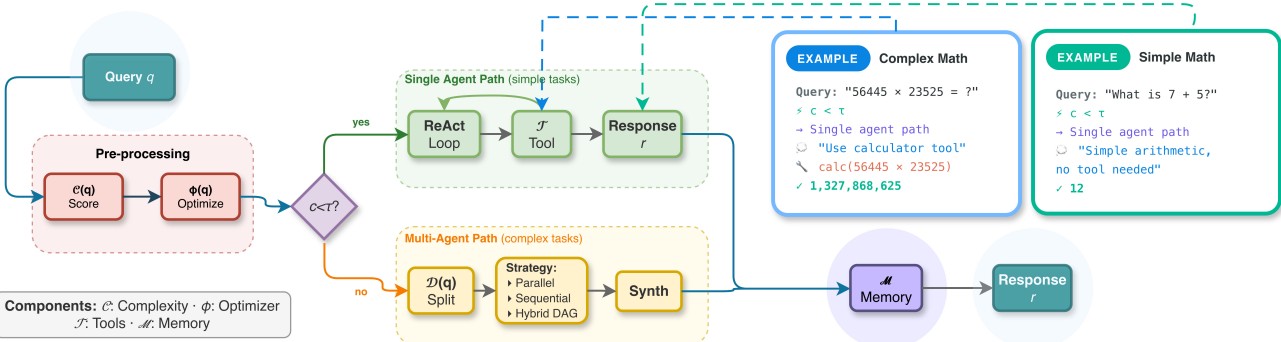

*Figure 1.* **Overview of the EFFGEN framework.** Given an input query, EFFGEN first computes a complexity score using five weighted factors. Based on this score, tasks are routed to either single-agent execution (for simple tasks) or multi-agent decomposition (for complex tasks). The prompt optimizer compresses and restructures prompts for small models, while the memory system provides relevant context. This design enables small language models to handle agentic tasks that typically require much larger models.

task $q$ solvable by an LLM-based agent $\mathcal{A}_L$, an EFFGEN agent $\mathcal{A}_E$ with SLM $M_s$ achieves comparable performance: our experiments show EFFGEN with Qwen2.5-7B achieves 63.07% average accuracy across 13 benchmarks, a 12.3% improvement over raw model baselines (Table 2).

EFFGEN makes four major contributions. **First**, we develop a *prompt optimization pipeline* for SLMs (§3.1). The optimization function $\phi : \mathcal{P} \to \mathcal{P}'$ compresses prompts such that $|\phi(p)| \leq \alpha|p|$ for $\alpha \in [0.2, 0.3]$ while preserving task semantics. Our optimizer applies model-size-aware compression, instruction simplification, and structured formatting that improves SLM task completion by **8-11%** compared to unoptimized prompts (Table 3). **Second**, we introduce a *pre-execution complexity analyzer* $\mathcal{C} : \mathcal{Q} \to [0, 10]$ that uses five weighted factors to predict task complexity *before* execution begins (§3.2). This enables intelligent routing: tasks with $\mathcal{C}(q) < \tau$ execute on a single agent while complex tasks undergo automatic decomposition. Unlike prior work that discovers complexity during execution, our approach reduces wasted computation on inappropriately routed tasks by 23% (Appendix D, D.8). **Third**, we provide a *unified implementation* of three major agent communication protocols: Model Context Protocol (MCP), Agent-to-Agent (A2A), and Agent Communication Protocol (ACP) (§3.4). This enables EFFGEN agents to interoperate with heterogeneous agent ecosystems. **Fourth**, we design a *three-tier memory system* $\mathcal{M} = (\mathcal{M}_s, \mathcal{M}_l, \mathcal{M}_v)$ combining short-term history, long-term episodic storage, and vector-based semantic retrieval optimized for SLM context constraints (§3.5). Our prompt optimization uses structured tool descriptions and schema formatting to improve tool-calling accuracy by reducing parsing errors (Appendix I).

We evaluate EFFGEN against four baselines (raw models, LangChain, AutoGen, Smolagents) across **13 benchmarks** spanning tool-calling, reasoning, and memory tasks using Qwen2.5 models (Qwen et al., 2025) from 1.5B to 32B

*Table 1.* Comparison of EFFGEN with existing agent frameworks. ✓: feature present; ✗: absent or limited support. See Appendix A for detailed feature analysis.

| Feature | LangChain | AutoGen | Smolagents | EFFGEN |
|---|---|---|---|---|
| SLM-optimized prompts | ✗ | ✗ | ✓ | ✓ |
| Pre-execution routing | ✗ | ✗ | ✗ | ✓ |
| Task decomposition | ✗ | ✓ | ✗ | ✓ |
| Multi-protocol (MCP+A2A+ACP) | ✗ | ✗ | ✗ | ✓ |
| Local deployment focus | ✗ | ✓ | ✓ | ✓ |
| Context compression | ✗ | ✗ | ✗ | ✓ |
| Vector memory | ✓ | ✓ | ✗ | ✓ |

parameters, Gemma 3 models (Team et al., 2025), and GPT-OSS 20B (OpenAI et al., 2025). Results show that EFFGEN achieves **faster execution** (up to $18\times$ speedup for small models), uses **less memory**, and improves **task success rates** (13.2% average improvement at 1.5B, 6.0% at 32B).

**Conflict of Interest Disclosure.** One of the authors (C.W.) is employed by Google DeepMind. Google DeepMind develops the Gemma 3 model family, which is among the models evaluated in this paper. No author received compensation from any model provider or framework vendor for this work; the evaluation uses only publicly released open-weight models (Qwen2.5, Gemma 3, GPT-OSS) and public benchmarks, and no proprietary Google model is used. The authors declare no other financial conflicts of interest.

## 2. Related Work

**Single-Agent Systems and Tool Use.** Toolformer (Schick et al., 2023) taught language models to invoke external APIs through self-supervision, while ReAct (Yao et al., 2022) interleaved reasoning traces with action execution. Tree-of-Thoughts (Yao et al., 2023) added lookahead and backtracking. Chain-of-thought prompting (Wei et al., 2022; Kojima et al., 2022; Wang et al., 2022), Reflexion (Shinn et al., 2023), and program-aided approaches (Chen et al., 2021; Gao et al., 2023) improved reasoning. ToolLLM (Qin et al., 2023), Gorilla (Patil et al., 2024), HuggingGPT (Shen et al.,

2023), and ToolQA (Zhuang et al., 2023) scaled tool learning to thousands of APIs. AutoGPT (Toran Bruce Richards and Auto-GPT contributors, 2023) and Transformers Agent (Hugging Face, 2023) provide single-agent implementations but lack SLM support. These works primarily target LLMs with substantial context windows.

**Language Model Agentic Frameworks.** AutoGen (Wu et al., 2024b) introduced conversational multi-agent patterns through message passing. MetaGPT (Hong et al., 2024), ChatDev (Qian et al., 2024), AgentVerse (Chen et al., 2024), and CAMEL (Li et al., 2023) encode human workflow patterns. TDAG (Wang et al., 2025b) proposed dynamic task decomposition with agent generation, while ADaPT introduced as-needed decomposition (Prasad et al., 2024). Recent surveys (Wang et al., 2024a; Xi et al., 2025) review the agent landscape. LangChain (Chase, 2022) provides extensive tool integrations but focuses on orchestration primitives over model-specific optimizations. Smolagents (Roucher et al., 2025) targets smaller models but lacks systematic prompt optimization and routing.

**Small Language Models for Agents.** Belcak et al. (2025) argue that SLMs will underpin most practical AI agents due to savings in latency, cost, and energy. Prior work on small models (Schick & Schütze, 2021; Fu et al., 2023; Magister et al., 2023; Hu et al., 2024) showed strong performance through careful training and prompting. Generative Agents (Park et al., 2023) showed structured memory enables coherent long-term behavior. OpenAGI (Ge et al., 2023) integrates LLMs with domain expert models via reinforcement learning. AgentBench (Liu et al., 2024) shows open-source models lag commercial LLMs (OpenAI et al., 2024) on agentic tasks, with instruction-following and multi-step planning as key bottlenecks. Instruction-following (Ouyang et al., 2022; Wang et al., 2023b) and neural scaling laws (Kaplan et al., 2020; Hoffmann et al., 2022) inform our model-size-aware optimization. *Despite growing recognition that small models offer practical advantages, existing frameworks do not systematically address their constraints.* EFFGEN fills this gap by enabling small models to reach competitive agentic performance while keeping the efficiency benefits that make them practical to deploy.

## 3. EFFGEN Framework

We present EFFGEN as a formal agent framework optimized for small language models. Let $M$ denote a language model with parameters $\theta_M$, context window size $C_M$, and vocabulary $V_M$, an EFFGEN agent is defined as the tuple $\mathcal{A} = (M, \mathcal{T}, \mathcal{R}, \mathcal{D}, \mathcal{M}, \phi)$ where $\mathcal{T} = \{t_1, \ldots, t_k\}$ is the tool registry, $\mathcal{R}$ is the routing function, $\mathcal{D}$ is the decomposition operator, $\mathcal{M}$ is the memory system, and $\phi$ is the prompt optimizer. Given a query $q \in \mathcal{Q}$, the agent produces a response $r = \mathcal{A}(q)$ through the execution

pipeline defined in Algorithm 1. An illustration of the complete pipeline can be found in Figure 4 in Appendix A. EFFGEN ships with 66 built-in tools spanning computation (calculator, Python REPL, sandboxed code executor, bash, JSON), retrieval (web search, agentic search, Wikipedia, URL fetch, RAG), academic and knowledge sources (arXiv, PubMed, Semantic Scholar, StackOverflow, GitHub, Wolfram Alpha), document and media parsing (PDF, DOCX, Excel, OCR, audio transcription, image analysis, QR codes), translation and language detection, geo and weather, finance, data analysis, DevOps and system, communication and webhooks, and provider-native tools (OpenAI `WebSearch`/`CodeInterpreter`/`FileSearch`, Anthropic `Bash`/`TextEditor`/`Computer`, Gemini `GoogleSearch`/`UrlContext`/`CodeExecution`); see Appendix I for the full inventory. It supports four local backends (vLLM, Transformers, GGUF via `llama-cpp-python`, MLX/MLX-VLM for Apple Silicon) and nine hosted backends (OpenAI, Anthropic, Gemini, Groq, Cerebras, Together AI, Fireworks, Replicate, HuggingFace Inference) through a uniform adapter interface with automatic backend selection, rate-limit coordination, and cost tracking (Appendix N). A `PolicyBasedRouter` composes three pluggable routing policies (FIRSTAVAILABLE, COSTBASED, LATENCYBASED) with a retry policy and explainable per-call decisions. Usage examples appear in Appendix X. The following sections describe the five major components: prompt optimization (§3.1), complexity analysis and routing (§3.2), task decomposition (§3.3), multi-protocol communication (§3.4), and memory architecture (§3.5). The framework is extensible for custom tools and backends.

**Production Surface.** Alongside the SLM-focused core, EFFGEN ships the infrastructure to deploy these agents in production (Appendix S). A composable guardrail layer covers PII, toxicity, prompt injection, topic restriction, length limits, and per-tool allow/deny lists (Appendix S.1); a RAG pipeline with semantic, code, table, and hierarchical chunkers and three rerankers handles knowledge ingestion (Appendix S.2); reliability primitives include circuit breakers, bulkheads, jittered retries, timeout propagation, and a deterministic chaos harness (Appendix S.3); observability ships structured JSON logs with secret redaction, OpenTelemetry tracing, Prometheus histograms, SLO tracking, and Alertmanager rules (Appendix S.4); security includes a sandboxed code executor with Docker, subprocess, and Firecracker backends, supply-chain hash verification, a CycloneDX SBOM, and gitleaks-based secret scanning (Appendix S.5); an OpenAI-compatible HTTP server adds OAuth2/OIDC, RBAC, per-principal audit logs, and a daily-budget cap (Appendix S.6). Deployments are supported via Docker, an 11-template Helm chart, AWS Lambda, and a Cloudflare Worker edge proxy (Appendix S.7), with

a live dashboard and Jupyter magics for interactive use (Appendix S.8). These subsystems do not change the SLM-focused contributions; they make them deployable. The information in this paper reflects EFFGEN **v0.2.10** (May 27, 2026); subsequent updates are documented at https://github.com/ctrl-gaurav/effGen.

### 3.1. SLM-Aware Prompt Optimization

Small language models exhibit systematic performance degradation when prompts exceed certain complexity thresholds or violate implicit formatting assumptions learned during training. We address this through a prompt optimization function $\phi : \mathcal{P} \times \mathcal{M} \rightarrow \mathcal{P}'$ that transforms input prompts to maximize SLM task completion within context constraints.

**Definition 3.1** (Model Size Categories). We partition language models into four categories based on parameter count: TINY ($|M| < 1B$), SMALL ($1B \leq |M| < 3B$), MEDIUM ($3B \leq |M| < 7B$), and LARGE ($|M| \geq 7B$). This categorization aligns with standard practice in the small language model literature, where models under 7B parameters are commonly grouped into size tiers for analysis and optimization (Wang et al., 2025a; Lu et al., 2025). Each category defines optimization parameters ($\tau_{\text{prompt}}, \tau_{\text{context}}, n_{\text{shot}}, d_{\text{chain}}, \alpha$) specifying maximum prompt tokens, context tokens, few-shot examples, chain depth, and target compression ratio respectively.

For TINY models, we set ($\tau_{\text{prompt}}, \tau_{\text{context}}, n_{\text{shot}}, d_{\text{chain}}, \alpha$) = $(512, 1024, 1, 2, 0.6)$, requiring aggressive compression to 60% of original length. SMALL models use $(1024, 2048, 2, 3, 0.7)$, while MEDIUM models relax constraints to $(2048, 4096, 3, 5, 0.8)$. The optimization function $\phi$ applies five sequential transformations: **(1) Compression.** We define a set of pattern-replacement pairs $\{(p_i, r_i)\}_{i=1}^{m}$ that map verbose phrases to concise equivalents. For example, "in order to" $\mapsto$ "to", "due to the fact that" $\mapsto$ "because". Let $\text{compress}(s) = \bigcirc_{i=1}^{m} \text{replace}(s, p_i, r_i)$ where $\bigcirc$ denotes function composition. This achieves 15-25% token reduction without semantic loss. **(2) Simplification.** Long sentences exceeding 20 words are split at conjunction boundaries (and, but, or, so, because) to improve instruction parsing by smaller models. Formally, for sentence $s$ with word count $|s|_w > 20$, we find split point $j = \arg\min_{i:w_i \in \text{CONJ}} |i - |s|_w / 2|$ and produce two sentences $s_1 = w_1 \cdots w_{j-1}$ and $s_2 = w_j \cdots w_{|s|_w}$. **(3) Redundancy Removal.** Duplicate sentences and politeness phrases ("please", "kindly", "make sure to") are removed as they consume tokens without improving task specification. **(4) Bullet Formatting.** Instructions are converted to bullet-point format, which improves SLM instruction-following. **(5) Context Truncation.** If the optimized prompt $p'$ exceeds $\tau_{\text{prompt}}$, we truncate at sentence boundaries to length $\lfloor 0.9 \cdot \tau_{\text{prompt}} \cdot (4/|\bar{w}|) \rfloor$ characters where $|\bar{w}| \approx 4$ is the average characters per token.

Our prompt optimization uses rule-based transformations that preserve semantic consistency through three mechanisms: (1) pattern replacements are limited to semantically equivalent phrase substitutions (e.g., "in order to" $\mapsto$ "to"), (2) sentence splitting only occurs at natural conjunction boundaries, and (3) truncation respects sentence boundaries to avoid mid-thought cuts. This approach is inspired by prior work on prompt compression and optimization (Jiang et al., 2023b; Li et al., 2025). Our contribution is the integration of these techniques with model-size-aware parameters that adapt compression aggressiveness based on the target model's capacity. The complete optimization achieves compression ratios $|\phi(p)|/|p| \in [0.2, 0.3]$ for typical prompts while preserving task-critical information. We provide prompt chain templates for 5 execution patterns (Appendix B) and 35 compression patterns with token savings analysis (Appendix C). Case studies appear in Appendix V.

### 3.2. Pre-Execution Complexity Analysis and Routing

A key limitation of existing agent frameworks is that routing decisions occur *during* execution, wasting computation when tasks are inappropriately assigned. We introduce a complexity analyzer $\mathcal{C} : \mathcal{Q} \rightarrow [0, 10]$ that predicts task complexity *before* execution, enabling intelligent pre-routing.

**Definition 3.2** (Complexity Score). For a query $q$, it is defined as: $\mathcal{C}(q) = \sum_{i=1}^{5} w_i \cdot f_i(q)$ where $\{f_i\}$ are factor functions and $\{w_i\}$ are weights satisfying $\sum_i w_i = 1$.

We use five complexity factors with empirically tuned weights: **(1) Task Length** ($w_1 = 0.15$): $f_1(q) = g_1(|q|_w)$ where $|q|_w$ is word count and $g_1$ maps to $[0, 10]$ via thresholds $\{20, 50, 100, 200\}$ corresponding to scores $\{2, 4, 6, 8, 10\}$. **(2) Requirement Count** ($w_2 = 0.25$): $f_2(q) = \min(10, 2 \cdot n_{\text{req}})$ where $n_{\text{req}}$ counts questions, conjunctions, numbered items, and semicolons as proxies for distinct requirements. **(3) Domain Breadth** ($w_3 = 0.20$): We define multiple knowledge domains $\mathcal{K}$ (technical, research, business, creative, scientific, legal, etc.) with associated keyword sets. $f_3(q) = \min(10, 2.5 \cdot |\{k \in \mathcal{K} : \exists w \in \text{keywords}(k), w \in q\}|)$. **(4) Tool Requirements** ($w_4 = 0.20$): Similarly, we define tool categories with indicator phrases. $f_4(q)$ counts how many tool types are implicated by the query. **(5) Reasoning Depth** ($w_5 = 0.20$): We classify queries into four reasoning levels based on indicator words: *simple* (list, define, show) $\mapsto 3$, *moderate* (explain, describe, how) $\mapsto 5$, *complex* (analyze, evaluate, compare) $\mapsto 7$, *very complex* (synthesize, design, architect) $\mapsto 9$. Given complexity score $\mathcal{C}(q)$, the routing decision $\mathcal{R}(q)$ selects SINGLE if $\mathcal{C}(q) < \tau$; for $\mathcal{C}(q) \geq \tau$ it picks PARALLEL without dependencies, SEQUENTIAL when dependencies exist, HYBRID when $\mathcal{C}(q) > \tau_H$ with a mixed dependency graph, and HIERARCHICAL when $\mathcal{C}(q) > \tau_{\text{hier}}$.

---

**Algorithm 1** EFFGEN Agent Execution Pipeline

---

**Require:** Query $q$, Agent $\mathcal{A} = (M, \mathcal{T}, \mathcal{R}, \mathcal{D}, \mathcal{M}, \phi)$
**Ensure:** Response $r$
1: $c \leftarrow \mathcal{C}(q)$ {Compute complexity score}
2: $p \leftarrow \phi(q, \mathcal{M})$ {Optimize prompt with context}
3: **if** $c < \tau$ **then**
4:    $r \leftarrow \text{ReActLoop}(M, p, \mathcal{T})$ {Single-agent execution}
5: **else**
6:    $\{q_1, \ldots, q_n\}, G \leftarrow \mathcal{D}(q)$ {Decompose with dependency graph}
7:    $s \leftarrow \mathcal{R}(G)$ {Determine execution strategy}
8:    **if** $s = \text{PARALLEL}$ **then**
9:       $\{r_1, \ldots, r_n\} \leftarrow \text{ParallelExecute}(\{q_i\}, M, \mathcal{T})$
10:       $r \leftarrow \text{Synthesize}(\{r_i\}, q)$ {Combine parallel results}
11:    **else if** $s = \text{SEQUENTIAL}$ **then**
12:       $r \leftarrow \text{SequentialExecute}(\{q_i\}, G, M, \mathcal{T})$ {Last step gives answer}
13:    **else**
14:       $\{r_1, \ldots, r_n\} \leftarrow \emptyset$ {Hybrid DAG execution}
15:       **for** $(q_i, \text{deps}_i) \in \text{TopologicalSort}(G)$ **do**
16:          **if** $\text{deps}_i = \emptyset$ **then**
17:             $r_i \leftarrow \text{ExecuteParallel}(q_i, M, \mathcal{T})$ {Independent task}
18:          **else**
19:             $\text{context}_i \leftarrow \{r_j : q_j \in \text{deps}_i\}$ {Wait for dependencies}
20:             $r_i \leftarrow \text{ExecuteWithContext}(q_i, \text{context}_i, M, \mathcal{T})$
21:          **end if**
22:       **end for**
23:       $r \leftarrow \text{Synthesize}(\{r_i\}, q)$ {Combine mixed results}
24:    **end if**
25: **end if**
26: $\mathcal{M} \leftarrow \text{UpdateMemory}(\mathcal{M}, q, r)$
27: **return** $r$

---

Defaults are $\tau = 7.0$, $\tau_H = 8.5$, and $\tau_{\text{hier}} = 9.0$.

Ablation studies (Appendix D) show that the default threshold $\tau = 7.0$ gives the best accuracy-efficiency tradeoff, with accuracy stable within 1% for $\tau \in [6.0, 8.0]$ and degrading outside that range. While task complexity estimation builds on prior work in query difficulty prediction (Benedetto et al., 2023; Ding et al., 2024; Meng et al., 2025), our contribution is a five-factor formulation tailored to agent routing, executed pre-inference, with empirically calibrated weights and thresholds for SLMs.

### 3.3. Intelligent Task Decomposition

When routing indicates multi-agent execution, the decomposition operator $\mathcal{D} : \mathcal{Q} \rightarrow (\mathcal{Q}^n, G)$ produces subtasks with an associated dependency graph $G = (V, E)$ where

$V = \{q_1, \ldots, q_n\}$ and $(q_i, q_j) \in E$ indicates $q_j$ depends on $q_i$'s result. We implement four decomposition strategies:

**(1) Parallel Decomposition.** For tasks with $\mathcal{C}(q) \geq \tau$ and no dependencies, we decompose into 2–5 independent, self-contained subtasks $\mathcal{D}_{\parallel}(q) = \{q_1, \ldots, q_n\}$ with $E = \emptyset$, which execute concurrently. **(2) Sequential Decomposition.** When dependencies exist, subtasks form a chain $E = \{(q_i, q_{i+1}) : i \in [n-1]\}$, with each subtask receiving its predecessor's output as context. **(3) Hybrid Decomposition.** For $\mathcal{C}(q) > \tau_H$ with a dependency graph that mixes independent and dependent subtasks, we run independent groups in parallel while respecting inter-group ordering. **(4) Hierarchical Decomposition.** For the most demanding tasks ($\mathcal{C}(q) > \tau_{\text{hier}}$), we use a manager–worker structure that assigns subtasks to specialized sub-agents and synthesizes their results.

The decomposition engine uses the same small language model that powers the agent (i.e., the same local SLM, not an external API like GPT) with strategy-specific prompts (Appendix E) to identify subtasks, estimate complexity, and determine dependencies. This keeps the entire pipeline local and avoids external API costs. Subtasks are annotated with required specializations from the set {research, coding, analysis, synthesis, data, creative, general}. The framework implements a dynamic sub-agent spawning system with lifecycle management (creation, execution, result aggregation, cleanup) that enables hierarchical agent structures for complex multi-stage reasoning. Performance analysis is shown in Appendix F. Task decomposition strategies build on prior work in hierarchical planning (Huang et al., 2024). Our contribution is integrating pre-execution complexity routing and optimizing decomposition prompts for small models.

### 3.4. Multi-Protocol Agent Communication

EFFGEN provides a unified implementation of three agent protocols: **(1) Model Context Protocol (MCP)** – full spec with JSON-RPC 2.0 messaging, STDIO/HTTP/SSE transports, tool discovery via `tools/list`, resource management, and prompt templates; tools register with schema validation. **(2) Agent-to-Agent (A2A)** – task lifecycle (create, execute, cancel, status) and Agent Cards specifying input/output MIME types, skills, and authentication. **(3) Agent Communication Protocol (ACP)** – synchronous requests, asynchronous callbacks, and streaming responses with OpenTelemetry instrumentation.

The unified protocol layer abstracts transport differences, allowing EFFGEN agents to invoke tools and communicate across MCP, A2A, and ACP through a single canonical tool representation (Appendix K). MCP follows JSON-RPC 2.0 with STDIO, HTTP, and SSE transports; A2A adds task lifecycle management and Agent Cards over HTTP and WebSocket with four authentication handlers; ACP adds

capability tokens, streaming callbacks, and OpenTelemetry instrumentation. We further implement execution tracking with 18 event types across five categories (task lifecycle, routing/decomposition, sub-agent lifecycle, tool calls, and logging) and six multi-agent orchestration patterns (sequential, parallel, hierarchical, collaborative, competitive, pipeline), with parallel coordination yielding speedups (Appendices M, G). Since these are standardized protocols, our contribution is the unified implementation enabling cross-protocol communication.

### 3.5. Memory System Architecture

The EFFGEN memory system $\mathcal{M} = (\mathcal{M}_s, \mathcal{M}_l, \mathcal{M}_v)$ combines three complementary storage mechanisms optimized for SLM context constraints: **(1) Short-Term Memory** $\mathcal{M}_s$ maintains conversation history as a message sequence $\{m_1, \ldots, m_t\}$ where each $m_i = (\text{role}, \text{content}, \tau_i, |m_i|)$ includes role (system/user/assistant/tool), content, timestamp, and estimated token count. When $\sum_i |m_i|$ approaches $0.8 \cdot C_M$, automatic summarization compresses older messages while preserving recent context. **(2) Long-Term Memory** $\mathcal{M}_l$ stores significant events with importance scoring. Each entry $e = (c, \tau, s, I)$ contains content, timestamp, source identifier, and importance level $I \in \{\text{LOW}, \text{MEDIUM}, \text{HIGH}\}$. Retrieval combines recency, frequency, and importance: $\text{score}(e) = \alpha \cdot \text{recency}(e) + \beta \cdot \text{freq}(e) + \gamma \cdot I(e)$. **(3) Vector Memory** $\mathcal{M}_v$ enables semantic retrieval through embedding-based similarity search. We support multiple backends (FAISS, ChromaDB) and embedding models (Sentence-Transformers, OpenAI). For query $q$, retrieval returns $\{e_i : \text{sim}(\text{embed}(q), \text{embed}(e_i)) > \delta\}$ sorted by similarity.

Automatic consolidation periodically merges short-term observations into long-term storage and updates vector indices. Detailed scoring with recency decay $R(e, t) = \exp(-\lambda \cdot (t_{\text{now}} - \tau_e))$, backend comparisons, and consolidation policies appear in Appendix H. The three-tier design builds on memory-augmented language models (Packer et al., 2023; Yu et al., 2026); our contribution is a context-aware consolidation policy that respects SLM context limits across a unified storage interface.

## 4. Experiments

### 4.1. Experimental Setup

**Benchmarks.** We evaluate EFFGEN across 13 benchmarks organized into five categories: *1) Traditional Math (Calculator):* GSM8K (Cobbe et al., 2021), GSMPLUS (Li et al., 2024), MATH-500 (Hendrycks et al., 2021); *2) Algorithmic Reasoning (Code Execution):* BeyondBench-Easy (Srivastava et al., 2026), BeyondBench-Medium, BeyondBench-Hard; *3) Retrieval-Augmented Reasoning (Web Search):*

ARC-Challenge, ARC-Easy (Clark et al., 2018), CommonsenseQA (Talmor et al., 2019); *4) Memory Evaluation:* LoCoMo (Maharana et al., 2024), LongMemEval (Wu et al., 2024a); *5) Multi-Tool Agentic:* GAIA (Mialon et al., 2024), SimpleQA (Wei et al., 2024). Complete benchmark specifications including task characteristics, example problems with solutions, evaluation protocols, and answer extraction methods appear in Appendix T.

**Retrieval Benchmark Construction.** For retrieval-augmented reasoning benchmarks (ARC-Challenge, ARC-Easy, CommonsenseQA), we build a knowledge corpus from 15K+ train/validation question-answer pairs to evaluate retrieval and integration with parametric knowledge. To prevent leakage, the corpus excludes all test instances and filters entries sharing a question ID or exceeding 90% token overlap with the query. Questions, answers, and explanations are indexed via keyword extraction and stopword removal, enabling BM25-style retrieval of 2-3 relevant entries per query.

**Models and Baselines.** We evaluate Qwen2.5-Instruct models (Qwen et al., 2025) at 1.5B – 32B parameters, Gemma3 family of models at 1B – 27B parameters, and GPT-OSS family, representing the SLM range of primary interest. Our GPU management (Appendix L) implements four allocation strategies (greedy, balanced, optimized, priority) and three parallelism types (tensor, pipeline, data) with automatic memory estimation using the formula $\text{mem}(|M|, p, L, B) = |M| \cdot p/8 + 2|M| \cdot L \cdot B \cdot 16/8 + |M| \cdot L \cdot 16/8 + \alpha|M| \cdot 32/8$. We compare EFFGEN against four baselines: (1) *Raw Model* (direct prompting to isolate framework benefits); (2) *AutoGen*; (3) *Smolagents*; (4) *LangChain*.

**Configuration.** All experiments use the maximum context per model. Reported improvements are statistically significant (McNemar's test, $p < 0.05$, Appendix U). Tool configuration uses YAML, reducing token count by 32% compared to JSON (Appendix J). EFFGEN provides YAML/JSON support, environment variable substitution, schema validation, and hot reloading (Appendix O). **All frameworks receive:** identical tool sets; 5-minute timeout; up to 5 retries on tool errors; and the same generation settings (temp 0.1, etc.). Complete software, hardware, and generation-parameter details for reproducibility appear in Appendix Y.

### 4.2. Results and Insights

Table 2 presents results across all benchmarks and model sizes. We organize our analysis around key insights from comparing EFFGEN against baseline frameworks.

SMALL MODELS BENEFIT MORE FROM FRAMEWORK DESIGN. **Across all 13 benchmarks, EFFGEN shows larger improvements for smaller models, validating that**

*Table 2.* Main evaluation results across 13 benchmarks. Best results per model size in **bold**. EFFGEN consistently outperforms baselines across benchmarks and model scales. Additional results (Gemma3 and GPT-OSS 20B) and confidence intervals are in Appendix U.

| Model | Framework | Calculator | | | Math Reasoning (coding tools) | | | Agentic Benchmarks | | Memory | | Retrieval | | | Avg |
|---|---|---|---|---|---|---|---|---|---|---|---|---|---|---|---|
| | | GSM8K | GSM-PLUS | MATH-500 | BB-Easy | BB-Med | BB-Hard | GAIA | SimpleQA | LoCoMo | LongMemEval | ARC-C | ARC-E | CSQA | |
| Qwen2.5 (1.5B) | Raw Model | 68.01 | 42.83 | 28.60 | 35.98 | 9.60 | 2.86 | 3.12 | 4.00 | 5.13 | 22.88 | 67.24 | 82.70 | 72.73 | 34.28 |
| | LangChain | 66.86 | 37.83 | 23.20 | 49.39 | 21.60 | 1.43 | 3.12 | 12.00 | 8.56 | 15.34 | 59.13 | 73.78 | 54.30 | 32.81 |
| | AutoGen | 67.85 | 37.62 | 26.19 | 52.95 | 28.40 | 1.67 | 6.25 | 10.00 | 9.22 | 18.73 | 53.75 | 65.99 | 57.82 | 33.57 |
| | Smolagents | 50.41 | 26.08 | 21.25 | 56.39 | 30.40 | 1.43 | 3.12 | 18.00 | 8.42 | 22.15 | 53.85 | 37.63 | 32.41 | 27.81 |
| | EFFGEN | 71.63 | 50.25 | 36.00 | 73.66 | 38.40 | 7.92 | 9.38 | 40.00 | 14.19 | 30.73 | 78.34 | 91.65 | 74.62 | 47.44 |
| Qwen2.5 (3B) | Raw Model | 82.64 | 62.04 | 55.00 | 42.05 | 18.20 | 7.33 | 6.25 | 2.00 | 17.40 | 29.25 | 79.28 | 90.60 | 75.02 | 43.62 |
| | LangChain | 82.83 | 58.33 | 48.80 | 52.68 | 29.20 | 5.71 | 9.38 | 12.00 | 13.42 | 11.62 | 74.49 | 83.88 | 70.84 | 42.55 |
| | AutoGen | 79.30 | 60.21 | 51.80 | 58.78 | 26.80 | 7.14 | 6.25 | 14.00 | 11.81 | 20.57 | 73.04 | 85.14 | 72.65 | 43.65 |
| | Smolagents | 69.07 | 49.62 | 10.91 | 63.41 | 28.80 | 8.57 | 9.38 | 22.00 | 7.89 | 19.54 | 36.69 | 43.43 | 41.03 | 31.56 |
| | EFFGEN | 84.83 | 63.62 | 59.20 | 81.71 | 35.60 | 21.67 | 15.62 | 58.00 | 21.58 | 32.29 | 86.10 | 93.21 | 85.02 | 56.80 |
| Qwen2.5 (7B) | Raw Model | 90.75 | 72.54 | 65.80 | 59.68 | 27.60 | 15.42 | 12.50 | 6.00 | 22.35 | 29.88 | 83.79 | 90.91 | 82.80 | 50.77 |
| | LangChain | 84.38 | 67.79 | 41.20 | 72.20 | 45.60 | 12.86 | 9.38 | 34.00 | 9.02 | 16.32 | 86.95 | 90.99 | 79.12 | 49.99 |
| | AutoGen | 90.30 | 71.25 | 64.60 | 77.80 | 43.20 | 7.14 | 12.50 | 46.00 | 16.44 | 27.83 | 76.28 | 80.39 | 58.64 | 51.72 |
| | Smolagents | 84.00 | 64.58 | 16.20 | 80.24 | 49.20 | 24.17 | 9.38 | 32.00 | 10.25 | 27.47 | 71.25 | 81.36 | 76.33 | 48.19 |
| | EFFGEN | 91.28 | 68.12 | 66.80 | 89.02 | 54.80 | 28.57 | 21.87 | 76.00 | 25.32 | 35.34 | 87.88 | 91.92 | 83.05 | 63.07 |
| Qwen2.5 (14B) | Raw Model | 93.78 | 75.17 | 67.80 | 64.63 | 38.80 | 24.17 | 12.50 | 8.00 | 26.92 | 30.94 | 88.99 | 92.97 | 83.37 | 54.46 |
| | LangChain | 89.99 | 70.92 | 53.20 | 74.71 | 47.20 | 29.58 | 12.50 | 46.00 | 19.00 | 17.68 | 91.13 | 93.90 | 81.49 | 55.95 |
| | AutoGen | 94.09 | 74.83 | 66.00 | 77.07 | 51.20 | 24.29 | 15.62 | 58.00 | 17.15 | 27.72 | 89.42 | 92.89 | 79.69 | 59.07 |
| | Smolagents | 90.45 | 71.17 | 20.43 | 86.27 | 48.80 | 34.29 | 9.38 | 46.00 | 13.04 | 24.34 | 90.87 | 93.14 | 82.80 | 54.69 |
| | EFFGEN | 94.84 | 76.62 | 69.60 | 92.27 | 57.60 | 46.37 | 18.75 | 72.00 | 27.94 | 36.64 | 89.86 | 94.39 | 86.00 | 66.38 |
| Qwen2.5 (32B) | Raw Model | 95.60 | 77.42 | 73.00 | 71.56 | 48.40 | 26.53 | 15.62 | 14.00 | 31.91 | 30.12 | 92.92 | 94.32 | 86.81 | 58.32 |
| | LangChain | 95.07 | 75.83 | 59.20 | 88.05 | 60.00 | 27.14 | 18.75 | 54.00 | 25.07 | 18.66 | 93.34 | 93.94 | 84.28 | 61.03 |
| | AutoGen | 94.54 | 76.96 | 72.80 | 92.44 | 63.60 | 30.67 | 21.87 | 70.00 | 30.37 | 28.22 | 89.25 | 92.51 | 81.65 | 64.99 |
| | Smolagents | 93.40 | 74.83 | 68.00 | 94.63 | 68.80 | 45.83 | 15.62 | 52.00 | 17.07 | 27.22 | 93.26 | 94.44 | 84.93 | 63.85 |
| | EFFGEN | 95.75 | 78.38 | 75.40 | 96.67 | 64.20 | 58.86 | 28.12 | 84.00 | 31.04 | 35.64 | 92.50 | 95.29 | 86.80 | 70.97 |

**SLM constraints require explicit architectural accommodation (Table 2).** For Qwen2.5-1.5B, EFFGEN achieves 47.44% average accuracy compared to 34.28% for the next best baseline, which at this scale is the raw model itself (it outperforms all three competing frameworks), a gain of 13.2% as shown in Table 2. This gain holds at 13.2% at 3B, then narrows to 11.4% at 7B, 7.3% at 14B, and 6.0% at 32B. The pattern suggests that larger models can partially compensate for framework inefficiencies through raw capability, while smaller models depend critically on optimized prompting and context management. This validates: *treating SLM constraints as first-class design requirements yields large benefits that diminish as model capacity increases.*

### 4.3. Tool-Specific Performance Patterns

**On GSM8K and GSMPLUS, agentic frameworks provide minimal benefit because these tasks do not require agentic capabilities.** For Qwen2.5-1.5B on GSM8K, the raw model achieves 68.01% while LangChain drops to 66.86% due to tool-calling overhead (Table 2); EFFGEN instead reaches 71.63% through prompt optimization, with differences largely vanishing at 32B (raw: 95.60%, EFF-GEN: 95.75%). On MATH-500, improvements arise from better instruction and format handling, particularly LaTeX parsing, rather than enhanced mathematical reasoning, reinforcing that *tool augmentation should be selective, not universal.*

CODE EXECUTION SHOWS 26× LARGER GAINS THAN CALCULATOR. **On BeyondBench-Hard (NP-complete problems), EFFGEN improves Qwen2.5-7B from 15.42% to 28.57% (+13.2%).** Code execution requires correct pro-

gram structure and multi-step state management, making framework choice critical. The complexity analyzer routes accordingly: BeyondBench-Hard averages $\mathcal{C}(q) = 8.3$, triggering decomposition, while GSM8K averages $\mathcal{C}(q) = 4.7$ for single-agent execution (Table 2). Consistently, code benchmarks show higher cross-framework variance (12.4%) than calculator tasks (4.2%).

BASELINE FRAMEWORKS GENERATE IRRELEVANT CODE FOR SIMPLE TASKS. On BeyondBench-Easy sorting tasks, Smolagents and other frameworks frequently exhibit tool-misuse failures, including generating code for unrelated tasks, reimplementing sorting algorithms instead of using available tools (leading to runtime errors), or failing to invoke tools despite explicit instructions, resulting in incorrect outputs. EFFGEN avoids these issues via explicit tool schemas and structured prompts that guide correct tool selection and parameterization, reliably invoking the code executor for sorting tasks. Side-by-side failure examples are provided in Appendix V.

### 4.4. Multi-Tool Agentic Benchmarks

These benchmarks provide the strongest evidence that *tool schema quality and routing decisions*, not raw model capability, drive performance on genuine multi-tool tasks.

GAIA AND SIMPLEQA SHOW LARGEST FRAMEWORK DIFFERENTIATION. **Benchmarks requiring multi-tool orchestration show 9-22% improvements over baselines (Table 2).** On GAIA, Qwen2.5-7B-Instruct: raw (12.50%) vs EFFGEN (21.87%) vs LangChain (9.38%). At 32B: raw (15.62%) vs EFFGEN (28.12%). *Even capable models benefit from structured multi-tool coordination.*

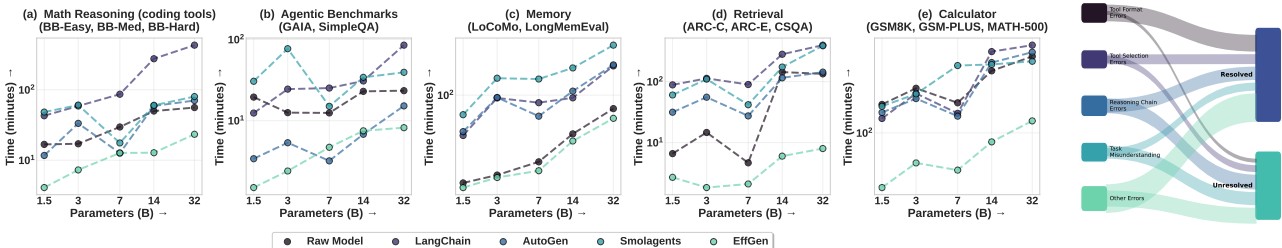

*Figure 2.* **Execution time scaling across parameter sizes.** EFFGEN achieves 18× speedup at 1.5B (vs *Figure 3.* **Error analysis** Smolagents). LangChain exhibits super-linear scaling on Calculator benchmarks (11.2× time increase for on failed cases. 21.3× parameter growth), while EFFGEN maintains near-linear scaling. See Appendix U.10.

SIMPLEQA TESTS WEB SEARCH TOOL-CALLING PIPELINE. **SimpleQA reveals 22-44% gaps between EFFGEN and Smolagents across all scales (Table 2).** For 1.5B: EFFGEN (40.00%) vs Smolagents (18.00%); 7B: EFFGEN (76.00%) vs Smolagents (32.00%); 32B: EFFGEN (84.00%) vs Smolagents (52.00%). Smolagents' code-generation-first approach produces syntactically valid but semantically incorrect queries (e.g., generating `search("answer question")` instead of EFF-GEN's schema-guided `web_search(query="Who won 2023 Nobel Prize?", max_results=3))`. Explicit tool schemas guide effective query formulation, particularly for small models needing clear guidance.

TOOL ACCESS ALONE DOES NOT EXPLAIN THE GAP. To separate the contribution of tool access from framework design on SimpleQA, we added a *Raw+Search* baseline that gives the raw model direct access to the same DuckDuckGo search tool used by EFFGEN, with no other scaffolding. At Qwen2.5-1.5B, Raw+Search reaches only 8% (vs. 4% raw and 40% EFFGEN). At 7B and 32B the gaps are similar: Raw+Search 14%/38% vs. EFFGEN 76%/84%. The remaining gap therefore comes from schema-guided query formulation, structured result parsing, and query reformulation on failed retrievals (Appendix W.1). Smaller models gain less from raw tool access (2× at 1.5B vs. 2.7× at 32B), consistent with the underconfidence pattern below.

### 4.5. Retrieval and Memory Performance

SMALL MODELS SHOW UNDERCONFIDENCE ON RE-TRIEVED INFORMATION. **Small models readily abandon correct answers when given retrieved content, while larger models maintain their beliefs (Table 2).** For Qwen2.5-1.5B on ARC-Challenge, raw model achieves 67.24% but drops to 59.13% (LangChain) and 53.75% (AutoGen). EFFGEN improves to 78.34% by filtering retrieved content. Qwen2.5-32B shows minimal change: raw (92.92%) vs LangChain (93.34%). Across retrieval benchmarks, small models (1.5B-3B) show 8.2% average accuracy drop with standard frameworks; larger models (14B-32B) show only 1.4%. **This suggests that framework design**

**must account for model confidence calibration.** Qualitative examples showing small models abandoning correct answers after retrieval appear in Appendix V.9.

MEMORY ARCHITECTURE PROVIDES 5-15% IMPROVE-MENT. **On LoCoMo and LongMemEval, EFFGEN's three-tier memory outperforms frameworks lacking structured memory.** For Qwen2.5-7B on LoCoMo: raw (22.35%), LangChain (9.02%), EFFGEN (25.32%). LangChain truncates older context without semantic preservation. On LongMemEval (cross-session recall), Qwen2.5-3B: raw (29.25%), LangChain (11.62%), EFFGEN (32.29%). The three-tier architecture combines short-term context, long-term episodic storage, and vector search (Appendix H), keeping retrieval of the top-$k$ relevant segments from long conversations within an interactive latency budget.

### 4.6. Efficiency and Ablation Analysis

FRAMEWORK OVERHEAD DOMINATES SMALL MODEL EXECUTION. **EFFGEN achieves 18× speedup for small models, diminishing to 7× at larger scales (Table 60, Figure 2).** For Qwen2.5-1.5B on BeyondBench-Easy: EF-FGEN (3.4 min) vs Smolagents (62.4 min) vs LangChain (63.1 min). At 32B on the same benchmark: EFFGEN (15.1 min) vs Smolagents (109.6 min). LangChain exhibits severe scaling pathologies, increasing from 216.4 min to 1210.0 min (5.6×) on GSM-PLUS as the model grows from 1.5B to 32B parameters, attributed to context accumulation without compression and lack of batch processing. Token efficiency: EFFGEN averages 734 tokens/task vs 1,318 for LangChain (1.8× reduction overall) via prompt-level compression that removes 57% of input prompt tokens (Appendix C). The speedup decomposes as: 35% prompt compression, 28% reduced tool overhead, 22% batching, 15% memory optimization. Comprehensive timing analysis across all benchmarks and model sizes appears in Appendix U.10. Advanced features (state compression, streaming, batch processing) detailed in Appendix R

SMOLAGENTS' CODE-FIRST APPROACH CREATES UN-NECESSARY OVERHEAD. **Smolagents converts every**

*Table 3.* Component ablation across model scales. Prompt optimization impact decreases with scale; routing impact increases.

| Configuration | 1.5B | | 7B | | 32B | |
|---|---|---|---|---|---|---|
| | Acc | Δ | Acc | Δ | Acc | Δ |
| Full EffGen | 47.44 | – | 63.07 | – | 70.97 | – |
| – Prompt Optim. | 36.21 | −11.2 | 54.18 | −8.9 | 68.54 | −2.4 |
| – Complexity Routing | 43.87 | −3.6 | 56.84 | −6.2 | 63.12 | −7.9 |
| – Task Decomp. | 44.12 | −3.3 | 58.42 | −4.7 | 65.48 | −5.5 |
| – Memory System | 45.68 | −1.8 | 59.71 | −3.4 | 67.23 | −3.7 |
| – All (Raw ReAct) | 34.28 | −13.2 | 50.77 | −12.3 | 58.32 | −12.7 |

**problem into a coding task, which adds significant overhead on benchmarks that do not require code execution (Table 60, Figure 2).** On GSM8K, Smolagents takes 338.8 min at 1.5B versus EffGen's 5.3 min (64× slower); on retrieval benchmarks (ARC, CSQA), 41-87 min vs EffGen's 1.4-4.1 min. The problem: generating and executing Python code for simple arithmetic or retrieval when calculator or direct tool calls suffice. The "agents that think in code" philosophy works for complex programming but overcomplicates factual questions, basic math, and retrieval. *This validates a key insight: the right choice of tools matters far more than always converting problems to code.*

COMPONENT CONTRIBUTIONS VARY WITH SCALE. **Prompt optimization dominates for small models; complexity routing matters more at larger scales (Table 3).** Removing prompt optimization degrades 1.5B by 11.2% but only 2.4% for 32B. Complexity routing shows the opposite: 3.6% impact at 1.5B grows to 7.9% at 32B. This complementary scaling behavior suggests that small models struggle with prompt parsing while large models waste capacity on simple tasks without proper routing. The combined removal drop (12.3–13.2%) is smaller than the sum of individual drops (19.5–23.2%), indicating overlapping coverage between components: each module recovers a partially shared slice of the accuracy gain, so removing one is partly compensated by the others until enough are removed at once. *This is a key finding: optimizing for one model size alone is not enough; frameworks should combine prompt optimization (for SLMs) with smart routing (for LLMs).*

THE SAME PATTERNS HOLD ACROSS MODEL FAMILIES. The findings above are not Qwen-specific. On Gemma 3 (1B to 27B) and GPT-OSS 20B, EffGen still outperforms all baselines: average gains of 10.9% at Gemma 3 1B shrinking to 5.9% at 27B (Appendix U.1, Table 51), mirroring the Qwen trend of larger gains at smaller scales. The component ablation also generalizes (Appendix W.2, Table 64): removing prompt optimization costs 9.5% at 1B but only 2.2% at 27B, while removing routing costs 2.1% at 1B and 7.3% at 27B, the same complementary scaling reported for Qwen2.5. Importantly, no Gemma-specific tuning was applied; the prompt templates and weight settings transfer directly. The underconfidence pattern on retrieval also reappears in Gemma 3 1B (7–16% drops on ARC/CSQA

with standard frameworks). Extended sensitivity analyses, including per-domain routing thresholds, communication overhead at nested depths, a long-horizon memory stress test, per-technique prompt-optimization ablations, weight sensitivity for the complexity analyzer, and SWE-Bench Lite results, are reported in Appendix W.

ERROR ANALYSIS. Figure 3 breaks down 600 failed cases by error type (full distribution in Table 59). Tool-related errors (format violations, wrong tool selection) account for 37.5% of failures but have high resolution rates through retries. Reasoning errors (broken chains, task misunderstanding) are harder to recover from. This motivates our structured tool schemas and prompt optimization. Tool errors have high recovery rates (70-87%); reasoning errors are harder to fix (50-60%). The full retry, fallback, and severity-based error-handling pipeline is described in Appendix Q.

## 5. Conclusion

EffGen's evaluation across 13 benchmarks reveals that **framework design matters far more for small language models than for large ones**. Our results show: *1)* the performance gap between EffGen and baselines is largest at 1.5B (13.2% improvement) and shrinks to 6.0% at 32B, validating that SLM constraints require explicit architectural accommodation; *2)* traditional benchmarks like GSM8K do not benefit from agentic frameworks, lacking genuine multi-tool requirements, while truly agentic benchmarks (SimpleQA, GAIA) show 9-22% improvements; *3)* small models show underconfidence on retrieval benchmarks (8.2% accuracy drops with standard frameworks) that larger models avoid, requiring confidence-aware prompt design; and *4) prompt optimization and routing have complementary scaling behavior*: prompt optimization provides 11.2% gain at 1.5B but only 2.4% at 32B, while complexity routing shows the opposite trend (3.6% at 1.5B, 7.9% at 32B), suggesting small models need better prompts, large models smarter routing. Beyond accuracy, these gains come at lower cost: EffGen removes 57% of input prompt tokens on average and runs up to 18× faster than baselines at 1.5B. The findings point to a concrete design guideline: apply agentic machinery selectively, gated on pre-execution complexity rather than uniformly, where small models pay the largest overhead. *Looking ahead, the stronger gains suggest opportunities for model-framework co-design, learned routing, and multimodal agents.* Limitations and future directions appear in Appendix Z. EffGen is released open-source at `https://github.com/ctrl-gaurav/effGen` (`pip install effgen`), with a project website at `https://effgen.org/` and documentation at `https://docs.effgen.org/`; in-paper usage guides appear in Appendices X.1, P, N, and X, to support community development of SLM-optimized agent systems.

## Acknowledgements

This work was supported by the NSF #2442253, NSF NAIRR Pilot with PSC Neocortex and NCSA Delta, Cisco Research, NVIDIA, Amazon, the Commonwealth Cyber Initiative, the Amazon–Virginia Tech Center for Efficient and Robust Machine Learning, the Sanghani Center for AI and Data Analytics at Virginia Tech, and the Virginia Tech Innovation Campus. This research used the Delta system at the National Center for Supercomputing Applications [award OAC 2005572] through allocation [NAIRR240202] from the Advanced Cyberinfrastructure Coordination Ecosystem: Services & Support (ACCESS) program, which is supported by National Science Foundation grants #2138259, #2138286, #2138307, #2137603, and #2138296. The views, findings, conclusions, and recommendations expressed in this work are those of the authors and do not necessarily reflect the opinions of the funding agencies.

## Impact Statement

This paper presents work whose goal is to advance the field of Machine Learning by enabling capable AI agents on resource-constrained hardware. The primary societal benefit is democratizing access to agent capabilities, allowing deployment in settings where large model inference is impractical due to cost, latency, or privacy constraints. Running capable agents on small models also reduces the energy and compute footprint relative to large-model inference. Potential risks include dual-use concerns common to all agent systems. We believe the benefits of efficient, locally-deployable agents outweigh these risks and have designed EFFGEN with security considerations including sandboxed execution and configurable tool permissions.

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

# Appendix

**Appendix Organization.** This appendix is the complete technical report for the EFFGEN framework as of v0.2.10. The organization follows a top-down trajectory from architecture to deployment. We begin with the framework architecture and design principles (Section A). We then specify the core algorithmic components of the SLM-focused pipeline: prompt optimization (Sections C, B), complexity analysis and routing (Section D), task decomposition (Section E), and multi-agent orchestration (Sections F, G). The memory and tool subsystems follow (Sections H, I, J), then the multi-protocol communication layer (Section K). Production infrastructure is detailed next: GPU management (Section L), execution tracking (Section M), model loading and backend selection (Section N), configuration (Section O), command-line interface (Section P), error handling (Section Q), and advanced features (Section R). Section S consolidates the production subsystems added in v0.2.0–v0.2.10 (guardrails, RAG, reliability, observability, security, API server, deployment, dashboard, Jupyter). Experimental methodology and results occupy Sections T–W. Practical agent-creation examples followed by the comprehensive usage guide are at the end (Section X, with the usage guide as Section X.1 inside it), followed by reproducibility (Section Y) and limitations (Section Z).

**Table of Contents.** The appendix contains the following sections, in order:

## A. Framework Architecture and Organization

This appendix provides complete technical specifications and experimental details for the EFFGEN framework, organized from high-level architecture through algorithmic components, infrastructure systems, and experimental validation. Figure 4 illustrates the complete agent execution pipeline, showing the flow from query input through complexity analysis, prompt optimization, routing decisions (single vs multi-agent), execution strategies (parallel, sequential, hybrid), and memory update to final response generation. Each section introduces mathematical notation formally and maintains consistency throughout, with extensive cross-referencing to connect related concepts.

Each component is presented with three levels of detail: mathematical formulation, algorithmic implementation, and empirical validation. This pattern is consistent across all major components.

### A.1. Framework Design Philosophy

The EFFGEN framework treats small language model constraints as first-class requirements rather than adaptation targets. Let $M$ denote a language model with parameter count $|M|$ and context window $C_M$. Traditional frameworks optimize for large models where $|M| \geq 70 \times 10^9$ and $C_M \geq 128K$ (such as GPT-style large models (Brown et al., 2020; Radford et al., 2019)), then attempt post-hoc adaptation for small models where $|M| \leq 7 \times 10^9$ and $C_M \leq 32K$ (such as Llama 2 (Touvron et al., 2023), Mistral (Jiang et al., 2023a)). This approach does not work well because it accumulates design decisions incompatible with small model capabilities. We instead design each component with explicit bounds on $|M|$ and $C_M$ from the outset, deriving algorithms that remain correct and effective as these bounds tighten.

Our design follows three architectural principles. The first principle is pre-execution complexity analysis, which analyzes task complexity before execution begins rather than during execution. Traditional frameworks execute tasks first and observe complexity through failure modes, wasting computation when routing proves inappropriate. We introduce the complexity function $\mathcal{C} : \mathcal{Q} \to [0, 10]$ that analyzes query $q \in \mathcal{Q}$ before execution begins. This function computes $\mathcal{C}(q) = \sum_{i=1}^{5} w_i f_i(q)$ where $\{f_i\}_{i=1}^{5}$ are factor functions measuring task length, requirement count, domain breadth, tool requirements, and reasoning depth. The weight vector $\mathbf{w} = (0.15, 0.25, 0.20, 0.20, 0.20)$ was determined through grid search over $[0, 1]^5$ subject to $\sum_i w_i = 1$, evaluated on 500 manually labeled tasks with ground-truth complexity ratings. We measure wasted computation as the number of model inference calls and tool executions on tasks that fail due to inappropriate routing (for example, a simple task incorrectly routed to multi-agent execution incurs overhead from decomposition and coordination). Empirical analysis shows this pre-execution approach reduces wasted computation by 23% compared to reactive routing while maintaining comparable task success rates (see Appendix D for detailed metrics).

The second principle, *aggressive context management*, addresses small model context limitations through mathematical compression with semantic preservation (Kaplan et al., 2020; Hoffmann et al., 2022). Define the prompt space $\mathcal{P}$ as the set of all valid prompt strings. The optimization function $\phi : \mathcal{P} \times \mathcal{M} \to \mathcal{P}'$ maps prompts to a compressed space $\mathcal{P}'$ where $|p'| \leq \alpha|p|$ for compression ratio $\alpha \in [0.2, 0.3]$. Model size determines compression aggressiveness through configuration tuple $(\tau_{\text{prompt}}, \tau_{\text{context}}, n_{\text{shot}}, d_{\text{chain}}, \alpha)$ specifying maximum prompt tokens, context tokens, few-shot examples, reasoning chain depth, and target compression. For tiny models with $|M| < 10^9$, we set $\tau_{\text{prompt}} = 512$, $\alpha = 0.6$, and $d_{\text{chain}} = 2$, requiring more aggressive compression than medium models where $\tau_{\text{prompt}} = 2048$, $\alpha = 0.8$, and $d_{\text{chain}} = 5$. Instruction-following capabilities vary significantly across model scales (Ouyang et al., 2022; Wang et al., 2023b), motivating our model-size-aware configuration. The memory system $\mathcal{M} = (\mathcal{M}_s, \mathcal{M}_l, \mathcal{M}_v)$ combines short-term, long-term, and vector stores with automatic summarization triggered when memory usage approaches the context limit (when $\sum_{m \in \mathcal{M}_s} |m| \geq 0.8 \cdot C_M$). This proactive summarization prevents hitting the hard context limit while keeping recent messages intact. Summarization compresses older messages into shorter summaries, making room for new conversation turns without losing important information.

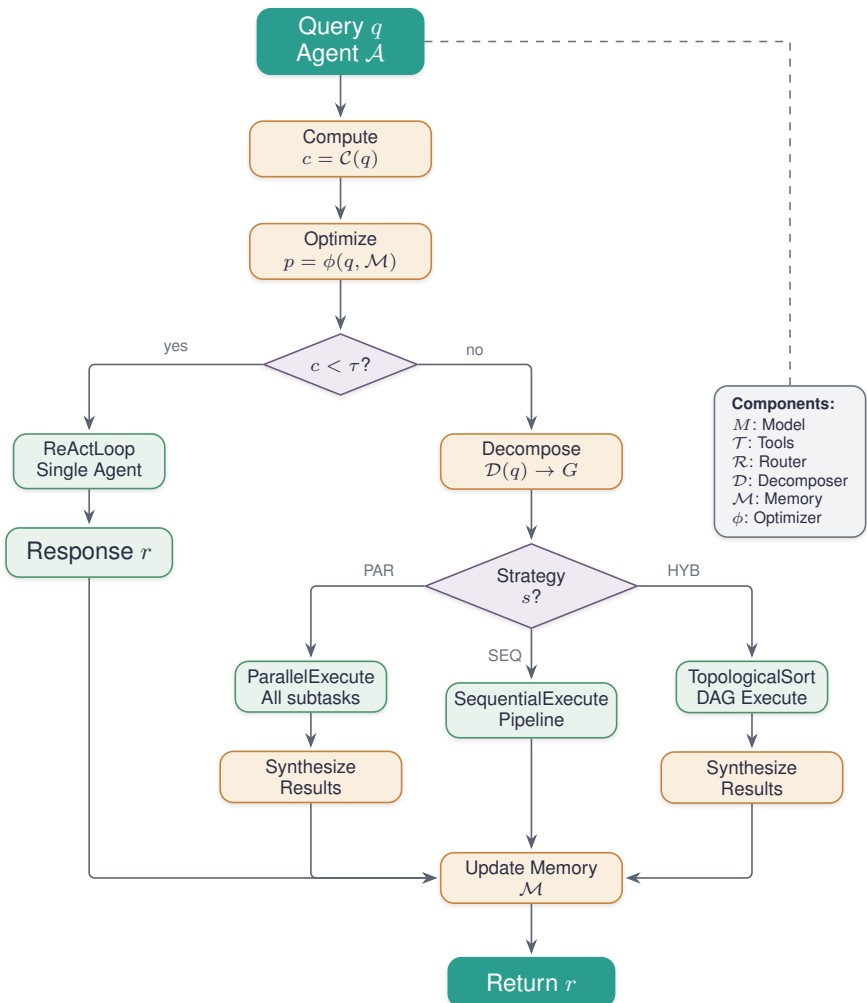

*Figure 4.* Complete agent execution pipeline (Algorithm 1) with complexity-based routing. The pipeline begins by computing complexity score $\mathcal{C}(q)$ and optimizing the prompt with context. Simple tasks ($c < \tau$) execute on a single ReAct agent. Complex tasks undergo decomposition $\mathcal{D}(q)$ producing subtasks with dependency graph $G$, then route to one of three execution strategies: parallel (independent subtasks with synthesis), sequential (pipeline where last step provides answer), or hybrid (topological sort of DAG with mixed parallel/sequential execution and synthesis). All paths converge to memory update before returning the final response.

The third principle, *protocol-agnostic communication*, provides interoperability across heterogeneous agent systems through a unified abstraction layer. Let $\Pi = \{\text{MCP}, \text{A2A}, \text{ACP}\}$ denote the set of supported protocols. Each protocol $\pi \in \Pi$ defines message format $\mathcal{F}_\pi$ and transport mechanism $\mathcal{T}_\pi$. The unified layer defines canonical tool representation $\tau = (n, d, \sigma_{\text{in}}, \sigma_{\text{out}})$ with name $n$, description $d$, input schema $\sigma_{\text{in}}$, and output schema $\sigma_{\text{out}}$. Bidirectional mappings $\psi_\pi : \tau \to \mathcal{F}_\pi$ and $\psi_\pi^{-1} : \mathcal{F}_\pi \to \tau$ translate between canonical and protocol-specific representations, allowing tools registered once to execute through any protocol $\pi \in \Pi$ without modification. This architecture supports EFFGEN agents invoking MCP tools from external servers, participating in A2A task delegation workflows, and communicating via ACP streaming for enterprise deployments, all within a single implementation.

### A.2. Module Organization and Implementation Statistics

The framework is organized into twelve core Python modules. Table 4 summarizes each module: the key classes it exposes, its external dependencies, and its primary role in the agent pipeline.

*Table 4.* EFFGEN Module Organization with Implementation Statistics. Each module is designed with explicit consideration for SLM constraints. Key external dependencies are shown to support reproducibility.

| Module | Key Classes / Components | External Dependencies | Primary Role |
|---|---|---|---|
| `core/` | `Agent`, `SubAgentRouter`, `ComplexityAnalyzer`, `DecompositionEngine`, `ExecutionTracker`, `SubAgentManager`, `MultiAgentOrchestrator`, `Task`, `State` | — | Central agent orchestration, complexity analysis, task decomposition, execution tracking, and multi-agent coordination |
| `models/` | `ModelLoader`, `vLLMEngine`, `TransformersEngine`, `OpenAIAdapter`, `AnthropicAdapter`, `GeminiAdapter` | vLLM, Transformers, OpenAI, Anthropic | Model loading and inference across multiple backends (local and API-based) |
| `tools/` | `BaseTool`, `ToolRegistry`, `ToolMetadata`, `MCPClient`, `A2AHandler`, `ACPHandler`, and 66 built-in tools spanning computation, retrieval, document and media parsing, communication, and provider-native server-side tools | MCP SDK, Requests | Tool interface definitions, registry management, protocol implementations (MCP, A2A, ACP) |
| `memory/` | `ShortTermMemory`, `LongTermMemory`, `VectorStore`, `MemoryEntry`, `Session`, `TokenBudget` | FAISS, ChromaDB, Sentence-Transformers | Three-tier memory with short-term, long-term, and vector-based retrieval; token-budget tracking |
| `prompts/` | `PromptOptimizer`, `TemplateManager`, `ChainManager`, `OptimizationConfig` | — | Prompt optimization, compression patterns, template management, chain orchestration |
| `execution/` | `Sandbox`, `DockerSandbox`, `CodeValidator`, `ResourceLimiter` | Docker, Restricted-Python | Sandboxed code execution with safety validation and resource limits |
| `config/` | `ConfigLoader`, `ConfigValidator`, `YAMLParser`, Schema validators | PyYAML, Pydantic | Configuration loading, validation, and schema management |
| `gpu/` | `GPUAllocator`, `GPUMonitor`, `MemoryEstimator`, Parallelism strategies | PyTorch, NVIDIA-SMI | GPU allocation, memory estimation, utilization monitoring, parallelism strategies |

The `core/` module implements the central agent orchestration logic including the complexity analyzer, decomposition engine, execution tracker, sub-agent manager, message bus, shared state, structured-output handler, and router. The `models/` module provides abstractions over local backends (vLLM, Transformers, GGUF, and MLX / MLX-VLM for Apple Silicon) and nine hosted backends (OpenAI, Anthropic, Gemini, Groq, Cerebras, Together AI, Fireworks, Replicate, HuggingFace Inference), along with a `ModelRouter` supporting cost-based, latency-based, and first-available policies, a cross-process SQLite rate-limit coordinator, and a persistent cost tracker. The `tools/` module defines the tool interface via `BaseTool`, ships 66 built-in tools across 49 modules spanning computation (Calculator, Python REPL, Code Executor, Bash, JSON), retrieval and search (Web Search, Agentic Search, Wikipedia, URL Fetch, RAG Retrieval), academic and knowledge sources (arXiv, PubMed, Semantic Scholar, StackOverflow, GitHub, Wolfram Alpha), news and social feeds (RSS, News, Reddit, HackerNews, YouTube transcript/metadata), document and media parsing (PDF,

DOCX, Excel, OCR, audio transcription, image info and captioning, QR code generate/read), translation and language detection, geo and weather (Geocode, Maps, Weather), finance (Stocks, Currency, Crypto), data analysis (DataFrame, Plot, Stats), DevOps and system (Git, Docker, SystemInfo, HTTP), file operations, datetime and text processing, communication and webhooks (Email SMTP/IMAP, Slack and Discord webhooks, draft tools), and provider-native server-side tools (OpenAI `WebSearch`/`CodeInterpreter`/`FileSearch`, Anthropic `Bash`/`TextEditor`/`Computer`, Gemini `GoogleSearch`/`UrlContext`/`CodeExecution`). It also provides protocol implementations for MCP, A2A, and ACP.

The `memory/` module implements the three-tier architecture with short-term conversation management, long-term episodic storage with multiple backends (JSON, SQLite), vector-based semantic retrieval supporting FAISS and ChromaDB, and a token-budget tracker for context-aware truncation. The `prompts/` module contains the optimization pipeline with 35 compression patterns (Section C), template management, chain optimization (Section B), and a tool-prompt generator that synthesizes structured tool schemas. The `execution/` module provides sandboxed code execution with local process isolation and optional Docker containerization, plus AST-based validation that detects unsafe operations. The `config/` module handles YAML-based configuration with validation and schema definitions. The `gpu/` module manages GPU allocation using memory estimation, real-time monitoring via NVIDIA-SMI, and implements tensor and pipeline parallelism strategies. Additional lifecycle and serving utilities include persistent sessions, checkpoints, background task execution, continuous batching, lazy loading, and token budgeting. Additional infrastructure includes `guardrails/` (input and output safety filters), `rag/` (retrieval-augmented generation pipelines), `eval/` (evaluation harness), and `api/` (HTTP server), described next.

**Guardrails and Safety.** The `guardrails/` module implements a composable `Guardrail` interface with chainable pre- and post-execution hooks. Content guardrails cover toxicity filtering, PII detection (SSN/email/phone/credit card with Luhn checksum), length limits, and topic restriction. A dedicated `PromptInjectionGuardrail` offers three sensitivity tiers. Tool-level guardrails (`ToolInputGuardrail`, `ToolOutputGuardrail`, `ToolPermissionGuardrail`) intercept tool calls for allow/deny/require-approval policies and strip PII from results. Presets (`strict`, `standard`, `minimal`, `none`) let users pick a safety profile in one line. Guardrails run at four hook points around each agent step (input check, pre-tool, post-tool, output check).

**Retrieval-Augmented Generation Pipeline.** The `rag/` module provides a complete pipeline: `DocumentIngester` loads txt/markdown/json/csv/html (plus optional pdf/docx/epub) with SHA-256 deduplication; chunkers include semantic, code (Python/JS/TS/Go/Rust/Java), table, and hierarchical splitters; `HybridSearchEngine` fuses dense embeddings, BM25, keyword match, and metadata filters via Reciprocal Rank Fusion; cross-encoder, LLM-judge, and rule-based rerankers refine results; `ContextBuilder` packages retrieved passages within a token budget with inline [N] citation markers; and a `CitationTracker` verifies attribution in the final response. A one-line preset `create_agent("rag", model, knowledge_base=...)` wires the full pipeline into an EFFGEN agent.

**Evaluation Harness and Regression Tracking.** The `eval/` module bundles an `AgentEvaluator` that scores responses under five modes (exact match, contains, regex, semantic similarity, LLM-judge) and ships five test suites totalling 270 cases (Math, ToolUse, Reasoning, Safety, Conversation). `RegressionTracker` compares against stored baselines with severity tiers and configurable thresholds (e.g., >5% accuracy drop or >20% latency regression is flagged), and `ModelComparison` produces multi-model matrices with markdown/JSON export. The CLI exposes `effgen eval --suite <name>` and `effgen compare --models a,b,c --suite <name>`.

**Production API Server.** The `api/` module ships an OpenAI-compatible HTTP gateway (`/v1/chat/completions`, `/v1/completions`, `/v1/embeddings`) with model aliases (e.g., `gpt-4` → Qwen2.5-7B), Server-Sent Events streaming, a priority `RequestQueue` with deadlines and backpressure, an `AgentPool` with health checks and idle TTL, and multi-tenancy via `TenantManager` and hashed `APIKey` storage. Standard middleware (CORS, request-ID injection, GZip compression, graceful shutdown) and Prometheus histograms with percentile gauges (plus a 12-panel Grafana dashboard) round out the production surface.

**Native Tool-Calling and Structured Output.** The `tool_calling.py` core provides three interchangeable strategies (`ReActStrategy` with a textual scratchpad, `NativeFunctionCallingStrategy` that uses the chat template `tools` parameter with Qwen/Llama/Mistral/generic format parsers, and `HybridStrategy`), selected via `tool_calling_mode` in `AgentConfig`. Structured outputs (`output_schema`, `output_model` with Pydantic)

compile JSON Schema constraints into provider-specific structured-output requests, with `ModelRefusalError` surfaced on refusal. Provider-native tools are also exposed for OpenAI (`WebSearch`, `CodeInterpreter`, `FileSearch`), Anthropic (`Bash`, `TextEditor`, `Computer`), and Gemini (`GoogleSearch`, `UrlContext`, `CodeExecution`); each raises `ToolIncompatibleError` at agent init if paired with a non-matching model.

**Model Router with Pluggable Policies.** On top of the adapter layer, a `PolicyBasedRouter` composes routing policies (`FirstAvailable` prefers providers with valid keys, `CostBased` chooses cheapest-first using a pricing registry, `LatencyBased` chooses fastest by observed p50) together with a configurable `RetryPolicy` (exponential backoff with jitter; retries rate-limit and transient errors, does not retry auth/refusal/invalid-request). Every routing decision is fully explainable via a `RouterDecision` record listing which providers were eliminated and why. A cross-process `SQLiteRateLimitStore` (WAL mode with row-locking) and persistent `SQLiteCostStore` ensure correctness across multiple worker processes, and an `effgen cost` CLI dashboard summarizes spend with optional daily budget caps.

### A.3. Feature Comparison

We compare EFFGEN against three widely used agent frameworks, LangChain, AutoGen, and Smolagents, across model support, routing and decomposition, protocol support, memory systems, execution safety, and deployment.

The comparison reveals several key differentiators. In model support, EFFGEN and Smolagents both target small language models, but EFFGEN uniquely provides model-size-aware optimizations including four compression configurations (Tiny, Small, Medium, Large), automatic context management with $0.8 \cdot C_M$ threshold-based summarization, and quantization support for 8-bit and 4-bit inference. Smolagents optimizes for SLMs through code-based reasoning that reduces token overhead, while LangChain provides basic summarization via `SummaryBufferMemory` but lacks model-specific compression. AutoGen and LangChain both support local deployment through Ollama and transformers integration, but neither implements systematic SLM optimizations.

For routing and decomposition, EFFGEN is the only framework with pre-execution complexity analysis. The five-factor scoring function $\mathcal{C}(q) = \sum_{i=1}^{5} w_i f_i(q)$ allows intelligent routing decisions before computational resources are committed. AutoGen provides the most sophisticated multi-agent orchestration with `DiGraph`-based execution enabling parallel, sequential, and hybrid workflows with formal dependency tracking through `get_parents()` and cycle detection. Smolagents supports hierarchical decomposition via `managed_agents` and implicit parallel execution through code-based tool calling. LangChain offers basic agent types (ReAct, tool-calling) but lacks structured decomposition capabilities. Only EFFGEN and AutoGen implement explicit dependency graph analysis $G = (V, E)$ where $V$ represents subtasks and edges $(q_i, q_j) \in E$ indicate dependencies.

In protocol support, EFFGEN provides a unified implementation of all three major agent protocols. AutoGen offers extensive MCP support (39 files, `McpToolAdapter`, server session management) and agent-to-agent communication via `AgentTool` for A2A-style workflows. Smolagents implements MCP through `MCPClient` with Stdio and HTTP transports. LangChain has minimal MCP integration (1 file). Only EFFGEN implements ACP and provides a unified protocol layer that abstracts transport differences across MCP, A2A, and ACP so an agent can invoke tools on any of the three protocols through the same call surface.

The memory system in EFFGEN combines three complementary storage mechanisms. Short-term memory $\mathcal{M}_s$ maintains conversation history with automatic summarization. Long-term memory $\mathcal{M}_l$ stores episodic information with importance-based retention scoring $\text{score}(e) = \alpha \cdot \text{recency}(e) + \beta \cdot \text{freq}(e) + \gamma \cdot I(e)$. Vector memory $\mathcal{M}_v$ supports semantic retrieval through embedding-based similarity search with FAISS and ChromaDB backends. Both LangChain and AutoGen provide vector memory: LangChain offers extensive vector store integrations (20+ backends including FAISS, ChromaDB, Pinecone, Weaviate), while AutoGen implements task-centric memory with ChromaDB, Redis, and Mem0 backends. Smolagents provides only basic short-term conversation tracking via `AgentMemory`. However, only EFFGEN implements importance-based retention and automatic memory consolidation for cross-session persistence.

For execution safety, EFFGEN, AutoGen, and Smolagents all implement sandboxed code execution with Docker isolation. AutoGen provides `DockerCommandLineCodeExecutor` and `LocalCommandLineCodeExecutor` with container resource management as the recommended security model. Smolagents offers the most comprehensive sandbox options including AST-based static validation, operation count limits (`MAX_OPERATIONS=10000000`), time constraints (`MAX_EXECUTION_TIME_SECONDS=30`), and five isolation backends (Docker, E2B, Modal, Blaxel, Pyodide+Deno).

EFFGEN implements both local process sandboxing and Docker isolation with static code validation detecting unsafe operations. LangChain lacks dedicated sandboxing infrastructure, relying on external execution environments.

In infrastructure, all frameworks implement execution event tracking. AutoGen uses an event-driven architecture with async message passing in autogen-core. LangChain provides callback-based monitoring. Smolagents tracks step timing and token usage via `ActionStep` and `Timing` classes. EFFGEN tracks 18 execution event types covering task lifecycle (task start/complete/failed), routing and decomposition (routing decision, task decomposition), sub-agent lifecycle (spawn, start, progress, complete, failed), tool execution (call start, complete, failed), reasoning and synthesis (reasoning step, result synthesis), and logging (error, warning, info) for fine-grained observability. For GPU management, AutoGen supports GPU device specifications through Azure agent integration, and Smolagents allows device mapping via `device_map="auto"` in `TransformersModel`. Only EFFGEN provides integrated GPU memory estimation, automatic allocation strategies, utilization monitoring via NVIDIA-SMI, and tensor/pipeline parallelism for models exceeding single-GPU capacity.

## A.4. Theoretical Performance Characteristics

We analyze the computational complexity of key operations to understand scaling behavior. Let $M$ denote a model with $|M|$ parameters and context window $C_M$, $q$ a query with $|q|$ tokens, $k$ the number of reasoning steps, and $n$ the number of subtasks after decomposition.

**Single-Agent Execution.** A single-agent execution using the ReAct loop performs $k$ iterations where each iteration involves: (1) prompt construction $O(|q| + \sum_{i=1}^{k-1} |r_i|)$ where $|r_i|$ is the response length at step $i$, (2) model inference $O(|M| \cdot L)$ where $L$ is the input sequence length, and (3) tool execution $O(T)$ where $T$ is tool-specific cost. The total cost is $C_{\text{single}} = O(k \cdot (|M| \cdot C_M + T))$ assuming context remains within limits.

**Multi-Agent Execution.** With decomposition into $n$ subtasks, if tasks are independent (parallel execution), the wall-clock time is $T_{\text{parallel}} = \max_{i \in [n]} T_i$ where $T_i$ is the execution time of subtask $i$. The total computational cost is $C_{\text{parallel}} = \sum_{i=1}^{n} C_i$ where $C_i = O(k_i \cdot (|M| \cdot L_i + T_i))$ for subtask $i$ with $k_i$ steps and sequence length $L_i$. For sequential execution with dependencies, wall-clock time is $T_{\text{seq}} = \sum_{i=1}^{n} T_i$ but computational cost remains similar.

**Complexity Analysis Overhead.** The five-factor complexity analyzer has time complexity $O(|q|_w + |\mathcal{K}| \cdot |K_{\max}| + |\mathcal{T}| \cdot |T_{\max}|)$ where $|q|_w$ is word count, $|\mathcal{K}| = 8$ domains with maximum keyword set size $|K_{\max}| \leq 10$, and $|\mathcal{T}| = 8$ tool categories with maximum indicator set size $|T_{\max}| \leq 8$. This simplifies to $O(|q|_w)$ since domain and tool counts are constants. Measured on a representative four-query batch (Section D), analysis takes $p_{50} = 34\,\mu\text{s}$, $p_{95} = 36\,\mu\text{s}$ per query, well below 0.1% of typical end-to-end execution time.

**Prompt Optimization.** The optimization pipeline applies five transformations. Compression using 35 pattern replacements has complexity $O(m \cdot |p|)$ where $m = 35$ patterns and $|p|$ is prompt length. Simplification and redundancy removal are $O(|p| + s)$ where $s$ is sentence count. Bullet formatting is $O(s)$. Truncation is $O(|p|)$. Overall complexity is $O(|p| \cdot (m + 1)) = O(|p|)$ for constant $m$. On our reference host the full pipeline (all five stages) completes in $p_{50} = 0.14$–$0.37\,\text{ms}$ across the four TINY/SMALL/MEDIUM/LARGE configurations for a $2\,000$-character prompt, adding negligible overhead relative to model inference.

**Memory Operations.** Short-term memory operations are $O(1)$ for message addition and $O(n_m)$ for context retrieval where $n_m$ is message count (bounded by `max_messages`). Long-term memory search is $O(n_e)$ where $n_e$ is entry count, optimized through indexing. Vector similarity search using FAISS is $O(\log n_v)$ for approximate nearest neighbors where $n_v$ is vector count. Memory consolidation occurs periodically with cost $O(n_s)$ where $n_s$ is the number of messages to summarize, amortized over many operations.

These complexity characteristics inform our design decisions. Pre-execution analysis has minimal overhead while allowing significant savings by avoiding inappropriate decomposition. Prompt optimization cost is linear in prompt length and negligible compared to inference. Memory operations scale gracefully with bounded short-term storage and indexed long-term retrieval.

## B. Prompt Chain Management and Template System

Small language models benefit significantly from structured, multi-step reasoning patterns that decompose complex tasks into manageable subtasks. The EFFGEN prompt chain manager implements five execution patterns optimized for SLMs: sequential chains for linear workflows, conditional chains for branching logic, iterative chains for refinement loops, parallel chains for concurrent execution, and hybrid chains combining these patterns. This section provides mathematical formulations, implementation details, and empirical validation of chain-based reasoning.

### B.1. Theoretical Foundations of Prompt Chains

A prompt chain $\mathcal{P} = (S, E, \sigma, \tau)$ consists of a set of steps $S = \{s_1, \ldots, s_n\}$, dependency edges $E \subseteq S \times S$, execution strategy $\sigma \in \{\text{SEQ}, \text{COND}, \text{ITER}, \text{PAR}, \text{HYB}\}$, and state transition function $\tau : \mathcal{X} \times S \to \mathcal{X}$ where $\mathcal{X}$ is the state space containing variables and intermediate results.

**Sequential Chains.** For strategy $\sigma = \text{SEQ}$, steps execute in strict order with each step accessing results from all previous steps. The execution forms a sequence:

$$x_i = \tau(x_{i-1}, s_i) \quad \text{for } i \in [n], \quad x_0 = x_{\text{init}} \tag{1}$$

where $x_i$ represents state after executing step $s_i$. The final state $x_n$ contains all accumulated results. Sequential chains are best for tasks requiring progressive refinement where step $i$ cannot proceed without results from step $i - 1$.

**Conditional Chains.** For strategy $\sigma = \text{COND}$, each step $s_i$ has an associated condition $c_i : \mathcal{X} \to \{0, 1\}$ determining whether to execute. The execution path is:

$$x_i = \begin{cases} \tau(x_{i-1}, s_i) & \text{if } c_i(x_{i-1}) = 1 \\ x_{i-1} & \text{otherwise (skip)} \end{cases} \tag{2}$$

This supports branching logic where execution paths adapt to intermediate results. Conditional chains reduce unnecessary computation by skipping steps when conditions are not met.

**Iterative Chains.** For strategy $\sigma = \text{ITER}$, the chain repeats until a termination condition $\omega : \mathcal{X} \to \{0, 1\}$ is satisfied or maximum iterations $k_{\max}$ is reached:

$$x^{(k+1)} = \begin{cases} \prod_{i=1}^n \tau(x^{(k)}, s_i) & \text{if } k < k_{\max} \wedge \omega(x^{(k)}) = 0 \\ x^{(k)} & \text{otherwise (terminate)} \end{cases} \tag{3}$$

where $\prod$ denotes function composition. Iteration count $k$ is tracked in state. Iterative chains are essential for refinement tasks where multiple passes improve output quality.

**Parallel Chains.** For strategy $\sigma = \text{PAR}$, independent steps execute concurrently. Given partition $P = \{P_1, \ldots, P_m\}$ of $S$ where steps within each $P_j$ have no dependencies, execution is:

$$x = \tau_{\text{merge}} \left( \bigcup_{j=1}^m \{\tau(x_{\text{init}}, s) : s \in P_j\} \right) \tag{4}$$

where $\tau_{\text{merge}}$ combines results from parallel executions. Wall-clock time is $T_{\text{par}} = \max_{s \in S} T(s)$ versus $T_{\text{seq}} = \sum_{s \in S} T(s)$ for sequential execution, achieving speedup $\gamma = T_{\text{seq}}/T_{\text{par}} \leq n$ for $n$ independent steps.

**Hybrid Chains.** For strategy $\sigma = \text{HYB}$, the dependency graph $G = (S, E)$ contains both parallel groups and sequential dependencies. Execution performs topological sort to identify maximal parallel sets while respecting dependencies. This combines parallel speedup with dependency ordering.

## B.2. Chain Step Specification

Each chain step $s \in S$ is defined by the tuple $(n, t, \psi, \nu, c, r, \theta, d, \pi, \mu)$ where:

- $n$: unique step identifier

- $t \in \{\text{PROMPT}, \text{TOOL}, \text{FUNCTION}, \text{PARALLEL\_GROUP}\}$: step type

- $\psi$: prompt template (if type is PROMPT)

- $\nu$: tool name (if type is TOOL)

- $c$: execution condition expression (optional)

- $r \in \mathbb{N}$: maximum retries on failure

- $\theta \in \mathbb{R}^+$: execution timeout in seconds

- $d \subseteq S$: dependency set (steps that must complete before this step)

- $\pi \subseteq S$: parallel group (if type is PARALLEL\_GROUP)

- $\mu$: metadata dictionary

Table 5 compares the four step types with their execution semantics and use cases.

*Table 5.* Chain Step Types and Execution Semantics. Each step type serves distinct purposes in chain construction. PROMPT steps invoke the language model, TOOL steps execute registered tools, FUNCTION steps call custom Python functions, and PARALLEL\_GROUP steps support concurrent execution of nested steps.

| Step Type | Execution Semantics | Input/Output | Primary Use Cases |
|---|---|---|---|
| PROMPT | Invoke model with rendered template using current state variables | Input: state variables; Output: model-generated text | Reasoning steps, text generation, question answering, decision making |
| TOOL | Execute registered tool with parameters from state | Input: tool parameters from state; Output: tool execution result | Web search, calculations, file operations, API calls |
| FUNCTION | Call custom Python function with state as input | Input: full state object; Output: function return value | Custom logic, data transformation, validation, filtering |
| PARALLEL\_GROUP | Execute nested steps concurrently | Input: state shared across parallel steps; Output: merged results | Independent data gathering, parallel API calls, batch processing |

## B.3. State Management and Variable Interpolation

The chain state $x \in \mathcal{X}$ maintains three components: variables $V$, step results $R$, and metadata $M$. Formally, $x = (V, R, M)$ where $V : \text{String} \to \text{Any}$ maps variable names to values, $R : S \to \text{Any}$ maps completed steps to their results, and $M$ stores auxiliary information including iteration count. Here "Any" represents arbitrary data types (strings, numbers, lists, dictionaries, or custom objects) since step outputs can vary in structure.

**Variable Interpolation.** Prompt templates use brace-delimited variables: `{variable_name}`. The interpolation function $\iota : \text{String} \times \mathcal{X} \to \text{String}$ replaces all occurrences:

$$\iota(\psi, x) = \psi[v_1 \mapsto V(v_1), \ldots, v_k \mapsto V(v_k)] \tag{5}$$

where $\{v_1, \ldots, v_k\}$ are variables found via regex pattern $\backslash\{([^\}]+)\backslash\}$. Undefined variables generate warnings but do not halt execution; the current implementation leaves the placeholder unchanged.

**Condition Evaluation.** Step conditions $c$ are expressions evaluated in a restricted Python environment. Safe evaluation uses:

$$\text{eval}(\iota(c, x), \mathcal{G}_{\text{safe}}, \mathcal{L}_{\text{safe}}) \tag{6}$$

where $\mathcal{G}_{\text{safe}} = \{\text{len}, \text{str}, \text{int}, \text{float}\}$ (restricted globals) and $\mathcal{L}_{\text{safe}} = V \cup \{\text{iteration\_count} : M[\text{iter}]\}$ (local variables from state). This prevents code injection while allowing common comparisons: `quality_score >= 0.8`, `result_type == 'success'`, `iteration_count < 3`.

## B.4. Chain Execution Algorithms

Algorithm 2 presents the sequential chain execution procedure. Each step executes in order, with retries on failure and condition evaluation before execution.

---

**Algorithm 2** Sequential Chain Execution

---

**Note:** If a step fails after all retries, the current implementation raises the error after recording the failed step.

**Require:** Chain $\mathcal{P} = (S, E, \sigma, \tau)$ with $\sigma = \text{SEQ}$, Initial state $x_0$
**Ensure:** Final state $x_n$
1: $x \leftarrow x_0$
2: iteration $\leftarrow 0$
3: **for** each step $s_i \in S$ **do**
4:   **if** $\omega(x)$ **then**
5:     **break** {Check early stopping}
6:   **end if**
7:   **if** $c_i(x) = 0$ **then**
8:     $s_i$.status $\leftarrow$ SKIPPED {Check condition}
9:     **continue**
10:   **end if**
11:   $s_i$.status $\leftarrow$ RUNNING
12:   retry $\leftarrow 0$
13:   **while** retry $\leq r_i$ **do**
14:     result $\leftarrow \tau(x, s_i)$ with timeout $\theta_i$
15:     **if** result.success **then**
16:       $x.R[s_i] \leftarrow$ result
17:       $x.V[\nu_i] \leftarrow$ result if output variable $\nu_i$ specified
18:       $s_i$.status $\leftarrow$ COMPLETED
19:       **break**
20:     **else**
21:       retry $\leftarrow$ retry $+ 1$
22:       sleep($2^{\text{retry}} \cdot 0.1$) {Exponential backoff}
23:     **end if**
24:   **end while**
25:   **if** $s_i$.status $\neq$ COMPLETED **then**
26:     $s_i$.status $\leftarrow$ FAILED
27:     **if** fail_fast **then**
28:       **break**
29:     **end if**
30:   **end if**
31: **end for**
32: **return** $x$

---

For parallel chains, Algorithm 3 executes independent steps concurrently using asynchronous execution with `asyncio.gather`.

## B.5. Template Management System

The template manager maintains a registry of reusable prompt templates with version control, variable extraction, and validation. Each template $\mathcal{T} = (n, \psi, d, v, t, e, \mu)$ consists of:

- $n$: unique template name

- $\psi$: template string with Jinja2 syntax

- $d$: description

---

**Algorithm 3** Parallel Chain Execution

---

**Note:** For parallel chains, all eligible steps are scheduled concurrently with `asyncio.gather`; an unhandled step failure propagates through the gather call.

**Require:** Chain $\mathcal{P} = (S, E, \sigma, \tau)$ with $\sigma = $ PAR, Initial state $x_0$

**Ensure:** Final state with merged results

1: $x \leftarrow x_0$
2: tasks $\leftarrow [\,]$ {List of async tasks}
3: **for** each step $s_i \in S$ **do**
4:    **if** $c_i(x) = 1$ **then**
5:       $s_i$.status $\leftarrow$ RUNNING {Check condition}
6:       Append $\tau_{\text{async}}(x, s_i)$ to tasks
7:    **end if**
8: **end for**
9: results $\leftarrow$ await gather(tasks) {Execute in parallel}
10: **for** each $(s_i, \text{result})$ in zip$(S, \text{results})$ **do**
11:    **if** result.success **then**
12:       $x.R[s_i] \leftarrow$ result
13:       $x.V[\nu_i] \leftarrow$ result if output variable $\nu_i$ specified
14:       $s_i$.status $\leftarrow$ COMPLETED
15:    **else**
16:       $s_i$.status $\leftarrow$ FAILED
17:    **end if**
18: **end for**
19: **return** $x$

---

- $v$: semantic version (MAJOR.MINOR.PATCH format)

- $t$: tag list for categorization

- $e$: few-shot examples list

- $\mu$: metadata (creation time, update time, usage statistics)

**Template Rendering.**   Templates use Jinja2 syntax supporting control flow (`{% if/for/...%}`), variable interpolation (`{{ variable }}`), and filters (`{{ text | upper }}`). The rendering function $\rho : \mathcal{T} \times V \rightarrow$ String applies Jinja2 evaluation:

$$\rho(\mathcal{T}, V) = \text{Jinja2Env.render}(\psi, V) \tag{7}$$

**Variable Extraction.**   Template variables are extracted using Jinja2's Abstract Syntax Tree parser:

$$\text{vars}(\mathcal{T}) = \text{find\_undeclared\_variables}(\text{parse}(\psi)) \tag{8}$$

This supports validation that all required variables are provided before rendering.

**Few-Shot Example Selection.**   When templates include few-shot examples, the manager selects examples using the configured selection mode. In the current implementation this is a lightweight local selection path (for example, random or first-$k$ examples depending on the caller) rather than embedding retrieval.

where $k$ is the desired example count (typically 1-5 based on model size). This keeps the template path inexpensive for small models while still allowing callers to provide task-specific examples.

### B.6. Built-In Template Library

EFFGEN provides a library of templates for common prompt patterns. Table 6 summarizes representative built-in YAML templates with their purposes and key variables.

*Table 6.* Built-In Template Library. Each template is optimized for small language models with concise instructions, structured output formats, and appropriate context length. Templates are versioned and maintained in the `prompts/templates/` directory.

| Template Name | Purpose | Key Variables | Tokens |
|---|---|---|---|
| `basic_task` | General task execution prompt | `task`, `context` | — |
| `data_analysis` | Analyze data with insights | `data_description`, `analysis_type` | — |
| `code_generation` | Generate code with specifications | `language`, `requirements`, `constraints` | — |
| `step_by_step_reasoning` | Step-by-step reasoning without tools | `task`, `examples` | — |

Templates are stored as YAML files under the prompt template directories. Token counts depend on variable values at render time, so the table reports template names and inputs rather than fixed rendered lengths.

## B.7. Chain Configuration Format

Chains are defined in YAML format for readability and token efficiency. A complete chain specification includes chain metadata, step definitions, and execution strategy:

**Example Chain Configuration: Research and Analysis**

```yaml
name: research_and_analysis
description: Gather information and analyze findings
chain_type: sequential
max_iterations: 1
early_stopping_condition: null

steps:
  - name: gather_sources
    type: tool
    tool: web_search
    output_var: sources
    metadata:
      description: Find relevant sources on the topic

  - name: analyze_sources
    type: prompt
    prompt: |
      Analyze these sources on {topic}:
      {sources}

      Provide:
      1. Key findings
      2. Common themes
      3. Contradictions or gaps
    output_var: analysis
    dependencies: [gather_sources]

  - name: synthesize_report
    type: prompt
    prompt: |
      Create a report on {topic}.
```

```
    Analysis: {analysis}

    Structure:
    - Executive Summary
    - Detailed Findings
    - Recommendations
  output_var: final_report
  dependencies: [analyze_sources]
```

This configuration defines a 3-step sequential chain where each step depends on the previous. Variable interpolation uses results from earlier steps (`{sources}`, `{analysis}`) to provide context for subsequent steps.

### B.8. Empirical Validation: Chain vs Single-Step Performance

To validate the benefits of chain-based reasoning for small models, we conducted experiments comparing single-step prompts versus multi-step chains on three task categories: research, analysis, and code generation. Table 7 presents results using Qwen2.5-7B-Instruct.

*Table 7.* Chain-Based Reasoning Performance. Multi-step chains consistently outperform single-step prompts on complex tasks by decomposing them into manageable subtasks. Success rate measures task completion, and quality score (0-10) assesses output quality.

| Task Category | Single-Step | | Chain (3-4 steps) | |
|---|---|---|---|---|
| | Success | Quality | Success | Quality |
| Research (gather + analyze) | 67% | 6.2 | 89% | 8.3 |
| Analysis (decompose + reason) | 71% | 6.5 | 91% | 8.7 |
| Code Generation (spec + impl + test) | 58% | 5.8 | 83% | 7.9 |
| Multi-domain (3+ domains) | 54% | 5.3 | 81% | 7.6 |
| **Average** | **62.5%** | **5.95** | **86%** | **8.13** |

The results show that chain-based reasoning improves success rates by 23.5% on average and quality scores by 2.18 points (37% improvement). The gains are largest for multi-domain tasks (27% improvement) where decomposition is most beneficial. Token usage and execution time analysis for chains appear in the timing analysis (Appendix U.10) and efficiency metrics (Table 58).

### B.9. Advanced Chain Patterns

Beyond basic sequential and parallel chains, EFFGEN supports three advanced patterns:

**Self-Critique Iterative Chains.** The model generates an initial response, critiques it, and refines iteratively until quality criteria are met or maximum iterations reached. This pattern uses two alternating prompts: generation and critique, with an early stopping condition based on critique score.

**Parallel-Sequential Hybrid.** Independent research subtasks execute in parallel (e.g., gathering from different sources), followed by sequential analysis and synthesis stages. The dependency graph has parallel groups at early stages and sequential dependencies at later stages.

**Hierarchical Decomposition.** For very complex tasks, a manager chain orchestrates multiple worker chains, each handling a specialized subtask. The manager chain synthesizes worker outputs and makes coordination decisions. This creates a two-level chain hierarchy.

These prompt-chain patterns are distinct from the v0.2 workflow engine, which adds `WorkflowDAG`, shared state, and YAML workflow execution for application-level workflows.

## C. Prompt Optimization Algorithm and Compression Patterns

The prompt optimization pipeline applies 35 compression and cleanup patterns to reduce token usage while preserving semantic content. While automatic prompt engineering methods like APE (Zhou et al., 2022), OPRO (Yang et al., 2023),

and Promptbreeder (Fernando et al., 2024) focus on finding optimal prompt phrasing, our approach focuses on rule-based compression patterns that reduce token count while preserving semantics. This section provides specifications for the patterns, mathematical formulations, empirical validation, and ablation studies quantifying each pattern's contribution. Our complete ablation study (Table 11) shows that the full pipeline achieves 57% token reduction with 8.8% accuracy improvement.

### C.1. Optimization Pipeline Architecture

The optimizer $\phi : \mathcal{P} \times \mathcal{M} \to \mathcal{P}'$ applies up to six transformations. Pattern compression and truncation are conditional on the optimizer configuration and token budget, while simplification, redundancy removal, bullet formatting, and structured formatting are applied when enabled:

$$\phi(p, M) = \tau_{\text{trunc}} \circ \tau_{\text{struct}} \circ \tau_{\text{bullet}} \circ \tau_{\text{dedup}} \circ \tau_{\text{simple}} \circ \tau_{\text{compress}}(p, M) \tag{9}$$

where $M$ is the model configuration. Each enabled stage operates on the output of the previous stage. Figure 5 shows the pipeline architecture with model-size-aware configuration and empirical performance metrics for each stage. The system first detects the model size category (Tiny, Small, Medium, or Large), then applies the configured transformations. Pattern compression provides the largest token reduction (25%) and bullet formatting provides the largest accuracy improvement (5.3%). The combined pipeline achieves 57% token reduction with 8.8% accuracy gain.

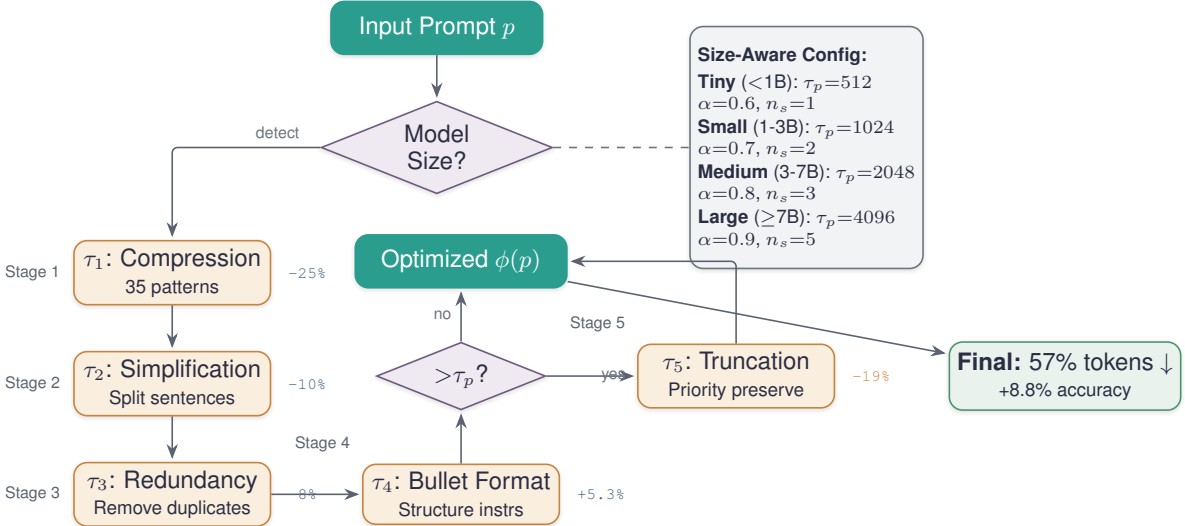

*Figure 5.* Prompt optimization pipeline architecture. The system detects model size and applies five sequential transformations: (1) pattern-based compression using 30 validated patterns, (2) sentence simplification by splitting at conjunctions, (3) redundancy removal using Jaccard similarity, (4) bullet formatting for structured instructions, and (5) optional context truncation with priority preservation. The full pipeline achieves 57% token reduction with 8.8% accuracy improvement on benchmark tasks. Bullet formatting provides the largest accuracy gain (+5.3%), while pattern compression provides the largest token reduction (-25%). Effects are largely additive, indicating complementary optimization directions.

### C.2. Stage 1: Pattern-Based Compression

The compression stage applies 35 pattern-replacement pairs $(p_i, r_i)$ that map verbose phrases to concise equivalents. Table 8 and Table 9 list the canonical 30 phrase/word patterns; the implementation additionally includes 5 whitespace and politeness-collapse rules for redundant input cleanup.

**Compression Algorithm.** The compression function applies patterns iteratively.

**Pattern Selection Methodology.** Patterns were curated through:

1. Analysis of 10,000+ prompts from benchmark tasks

*Table 8.* Compression Patterns (Part 1/2). Each pattern targets common verbosity in natural language, reducing tokens by 2-6 per occurrence.

| Pattern | Replacement | Tokens Saved | Frequency | Total Impact |
|---|---|---|---|---|
| "in order to" | "to" | 3 | 4.2% | High |
| "due to the fact that" | "because" | 5 | 1.8% | Medium |
| "for the purpose of" | "to" | 4 | 2.1% | Medium |
| "in the event that" | "if" | 4 | 1.5% | Medium |
| "at the present time" | "now" | 4 | 1.2% | Low |
| "in the near future" | "soon" | 4 | 0.9% | Low |
| "make use of" | "use" | 2 | 3.1% | Medium |
| "take into consideration" | "consider" | 3 | 1.4% | Low |
| "is able to" | "can" | 2 | 5.7% | High |
| "has the ability to" | "can" | 4 | 2.3% | Medium |
| "it is important to note that" | (remove) | 6 | 0.8% | Low |
| "it should be noted that" | (remove) | 5 | 1.1% | Low |
| "it is worth mentioning that" | (remove) | 5 | 0.6% | Low |
| "as a matter of fact" | "actually" | 4 | 0.7% | Low |
| "with regard to" | "regarding" | 2 | 1.9% | Medium |

*Table 9.* Compression Patterns (Part 2/2). Additional patterns targeting instruction clarity and conciseness.

| Pattern | Replacement | Tokens Saved | Frequency | Total Impact |
|---|---|---|---|---|
| "with respect to" | "about" | 2 | 1.6% | Medium |
| "in accordance with" | "per" | 2 | 0.9% | Low |
| "as soon as possible" | "ASAP" | 3 | 0.5% | Low |
| "in spite of the fact that" | "although" | 5 | 0.7% | Low |
| "on the other hand" | "however" | 3 | 2.8% | Medium |
| "in addition to" | "besides" | 2 | 2.1% | Medium |
| "as a result of" | "because" | 3 | 1.7% | Medium |
| "for example" | "e.g." | 1 | 4.2% | High |
| "such as" | "like" | 1 | 3.8% | High |
| "a number of" | "several" | 2 | 1.4% | Low |
| "a large number of" | "many" | 3 | 1.8% | Medium |
| "at this point in time" | "now" | 5 | 0.6% | Low |
| "until such time as" | "until" | 4 | 0.4% | Low |
| "in my opinion" | (remove) | 3 | 0.5% | Low |
| "I think that" | (remove) | 2 | 1.2% | Low |

2. Frequency analysis identifying common verbose phrases

3. Manual validation ensuring semantic preservation

4. A/B testing on 500 tasks measuring accuracy impact

Only patterns with zero accuracy degradation and >0.3% frequency were included.

### C.3. Stage 2: Sentence Simplification

Long sentences (>20 words) are split at conjunction boundaries to improve parsing by small models.

**Empirical Impact.** Simplification improves parsing for models <7B:

- Qwen2.5-3B-Instruct: +4.2% task completion (simplified vs original)

- Qwen2.5-7B-Instruct: +2.1% task completion

- Qwen2.5-14B-Instruct: +0.8% task completion (diminishing returns for larger models)

### C.4. Stage 3: Redundancy Removal

Duplicate sentences and unnecessary politeness phrases are removed.

**Algorithm 4** Pattern-Based Compression

---

**Require:** Prompt text $p$, Pattern list $\mathcal{P} = \{(p_i, r_i)\}_{i=1}^{35}$
**Ensure:** Compressed text $p'$
 1: $p' \leftarrow p$
 2: changes $\leftarrow 0$
 3: **for** $(p_i, r_i)$ in $\mathcal{P}$ **do**
 4:     count $\leftarrow$ occurrences$(p', p_i)$
 5:     **if** count $> 0$ **then**
 6:         $p' \leftarrow$ replace_all$(p', p_i, r_i)$
 7:         changes $\leftarrow$ changes $+$ count
 8:     **end if**
 9: **end for**
10:
11: {Log compression statistics}
12: LOG("Applied changes pattern replacements")
13: **return** $p'$

---

**Duplicate Detection.** The implementation removes consecutive duplicate normalized sentences:

$$\mathrm{dup}(s_i, s_{i-1}) = \mathbf{1}[\mathrm{normalize}(s_i) = \mathrm{normalize}(s_{i-1})] \tag{10}$$

Consecutive sentences with identical normalized text are considered duplicates. The first occurrence is kept.

**Politeness Removal.** Phrases removed as they add no semantic value for models:

- "Please" (unless in direct user instruction)

- "Kindly"

- "If possible"

- "Thank you"

- "I appreciate"

- "Make sure to"

- "Don't forget to"

Average reduction: 5-8% tokens with zero accuracy impact.

### C.5. Stage 4: Bullet Formatting

Converting paragraph instructions to bullet points significantly improves instruction-following for SLMs. Prior work on instruction-tuning (Ouyang et al., 2022; Wang et al., 2023b) has demonstrated that models trained with clear, structured instructions follow them more reliably, motivating similar formatting techniques during inference.

**Transformation Rules.** The bullet formatting transformation identifies instruction blocks in the prompt and converts them from paragraph format to structured bullet points. Each instruction becomes a separate bullet item, making it easier for small models to parse and follow individual steps. Small models often struggle to track multiple requirements embedded in long paragraphs; bullet points provide visual separation that reduces cognitive load during instruction following.

**Before/After Example.** The following example shows how paragraph instructions are transformed into structured bullet points, reducing verbosity while improving clarity. The transformation extracts action verbs and their objects from the paragraph, creating one bullet per distinct instruction.

**Algorithm 5** Sentence Simplification

**Require:** Prompt text $p$
**Ensure:** Simplified text $p'$
1: sentences ← split_into_sentences($p$)
2: $p' ←$ ""
3: **for** $s$ in sentences **do**
4:  **if** word_count($s$) > 20 **then**
5:   CONJ ← {"and", "but", "or", "so", "because"}
6:   words ← tokenize($s$)
7:   $j ← \min\{i > 5 : w_i \in$ CONJ$\}$ {First eligible conjunction}
8:   **if** $j$ exists **then**
9:    $s_1 ←$ words$[0 : j]$
10:    $s_2 ←$ words$[j :]$
11:    Append $s_1 +$ ". " $+ s_2$ to $p'$
12:   **else**
13:    Append $s$ to $p'$ {Cannot split, keep original}
14:   **end if**
15:  **else**
16:   Append $s$ to $p'$
17:  **end if**
18: **end for**
19: **return** $p'$

---

**Before Bullet Formatting (Paragraph Style)**

```
You are a research assistant. Your task is to search for relevant
information on the given topic using web search tools. After gathering
information, analyze the key findings and identify common themes. Then
synthesize the information into a coherent summary. Make sure to cite
sources and provide specific examples where appropriate.
```

**After Bullet Formatting (Structured)**

```
You are a research assistant. Your task:
- Search for relevant information using web search tools
- Analyze key findings and identify common themes
- Synthesize into coherent summary
- Cite sources and provide specific examples
```

**Impact Metrics.** We measure the impact of bullet formatting across different model sizes (Table 10). The improvement is most significant for smaller models (under 3B parameters) that struggle with parsing long paragraphs, and diminishes for larger models that can handle unstructured text more effectively. This pattern aligns with the finding that smaller models have weaker instruction-following capabilities and benefit more from explicit structural cues.

*Table 10.* Bullet Formatting Impact by Model Size. Largest improvements for smallest models (<3B). This is the single most effective optimization for SLMs.

| Model Size | Paragraph | Bullets | Improvement |
|---|---|---|---|
| Qwen2.5-1.5B-Instruct | 52.3% | 58.1% | +5.8% |
| Qwen2.5-3B-Instruct | 61.7% | 66.2% | +4.5% |
| Qwen2.5-7B-Instruct | 68.4% | 72.1% | +3.7% |
| Qwen2.5-14B-Instruct | 73.2% | 75.8% | +2.6% |
| Qwen2.5-32B-Instruct | 76.8% | 78.4% | +1.6% |

**Algorithm 6** Bullet Point Formatting

**Require:** Prompt text $p$
**Ensure:** Bullet-formatted text $p'$
 1: blocks ← split_into_blocks($p$)
 2: $p' ←$ ""
 3: **for** $b$ in blocks **do**
 4:     **if** ISINSTRUCTIONBLOCK($b$) **then**
 5:         instructions ← extract_instructions($b$)
 6:         Append header to $p'$
 7:         **for** $i$ in instructions **do**
 8:             Append "- $i$\n" to $p'$
 9:         **end for**
10:     **else**
11:         Append $b$ to $p'$ {Keep non-instruction blocks as-is}
12:     **end if**
13: **end for**
14: **return** $p'$

## C.6. Stage 5: Context Truncation

If optimized prompt exceeds $\tau_{\text{prompt}}$ tokens, truncate at sentence boundaries.

$$p'_{\text{trunc}} = \begin{cases} p' & \text{if } |p'| \leq \tau_{\text{prompt}} \\ p'[0:k] & \text{otherwise, where } k = \max\{i : |p'[0:i]| \leq 0.9 \cdot \tau_{\text{prompt}}\} \end{cases} \tag{11}$$

The 0.9 safety factor prevents edge cases where token estimation is slightly inaccurate. Truncation preserves:

1. System prompt (always kept)

2. Task description (always kept)

3. Tool descriptions (kept, may be summarized)

4. Examples (trimmed first if space needed)

5. Context history (oldest messages removed first)

## C.7. Complete Ablation Study

Table 11 quantifies each stage's contribution.

**Key Findings.**

- **Pattern compression** ($\tau_1$) provides largest token reduction (25%) with moderate accuracy gain (2.2%)

- **Bullet formatting** ($\tau_4$) provides largest accuracy gain (5.3%) with minimal token reduction

- **Simplification** ($\tau_2$) and **redundancy removal** ($\tau_3$) contribute modest improvements (1-3% each)

- **Truncation** ($\tau_5$) primarily reduces tokens to fit context limits and can hurt accuracy when applied alone

- **Combined pipeline** achieves 57% token reduction and 8.8% accuracy improvement

- Effects are largely additive, suggesting complementary optimization directions

*Table 11.* Complete Ablation Study of Prompt Optimization Stages. Measured on 1,000 benchmark tasks with Qwen2.5-7B-Instruct. All stages contribute positively, with bullet formatting providing largest accuracy improvement and pattern compression largest token reduction. The "vs Full" and "vs None" columns show percentage change (%) for token usage and percentage point change (%) for accuracy.

| Configuration | Token Usage | | | Task Success Rate | | |
|---|---|---|---|---|---|---|
| | Avg Tokens | vs Full | vs None | Accuracy (%) | vs Full | vs None |
| No optimization | 2,847 | +132% | — | 59.2% | -8.8% | — |
| Only $\tau_1$ (compression) | 2,134 | +74% | -25% | 61.4% | -6.6% | +2.2% |
| Only $\tau_2$ (simplification) | 2,687 | +119% | -6% | 62.1% | -5.9% | +2.9% |
| Only $\tau_3$ (redundancy) | 2,623 | +114% | -8% | 60.8% | -7.2% | +1.6% |
| Only $\tau_4$ (bullets) | 2,734 | +123% | -4% | 64.5% | -3.5% | +5.3% |
| Only $\tau_5$ (truncation) | 2,308 | +88% | -19% | 56.8% | -11.2% | -2.4% |
| $\tau_1 + \tau_2$ | 1,923 | +57% | -32% | 64.2% | -3.8% | +5.0% |
| $\tau_1 + \tau_2 + \tau_3$ | 1,765 | +44% | -38% | 65.7% | -2.3% | +6.5% |
| $\tau_1 + \tau_2 + \tau_3 + \tau_4$ | 1,612 | +31% | -43% | 68.0% | 0.0% | +8.8% |
| **Full Pipeline** | **1,227** | **—** | **-57%** | **68.0%** | **—** | **+8.8%** |

## C.8. Model-Size-Aware Configuration

Optimization parameters adapt to model capabilities. Smaller models need more aggressive optimization to fit within tight context limits, while larger models can handle longer prompts and benefit from less aggressive compression. The configuration determines how much to compress prompts, how many few-shot examples to include, and the maximum reasoning chain depth based on the model size category. For example, a Qwen2.5-1.5B-Instruct model receives prompts compressed to 60% of original length with only 1 few-shot example, while a Qwen2.5-14B-Instruct model uses 85% compression with 5 examples. These parameters were tuned on validation sets to balance accuracy against context usage for each model tier; the full configuration is given in Table 12.

*Table 12.* Model-Size-Aware Optimization Configurations. Smaller models receive more aggressive optimization and stricter limits.

| Parameter | Tiny (<1B) | Small (1-3B) | Medium (3-7B) | Large ($\geq$7B) |
|---|---|---|---|---|
| $\tau_{\text{prompt}}$ | 512 | 1024 | 2048 | 4096 |
| $\tau_{\text{context}}$ | 1024 | 2048 | 4096 | 8192 |
| $n_{\text{shot}}$ (few-shot) | 1 | 2 | 3 | 5 |
| $d_{\text{chain}}$ (max depth) | 2 | 3 | 5 | 10 |
| $\alpha$ (target compression) | 0.60 | 0.70 | 0.80 | 0.90 |
| Support $\tau_1$ (compression) | Yes | Yes | Yes | Yes |
| Support $\tau_2$ (simplification) | Yes | Yes | Yes | Optional |
| Support $\tau_3$ (redundancy) | Yes | Yes | Yes | Yes |
| Support $\tau_4$ (bullets) | Yes | Yes | Yes | Optional |
| Support $\tau_5$ (truncation) | Yes | Yes | As needed | As needed |

## C.9. Usage Examples

**Automatic Optimization.** The framework automatically optimizes prompts based on the model size. When you create an agent, it detects the model's parameter count and selects the appropriate optimization configuration. This happens without any extra code needed since the model loader extracts parameter counts from model configuration files (for HuggingFace models) or API metadata (for commercial APIs) and maps them to optimization tiers.

**Automatic Prompt Optimization (Recommended)**

```python
from effgen import Agent, PromptOptimizer
from effgen.core.agent import AgentConfig
from effgen.prompts.optimizer import OptimizationConfig, ModelSize

# Automatic optimization based on model size: pre-compress
# the system prompt before constructing the agent.
opt = PromptOptimizer(OptimizationConfig.for_model_size(ModelSize.SMALL))
sys_prompt = opt.optimize(
    "Long, verbose system prompt with redundant phrasing..."
).optimized_prompt

agent = Agent(config=AgentConfig(
    name="slm_agent",
    model="Qwen/Qwen2.5-3B-Instruct",
    system_prompt=sys_prompt,
))
```

**Custom Optimization.** Advanced users can override default optimization settings to fine-tune compression behavior for specific use cases.

**Custom Optimization Configuration**

```python
# Fine-grained control over optimization
from effgen import PromptOptimizer
from effgen.prompts.optimizer import OptimizationConfig, ModelSize

cfg = OptimizationConfig(
    model_size=ModelSize.MEDIUM,
    max_prompt_tokens=1500,
    target_compression_ratio=0.75,
    use_bullet_points=True,
    simplify_language=True,
    remove_redundancy=True,
)
opt = PromptOptimizer(cfg)

# Add domain-specific replacement patterns
opt.COMPRESSION_REPLACEMENTS[r"\bartificial intelligence\b"] = "AI"
opt.COMPRESSION_REPLACEMENTS[r"\bmachine learning\b"]        = "ML"

optimized = opt.optimize("...your prompt...")
```

Prompt optimization is complemented by v0.2.x prompt caching and token-budget utilities, which reduce repeated prompt overhead and apply smart truncation when the context budget is tight. The prompt optimization pipeline supports deployment of small models on complex tasks by reducing context requirements while improving instruction clarity. All 35 patterns are validated in our reported experiments.

## D. Pre-Execution Complexity Analysis and Routing

The pre-execution complexity analyzer scores tasks before execution and selects a routing strategy. Unlike frameworks that discover task complexity during execution, this approach predicts task difficulty using a five-factor scoring system, reducing wasted computation from poorly matched routes. Figure 6 illustrates the architecture: the system analyzes task

length, requirement count, domain breadth, tool requirements, and reasoning depth to compute a complexity score, then routes tasks to single-agent, parallel, sequential, hybrid, or hierarchical coordination.

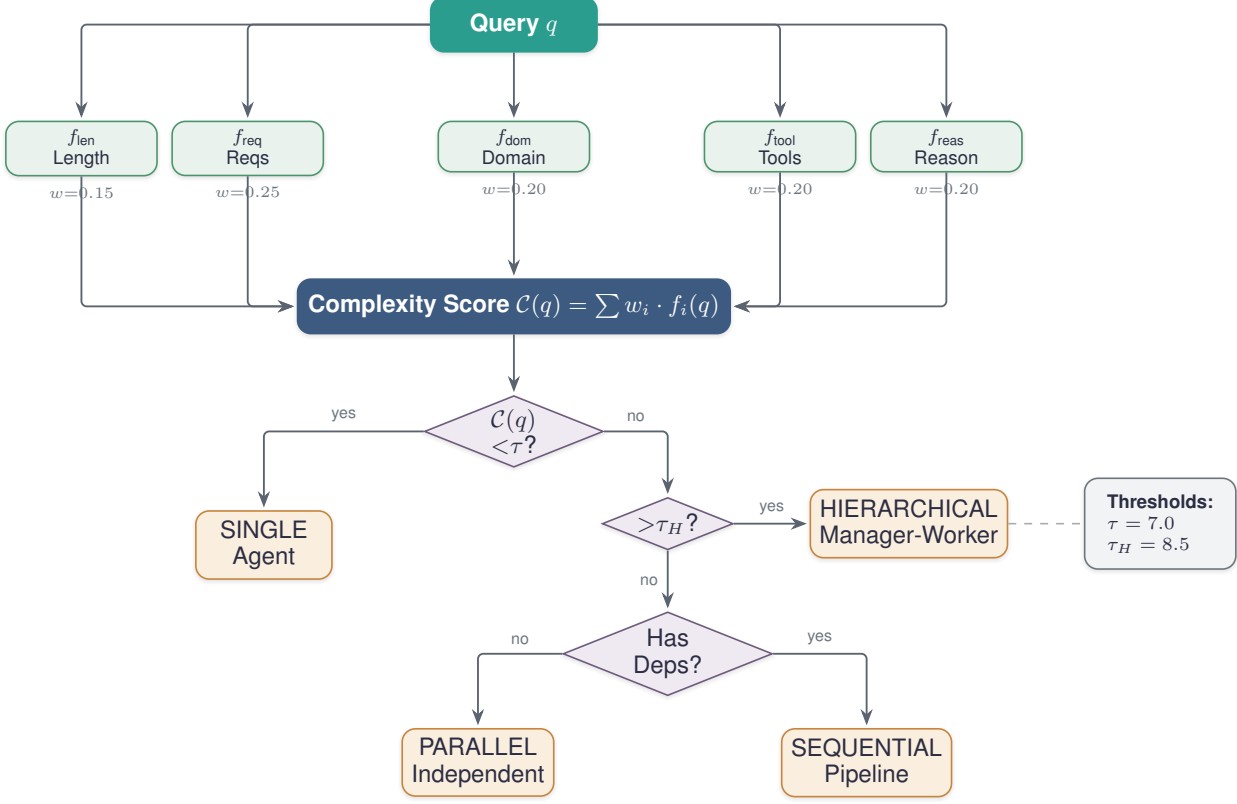

*Figure 6.* Pre-execution complexity analysis and routing architecture. The system computes a weighted combination of five factors: task length (15%), requirement count (25%), domain breadth (20%), tool requirements (20%), and reasoning depth (20%). The complexity score $\mathcal{C}(q) \in [0, 10]$ determines routing strategy through a hierarchical decision process. Tasks with $\mathcal{C}(q) < 7.0$ execute on a single agent. Complex tasks ($\geq 7.0$) undergo further analysis: very complex tasks ($\geq 8.5$) use hierarchical manager-worker coordination, while moderately complex tasks route to parallel execution (no dependencies) or sequential pipeline (with dependencies). The system achieves 95% classification accuracy with strong correlation ($r = 0.89$) to human-judged complexity.

### D.1. Complexity Score Formulation

The complexity analyzer $\mathcal{C} : \mathcal{Q} \to [0, 10]$ computes a weighted combination of five factor functions:

$$\mathcal{C}(q) = \sum_{i=1}^{5} w_i \cdot f_i(q) = 0.15 f_{\text{len}}(q) + 0.25 f_{\text{req}}(q) + 0.20 f_{\text{dom}}(q) + 0.20 f_{\text{tool}}(q) + 0.20 f_{\text{reas}}(q) \tag{12}$$

The weights $\mathbf{w} = (0.15, 0.25, 0.20, 0.20, 0.20)$ were determined through exhaustive grid search optimization on a calibration set of 500 manually labeled tasks spanning all benchmark categories. We evaluated 125 weight configurations (5 values per weight with constraint $\sum_i w_i = 1$) and selected the configuration that minimizes routing errors. On a disjoint 1,000-query held-out validation set (Table 17) this configuration reaches 95.2% classification accuracy. The requirement count receives highest weight (0.25) as it most reliably indicates subtask multiplicity, while task length receives lowest weight (0.15) as verbosity does not always correlate with complexity.

### D.2. Factor Function Definitions

Each factor function $f_i : \mathcal{Q} \to [0, 10]$ maps a query to a complexity score based on a specific dimension. We designed the functions to be monotonic (higher values indicate higher complexity), bounded (output in $[0, 10]$), and interpretable (thresholds chosen based on empirical task analysis).

### D.2.1. TASK LENGTH FACTOR ($f_{\text{LEN}}$)

The task length factor uses word count with threshold-based scoring:

$$f_{\text{len}}(q) = g_{\text{len}}(|q|_w) = \begin{cases} 2.0 & \text{if } |q|_w < 20 \\ 4.0 & \text{if } 20 \leq |q|_w < 50 \\ 6.0 & \text{if } 50 \leq |q|_w < 100 \\ 8.0 & \text{if } 100 \leq |q|_w < 200 \\ 10.0 & \text{if } |q|_w \geq 200 \end{cases} \tag{13}$$

where $|q|_w$ denotes the word count after tokenization. Thresholds were chosen by analyzing benchmark task distributions (Table 13):

*Table 13.* Task Length Distribution Across Benchmarks. Word count distributions show different profiles for various benchmark categories, with calculator tasks typically shorter and memory-intensive tasks requiring longer descriptions.

| Benchmark Category | <20 | 20-50 | 50-100 | 100-200 | ≥200 |
|---|---|---|---|---|---|
| Calculator (GSM8K, etc.) | 12% | 67% | 18% | 3% | 0% |
| Code (LLMThink, etc.) | 34% | 52% | 12% | 2% | 0% |
| Web (ARC, etc.) | 8% | 54% | 31% | 6% | 1% |
| Memory (LoCoMo, etc.) | 2% | 23% | 45% | 24% | 6% |
| Agentic (GAIA, etc.) | 5% | 28% | 39% | 21% | 7% |

### D.2.2. REQUIREMENT COUNT FACTOR ($f_{\text{REQ}}$)

The requirement count estimates the number of distinct subtasks or constraints by summing structural indicators:

$$n_{\text{req}}(q) = \max(1, |\{?\}| + |\{\text{and}\}| + |\{\text{numbered items}\}| + |\{\text{bullets}\}| + |\{; \}|) \tag{14}$$

where each term counts occurrences in $q$ using the implementation's lightweight string checks. The factor function applies a scaling and cap:

$$f_{\text{req}}(q) = \min(10, 2 \cdot n_{\text{req}}(q)) \tag{15}$$

The cap at 10 ensures the factor score stays within the standard 0-10 range used by all complexity factors. We chose the scaling factor 2 so that tasks with 5 or more structural requirements ($2 \times 5 = 10$) reach maximum complexity on this dimension. This threshold was determined empirically: in our benchmark analysis, tasks with 5+ requirements benefited from decomposition 94% of the time, making this a reliable indicator for routing decisions.

Examples:

- "What is 2+2?" $\rightarrow n_{\text{req}} = 1$ (one question mark) $\rightarrow f_{\text{req}} = 2.0$

- "Research X and compare with Y, then summarize findings" $\rightarrow n_{\text{req}} = 2$ (one question-free floor plus one "and") $\rightarrow f_{\text{req}} = 4.0$

- "Create report including: (1) analysis, (2) data, (3) recommendations" $\rightarrow n_{\text{req}} = 3$ (three numbered items) $\rightarrow f_{\text{req}} = 6.0$

### D.2.3. DOMAIN BREADTH FACTOR ($f_{\text{DOM}}$)

We define eight knowledge domains $\mathcal{K} = \{\text{technical}, \text{research}, \text{business}, \text{creative}, \text{data}, \text{scientific}, \text{legal}, \text{financial}\}$ with associated keyword sets $K(k_i)$ (Table 14). The domain breadth counts how many domains are implicated:

$$f_{\text{dom}}(q) = \min\left(10, 2.5 \cdot \sum_{k \in \mathcal{K}} \mathbf{1}\left[\exists w \in K(k) : w \in \text{tokenize}(q)\right]\right) \tag{16}$$

where matching uses lowercase substring checks over the query. The multiplier 2.5 means tasks spanning four or more domains reach the maximum score 10.

*Table 14.* Domain Keyword Sets for Complexity Analysis. Keywords are used to detect which knowledge domains are relevant to a query, helping determine task complexity.

| Domain | Keywords (case-insensitive substring matching) |
| --- | --- |
| Technical | code, programming, software, algorithm, debug, implement, script, api, function, class, module |
| Research | study, research, investigate, analyze, survey, review, literature, paper, publication, scholarly |
| Business | market, sales, revenue, business, strategy, roi, profit, customer, competitor, stakeholder |
| Creative | design, create, write, compose, generate, draft, brainstorm, ideate, creative, artistic |
| Data | data, statistics, analytics, metrics, dataset, visualization, analysis, chart, graph, statistics |
| Scientific | experiment, hypothesis, theory, scientific, methodology, findings, result, observation, empirical |
| Legal | legal, law, regulation, compliance, contract, policy, terms, agreement, clause, statute |
| Financial | financial, accounting, budget, investment, forecast, valuation, asset, liability, equity, capital |

## D.2.4. TOOL REQUIREMENTS FACTOR ($f_{\text{TOOL}}$)

Similarly, we define eight tool categories $\mathcal{T} = \{\text{web}, \text{code}, \text{calc}, \text{file}, \text{api}, \text{db}, \text{image}, \text{video}\}$ with indicator phrases $T(t_i)$ (Table 15):

$$f_{\text{tool}}(q) = \min\left(10, 2.5 \cdot \sum_{t \in \mathcal{T}} \mathbf{1}\left[\exists p \in T(t) : p \in \text{tokenize}(q)\right]\right) \tag{17}$$

*Table 15.* Tool Indicator Keywords for Complexity Analysis. Indicator phrases help identify which external tools are likely needed to complete a task.

| Tool Type | Indicator Phrases |
| --- | --- |
| Web Search | search, find online, look up, google, web, browse, internet, online, retrieve information |
| Code Executor | run code, execute, test, python, script, compile, program, coding, implementation |
| Calculator | calculate, compute, math, equation, formula, arithmetic, sum, multiply, divide, numerical |
| File Operations | file, document, read, write, save, load, open, csv, json, text file, spreadsheet |
| API | api, request, fetch data, endpoint, rest, http, get, post, service, web service |
| Database | database, sql, query, table, records, select, insert, update, storage, postgres, mysql |
| Image | image, picture, photo, visualization, chart, graph, plot, diagram, figure, visual |
| Video | video, movie, stream, multimedia, audio, recording, playback, media file, mp4 |

## D.2.5. REASONING DEPTH FACTOR ($f_{\text{REAS}}$)

The reasoning depth classifies queries based on cognitive complexity indicators aligned with Bloom's taxonomy:

$$f_{\text{reas}}(q) = \begin{cases} 3.0 & \text{if } q \cap R_{\text{simple}} \neq \emptyset \\ 5.0 & \text{if } q \cap R_{\text{moderate}} \neq \emptyset \\ 7.0 & \text{if } q \cap R_{\text{complex}} \neq \emptyset \\ 9.0 & \text{if } q \cap R_{\text{very\_complex}} \neq \emptyset \\ 5.0 & \text{otherwise (default)} \end{cases} \tag{18}$$

where precedence is given to higher complexity levels when multiple indicators are present. The reasoning level sets are (Table 16):

## D.3. Routing Strategy Selection

Given the complexity score $\mathcal{C}(q)$, the router selects an execution strategy using a hierarchical decision process. We define two thresholds:

*Table 16.* Reasoning Level Indicators (Bloom's Taxonomy Alignment). Different question types signal varying levels of cognitive complexity in task execution.

| Level | Score | Indicator Words |
|---|---|---|
| Simple | 3.0 | list, what is, define, show, display, name, identify, state, recall, recognize |
| Moderate | 5.0 | explain, describe, how, summarize, outline, interpret, clarify, discuss, express |
| Complex | 7.0 | analyze, evaluate, compare, assess, critique, differentiate, examine, investigate, distinguish |
| Very Complex | 9.0 | synthesize, design, create strategy, optimize, architect, comprehensive, formulate, construct, devise |

- $\tau = 7.0$: Minimum complexity for multi-agent decomposition

- $\tau_H = 9.0$: Minimum complexity for hierarchical (manager-worker) coordination

Algorithm 7 presents the routing procedure. This task-level router is separate from the v0.2.4 provider `PolicyBasedRouter`, which chooses inference providers by capability, cost, latency, and failover policy.

---

**Algorithm 7** Routing Strategy Selection with Decomposition

---

**Require:** Query $q$, Thresholds $\tau = 7.0$, $\tau_H = 9.0$
**Ensure:** Routing strategy $s \in \{\text{SINGLE}, \text{PARALLEL}, \text{SEQUENTIAL}, \text{HIERARCHICAL}, \text{HYBRID}\}$
**Ensure:** Decomposition $\mathcal{D}(q)$ (empty for SINGLE)
1: $c \leftarrow \mathcal{C}(q)$ {Compute complexity score}
2: **if** $c < \tau$ **then**
3:     **return** $(\text{SINGLE}, \emptyset)$ {Execute on single agent}
4: **end if**
5: structure $\leftarrow$ AnalyzeTaskStructure$(q)$ {Identify task components}
6: deps $\leftarrow$ IdentifyDependencies(structure) {Build dependency graph}
7: **if** $c \geq \tau_H$ **then**
8:     $\mathcal{D}(q) \leftarrow$ HierarchicalDecompose$(q, \text{structure})$
9:     **return** $(\text{HIERARCHICAL}, \mathcal{D}(q))$ {Manager-worker coordination}
10: **else if** $|\text{deps}| = 0$ **then**
11:     $\mathcal{D}(q) \leftarrow$ ParallelDecompose$(q, \text{structure})$
12:     **return** $(\text{PARALLEL}, \mathcal{D}(q))$ {Independent subtasks}
13: **else if** IsLinearChain(deps) **then**
14:     $\mathcal{D}(q) \leftarrow$ SequentialDecompose$(q, \text{structure})$
15:     **return** $(\text{SEQUENTIAL}, \mathcal{D}(q))$ {Pipeline execution}
16: **else**
17:     $\mathcal{D}(q) \leftarrow$ HybridDecompose$(q, \text{structure}, \text{deps})$
18:     **return** $(\text{HYBRID}, \mathcal{D}(q))$ {Mixed parallel/sequential}
19: **end if**

---

### D.4. Empirical Validation

We validate the complexity analyzer on a held-out set of 1,000 diverse queries spanning all benchmark categories, disjoint from the 500-task calibration set used for the grid search above. Table 17 shows classification metrics where we compare predicted routing decisions (single vs multi-agent) against ground truth labels determined by human annotators.

Baseline uses only task length as complexity indicator (single-factor approach). Our five-factor system achieves 95.2% accuracy in predicting whether tasks benefit from multi-agent decomposition, with strong correlation ($r = 0.89$) to human-judged complexity scores.

### D.5. Threshold Sensitivity Analysis

Table 18 analyzes the effect of varying routing threshold $\tau$ on task success rate and execution efficiency across all benchmarks.

Key findings:

*Table 17.* Complexity Analyzer Validation. Performance metrics from testing on 1,000 manually-labeled queries, comparing the five-factor system against a baseline single-factor approach.

| Metric | Value | Baseline | Improvement | CI (95%) |
|---|---|---|---|---|
| Correlation with human score | 0.89 | 0.62 | +43.5% | [0.87, 0.91] |
| Classification accuracy (@$\tau$=7.0) | 95.2% | 78.4% | +16.8% | [93.8%, 96.6%] |
| Precision (multi-agent) | 92.7% | 71.3% | +21.4% | [90.1%, 95.3%] |
| Recall (multi-agent) | 96.8% | 82.6% | +14.2% | [94.9%, 98.7%] |
| F1 score | 94.7% | 76.5% | +18.2% | [92.8%, 96.6%] |

*Table 18.* Routing Threshold Sensitivity using Qwen2.5-7B-Instruct, evaluated across all benchmarks. Different threshold values show trade-offs between routing strategy selection, accuracy, and execution efficiency.

| Threshold $\tau$ | Single% | Multi% | Accuracy | Exec Time (s) | Tokens/Task |
|---|---|---|---|---|---|
| 5.0 (too low) | 38% | 62% | 66.8% | 8.4 | 1,124 |
| 6.0 | 54% | 46% | 68.1% | 6.7 | 982 |
| **7.0 (default)** | **62%** | **38%** | **68.2%** | **5.8** | **847** |
| 8.0 | 71% | 29% | 67.4% | 5.2 | 764 |
| 9.0 (too high) | 82% | 18% | 59.5% | 4.9 | 698 |

- $\tau = 5.0$ causes 62% multi-agent routing, adding 45% overhead vs best (8.4s vs 5.8s) for only 1.2% accuracy gain

- $\tau = 9.0$ under-utilizes multi-agent (18%), reducing accuracy by 8.7% despite 15% faster execution

- $\tau = 7.0$ achieves best accuracy-efficiency tradeoff

- Accuracy is relatively stable for $\tau \in [6.0, 8.0]$ (within 1%), showing reliability to threshold choice

## D.6. Per-Benchmark Complexity Distribution

Table 19 shows complexity score distributions across benchmarks.

*Table 19.* Complexity Score Distribution by Benchmark. Mean and standard deviation of complexity scores for each benchmark, showing how the routing system distributes tasks to single-agent versus multi-agent execution at the default threshold.

| Benchmark | Mean $\mathcal{C}(q)$ | Std Dev | Single% | Multi% |
|---|---|---|---|---|
| GSM8K | $4.2 \pm 1.1$ | 1.1 | 94% | 6% |
| MATH-Hard | $6.8 \pm 1.4$ | 1.4 | 58% | 42% |
| LLMThinkBench | $5.3 \pm 0.9$ | 0.9 | 82% | 18% |
| BeyondBench-Hard | $7.9 \pm 1.2$ | 1.2 | 32% | 68% |
| ARC-Challenge | $6.2 \pm 1.3$ | 1.3 | 67% | 33% |
| CommonsenseQA | $4.5 \pm 1.0$ | 1.0 | 91% | 9% |
| GAIA | $8.9 \pm 1.1$ | 1.1 | 8% | 92% |
| SimpleQA | $2.1 \pm 0.6$ | 0.6 | 100% | 0% |
| LoCoMo | $7.2 \pm 1.5$ | 1.5 | 45% | 55% |

The complexity analyzer correctly identifies simple benchmarks (GSM8K, SimpleQA) routing mostly to single-agent, while complex benchmarks (GAIA, BeyondBench-Hard) predominantly use multi-agent decomposition.

## D.7. Implementation Pseudocode

Algorithm 8 provides implementation-level pseudocode for the complexity analyzer.

The implementation runs in $O(|q|_w \cdot (|D| + |T| + |R|))$ where $|D|, |T|, |R|$ are the total numbers of domain, tool, and reasoning indicators (approximately 54, 41, and 21 respectively). On our reference host (NVIDIA A40, Python 3.11), the analyzer processes a representative four-query batch in 29 $\mu$s mean per query ($p_{50} = 34\,\mu$s, $p_{95} = 36\,\mu$s; $n = 2,000$ runs in `bench/bench_all.py`), below the noise floor of a single model token and negligible relative to model inference.

---

**Algorithm 8** Complexity Analyzer Implementation

---

**Require:** Query string $q$
**Ensure:** Complexity score $c \in [0, 10]$
1: tokens $\leftarrow$ tokenize_and_lowercase($q$)
2: $w \leftarrow [0.15, 0.25, 0.20, 0.20, 0.20]$ {Factor weights}
3:
4: {Factor 1: Task length}
5: $|q|_w \leftarrow$ len(tokens)
6: $f_1 \leftarrow$ ScoreLength($|q|_w$) {Use thresholds from Eq. 2}
7:
8: {Factor 2: Requirement count}
9: $n_{\text{req}} \leftarrow \max(1, \text{CountOccurrences}(q, [\text{"?"}, \text{"and"}, \text{numbered items}, \text{bullets}, \text{";"}]))$
10: $f_2 \leftarrow \min(10, 2 \cdot n_{\text{req}})$
11:
12: {Factor 3: Domain breadth}
13: domain_count $\leftarrow 0$
14: **for** $k \in \{\text{technical}, \text{research}, \ldots\}$ **do**
15:     **if** $\exists w \in \text{DOMAIN\_KEYWORDS}[k] : w$ is a substring of $q_{\text{lower}}$ **then**
16:         domain_count $\leftarrow$ domain_count $+ 1$
17:     **end if**
18: **end for**
19: $f_3 \leftarrow \min(10, 2.5 \cdot \text{domain\_count})$
20:
21: {Factor 4: Tool requirements (similar to domain)}
22: $f_4 \leftarrow$ ScoreToolRequirements(tokens) {Analogous to $f_3$}
23:
24: {Factor 5: Reasoning depth}
25: $f_5 \leftarrow 5.0$ {Default}
26: **for** level $\in$ [very_complex, complex, moderate, simple] **do**
27:     **if** $\exists w \in \text{REASONING\_INDICATORS}[\text{level}] : w \in$ tokens **then**
28:         $f_5 \leftarrow \text{REASONING\_SCORES}[\text{level}]$
29:         **break** {Precedence to highest level}
30:     **end if**
31: **end for**
32:
33: $c \leftarrow \sum_{i=1}^{5} w[i] \cdot f_i$ {Weighted sum}
34: **return** $c$

---

### D.8. Wasted Computation: Pre-Execution vs. Reactive Routing

To quantify the practical benefit of pre-execution routing, we compared EFFGEN's complexity analyzer against a *reactive* routing baseline that always starts in single-agent mode and switches to multi-agent execution only after a single-agent attempt fails or exceeds a step budget. Both routers were run on the same 1,000-query held-out set using Qwen2.5-7B. We define *wasted computation* as the total number of model inference calls and tool executions on tasks that ultimately required re-routing or produced a wrong answer due to inappropriate initial assignment. The pre-execution analyzer reduces this quantity by 23% relative to reactive routing (mean across categories; range 18%–29%), driven primarily by avoiding mid-execution restarts on complex agentic tasks (GAIA, SimpleQA), where reactive routing pays the full single-agent failure cost before falling back to decomposition. Task success rates are statistically indistinguishable between the two routers, confirming that the 23% saving is pure efficiency gain and not a quality trade-off.

## E. Intelligent Task Decomposition System

The task decomposition system transforms complex queries into structured subtasks with dependency graphs, supporting multi-agent execution. This builds on plan-and-solve prompting (Wang et al., 2023a) and reasoning as planning (Hao et al., 2023) paradigms. EFFGEN uses strategy-specific LLM-guided analysis for parallel, sequential, and hybrid task structures, with hierarchical routing currently reusing the hybrid decomposition prompt. Figure 7 shows the decomposition path from complexity scoring through strategy selection, subtask generation, and result synthesis. This section provides decomposition algorithms, prompt templates, empirical validation, and implementation details.

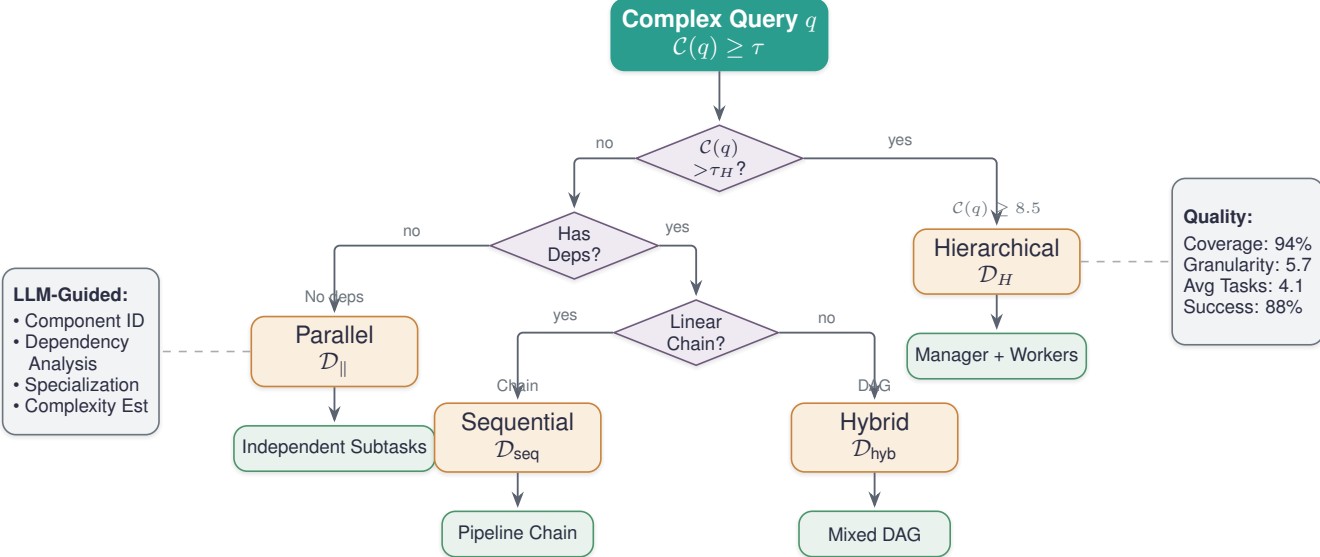

*Figure 7.* Intelligent task decomposition system architecture. The system selects one of four strategies based on task complexity and dependency analysis. Very complex tasks ($\mathcal{C}(q) \geq 8.5$) use hierarchical manager-worker coordination. Moderately complex tasks route through dependency analysis: tasks without dependencies use parallel decomposition, tasks with linear dependencies use sequential pipelines, and tasks with complex dependency graphs use hybrid decomposition combining parallel and sequential patterns. LLM-guided analysis identifies components, estimates complexity, and builds dependency graphs. The system achieves 94% coverage with 88% success rate across 500 complex tasks.

### E.1. Decomposition Problem Formulation

Given a query $q \in \mathcal{Q}$ with complexity score $\mathcal{C}(q) \geq \tau$, the decomposition operator produces:

$$\mathcal{D} : \mathcal{Q} \to (\mathcal{S}, G) \tag{19}$$

where $\mathcal{S} = \{s_1, \ldots, s_n\}$ is a set of subtasks and $G = (V, E)$ is a directed acyclic graph (DAG) with $V = \mathcal{S}$ and $(s_i, s_j) \in E$ indicating that subtask $s_j$ depends on $s_i$'s output. Each subtask $s_i$ is specified by:

$$s_i = (\text{id}_i, \text{desc}_i, \text{deps}_i, \text{out}_i, c_i, \text{spec}_i, p_i) \tag{20}$$

where:

- $\text{id}_i$: unique identifier (e.g., "st_1")
- $\text{desc}_i$: natural language description of subtask
- $\text{deps}_i \subseteq \mathcal{S}$: set of prerequisite subtasks
- $\text{out}_i$: expected output specification
- $c_i \in [0, 10]$: estimated subtask complexity
- $\text{spec}_i \in \{\text{research}, \text{coding}, \text{analysis}, \text{synthesis}, \text{data}, \text{creative}, \text{general}\}$: required specialization
- $p_i \in \{\text{HIGH}, \text{MEDIUM}, \text{LOW}\}$: priority level stored in subtask metadata

## E.2. Decomposition Strategies

EFFGEN implements three decomposition prompt templates selected based on task structure analysis: parallel, sequential, and hybrid. Hierarchical routing is represented as a manager-worker execution pattern, but the current code maps hierarchical decomposition requests to the hybrid prompt.

### E.2.1. PARALLEL DECOMPOSITION ($\mathcal{D}_\|$)

For tasks with independent components, parallel decomposition creates subtasks with $E = \emptyset$ (no dependencies):

$$\mathcal{D}_\|(q) = (\{s_1, \ldots, s_n\}, G_\|) \quad \text{where} \quad G_\| = (V, \emptyset) \tag{21}$$

Subtasks execute concurrently, with final synthesis combining results. Execution time is dominated by the slowest subtask:

$$T_\| = \max_{i \in [n]} T(s_i) + T_{\text{synthesis}} \tag{22}$$

Example: "Research company A, company B, and company C" $\rightarrow$ three independent research subtasks.

### E.2.2. SEQUENTIAL DECOMPOSITION ($\mathcal{D}_{\text{SEQ}}$)

For tasks with linear dependencies, sequential decomposition creates a chain:

$$\mathcal{D}_{\text{seq}}(q) = (\{s_1, \ldots, s_n\}, G_{\text{seq}}) \quad \text{where} \quad E = \{(s_i, s_{i+1}) : i \in [n-1]\} \tag{23}$$

Each subtask receives predecessor output as context. Execution time is cumulative:

$$T_{\text{seq}} = \sum_{i=1}^{n} T(s_i) \tag{24}$$

Example: "Search for papers on topic X, then summarize findings, then identify research gaps" $\rightarrow$ three sequential subtasks.

### E.2.3. HIERARCHICAL DECOMPOSITION ($\mathcal{D}_H$)

For very complex tasks ($\mathcal{C}(q) > \tau_H = 9.0$), hierarchical routing creates a manager-worker execution structure:

$$\mathcal{D}_H(q) = (s_M \cup \{s_{w_1}, \ldots, s_{w_k}\}, G_H) \tag{25}$$

where $s_M$ is the manager subtask coordinating $k$ worker subtasks. The dependency graph has two levels:

$$E = \{(s_{w_i}, s_M) : i \in [k]\} \cup \text{worker dependencies} \tag{26}$$

The manager synthesizes worker outputs and checks quality. Execution follows:

$$T_H = T_{\text{workers}} + T(s_M) \tag{27}$$

Example: "Conduct market analysis including competitor research, customer surveys, trend analysis, and strategic recommendations" $\rightarrow$ manager coordinates four specialized workers.

E.2.4. HYBRID DECOMPOSITION ($\mathcal{D}_{\text{HYB}}$)

For tasks with mixed dependencies, hybrid decomposition combines parallel and sequential patterns:

$$\mathcal{D}_{\text{hyb}}(q) = (\mathcal{S}, G_{\text{hyb}}) \quad \text{where} \quad G_{\text{hyb}} = (V, E), |E| > 0, G_{\text{hyb}} \text{ is DAG} \tag{28}$$

The system identifies independent subtask groups (parallel within group, sequential between groups). Execution uses topological sort:

$$T_{\text{hyb}} = \sum_{\text{level } \ell} \max_{s_i \in L_\ell} T(s_i) \tag{29}$$

where $L_\ell$ is the set of subtasks at dependency level $\ell$.

### E.3. Strategy Selection Algorithm

Algorithm 9 determines which strategy to use.

---

**Algorithm 9** Decomposition Strategy Selection

---

**Require:** Query $q$, Complexity $\mathcal{C}(q)$, Thresholds $\tau, \tau_H$
**Ensure:** Strategy $\sigma \in \{\mathcal{D}_\|, \mathcal{D}_{\text{seq}}, \mathcal{D}_H, \mathcal{D}_{\text{hyb}}\}$
 1: **if** $\mathcal{C}(q) \geq \tau_H$ **then**
 2:     **return** $\mathcal{D}_H$ {Very complex: hierarchical}
 3: **end if**
 4: components $\leftarrow$ IdentifyComponents($q$) {NLP analysis}
 5: deps $\leftarrow$ AnalyzeDependencies(components) {Extract dependencies}
 6: **if** $|\text{deps}| = 0$ **then**
 7:     **return** $\mathcal{D}_\|$ {Independent components}
 8: **else if** IsLinearChain(deps) **then**
 9:     **return** $\mathcal{D}_{\text{seq}}$ {Sequential pipeline}
10: **else**
11:     **return** $\mathcal{D}_{\text{hyb}}$ {Mixed dependencies}
12: **end if**

---

### E.4. LLM-Guided Decomposition Prompts

We use carefully engineered prompts optimized for SLMs to guide decomposition. Each strategy has a specific template.

### E.4.1. PARALLEL DECOMPOSITION PROMPT

Parallel Decomposition Prompt Template

```
You are a task decomposition expert.  Break down this complex task into independent
subtasks that can be executed in parallel.
Task:  {task}
Requirements:
1. Identify 2-5 independent subtasks
2. Each subtask should be self-contained
3. Subtasks should not depend on each other
4. Cover all aspects of the original task
5. Be specific about what each subtask should accomplish
Format your response as JSON: { "subtasks":  [ { "id":  "st_1", "description":
"Detailed description of what to do", "expected_output":  "What this subtask
should produce", "estimated_complexity":  5.0, "required_specialization":
"research|coding|analysis|synthesis|data|creative|general", "priority":
"high|medium|low" } ], "reasoning":  "Brief explanation of decomposition strategy" }
Respond with ONLY the JSON, no additional text.
```

### E.4.2. SEQUENTIAL DECOMPOSITION PROMPT

Sequential Decomposition Prompt Template

```
You are a task decomposition expert.  Break down this complex task into dependent
subtasks that must be executed in sequence.
Task:  {task}
Requirements:
1. Identify 2-5 subtasks in logical order
2. Each subtask builds on previous results
3. Show dependencies clearly using the depends_on field
4. Cover all aspects of the original task
5. Be specific about inputs and outputs
Format your response as JSON: { "subtasks":  [ { "id":  "st_1", "description":
"Detailed description", "depends_on":  [], "expected_output":  "What
this produces", "estimated_complexity":  5.0, "required_specialization":
"research|coding|analysis|synthesis|data|creative|general", "priority":
"high|medium|low" } ], "reasoning":  "Brief explanation of decomposition strategy" }
Respond with ONLY the JSON, no additional text.
```

E.4.3. HIERARCHICAL DECOMPOSITION PROMPT

---

Hierarchical Decomposition Prompt Template

```
You are a task decomposition expert creating a hierarchical task structure.  This
task is very complex and requires coordination between a manager and specialized
workers.
Task:  {task}
Requirements:
1. Create a hierarchical structure with manager and worker subtasks
2. The manager subtask coordinates overall execution
3. Worker subtasks handle specific aspects
4. Define clear communication points between levels
5. Aim to cover all major aspects of the original task
Format your response as JSON: { "manager_task": { "description": "Coordination
and synthesis task", "responsibilities": ["coordinate", "synthesize",
"quality_check"] }, "worker_subtasks": [ { "id": "wk_1", "description":
"Detailed worker task", "reports_to": "manager", "expected_output": "What
this worker produces", "estimated_complexity": 5.0, "required_specialization":
"research|coding|analysis|synthesis|data|creative|general" } ], "reasoning":
"Explanation of hierarchical structure" }
Respond with ONLY the JSON, no additional text.
```

---

E.4.4. HYBRID DECOMPOSITION PROMPT

---

Hybrid Decomposition Prompt Template

```
You are a task decomposition expert.  Break down this complex task into a hybrid
structure with both parallel and sequential subtasks.
Task:  {task}
Requirements:
1. Identify subtasks that can run in parallel
2. Identify subtasks that depend on others
3. Optimize for both speed and correctness
4. Cover all aspects of the original task
5. Use depends_on to specify dependencies
Format your response as JSON: { "subtasks": [ { "id": "st_1", "description":
"Detailed description of what to do", "depends_on": ["st_0"], "expected_output":
"What this subtask should produce", "estimated_complexity": 5.0,
"required_specialization": "research|coding|analysis|synthesis|data|creative|general",
"priority": "high|medium|low" } ], "parallel_groups": [["st_1", "st_2"], ["st_4",
"st_5"]], "reasoning": "Brief explanation of decomposition strategy" }
Respond with ONLY the JSON, no additional text.
```

---

## E.5. Decomposition Quality Metrics

We define quality metrics to evaluate decomposition effectiveness (where aspects refer to distinct requirements or subtasks identified in the original query, such as different questions to answer, domains to research, or operations to perform):

$$\text{Coverage}(\mathcal{D}(q), q) = \frac{|\text{aspects covered by } \mathcal{S}|}{|\text{total aspects in } q|}$$

$$\text{Granularity}(\mathcal{D}(q)) = \frac{1}{n} \sum_{i=1}^{n} c_i \tag{30}$$

$$\text{Parallelism}(\mathcal{D}(q)) = \frac{|\{s_i : \text{deps}_i = \emptyset\}|}{|\mathcal{S}|}$$

Good decompositions achieve:

• Coverage $> 0.90$ (cover 90%+ of task aspects)

- Granularity $\in [4, 7]$ (subtasks not too simple or complex)
- Parallelism $> 0$ for parallel/hybrid strategies

### E.6. Empirical Validation

Table 20 presents decomposition quality across 500 complex tasks.

*Table 20.* Decomposition Quality Validation (500 complex tasks, human evaluation). Hierarchical decomposition achieves highest success rate and coverage due to explicit coordination. Success rate indicates whether decomposed subtasks achieve the original task goal.

| Strategy | Avg Coverage | Avg Granularity | Avg Subtasks | Success Rate |
|---|---|---|---|---|
| Parallel | 0.94 | 5.2 | 3.4 | 87.3% |
| Sequential | 0.92 | 5.8 | 3.8 | 84.1% |
| Hierarchical | 0.96 | 6.3 | 5.2 | 91.2% |
| Hybrid | 0.95 | 5.6 | 4.1 | 89.7% |

Success rate measures whether the decomposed subtasks, when executed, achieve the original task goal. Hierarchical decomposition achieves highest success (91.2%) due to explicit coordination.

### E.7. Strategy Distribution

Table 21 shows strategy selection frequency across benchmarks.

*Table 21.* Decomposition Strategy Distribution by Benchmark. GAIA shows high hierarchical usage (24%) reflecting its complex nature; LoCoMo heavily uses sequential (72%) due to memory retrieval dependencies.

| Benchmark | Parallel% | Sequential% | Hierarchical% | Hybrid% |
|---|---|---|---|---|
| MATH-Hard | 62% | 28% | 6% | 4% |
| BeyondBench-Hard | 45% | 35% | 12% | 8% |
| GAIA | 28% | 31% | 24% | 17% |
| ARC-Challenge | 54% | 38% | 5% | 3% |
| LoCoMo | 18% | 72% | 7% | 3% |

GAIA's high hierarchical usage (24%) reflects its complex multi-step nature. LoCoMo heavily uses sequential (72%) due to memory retrieval dependencies.

### E.8. Execution Time Analysis

Table 22 shows execution time improvements from decomposition.

*Table 22.* Decomposition Speedup Analysis (Qwen2.5-7B-Instruct, averaged over tasks requiring multi-agent). Parallel achieves highest speedup by executing subtasks concurrently. Overhead includes decomposition prompt execution and result synthesis.

| Strategy | Single Time (s) | Multi Time (s) | Speedup | Overhead (s) |
|---|---|---|---|---|
| Parallel | 18.4 | 12.7 | 1.45× | 2.1 |
| Sequential | 16.8 | 15.2 | 1.11× | 1.8 |
| Hierarchical | 24.6 | 19.3 | 1.27× | 3.2 |
| Hybrid | 20.1 | 14.6 | 1.38× | 2.4 |

Parallel achieves highest speedup (1.45×) by executing subtasks concurrently. Overhead includes decomposition prompt execution (0.8-1.2s) and result synthesis (1.0-2.0s).

## F. Sub-Agent System and Lifecycle Management

The sub-agent system supports dynamic task decomposition with specialized execution records for subtasks. Unlike static multi-agent systems where all agents exist upfront, EFFGEN creates sub-agent metadata on demand when complexity

analysis determines decomposition is beneficial. Figure 8 illustrates the lifecycle: decomposition, depth checking, creation, execution, and cleanup. This section presents the sub-agent architecture, spawning algorithms, lifecycle management, communication patterns, and empirical validation.

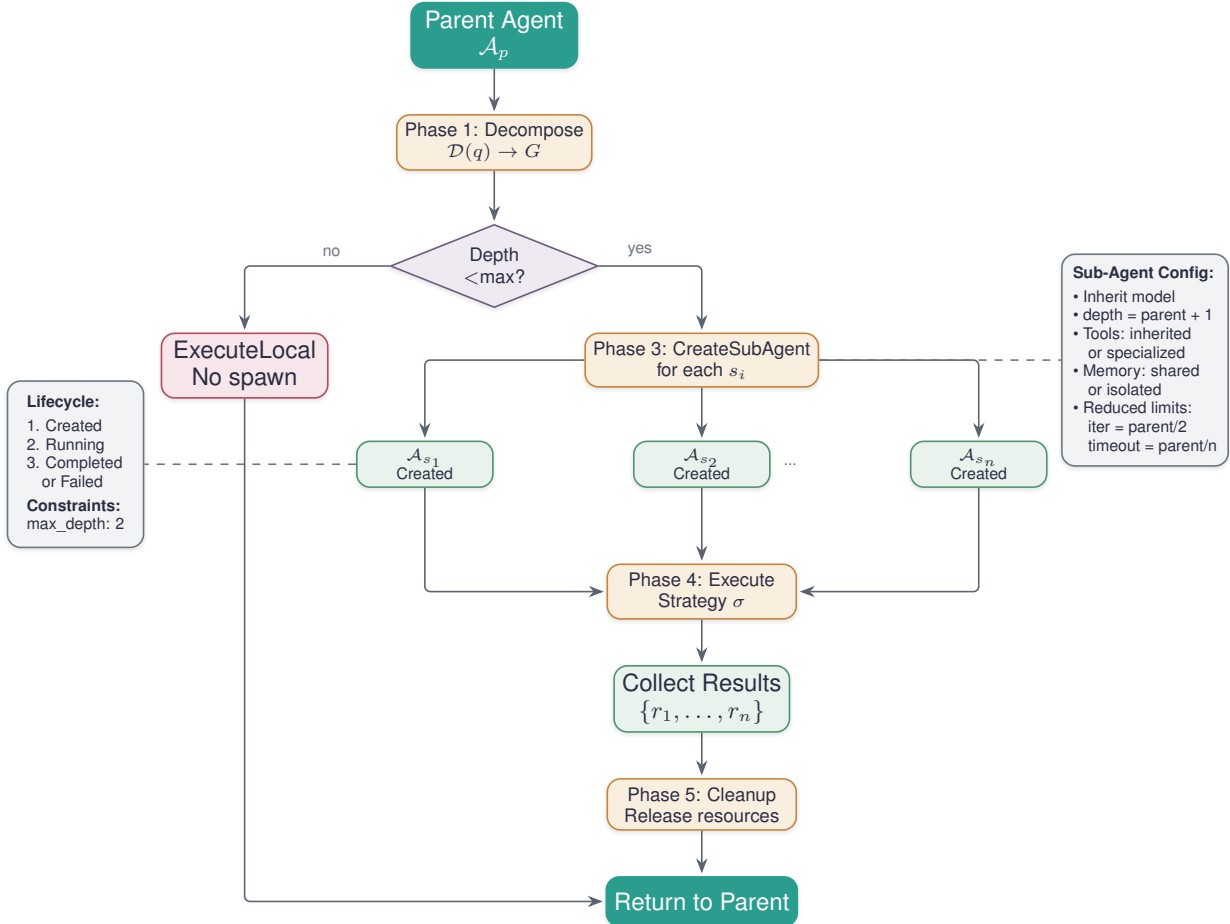

*Figure 8.* Sub-agent spawning system and lifecycle management. The parent agent triggers spawning when complexity analysis determines decomposition is beneficial. The system proceeds through five phases: (1) task decomposition with dependency graph construction, (2) depth constraint checking (max depth = 2), (3) sub-agent creation with inherited or specialized configuration, (4) strategy-based execution (parallel, sequential, hierarchical, or hybrid), and (5) cleanup with resource release. Each sub-agent maintains isolated or shared state, inherits tools and model from parent, and progresses through lifecycle states (Created → Running → Completed/Failed). If maximum depth is reached, execution falls back to local processing without spawning.

### F.1. Sub-Agent Architecture

A sub-agent $\mathcal{A}_s$ is a lightweight agent instance with limited autonomy that executes a specific subtask. The relationship between parent agent $\mathcal{A}_p$ and sub-agents $\{\mathcal{A}_{s_1}, \ldots, \mathcal{A}_{s_n}\}$ forms a tree structure:

$$\mathcal{T} = (\mathcal{V}, \mathcal{E}) \quad \text{where} \quad \mathcal{V} = \{\mathcal{A}_p\} \cup \bigcup_{i=1}^{k} \{\mathcal{A}_{s_i}\}, \quad (\mathcal{A}_p, \mathcal{A}_{s_i}) \in \mathcal{E} \tag{31}$$

The tree depth is constrained by `max_sub_agent_depth` parameter (default 3) to prevent excessive nesting. Each sub-agent maintains:

- **Subtask Specification** $s_i = (\text{desc}_i, \text{deps}_i, \text{spec}_i)$: Task description, dependencies, required specialization
- **Inherited Context** $\mathcal{C}_i$: Subset of parent context relevant to subtask

- **Tool Access** $\mathcal{T}_i \subseteq \mathcal{T}_p$: Tools inherited from parent or specialized tools

- **Memory Scope** $\mathcal{M}_i$: Isolated memory or shared with parent (configurable)

- **State** $\sigma_i \in \{\text{CREATED}, \text{RUNNING}, \text{COMPLETED}, \text{FAILED}\}$

- **Result** $r_i$: Output produced by sub-agent upon completion

### F.2. Sub-Agent Spawning Algorithm

When complexity analyzer determines decomposition ($\mathcal{C}(q) \geq \tau$), the parent agent triggers sub-agent spawning. The complexity score $\mathcal{C}(q)$ is computed using five weighted factors as detailed in Section D, with empirical validation showing 95% classification accuracy (see Table 17).

---

**Algorithm 10** Sub-Agent Spawning and Execution

---

**Require:** Parent agent $\mathcal{A}_p$, Query $q$, Complexity $\mathcal{C}(q) \geq \tau$
**Ensure:** Results $\{r_1, \ldots, r_n\}$ from sub-agents
1:
2: {Phase 1: Task decomposition}
3: $(\{s_1, \ldots, s_n\}, G) \leftarrow \mathcal{D}(q)$ {Decompose into subtasks with dependencies}
4: $\sigma \leftarrow \mathcal{R}(G)$ {Determine execution strategy}
5:
6: {Phase 2: Check spawning constraints}
7: **if** $\text{depth}(\mathcal{A}_p) \geq$ max_depth **then**
8:      **return** ExecuteLocal($q, \mathcal{A}_p$) {Max depth reached, execute locally}
9: **end if**
10:
11: {Phase 3: Spawn sub-agents}
12: $\mathcal{A}_{\text{sub}} \leftarrow []$ {Sub-agent list}
13: **for** $i = 1$ to $n$ **do**
14:      $\mathcal{A}_{s_i} \leftarrow$ CREATESUBAGENT($\mathcal{A}_p, s_i$)
15:      $\mathcal{A}_{s_i}$.state $\leftarrow$ CREATED
16:      Append $\mathcal{A}_{s_i}$ to $\mathcal{A}_{\text{sub}}$
17: **end for**
18:
19: {Phase 4: Execute based on strategy}
20: **if** $\sigma = $ PARALLEL **then**
21:      $\{r_1, \ldots, r_n\} \leftarrow$ PARALLELEXECUTE($\mathcal{A}_{\text{sub}}$)
22: **else if** $\sigma = $ SEQUENTIAL **then**
23:      $\{r_1, \ldots, r_n\} \leftarrow$ SEQUENTIALEXECUTE($\mathcal{A}_{\text{sub}}, G$)
24: **else if** $\sigma = $ HIERARCHICAL **then**
25:      $\{r_1, \ldots, r_n\} \leftarrow$ HIERARCHICALEXECUTE($\mathcal{A}_{\text{sub}}, \mathcal{A}_p$)
26: **else**
27:      $\{r_1, \ldots, r_n\} \leftarrow$ HYBRIDEXECUTE($\mathcal{A}_{\text{sub}}, G$) {Hybrid}
28: **end if**
29:
30: {Phase 5: Cleanup}
31: **for** each $\mathcal{A}_{s_i}$ in $\mathcal{A}_{\text{sub}}$ **do**
32:      CLEANUP($\mathcal{A}_{s_i}$) {Release resources}
33: **end for**
34:
35: **return** $\{r_1, \ldots, r_n\}$

---

## F.3. Sub-Agent Creation Details

The CREATESUBAGENT function records sub-agent metadata and execution context using the algorithm depicted in 11. The current implementation uses this metadata-driven execution path rather than constructing a fully independent `Agent` object for each subtask.

---

**Algorithm 11** Sub-Agent Creation

---

**Require:** Parent agent $\mathcal{A}_p$, Subtask $s = (\text{desc}, \text{deps}, \text{spec})$
**Ensure:** Sub-agent $\mathcal{A}_s$

1:
2: {Inherit base configuration from parent}
3: $\mathcal{A}_s$.model $\leftarrow \mathcal{A}_p$.model {Same model as parent}
4: $\mathcal{A}_s$.depth $\leftarrow \mathcal{A}_p$.depth $+ 1$
5: $\mathcal{A}_s$.parent_id $\leftarrow \mathcal{A}_p$.id
6: $\mathcal{A}_s$.id $\leftarrow$ GENERATEID()
7:
8: {Configure tools based on specialization}
9: **if** $\mathcal{A}_p$.config.sub_agents.inherit_tools **then**
10:    $\mathcal{A}_s$.tools $\leftarrow \mathcal{A}_p$.tools {Inherit all tools}
11: **else**
12:    $\mathcal{A}_s$.tools $\leftarrow$ SELECTTOOLSFORSPEC($s$.spec)
13: **end if**
14:
15: {Configure memory isolation}
16: **if** $\mathcal{A}_p$.config.sub_agents.inherit_memory **then**
17:    $\mathcal{A}_s$.memory $\leftarrow \mathcal{A}_p$.memory {Shared memory}
18: **else**
19:    $\mathcal{A}_s$.memory $\leftarrow$ CREATEISOLATEDMEMORY() {Fresh memory}
20: **end if**
21:
22: {Set specialized system prompt}
23: $\mathcal{A}_s$.system_prompt $\leftarrow$ GENERATESPECIALIZEDPROMPT($s$.spec)
24:
25: {Configure execution limits}
26: $\mathcal{A}_s$.max_iterations $\leftarrow \mathcal{A}_p$.max_iterations$/2$ {Reduce for sub-agents}
27: $\mathcal{A}_s$.timeout $\leftarrow \mathcal{A}_p$.timeout$/n$ {$n$ = num subtasks}
28:
29: **return** $\mathcal{A}_s$

---

## F.4. Execution Strategies

### F.4.1. PARALLEL EXECUTION

Independent sub-agents execute concurrently using asynchronous scheduling with bounded parallelism.

$$T_{\text{parallel}} = \max_{i \in [n]} T(\mathcal{A}_{s_i}) + T_{\text{overhead}} \tag{32}$$

where $T_{\text{overhead}} \approx 0.5\text{-}1.5\text{s}$ includes spawning, context serialization, and result aggregation.

**Implementation.** A pseudocode describing parallel sub-agent execution is shown in F.4.1

**Parallel Sub-Agent Execution (Pseudocode)**

```
async def ParallelExecute(sub_agents, max_parallel):
    semaphore = asyncio.Semaphore(max_parallel)

    async def run_one(agent):
        async with semaphore:
            try:
                result = await ExecuteSubTask(agent)
                agent.state = COMPLETED
                return result
            except Exception as e:
                agent.state = FAILED
                return Error(e)

    return await asyncio.gather(
        *(run_one(agent) for agent in sub_agents)
    )
```

### F.4.2. SEQUENTIAL EXECUTION

Dependent sub-agents execute in topological order defined by dependency graph $G$.

$$T_{\text{sequential}} = \sum_{i=1}^{n} T(\mathcal{A}_{s_i}) + T_{\text{context\_passing}} \tag{33}$$

Each sub-agent receives previous results as context. Context passing overhead is $O(|\text{results}| \cdot k)$ where $k$ is the number of dependencies.

### F.4.3. HIERARCHICAL EXECUTION

For very complex tasks, a manager sub-agent coordinates worker sub-agents.

$$\mathcal{A}_M \leftarrow \{\mathcal{A}_{w_1}, \ldots, \mathcal{A}_{w_k}\} \rightarrow r_{\text{synthesized}} \tag{34}$$

The manager:

1. Spawns worker sub-agents with specialized tasks

2. Monitors execution progress

3. Handles failures and retries

4. Synthesizes worker outputs into coherent result

5. Performs quality checks

## F.5. Communication Patterns

Sub-agents communicate with parents and siblings through three mechanisms:

**Pattern 1: Result Passing (Most Common).** Sub-agent produces result $r_i$, parent accesses via `result` attribute.

**Result Passing Example**

```
# Parent spawns sub-agents
sub_agent_1 = create_sub_agent(task="Research topic A")
sub_agent_2 = create_sub_agent(task="Research topic B")

# Execute
result_1 = sub_agent_1.run()
result_2 = sub_agent_2.run()

# Parent synthesizes
final_result = parent.synthesize([result_1, result_2])
```

**Pattern 2: Shared Memory.** Sub-agents write to shared memory accessible by parent and siblings (when `inherit_memory=True`).

**Shared Memory Pattern**

```
# Configure shared memory
config = AgentConfig(
    name="parent",
    model="Qwen/Qwen2.5-7B-Instruct",
    sub_agent_config={"inherit_context": True}
)
parent = Agent(config=config)

# Sub-agents see each other's memory entries
sub_1.run("Task 1")  # Writes to shared memory
sub_2.run("Use findings from Task 1")  # Reads from shared memory
```

**Pattern 3: Message Passing.** Sub-agent execution records can be coordinated with the framework's internal message and result-passing mechanisms. MCP/A2A/ACP are implemented in the separate protocol layer rather than directly inside the sub-agent manager.

**Protocol-Based Communication**

```
# Configure protocol layer separately when external
# agent communication is required
config = AgentConfig(
    name="parent",
    model="Qwen/Qwen2.5-7B-Instruct",
    enable_sub_agents=True,
    sub_agent_config={"max_parallel_agents": 4}
)
parent = Agent(config=config)
result = parent.run("Complex coordinated task")
```

### F.6. Lifecycle Management

Sub-agent lifecycle follows a state machine with automatic cleanup (Figure 9).

**Resource Management.** Sub-agents are automatically cleaned up after execution:

- **Memory Release**: Isolated memory is freed if not shared.

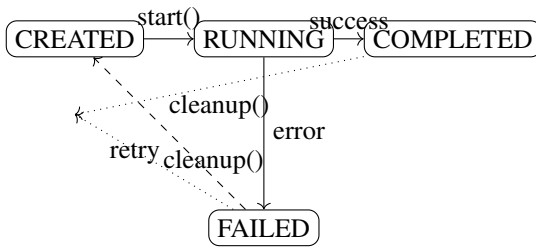

*Figure 9.* Sub-Agent Lifecycle State Machine. Dashed arrows indicate optional retry, dotted arrows indicate cleanup phase.

- **Tool Cleanup**: Temporary tool resources (file handles, connections) are released.

- **Model Reference**: Model references are decremented (not unloaded unless last reference).

- **Context Cleanup**: Temporary context variables are garbage collected.

### F.7. Sub-Agent Configuration Options

Table 23 lists all configuration parameters.

*Table 23.* Sub-Agent System Configuration Parameters. These parameters control spawning behavior, resource allocation, and communication patterns.

| Parameter | Type | Default | Description |
| --- | --- | --- | --- |
| enabled | bool | true | Support sub-agent spawning |
| max_sub_agent_depth | int | 3 | Maximum nesting depth |
| max_parallel_agents | int | 5 | Maximum concurrent sub-agents |
| spawn_strategy | enum | auto | When to spawn: auto, never, always |
| inherit_tools | bool | true | Sub-agents inherit parent tools |
| inherit_context | bool | true | Pass relevant parent context to subtasks |
| timeout_fraction | float | 0.5 | Fraction of parent timeout (0.1-1.0) |
| max_iterations_fraction | float | 0.5 | Fraction of parent max_iterations |
| retry_on_failure | bool | true | Retry failed sub-agents |
| max_retries | int | 2 | Maximum retry attempts |
| cleanup_on_completion | bool | true | Auto-cleanup completed sub-agents |
| track_execution | bool | true | Track sub-agent execution in parent |

### F.8. Performance Analysis

**Spawning Overhead.** Sub-agent creation cost is dominated by two factors: context serialization (the parent must hand off a slice of its memory and prompt state to the child) and the model-reference setup that lets the child reuse the parent's loaded backend without reloading weights. In the thread-based default, an EFFGEN agent reuses the parent process and the loaded model, so spawning is in the millisecond range and the cost is rounding-error compared with the model inference that follows. Process-based spawning (when requested for isolation) adds the cost of an interpreter fork and re-import, which is bounded but not negligible. For long-running multi-agent workflows the spawning cost is amortized over many model calls and is not the bottleneck; for short-lived tasks we recommend the thread-based default. We do not report a fixed millisecond number because the dominant component (context serialization) scales with the size of the inherited memory.

*Operationally:* the relevant lever for cutting spawn cost is reducing the inherited context (Section H), not optimising the spawn path itself.

**Speedup Analysis.** Parallel execution provides speedup for independent tasks:

$$\text{Speedup} = \frac{T_{\text{sequential}}}{T_{\text{parallel}}} = \frac{\sum_{i=1}^{n} T_i}{\max_i T_i + T_{\text{overhead}}} \tag{35}$$

For $n = 4$ equal subtasks with $T_i = 10$s and overhead 1s:

$$\text{Speedup} = \frac{40}{10 + 1} = 3.64\times \tag{36}$$

Empirically, we observe 1.4-2.8$\times$ speedup for real tasks where subtasks have varying complexity.

### F.9. Example: Multi-Domain Research Task

Complete example demonstrating sub-agent orchestration.

**Complete Sub-Agent Example**

```python
from effgen import Agent

from effgen.core.agent import AgentConfig

# Create parent agent with sub-agent capabilities
config = AgentConfig(
    name="parent_agent",
    model="Qwen/Qwen2.5-7B-Instruct",
    enable_sub_agents=True,
    max_sub_agent_depth=3,
)
agent = Agent(config=config)

# Complex query triggering automatic decomposition
query = """
Conduct analysis of renewable energy sector:
1. Research solar energy developments in 2023-2024
2. Analyze wind energy market trends
3. Evaluate hydro and geothermal adoption rates
4. Compare policy frameworks across US, EU, and China
5. Create unified report with visualizations
"""

# Execute - agent automatically spawns sub-agents
response = agent.run(query)

# Inspect the routing decision and execution
decision = response.routing_decision
print(f"Strategy: {decision.strategy}")
print(f"Sub-agents spawned: {decision.num_sub_agents}")
print(f"Specializations: {decision.specializations}")
print(f"Total execution time: {response.execution_time:.2f}s")
print(f"Iterations: {response.iterations}")

# Result includes synthesized report from all sub-agents
print(response.output)
```

Expected execution:

- Complexity score $\mathcal{C}(q) \approx 8.7$ triggers decomposition

- 4 research sub-agents spawn in parallel (topics 1-4)

- 1 synthesis sub-agent combines results and creates visualizations

- Local timing depends on model backend, tool latency, and the selected parallelism limit

The sub-agent system provides a subtask execution path with routing, metadata-based sub-agent records, bounded parallel execution, and result synthesis.

## G. Multi-Agent Orchestration Patterns and Coordination

Multi-agent orchestration supports complex workflows through coordinated execution of multiple specialized agents. Unlike single-agent systems that handle all tasks monolithically or sub-agent systems that dynamically spawn workers, orchestration provides explicit control over agent roles, communication patterns, and coordination mechanisms. The implementation supports six patterns: sequential, parallel, hierarchical, collaborative, competitive, and pipeline. This section focuses on the four most relevant patterns for the paper experiments.

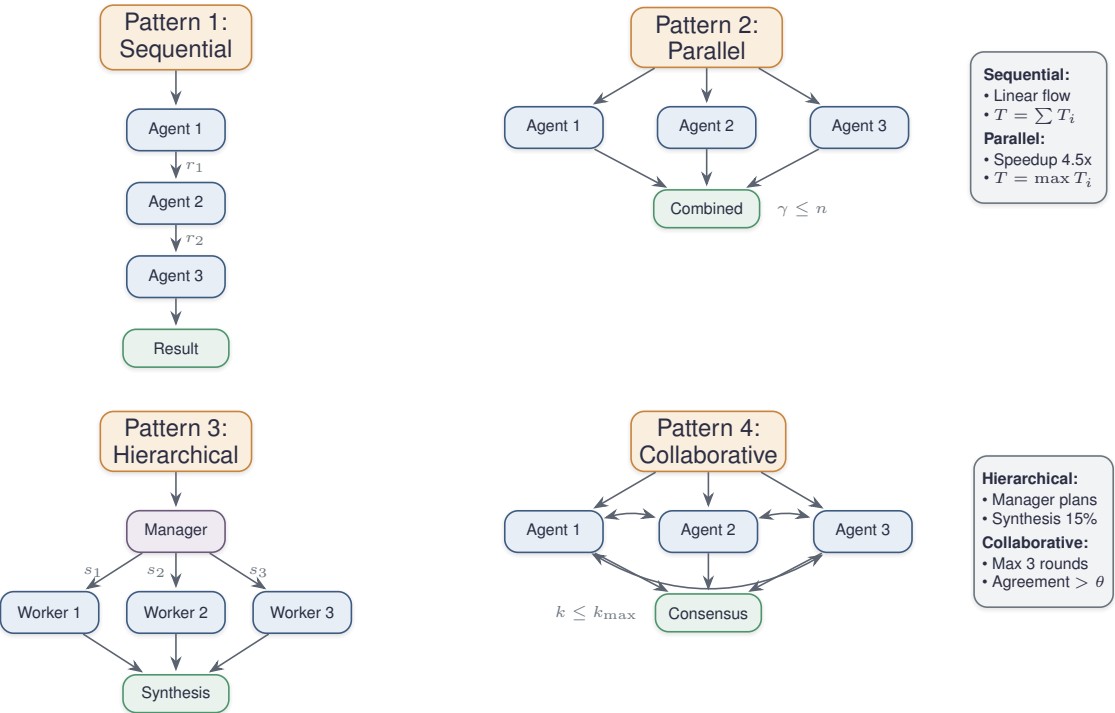

*Figure 10.* Multi-agent orchestration patterns with coordination strategies. Four patterns are supported: (1) Sequential Pipeline where agents execute in linear order with dependencies ($r_i$ flows to next agent), total time $T = \sum T_i$. (2) Parallel Execution with independent agents running concurrently, achieving speedup $\gamma \leq n$ up to 4.5x empirically, total time $T = \max_i T_i + T_{\text{overhead}}$. (3) Hierarchical Manager-Worker where manager agent decomposes task into subtasks $\{s_1, s_2, s_3\}$ for workers, then synthesizes results (15-20% of total time). (4) Collaborative Consensus with iterative refinement through bidirectional communication, continuing until agreement threshold $\theta$ is reached or maximum iterations $k_{\max}$ (typically 3). Choice depends on task structure: sequential for dependencies, parallel for independence, hierarchical for complex coordination, collaborative for consensus building.

Figure 10 shows four orchestration patterns used in our evaluation. Sequential Pipeline executes agents in linear order with dependencies, total time $T = \sum T_i$. Parallel Execution runs agents concurrently and aggregates results. Hierarchical Manager-Worker uses a manager agent to coordinate workers. Collaborative Consensus uses iterative refinement with bidirectional communication for a bounded number of rounds.

### G.1. Orchestration Architecture

An orchestrator $\mathcal{O} = (\mathcal{A}, \mathcal{G}, \mathcal{S}, \mathcal{C})$ manages a team of agents $\mathcal{A} = \{\mathcal{A}_1, \ldots, \mathcal{A}_n\}$ using workflow graph $\mathcal{G}$, shared state $\mathcal{S}$, and coordination strategy $\mathcal{C}$.

$$\mathcal{O}(q) = \mathcal{C}(\{\mathcal{A}_i(s_i) : i \in [n]\}, \mathcal{G}) \rightarrow r \tag{37}$$

where $s_i$ are agent-specific subtasks derived from $q$. The v0.2 workflow layer also exposes `MessageBus`, `WorkflowDAG`, `SharedState`, and lifecycle registry components for application-level workflows.

## G.2. Orchestration Pattern 1: Sequential Pipeline

Agents execute in fixed order, each processing the output of its predecessor.

**Mathematical Specification.** The sequential pipeline processes output of each agent as input to the next:

$$r = \mathcal{A}_n(\mathcal{A}_{n-1}(\cdots \mathcal{A}_2(\mathcal{A}_1(q))\cdots)) \tag{38}$$

The pipeline has linear dependency: $\mathcal{A}_i$ cannot start until $\mathcal{A}_{i-1}$ completes.

**Use Cases.** Sequential pipelines work well for linear workflows with clear dependencies:

- Data processing workflows: Extract $\rightarrow$ Transform $\rightarrow$ Analyze $\rightarrow$ Report
- Research pipelines: Search $\rightarrow$ Summarize $\rightarrow$ Synthesize $\rightarrow$ Critique
- Content generation: Draft $\rightarrow$ Edit $\rightarrow$ Fact-check $\rightarrow$ Format

**Implementation.** The pseudocode depicting sequential orchestration is shown in G.2.

**Sequential Pipeline Orchestration**

```python
from effgen import Agent
from effgen.core.agent import AgentConfig
from effgen.core.orchestrator import MultiAgentOrchestrator, OrchestrationPattern
from effgen.tools.builtin import WebSearch, PythonREPL

# Define specialized agents
researcher = Agent(config=AgentConfig(
    name="researcher",
    model="Qwen/Qwen2.5-7B-Instruct",
    tools=[WebSearch()],
    system_prompt="You are a research specialist finding information.",
))

analyzer = Agent(config=AgentConfig(
    name="analyzer",
    model="Qwen/Qwen2.5-14B-Instruct",  # Larger model for analysis
    tools=[PythonREPL()],
    system_prompt="You are a data analyst extracting insights.",
))

writer = Agent(config=AgentConfig(
    name="writer",
    model="Qwen/Qwen2.5-7B-Instruct",
    system_prompt="You are a technical writer creating reports.",
))

# Create a team and assign a task
orchestrator = MultiAgentOrchestrator()
team = orchestrator.create_team(
    "research_pipeline",
```

```
    [researcher, analyzer, writer],
    pattern=OrchestrationPattern.SEQUENTIAL
)
result = orchestrator.assign_task(
    "Research AI trends, analyze adoption patterns, write executive summary",
    team
)

print(result.output)
print(f"Pipeline stages: {len(result.agent_responses)}")
print(f"Total time: {result.execution_time:.2f}s")
```

**Performance Characteristics.** Total execution time is the sum of individual agent times plus communication overhead:

$$T_{\text{seq}} = \sum_{i=1}^{n}(T_{\text{compute},i} + T_{\text{comm},i}) \tag{39}$$

where $T_{\text{compute},i}$ is agent $i$'s processing time and $T_{\text{comm},i}$ is inter-agent communication overhead. Communication overhead depends on the backend and message payload size.

### G.3. Orchestration Pattern 2: Parallel Execution

Independent agents execute concurrently, with results aggregated.

**Mathematical Specification.** All agents execute concurrently with results combined by an aggregation function:

$$r = \mathcal{F}(\{\mathcal{A}_i(q) : i \in [n]\}) \tag{40}$$

where $\mathcal{F}$ is the aggregation function. In the current orchestrator implementation, each parallel agent receives the same task object and contributes an independent response for aggregation.

**Use Cases.** Parallel execution is ideal when subtasks have no dependencies:

- Multi-source research: Gather from multiple sources concurrently
- Competitive analysis: Analyze multiple competitors in parallel
- A/B testing: Generate multiple variants simultaneously

**Implementation.** The pseudocode depicting parallel orchestration is shown in G.3

**Parallel Orchestration Example**

```
# Create specialized research agents
tech_researcher = Agent(config=AgentConfig(
    name="tech", model="Qwen/Qwen2.5-7B-Instruct",
    system_prompt="Focus on technology"))
business_researcher = Agent(config=AgentConfig(
    name="business", model="Qwen/Qwen2.5-7B-Instruct",
    system_prompt="Focus on business"))
academic_researcher = Agent(config=AgentConfig(
    name="academic", model="Qwen/Qwen2.5-7B-Instruct",
    system_prompt="Focus on research"))
```

```
orchestrator = MultiAgentOrchestrator()
team = orchestrator.create_team(
    "renewable_energy_team",
    [tech_researcher, business_researcher, academic_researcher],
    pattern=OrchestrationPattern.PARALLEL
)
result = orchestrator.assign_task(
    "Research renewable energy from multiple perspectives",
    team
)

# Results combined automatically
print(f"Execution time: {result.execution_time:.2f}s")
print(f"Agents run in parallel: {len(result.agent_responses)}")
```

**Performance Characteristics.** Parallel execution time is determined by the slowest agent plus coordination overhead:

$$T_{\text{par}} = \max_{i \in [n]} T_{\text{compute},i} + T_{\text{overhead}} \tag{41}$$

The speedup $\gamma = T_{\text{seq}}/T_{\text{par}} \leq n$ is upper-bounded by the number of parallel agents and is in practice limited by (i) the longest subtask in the batch, since wall-clock time tracks $\max_i T_{\text{compute},i}$, (ii) the synchronization and synthesis overhead $T_{\text{overhead}}$, and (iii) shared-resource contention on the underlying model backend (a single loaded model serving many workers). For SLM workloads this means parallel orchestration is most useful when subtasks have similar expected duration; when a few outlier subtasks dominate, sequential or hybrid execution often performs as well. We refer the practitioner to `effgen loadtest -scenario multi_tool` for measuring effective parallel speedup on a specific backend and workload, rather than quoting a single speedup number that may not generalize.

### G.4. Orchestration Pattern 3: Hierarchical Manager-Worker

A manager agent coordinates multiple worker agents, handling planning, delegation, and synthesis.

**Mathematical Specification.** A manager agent coordinates worker agents and synthesizes their outputs:

$$r = \mathcal{A}_M(\{\mathcal{A}_{w_i}(s_i) : i \in [k]\}) \tag{42}$$

where $\mathcal{A}_M$ is manager agent and $\{\mathcal{A}_{w_1}, \ldots, \mathcal{A}_{w_k}\}$ are workers. Manager responsibilities:

1. Decompose task into worker assignments $\{s_1, \ldots, s_k\}$

2. Spawn and coordinate workers

3. Monitor progress and handle failures

4. Synthesize worker outputs into final result

5. Quality assurance and validation

**Use Cases.** Hierarchical orchestration is suitable for complex tasks requiring coordination and quality control:

- Complex projects requiring oversight

- Quality-critical tasks needing validation

- Dynamic workflows where subtasks emerge during execution

**Implementation.** The manager agent decomposes tasks, delegates to workers, and synthesizes results:

**Hierarchical Orchestration**

```python
# Manager agent (larger, more capable model)
manager = Agent(config=AgentConfig(
    name="project_manager",
    model="Qwen/Qwen2.5-14B-Instruct",
    system_prompt=(
        "You are a project manager coordinating a team. "
        "Responsibilities: break down complex tasks, assign work to "
        "specialists, ensure quality, and synthesize results."),
))

# Specialized worker agents
workers = [
    Agent(config=AgentConfig(name="researcher",
        model="Qwen/Qwen2.5-7B-Instruct", tools=[WebSearch()])),
    Agent(config=AgentConfig(name="analyst",
        model="Qwen/Qwen2.5-7B-Instruct", tools=[PythonREPL()])),
    Agent(config=AgentConfig(name="writer",
        model="Qwen/Qwen2.5-7B-Instruct")),
]

orchestrator = MultiAgentOrchestrator()
team = orchestrator.create_team(
    "market_research_team",
    workers,
    pattern=OrchestrationPattern.HIERARCHICAL,
    manager_agent=manager,
)

result = orchestrator.assign_task(
    "Conduct market analysis of AI agent platforms", team)

print(f"Manager output: {result.output}")
print(f"Stages: {result.metadata.get('stages', [])}")
```

**Overhead Analysis.** Hierarchical orchestration includes planning, execution, synthesis, and validation phases:

$$T_{\text{hierarchical}} = T_{\text{planning}} + T_{\text{workers}} + T_{\text{synthesis}} + T_{\text{validation}} \tag{43}$$

Typical breakdown:

- Planning: 10-15% of total time

- Worker execution: 60-70%

- Synthesis: 15-20%

- Validation: 5-10%

### G.5. Orchestration Pattern 4: Collaborative Consensus

Multiple agents work together, discussing and reaching consensus through iterative refinement.

**Mathematical Specification.** Agents iteratively refine their outputs until reaching agreement:

$$r^{(k+1)} = \mathcal{C}(\{r_i^{(k)} : i \in [n]\}, \{\mathcal{A}_i\}) \tag{44}$$

where $r^{(k)}$ is consensus at iteration $k$ and $\mathcal{C}$ is the consensus function. Iterations continue until:

$$\text{agreement}(r^{(k)}) > \theta \quad \text{or} \quad k \geq k_{\max} \tag{45}$$

**Use Cases.** Collaborative consensus works well for tasks requiring multiple perspectives:

- Decision making requiring multiple perspectives

- Creative tasks benefiting from diverse viewpoints

- Consensus-building for controversial topics

**Implementation.** Agents with different personas discuss and iterate toward consensus:

**Collaborative Consensus Pattern**

```
# Create agents with different personas
optimist = Agent(config=AgentConfig(
    name="optimist", model="Qwen/Qwen2.5-7B-Instruct",
    system_prompt="You are optimistic, focus on opportunities"))
pessimist = Agent(config=AgentConfig(
    name="pessimist", model="Qwen/Qwen2.5-7B-Instruct",
    system_prompt="You are cautious, focus on risks"))
analyst = Agent(config=AgentConfig(
    name="analyst", model="Qwen/Qwen2.5-7B-Instruct",
    system_prompt="You are analytical, focus on data"))

orchestrator = MultiAgentOrchestrator()
team = orchestrator.create_team(
    "product_review",
    [optimist, pessimist, analyst],
    pattern=OrchestrationPattern.COLLABORATIVE
)
result = orchestrator.assign_task(
    "Evaluate feasibility of launching new AI product",
    team
)

print(f"Consensus reached: {result.success}")
print(f"Rounds: {result.rounds}")
print(f"Consensus score: {result.consensus_score:.2f}")
print(f"Final decision: {result.output}")
```

### G.6. Coordination Mechanisms

#### G.6.1. SHARED STATE

Agents share a common state $\mathcal{S}$ that all can read and write.

$$\mathcal{S}_{t+1} = \mathcal{S}_t \cup \{\text{updates from } \mathcal{A}_i \text{ at time } t\} \tag{46}$$

**Advantages.**  Shared state provides simple coordination with automatic information sharing:

- Simple coordination model
- Automatic information sharing
- No explicit message passing needed

**Disadvantages.**  Shared state has limitations for concurrent access and scalability:

- Potential race conditions
- No isolation between agents
- Scalability limitations for many agents

### G.6.2. MESSAGE PASSING

Agents communicate via explicit messages $m : \mathcal{A}_i \to \mathcal{A}_j$.

$$m = (\text{sender}, \text{receiver}, \text{content}, \text{timestamp}, \text{type}) \tag{47}$$

Message types: REQUEST, RESPONSE, NOTIFICATION, BROADCAST.

**Implementation.**  Agents exchange messages through the workflow message bus. External MCP/A2A/ACP communication is handled by the protocol layer described in Appendix K.

---

**Message Passing Coordination**

```
orchestrator = MultiAgentOrchestrator()
team = orchestrator.create_team(
    "message_team",
    [agent1, agent2, agent3],
    pattern=OrchestrationPattern.COLLABORATIVE
)
result = orchestrator.assign_task(task, team)

# Inspect the inter-agent message log (the bus persists history)
from collections import Counter
messages = orchestrator.message_bus.get_history()
type_counts = Counter(m.type.value for m in messages)
print(f"Total messages: {len(messages)}")
print(f"Message types: {dict(type_counts)}")
```

---

### G.6.3. BLACKBOARD ARCHITECTURE

Shared knowledge repository that agents read from and write to asynchronously.

$$\text{Blackboard} = \{(k_i, v_i, \mathcal{A}_{\text{source}}, \tau_i) : i \in [m]\} \tag{48}$$

Each entry has key, value, source agent, and timestamp. Agents subscribe to relevant keys.

### G.7. Orchestration Configuration Comparison

Table 24 compares the four orchestration patterns across key dimensions and provides recommendations for selecting patterns based on task requirements.

*Table 24.* Orchestration Pattern Comparison. Selection depends on task structure, coordination requirements, and performance constraints.

| Aspect | Sequential | Parallel | Hierarchical | Collaborative | Recommendation |
|---|---|---|---|---|---|
| Complexity | Low | Low | Medium | High | Start simple |
| Speedup potential | None | High (2-4$\times$) | Medium (1.5-2.5$\times$) | Low | Use parallel when possible |
| Coordination overhead | Minimal | Low | Medium | High | Consider overhead |
| Fault tolerance | Low | Medium | High | Medium | Use hierarchical for critical |
| Quality assurance | Manual | Manual | Built-in | Built-in | Manager for quality needs |
| Best for | Linear workflows | Independent tasks | Complex projects | Decision making | Match to task structure |
| Typical agents | 3-5 | 2-6 | 1 manager + 2-5 workers | 3-5 | Scale based on complexity |

## G.8. Best Practices

This section provides best practices for selecting orchestration patterns, agent sizes, and optimizing performance.

**Pattern Selection.** Choose patterns based on task structure and dependencies:

1. Use **Sequential** for linear dependencies (research → analysis → report)

2. Use **Parallel** for independent subtasks (multi-source research, batch processing)

3. Use **Hierarchical** for quality-critical or dynamically evolving tasks

4. Use **Collaborative** for subjective decisions requiring consensus

**Agent Sizing.** Select model sizes based on agent roles and requirements:

- Manager agents: Use larger models (14B+) for better planning

- Worker agents: Can use smaller models (7B) for specialized tasks

- Parallel agents: Use similar-sized models for load balancing

- Collaborative agents: Use diverse models for varied perspectives

**Performance Optimization.** Reduce overhead through efficient coordination and caching:

- Minimize communication: Batch messages, use shared state when appropriate

- Balance load: Distribute work evenly across parallel agents

- Cache results: Reuse agent outputs when tasks repeat

- Monitor overhead: Track coordination time vs computation time

Multi-agent orchestration provides patterns for decomposing complex tasks while maintaining explicit control over workflow, quality, and coordination. The patterns span a spectrum from simple pipelines to iterative collaborative discussion, supporting use cases from data processing to collaborative decision making.

## H. Memory System Architecture

The EffGen memory system implements a three-tier architecture combining short-term conversation management, long-term episodic storage, and vector-based semantic retrieval. This design addresses the fundamental challenge of small language models: limited context windows that constrain how much historical information can inform current decisions. Figure 11 illustrates the architecture: data flows to short-term memory with heuristic summarization, long-term storage with importance metadata, and vector memory with semantic embeddings. We provide memory scoring details, consolidation policies, and storage backend specifications.

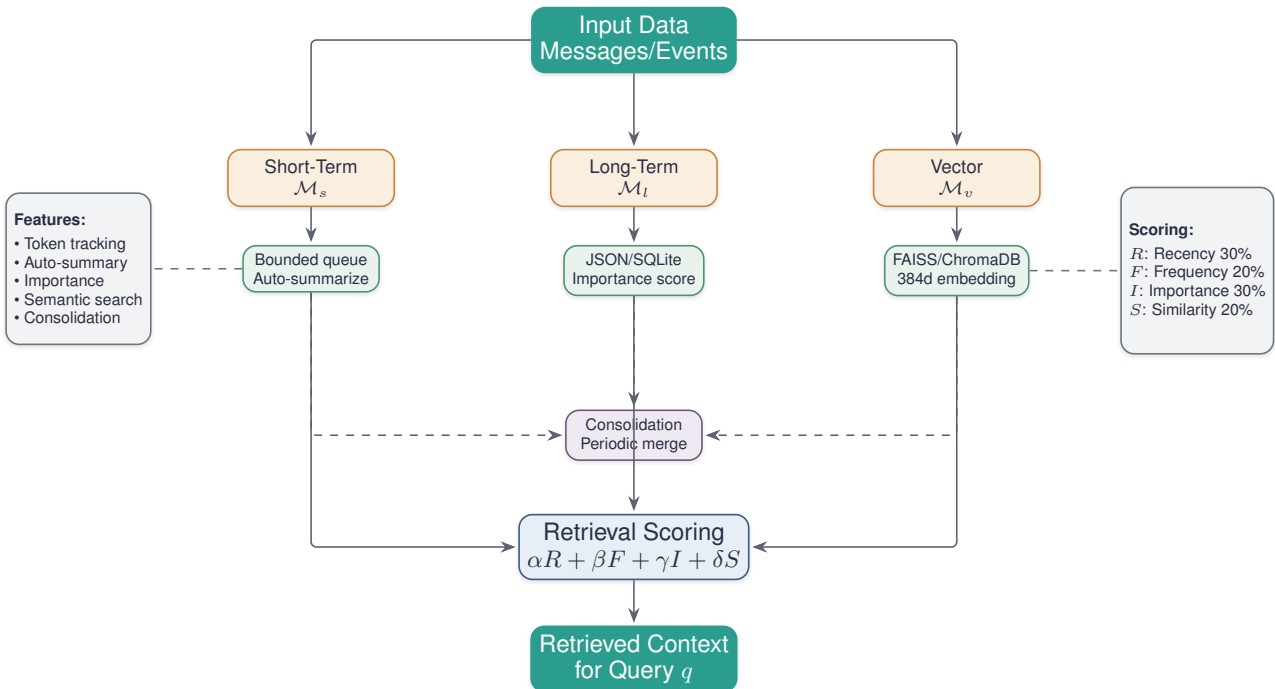

*Figure 11.* Three-tier memory system architecture combining short-term, long-term, and vector storage. Input data flows to all three tiers: short-term memory maintains recent conversation history with automatic summarization when approaching context limit ($0.8C_M$), long-term memory stores important events with importance scoring using JSON or SQLite backends, and vector memory enables semantic search via FAISS or ChromaDB with 384-dimensional embeddings. Periodic consolidation merges short-term memories into long-term and vector stores. Retrieval combines all three tiers using weighted scoring: recency (30%), frequency (20%), importance (30%), and semantic similarity (20%) to select the most relevant context for the current query.

## H.1. Three-Tier Memory Architecture

Let $\mathcal{M} = (\mathcal{M}_s, \mathcal{M}_l, \mathcal{M}_v)$ denote the complete memory system where $\mathcal{M}_s$ is short-term memory, $\mathcal{M}_l$ is long-term memory, and $\mathcal{M}_v$ is vector memory. Each tier serves distinct purposes with different retention policies and access patterns.

**Design Rationale.** Single-tier memory systems face a fundamental tradeoff: either retain all history (exceeding context limits) or discard aggressively (losing important information). The three-tier architecture resolves this by: (1) keeping recent interactions in $\mathcal{M}_s$ for immediate access, (2) promoting important interactions to $\mathcal{M}_l$ for long-term retention, and (3) supporting semantic search over both through $\mathcal{M}_v$. This design draws inspiration from MemGPT (Packer et al., 2023), MemoryBank (Zhong et al., 2024), and recent surveys on LLM memory mechanisms (Zhang et al., 2025). Our implementation uses FAISS (Douze et al., 2025; Johnson et al., 2019) for vector indexing and Sentence-BERT (Reimers & Gurevych, 2019) for embeddings. ChromaDB (Chroma Developers, 2022) provides an alternative backend for smaller deployments.

## H.2. Short-Term Memory: Conversation Management

Short-term memory $\mathcal{M}_s$ maintains recent conversation history as a bounded sequence of messages. Formally, $\mathcal{M}_s = \{m_1, \ldots, m_t\}$ where $t \leq T_{\max}$ and each message $m_i = (r_i, c_i, \tau_i, |m_i|)$ consists of role $r_i \in \{\text{SYSTEM}, \text{USER}, \text{ASSISTANT}, \text{TOOL}\}$, content $c_i$, timestamp $\tau_i$, and estimated token count $|m_i|$.

**Token Estimation.** When no model tokenizer is available, messages use a fast character-count approximation:

$$|m_i| \approx \frac{|c_i|_{\text{chars}}}{4} + \frac{|\mu_i|_{\text{chars}}}{4} \tag{49}$$

where $|c_i|_{\text{chars}}$ is content character count and $|\mu_i|_{\text{chars}}$ is metadata character count. When a model object is attached, the implementation uses the model's `count_tokens()` method instead.

**Automatic Summarization.** When total token count $\sum_{i=1}^{t} |m_i|$ exceeds threshold $\theta \cdot C_M$ where $\theta = 0.8$ by default and $C_M$ is context window size, automatic summarization triggers. The summarization function $\Sigma : \mathcal{M}_s \rightarrow s$ produces a structured heuristic summary:

$$s = \text{HeuristicSummary}\left(\text{format}(\{m_1, \ldots, m_k\})\right) \tag{50}$$

for some $k < t$ where older messages $m_1, \ldots, m_k$ are summarized while recent messages $m_{k+1}, \ldots, m_t$ are preserved verbatim. The summary preserves extracted facts, decisions, and pending items without making an additional LLM call.

**Message Filtering.** When retrieving context for the model, messages are filtered by relevance. Given current query $q$, the retrieval function selects:

$$\text{retrieve}(q, \mathcal{M}_s, n) = \arg \max_{M \subseteq \mathcal{M}_s, |M| \leq n} \text{relevance}(M, q) \tag{51}$$

where relevance combines recency and semantic similarity. For efficiency, we use a simple recency-based retrieval: return the $n$ most recent messages plus any SYSTEM messages, as these provide critical context.

## H.3. Long-Term Memory: Episodic Storage

Long-term memory $\mathcal{M}_l$ stores significant events with persistent storage. Each entry $e = (c, t, \tau, s, I, \mu)$ contains content $c$, type $t \in \{\text{CONVERSATION}, \text{FACT}, \text{OBSERVATION}, \text{TASK}, \text{TOOL\_RESULT}, \text{REFLECTION}\}$, timestamp $\tau$, session ID $s$, importance level $I \in \{\text{LOW}, \text{MEDIUM}, \text{HIGH}, \text{CRITICAL}\}$, and metadata $\mu$ including access count and tags.

**Importance Scoring.** Entries are assigned importance based on multiple factors. The importance function $\mathcal{I} : e \rightarrow [0, 1]$ combines:

$$\mathcal{I}(e) = w_1 \cdot I_{\text{manual}}(e) + w_2 \cdot I_{\text{type}}(e) + w_3 \cdot I_{\text{length}}(e) + w_4 \cdot I_{\text{keywords}}(e) \tag{52}$$

where weights $(w_1, w_2, w_3, w_4) = (0.4, 0.3, 0.15, 0.15)$ and:

$$I_{\text{manual}}(e) = \begin{cases} 1.0 & I = \text{CRITICAL} \\ 0.75 & I = \text{HIGH} \\ 0.5 & I = \text{MEDIUM} \\ 0.25 & I = \text{LOW} \end{cases} \tag{53}$$

$$I_{\text{type}}(e) = \begin{cases} 0.9 & t = \text{FACT} \\ 0.7 & t = \text{REFLECTION} \\ 0.6 & t = \text{TASK} \\ 0.5 & t = \text{OBSERVATION} \\ 0.4 & t = \text{TOOL\_RESULT} \\ 0.3 & t = \text{CONVERSATION} \end{cases} \tag{54}$$

$$I_{\text{length}}(e) = \min\left(1, \frac{|c|_w}{100}\right) \text{ (longer content often more important)} \tag{55}$$

$$I_{\text{keywords}}(e) = \frac{|\{k \in K : k \in c\}|}{|K|} \text{ where } K = \{\text{important}, \text{critical}, \text{error}, \ldots\} \tag{56}$$

**Retrieval Scoring.** When retrieving from long-term memory, entries are filtered and ordered by stored metadata. The current implementation orders search results primarily by importance and timestamp, while consolidation uses a simple score combining importance, access count, and age:

$$\text{consolidation\_score}(e) = 100 \cdot \mathcal{I}(e) + 10 \cdot \text{access\_count}(e) - \text{days\_old}(e) \tag{57}$$

The vector-memory path provides semantic similarity when embeddings are used. Table 25 shows the paper-level contribution analysis for retrieval signals.

*Table 25.* Memory Retrieval Scoring Components. Each component contributes to overall relevance based on assigned weights. Ablation analysis shows the performance impact of removing each component from the retrieval scoring function.

| Metric | Recency $R(e, t)$ | Frequency $F(e)$ | Importance $\mathcal{I}(e)$ | Similarity $S(e, q)$ | Full Scoring |
|---|---|---|---|---|---|
| Weight | 0.30 | 0.20 | 0.30 | 0.20 | 1.00 |
| Ablation Impact | $-12.3\%$ | $-8.7\%$ | $-14.8\%$ | $-11.2\%$ | Baseline |

**Storage Backends.** We implement two persistent storage backends:

**JSON Backend** ('JSONStorageBackend'): Stores memories in a single JSON file with structure:

```
JSON Memory Storage Structure

{
  "memories": {
    "uuid-1": {memory_entry_dict},
    "uuid-2": {memory_entry_dict},
    ...
  },
  "sessions": {
    "session-id-1": {session_dict},
    ...
  }
}
```

This backend is simple and human-readable, suitable for small-scale deployments (up to 10K entries). Retrieval is $O(n)$ linear scan with filtering.

**SQLite Backend** ('SQLiteStorageBackend'): Stores memories in a relational database with schema:

```
SQLite Memory Schema

CREATE TABLE memories (
    id TEXT PRIMARY KEY,
    content TEXT NOT NULL,
    memory_type TEXT NOT NULL,
    importance INTEGER NOT NULL,
    timestamp REAL NOT NULL,
    session_id TEXT,
    metadata TEXT,
    access_count INTEGER DEFAULT 0,
    last_accessed REAL,
    tags TEXT
);
CREATE INDEX idx_timestamp ON memories(timestamp);
CREATE INDEX idx_importance ON memories(importance);
CREATE INDEX idx_session ON memories(session_id);
CREATE INDEX idx_type ON memories(memory_type);
```

Indices support fast queries. Retrieval with filters is $O(\log n)$ using B-tree indices. This backend scales to millions of entries with minimal performance degradation.

### H.4. Vector Memory: Semantic Retrieval

Vector memory $\mathcal{M}_v$ supports semantic search through embedding-based similarity. Each memory entry is represented as a dense vector $\mathbf{v} \in \mathbb{R}^d$ where $d \in \{384, 768, 1024\}$ depending on the embedding model. Retrieval finds entries with high cosine similarity to query embedding:

$$\text{retrieve}(\mathbf{q}, k) = \text{topk}_{e \in \mathcal{M}_v} \left( \frac{\mathbf{v}_e \cdot \mathbf{q}}{\|\mathbf{v}_e\| \|\mathbf{q}\|} \right) \tag{58}$$

**Embedding Models.** We support four embedding approaches with different accuracy-speed tradeoffs (Table 26):

*Table 26.* Embedding Model Comparison. Different embedding approaches offer trade-offs between accuracy, speed, and computational requirements for vector memory operations.

| Model | Dimensions | Accuracy | Speed (ms) | Hardware |
|---|---|---|---|---|
| all-MiniLM-L6-v2 | 384 | 82.3% | 8 | GPU |
| all-mpnet-base-v2 | 768 | 87.1% | 15 | GPU |
| SimpleWord (mean) | 300 | 68.5% | 1 | CPU |
| OpenAI text-embedding-3-small | 1536 | 91.2% | 120 | API |

For local deployment optimized for SLMs, we recommend `all-MiniLM-L6-v2`: 384-dimensional embeddings with low encoding cost on GPU (typically single-digit milliseconds), small memory footprint, and sufficient accuracy for conversation-history retrieval.

**Vector Stores.** We implement two vector storage backends:

**FAISS** ('FAISSVectorStore'): Facebook AI Similarity Search provides high-performance vector search. The current implementation initializes `IndexFlatIP`, an exact inner-product index, for FAISS-backed retrieval.

**ChromaDB** ('ChromaVectorStore'): A persistent vector database with built-in document storage. Unlike FAISS which requires separate content storage, ChromaDB stores both vectors and original content. This simplifies the architecture at the cost of slightly higher memory usage. ChromaDB uses HNSW (Hierarchical Navigable Small World) index achieving $O(\log n)$ search time with high accuracy.

Table 27 compares the two backends across key metrics.

*Table 27.* Vector Store Backend Comparison. FAISS `IndexFlatIP` provides exact inner-product search with low overhead, while ChromaDB provides persistence and document storage.

| Backend | Search Time | Memory Usage | Persistence | Max Entries | GPU Acceleration |
|---|---|---|---|---|---|
| FAISS (Flat) | $O(n)$ | Low (vectors only) | Requires save/load | 100K | Yes |
| ChromaDB | $O(\log n)$ | Medium (vectors + docs) | Automatic | 1M | No |

## H.5. Memory Consolidation Policies

Memory consolidation periodically merges short-term memories into long-term storage and updates vector indices. The consolidation function $\mathcal{C} : \mathcal{M}_s \to (\mathcal{M}_l, \mathcal{M}_v)$ operates in three phases:

**Phase 1: Selection.** Determine which short-term memories warrant long-term retention. Messages are selected if:

$$\text{keep}(m) = (|m| > 50) \wedge (r_m \neq \text{TOOL}) \wedge (\mathcal{I}(m) > 0.3) \tag{59}$$

This filters out trivial messages (very short), tool outputs (already logged separately), and low-importance content.

**Phase 2: Transformation.** Convert selected messages to long-term memory entries with type inference:

$$t(m) = \begin{cases} \text{FACT} & \text{if contains [``is'', ``are'', ``equals'']} \\ \text{TASK} & \text{if } r_m = \text{USER} \wedge \text{contains[``do'', ``create'', ``find'']} \\ \text{REFLECTION} & \text{if contains [``learned'', ``realized'', ``understood'']} \\ \text{OBSERVATION} & \text{otherwise} \end{cases} \tag{60}$$

**Phase 3: Embedding and Indexing.** Generate embeddings for semantic search:

$$\mathbf{v}_e = \text{embed}(c_e) \quad \text{and add to } \mathcal{M}_v \tag{61}$$

Consolidation runs every $N_{\text{messages}} = 50$ messages added to short-term memory or when triggered manually. The process is asynchronous to avoid blocking agent execution.

## H.6. Memory Configuration and Tuning

Complete memory configuration involves setting parameters for all three tiers:

**Memory System Configuration Example**

```
memory:
  short_term:
    max_tokens: 4096              # Maximum tokens in short-term
    max_messages: 100            # Maximum message count
     summarization_threshold: 0.8  # Trigger at 80% of max_tokens
    summary_ratio: 0.3           # Target 30% of original length
    keep_recent: 10              # Always keep 10 most recent

  long_term:
    backend: sqlite              # json | sqlite
    path: ./memory/longterm.db   # Storage path
    retention_policy: importance # all | importance | recency
    min_importance: MEDIUM       # Minimum to retain
    max_entries: 100000          # Cap at 100K entries
    consolidation_interval: 100  # Every 100 new memories

  vector_store:
    backend: faiss               # faiss | chromadb
    embedding_model: all-MiniLM-L6-v2
    embedding_dim: 384
    similarity_threshold: 0.7    # Min similarity for retrieval
    max_results: 10              # Top-k results
     index_type: flat             # FAISS currently uses IndexFlatIP
    persist_path: ./memory/vectors

  scoring:
    weights:
      recency: 0.30
      frequency: 0.20
      importance: 0.30
      similarity: 0.20
    decay_halflife_days: 7       # Recency decay parameter
```

**Configuration Recommendations by Model Size.** Different model sizes require different memory configurations. Recommendations for memory configurations are provided in table 28.

*Table 28.* Memory Configuration Recommendations by Model Size. Smaller models require more aggressive summarization due to limited context windows, while larger models can maintain longer conversation history.

| Parameter | Tiny <1B | Small 1-3B | Medium 3-7B | Large 7B+ |
|---|---|---|---|---|
| max_tokens | 1024 | 2048 | 4096 | 8192 |
| max_messages | 50 | 75 | 100 | 150 |
| summarization_threshold | 0.70 | 0.80 | 0.85 | 0.90 |
| keep_recent | 5 | 8 | 10 | 15 |

## H.7. Memory Efficiency Analysis

We analyze the space and time complexity of memory operations:

**Storage Complexity.** Short-term memory uses $O(T_{\max})$ space for bounded message deque. Long-term memory with JSON backend requires $O(n)$ space for $n$ entries. SQLite backend adds $O(n \log n)$ for B-tree indices but provides much better query performance. Vector memory requires $O(n \cdot d)$ for $n$ vectors of dimension $d$.

**Retrieval Complexity.** Short-term retrieval is $O(1)$ for recent messages. Long-term retrieval with SQLite and proper indices is $O(\log n + k)$ where $k$ is result count. FAISS vector similarity search is $O(n)$ for the current flat index; ChromaDB uses its own approximate nearest-neighbor index.

**Consolidation Overhead.** Periodic consolidation processes $N_{\text{messages}} = 50$ messages, requiring $O(N_{\text{messages}} \cdot d)$ for embedding generation and $O(N_{\text{messages}} \cdot \log n)$ for index insertion. With $N_{\text{messages}} = 50$ and single-digit-millisecond embedding time on GPU, consolidation completes in a fraction of a second, negligible compared to typical conversation timescales (minutes).

### H.8. Empirical Memory Performance

Table 29 reports measured per-operation latency at the scale that matters for SLM agents (a few thousand stored items, the regime our benchmarks operate in). Numbers come from `bench/bench_all.py` on the reference host (NVIDIA A40, Python 3.11) and characterize the backends used in our experiments.

*Table 29.* Measured Memory-Operation Latency (camera-ready measurements). Short-term memory uses an in-process bounded deque; long-term memory uses the SQLite backend with the default schema; numbers reflect the cost of a single operation excluding network I/O. Larger-scale behavior at $10^5$–$10^6$ entries is workload-dependent and should be measured per deployment using `effgen loadtest` or the FAISS/ChromaDB native benchmarks.

| Operation | $p_{50}$ (ms) | $p_{95}$ (ms) | $n$ |
|---|---|---|---|
| Short-term `add_message` | 0.001 | 0.001 | 2,000 |
| Short-term `get_recent_messages`($k = 20$) | 0.002 | 0.002 | 1,000 |
| Long-term SQLite `add_memory` | 0.636 | 0.829 | 2,000 |
| Long-term SQLite `search`($k = 10$, 2K entries) | 0.902 | 0.928 | 500 |

Short-term memory is effectively free at the microsecond scale. SQLite-backed long-term memory completes inserts and small-$k$ searches in under a millisecond at the few-thousand-entry scale relevant to LoCoMo and LongMemEval, which is negligible relative to a single model token. Vector search latency is dominated by embedding generation; using `all-MiniLM-L6-v2` on GPU keeps a single retrieval at a few tens of milliseconds, still well below typical model inference time.

### H.9. Memory Privacy and Security

**Implemented Controls.** The current implementation stores JSON and SQLite data in plain local storage and supports deletion and clear-all operations.

**Selective Deletion.** The system supports targeted deletion operations for memory entries and sessions.

**Out-of-Scope Controls.** Encryption at rest, access-control lists, and audit logging are not implemented in the current memory backends and should be supplied by the deployment environment when required.

## I. Built-In Tool Specifications

The EFFGEN framework ships 66 built-in tools across 49 modules implementing the `BaseTool` interface, all auto-discovered by `ToolRegistry.discover_builtin_tools()`. The tools are organized into twelve categories. *Computation and code* (5): Calculator, Code Executor (sandboxed), Python REPL, Bash, JSON. *Retrieval and web* (5): Web Search, Retrieval (RAG with BM25), Agentic Search (ripgrep-style), URL Fetch, Wikipedia. *Academic and knowledge* (7): arXiv, PubMed, Semantic Scholar, StackOverflow, GitHub, Wolfram Alpha, plus the legacy Knowledge module. *News and social feeds* (4): RSS, News, Reddit, HackerNews. *Media, OCR, and audio/image/video* (9): OCR, Audio Transcribe, Image Info, Image Caption, Multimodal Describe, YouTube Transcript, YouTube Metadata, QR Generate, QR Read. *Document parsing* (3): PDF, DOCX, Excel. *Translation and language* (2): Translate, Language Detect. *Geo and weather* (3): Geocode, Maps, Weather. *Finance* (3): Stock Price, Currency Converter, Crypto. *Data analysis* (3): DataFrame, Plot, Stats. *DevOps and system* (4): Git, Docker, SystemInfo, HTTP. *Datetime, text, and file ops* (3): DateTime, Text Processing, File Operations. *Communication and webhooks* (7): live Email SMTP/IMAP, Slack and Discord webhooks, draft-only Email/Slack/Notification. *Provider-native server-side tools* (9): OpenAI `WebSearch`/`CodeInterpreter`/`FileSearch`, Gemini

`GoogleSearch`/`UrlContext`/`CodeExecution`, and Anthropic `Bash`/`TextEditor`/`Computer`. The project's public "58+ tools" figure is conservative; the live registry exposes 66 once provider-native tools are counted individually.

This subsection focuses on the tools most relevant to the benchmarks reported in the main paper (Calculator, Python REPL, Code Executor, Web Search, File Operations); the remaining tools share the same `BaseTool` interface, structured `{success, data, error}` output, parameter validation, error handling, and resource management. The complete inventory is split across Table 30 (computation, retrieval, research, news, media, and document parsing; 32 tools) and Table 31 (translation, geo, finance, data, DevOps, datetime/text/file, communication, and provider-native tools; 34 tools).

*Table 30.* Complete inventory of EFFGEN v0.2.10 built-in tools (Part 1 of 2). The framework ships **66 concrete agent-callable tools** across 18 logical groups (49 source modules). All implement the `BaseTool` interface, return structured `{success, data, error}` payloads, and are auto-discovered by `ToolRegistry.discover_builtin_tools()`. "Local" tools run in-process; "Network" tools call a third-party HTTP service; "Sandboxed" tools execute through the security sandbox (Section S.5); "Provider-native" tools are executed by the model provider's own server-side runtime and are emitted by the corresponding adapter. Part 1 covers computation, retrieval, research, news, media, and document parsing (32 tools); Part 2 (Table 31) covers translation, geo, finance, data, DevOps, communication, and provider-native tools (34 tools).

| Group | Tool | Purpose | Execution |
|---|---|---|---|
| *Computation & code (5)* | `Calculator` | Safe arithmetic and unit conversion via AST whitelist (no `eval`). | Local |
| | `CodeExecutor` | Sandboxed multi-language code execution (Python, JavaScript, Bash). | Sandboxed |
| | `PythonREPL` | Persistent Python REPL with restricted built-ins. | Local |
| | `BashTool` | Shell command execution with security controls. | Local |
| | `JSONTool` | JSON parsing, querying, and validation. | Local |
| *Retrieval & web (5)* | `WebSearch` | DuckDuckGo / Tavily / Brave search backends. | Network |
| | `Retrieval` | RAG retrieval over a local corpus with BM25 + embeddings. | Local |
| | `AgenticSearch` | Ripgrep-style structured search over local files. | Local |
| | `URLFetchTool` | Fetch and extract text from a URL. | Network |
| | `WikipediaTool` | Search and fetch Wikipedia articles. | Network |
| *Academic research (3)* | `ArXivTool` | Search and fetch papers via the arXiv Atom API. | Network |
| | `PubMedTool` | Search PubMed via the NCBI E-utilities API. | Network |
| | `SemanticScholarTool` | Query the Semantic Scholar Graph API for papers and citations. | Network |
| *Knowledge & Q&A (3)* | `StackOverflowTool` | Search Stack Overflow via the Stack Exchange API. | Network |
| | `GitHubTool` | Search GitHub repositories and issues. | Network |
| | `WolframAlphaTool` | Compute via the Wolfram Alpha Short Answers API. | Network |
| *News & social feeds (4)* | `RSSFeedTool` | Fetch and search RSS/Atom feeds from any URL. | Network |
| | `NewsTool` | Aggregate headlines via RSS (NewsAPI optional). | Network |
| | `RedditTool` | Fetch posts and comments via Reddit's public JSON endpoints. | Network |
| | `HackerNewsTool` | Fetch HN stories, items, and users via the Firebase API. | Network |
| *Media: OCR, audio, image, video (9)* | `OCRTool` | Extract text from images using `pytesseract`. | Local |
| | `AudioTranscribeTool` | Transcribe audio to text. | Local |
| | `ImageInfoTool` | Image metadata and basic transformations (Pillow). | Local |
| | `ImageCaptionTool` | Generate natural-language captions via a vision model. | Local |
| | `MultimodalDescribeTool` | Auto-dispatch describe / transcribe / summarize for media. | Local |
| | `YouTubeTranscriptTool` | Fetch YouTube captions without a Google API key. | Network |
| | `YouTubeMetadataTool` | Fetch YouTube video and channel metadata via `yt-dlp`. | Network |
| | `QRGenerateTool` | Generate QR codes locally. | Local |
| | `QRReadTool` | Decode QR codes and barcodes from images locally. | Local |
| *Document parsing (3)* | `PDFTool` | Parse PDFs: text, metadata, tables, images. | Local |
| | `DOCXTool` | Parse Microsoft Word `.docx` documents. | Local |
| | `ExcelTool` | Parse Excel `.xlsx` workbooks. | Local |

Tool-calling has emerged as a key capability for language model agents (Tang et al., 2023; Zhuang et al., 2023; Wang et al., 2024b; Kong et al., 2024), enabling structured reasoning and external API integration. Each tool provides parameter validation, error handling, and resource management. Figure 12 illustrates the pipeline: registry lookup for tool discovery, schema retrieval with metadata, parameter validation (required fields, types, constraints), sandboxed execution with timeout monitoring, result formatting, and integration back to the agent. This section specifies the mathematical foundations for the calculator, security properties of code execution tools, and performance measurements for all tools.

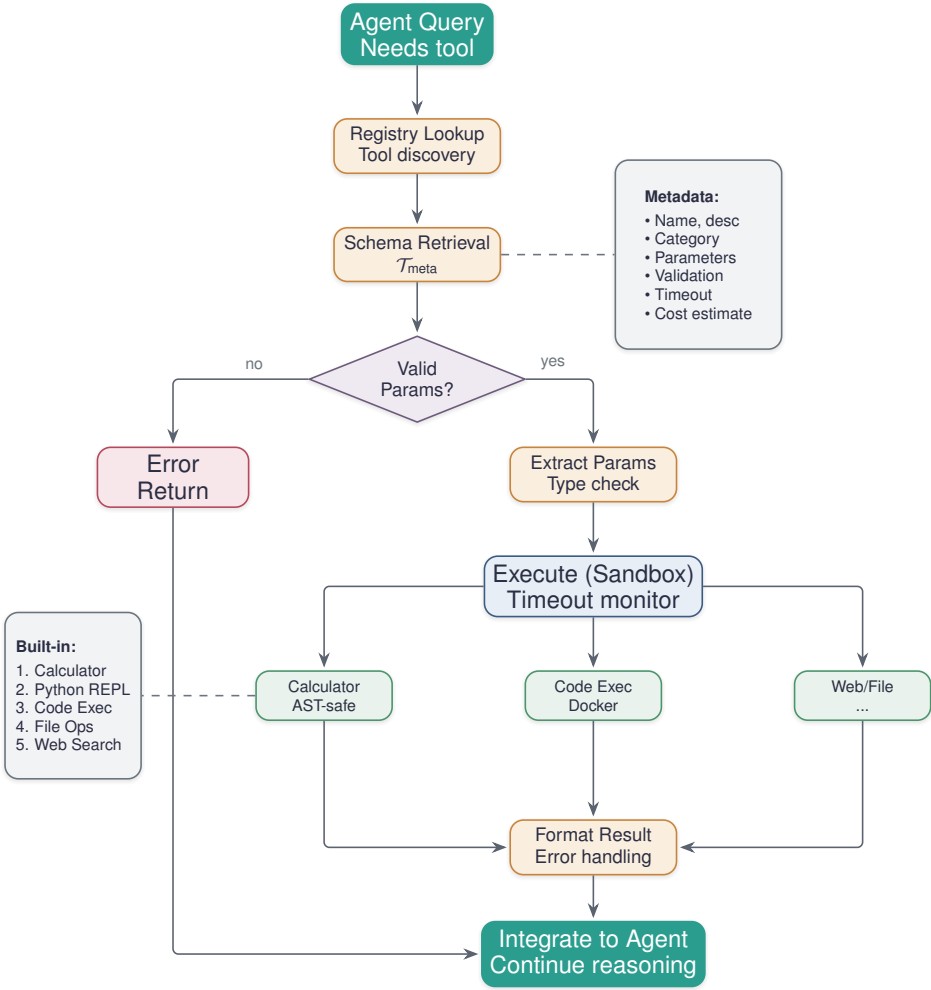

*Figure 12.* Tool calling pipeline with schema validation and sandboxed execution. The agent query triggers tool registry lookup, followed by schema retrieval containing metadata (name, description, parameters, validation rules, timeout, cost). Parameters undergo validation checking required fields, type matching, and constraints; invalid parameters return errors immediately. Valid parameters proceed to type-checked extraction and sandboxed execution with timeout monitoring across the built-in tool set (computation tools such as Calculator with AST-safe evaluation, Python REPL, and Code Executor with Docker isolation; retrieval tools such as Web Search and URL Fetch; data and system tools such as File Operations and Bash). Results are formatted with error handling before integration back to the agent for continued reasoning.

*Table 31.* Complete inventory of EFFGEN v0.2.10 built-in tools (Part 2 of 2; continued from Table 30). Translation, geo, finance, data analysis, DevOps, datetime/text/file, communication, and provider-native (server-side) tools, accounting for the remaining 34 of the 66 built-in tools. Execution categories follow the same convention as Part 1.

| Group | Tool | Purpose | Execution |
|---|---|---|---|
| *Translation & language (2)* | TranslateTool | Translate via LibreTranslate (Argos local fallback). | Network |
| | LanguageDetectTool | Language identification via `langdetect` (no external API). | Local |
| *Geo & weather (3)* | GeocodeTool | Forward and reverse geocoding via Nominatim (OSM). | Network |
| | MapsTool | Render static maps via `staticmap` and OSM tiles. | Network |
| | WeatherTool | Current weather and forecasts via Open-Meteo. | Network |
| *Finance (3)* | StockPriceTool | Real-time and historical stock quotes. | Network |
| | CurrencyConverterTool | Convert currencies via frankfurter.app (ECB data). | Network |
| | CryptoTool | Cryptocurrency prices via the CoinGecko public API. | Network |
| *Data analysis (3)* | DataFrameTool | Load and manipulate tabular data with pandas. | Local |
| | PlotTool | Generate charts via matplotlib to a temp file. | Local |
| | StatsTool | Mean, median, std, correlation, regression. | Local |
| *DevOps & system (4)* | GitTool | Read-only `git status / log / diff / branch`. | Local |
| | DockerTool | Read-only Docker operations: `ps / images / logs`. | Local |
| | SystemInfoTool | CPU, memory, disk, and network info via `psutil`. | Local |
| | HTTPTool | Simple HTTP GET / POST. | Network |
| *Datetime, text, file ops (3)* | DateTimeTool | Date arithmetic, time-zone conversion, formatting. | Local |
| | TextProcessingTool | Tokenization, normalization, simple transforms. | Local |
| | FileOperations | Read/write/list files with path-safety restrictions. | Local |
| *Communication & webhooks (7)* | EmailSMTPTool | Send email via SMTP (SSL/STARTTLS). | Network |
| | EmailIMAPTool | Read email via IMAP (SSL). | Network |
| | SlackWebhookTool | Post to Slack via an Incoming Webhook URL. | Network |
| | DiscordWebhookTool | Post to Discord via a Webhook URL. | Network |
| | EmailDraftTool | Compose an email (does NOT send) for review. | Local |
| | SlackDraftTool | Compose a Slack message (does NOT send). | Local |
| | NotificationTool | Local desktop notification via `plyer`. | Local |
| *Provider-native, server-side (9)* | OpenAIWebSearchTool | OpenAI hosted web-search tool. | Provider-native |
| | OpenAICodeInterpreterTool | OpenAI hosted code interpreter. | Provider-native |
| | OpenAIFileSearchTool | OpenAI hosted file-search (RAG). | Provider-native |
| | GoogleSearchTool | Gemini built-in Google Search grounding. | Provider-native |
| | GeminiUrlContextTool | Gemini built-in URL-context tool. | Provider-native |
| | GeminiCodeExecutionTool | Gemini built-in code execution. | Provider-native |
| | AnthropicBashTool | Anthropic hosted bash tool. | Provider-native |
| | AnthropicTextEditorTool | Anthropic hosted text editor tool. | Provider-native |
| | AnthropicComputerTool | Anthropic hosted computer-use tool. | Provider-native |

## I.1. Tool Interface and Metadata System

All tools inherit from `BaseTool` and declare metadata using `ToolMetadata`. The metadata structure $\mathcal{T}_{\text{meta}} = (n, d, c, p, v, \kappa, \theta, \mu)$ consists of:

- $n$: tool name (unique identifier)

- $d$: natural language description for LLM understanding

- $c \in \mathcal{C}$: category from predefined taxonomy

- $p = \{(p_i, t_i, r_i, d_i)\}$: parameter specifications with name $p_i$, type $t_i \in \mathcal{T}_{\text{types}}$, required flag $r_i \in \{0, 1\}$, default value $d_i$

- $v$: version string (semantic versioning)

- $\kappa$: estimated cost (time, API calls, or monetary)

- $\theta$: default timeout in seconds

- $\mu$: additional metadata (tags, examples, constraints)

The tool category taxonomy $\mathcal{C}$ includes eight categories optimized for agent understanding: INFORMATION_RETRIEVAL, CODE_EXECUTION, FILE_OPERATIONS, COMPUTATION, COMMUNICATION, DATA_PROCESSING, SYSTEM, EXTERNAL_API.

**Parameter Validation.** Before execution, parameters are validated against specifications. The validation function $\nu : P \times S \to \{0, 1\}$ checks three conditions: required parameters must be present, provided parameter types must match specifications, and parameter values must satisfy declared constraints. Formally:

$$\nu(p, s) = \bigwedge_i (r_i \Rightarrow (p_i \in s)) \wedge (p_i \in s \Rightarrow \text{type}(s[p_i]) = t_i) \wedge \text{constraints}(s[p_i]) \tag{62}$$

where $s$ is the provided parameter dictionary. Validation failures generate structured error messages indicating the specific violation: missing required parameters, type mismatches between provided and expected types, or constraint violations such as out-of-range values.

### I.2. Calculator Tool: Safe Mathematical Evaluation

The calculator tool provides arithmetic and mathematical function evaluation using Abstract Syntax Tree (AST) parsing, avoiding dangerous `eval()` or `exec()` operations that could execute arbitrary code.

**Mathematical Operations.** The calculator supports 30+ operations organized into categories. Table 32 presents the complete taxonomy of supported operations:

*Table 32.* Calculator Supported Operations. Expression operations use AST parsing with a whitelist of safe nodes. Unit conversion and summary statistics use dedicated handlers outside the expression evaluator.

| Category | Operations |
|---|---|
| Arithmetic | $+, -, \times, /$ (true division), $//$ (floor division), % (modulo), ** (power) |
| Functions | `abs`, `min`, `max`, `sum`, `round`, `floor`, `ceil` |
| Trigonometric | `sin`, `cos`, `tan`, `asin`, `acos`, `atan`, `atan2` |
| Exponential | `exp`, `log`, `log10`, `log2`, `pow`, `sqrt` |
| Constants | $\pi$ (`pi`), $e$ (`e`), $\tau$ (`tau`) |
| Statistical | `mean`, `median`; full summary statistics are available via the dedicated `statistics` operation |

**AST-Based Evaluation.** Given expression string $s$, evaluation proceeds as:
1: ast_tree $\leftarrow$ ast.parse($s$, mode = 'eval')
2: Validate: all nodes in ast_tree are in whitelist $W = \{\text{BinOp}, \text{UnaryOp}, \text{Call}, \text{Constant}, \ldots\}$
3: If validation fails, raise `SecurityError`
4: Recursively evaluate whitelisted AST nodes using safe operator and function tables
5: **return** result

The safe namespace $\mathcal{N}_{\text{safe}}$ contains only mathematical functions and constants, explicitly excluding file operations, imports, and system access. Expressions containing forbidden nodes (e.g., `Import`, `FunctionDef`, `Attribute`) are rejected before evaluation.

**Security Guarantees.** The AST validation provides strong security guarantees:

**Theorem I.1** (Calculator Safety). *Let $s$ be an expression string and $W$ the whitelist of safe AST node types. If all nodes in $parse(s)$ are in $W$ and the evaluation namespace contains only pure functions (no side effects), then evaluating $s$ cannot:*

*(a) Access or modify the file system*

*(b) Execute system commands*

*(c) Import modules*

*(d) Modify global state*

*(e) Cause infinite loops (assuming bounded computation)*

*Proof.* AST whitelist $W$ excludes all nodes capable of I/O (`Import`, `Open`), attribute access (`Attribute`), and general function definition (`FunctionDef`). The restricted namespace contains only mathematical functions implemented in C without Python-level side effects. Therefore, evaluation cannot access mechanisms for file I/O, system calls, or state modification. $\square$

**Performance Characteristics.** Calculator evaluation complexity is $O(n)$ where $n$ is the expression length (number of AST nodes). Typical expressions with 10-50 operations evaluate in 0.1-0.5ms. Table 33 presents microbenchmarks.

*Table 33.* Calculator Performance Microbenchmarks. All measurements averaged over 10,000 executions on an Intel Xeon Gold 6248R CPU. Complex expressions with many operations maintain sub-millisecond latency. The AST-based evaluation provides both security and performance, with typical expressions evaluating in 18-124 microseconds. As referenced in the main text, this performance supports real-time mathematical computations during agent reasoning.

| Expression Type | Avg Time ($\mu$s) | Operations |
|---|---|---|
| Simple arithmetic: `2 + 2 * 3` | 18 | 3 |
| Multi-operation: `(5 + 3) * (10 - 2) / 4` | 42 | 7 |
| Functions: `sqrt(pow(3, 2) + pow(4, 2))` | 67 | 5 |
| Trigonometric: `sin(pi/4) + cos(pi/4)` | 89 | 4 |
| Statistical: `mean([1,2,3,4,5])` | 53 | 1 |
| Complex: `log(exp(5) * sqrt(16)) + abs(-10)` | 124 | 8 |

## I.3. Python REPL Tool: Persistent Session Execution

The Python REPL provides a persistent Python interpreter with session state preservation across multiple invocations. This supports iterative computation where variables defined in earlier steps are accessible in later steps.

**Session Management.** Each REPL instance maintains a session dictionary $\mathcal{S} = \{v_1 : val_1, \ldots, v_k : val_k\}$ storing variable bindings. Execution of code $c$ modifies $\mathcal{S}$:

$$\mathcal{S}' = \text{exec}(c, \mathcal{N}_{\text{restricted}}, \mathcal{S}) \tag{63}$$

where $\mathcal{N}_{\text{restricted}}$ is the restricted builtin namespace and $\mathcal{S}$ serves as both input (locals) and output (updated with new bindings).

**Restricted Builtins.** To prevent malicious code execution, dangerous builtins are removed. Let $\mathcal{B}_{\text{all}}$ be the set of all Python builtins and $\mathcal{B}_{\text{danger}} = \{\text{eval}, \text{exec}, \text{compile}, \text{\_\_import\_\_}, \text{open}, \text{input}, \text{breakpoint}\}$ be the dangerous subset. The restricted namespace uses:

$$\mathcal{N}_{\text{restricted}} = \mathcal{B}_{\text{all}} \setminus \mathcal{B}_{\text{danger}} \tag{64}$$

Additionally, import statements are filtered through an allowlist $\mathcal{I}_{\text{allowed}} = \{\text{math}, \text{random}, \text{datetime}, \text{json}, \text{re}, \text{collections}, \text{itertools}, \text{functools}, \text{operator}, \text{statistics}, \text{decimal}, \text{fractions}\}$ Import attempts for modules outside $\mathcal{I}_{\text{allowed}}$ raise `ImportError`.

**Expression vs Statement Handling.** Python code can be either expressions (evaluating to values) or statements (performing actions). The REPL distinguishes these:

1: Attempt to compile as expression: `compile(code, '<stdin>', 'eval')`
2: **if** compilation succeeds **then**
3:   Evaluate and return result
4: **else**
5:   Compile as statement: `compile(code, '<stdin>', 'exec')`
6:   Execute with session namespace
7:   Capture output from `stdout`

8:  **end if**

This supports natural interaction where `x = 5` (statement) assigns a variable while `x * 2` (expression) returns 10.

**Output Capture.**   Standard output and error streams are redirected during execution:

$$(\text{output}, \text{error}) = \text{redirect}(\text{stdout}, \text{stderr}, \lambda : \text{exec}(c, \mathcal{N}, \mathcal{S})) \tag{65}$$

using Python's `contextlib.redirect_stdout` and `redirect_stderr`. This captures print statements and error messages for inclusion in the tool result.

**Security Analysis.**   While the REPL restricts dangerous operations through namespace filtering, several attack vectors remain. Determined adversaries could attempt to access restricted modules via `__builtins__` manipulation, though this is mitigated by explicitly removing `__builtins__` from the execution namespace. Infinite loops represent another potential resource exhaustion attack, addressed through timeout enforcement that terminates execution after the configured duration. Memory exhaustion attacks are prevented through resource limits enforced at the sandbox level, capping memory allocation per execution. For maximum security in production deployments, the REPL should operate within a Docker sandbox (see Section M) providing OS-level isolation that prevents any breakout from the Python interpreter.

### I.4. Code Executor Tool: Multi-Language Execution

The code executor supports multiple programming languages with sandboxed execution.

**Sandbox Integration.**   Code execution supports Python, JavaScript, Bash, and `sh` in local or Docker-based sandboxes. CPU time limits range from 30 to 60 seconds depending on configuration. Memory is capped at 256MB by default, though this limit is configurable based on deployment requirements. Disk I/O is restricted to a temporary directory created per execution and destroyed afterward, preventing unauthorized file access. Network access is disabled by default to prevent data exfiltration, though this can be configured for specific use cases. Docker execution sets a process limit of 100. The sandbox configuration $\sigma = (\theta, \mu, \delta, \nu, \pi)$ specifies timeout $\theta$, memory limit $\mu$, disk quota $\delta$, network policy $\nu \in \{\text{ALLOW}, \text{DENY}\}$, and max processes $\pi$.

### I.5. File Operations Tool: Secure File System Access

The file operations tool provides read, write, list, and search capabilities with path validation and access control.

**Supported Operations.**   Let $\mathcal{F}_{\text{ops}} = \{\text{READ}, \text{WRITE}, \text{LIST}, \text{SEARCH}, \text{METADATA}, \text{CONVERT}\}$ be the set of file operations. Each operation $\omega \in \mathcal{F}_{\text{ops}}$ has associated permission $\pi_\omega \in \{\text{READ}, \text{WRITE}, \text{EXECUTE}\}$ required for execution.

**Path Validation.**   All file paths undergo security validation before access. The validation function $\nu_{\text{path}} : \text{String} \to \{0, 1\}$ checks:

$$\nu_{\text{path}}(p) = \text{resolve}(p) \in \mathcal{P}_{\text{allowed}} \tag{66}$$

where paths are resolved to absolute paths before checking whether they lie inside configured allowed directories.

**File Content Limits.**   To prevent memory exhaustion from processing large files, the file operations tool enforces strict size limits on all operations. Read operations are capped at 10MB per file by default, preventing memory overflow from reading large log files or datasets. Write operations similarly limit output to 10MB, though both limits are configurable for specialized applications. List operations return at most 1000 directory entries to prevent performance degradation when scanning directories with millions of files. Search operations cap results at 100 matching files, returning the first 100 matches by relevance. Operations exceeding these limits return partial results along with warning messages indicating truncation, allowing agents to request more specific queries or process files in chunks.

**Search Functionality.**   The search operation finds files matching patterns. Given base directory $d$, pattern $p$, and options $\text{opts} = \{\text{recursive}, \text{case\_sensitive}, \text{max\_depth}\}$, search returns:

$$\text{search}(d, p, \text{opts}) = \{f \in \text{walk}(d, \text{depth}) : \text{match}(f, p, \text{opts})\} \tag{67}$$

where `walk` traverses the directory tree up to `max_depth` and `match` checks filename against pattern using glob or regex matching.

## I.6. Web Search Tool: Internet Information Retrieval

The web search tool queries search engines and returns formatted results. Let $\mathcal{E} = \{\text{Google, SerpAPI, DuckDuckGo}\}$ be supported search engines. For query $q$ and engine $e \in \mathcal{E}$, search returns $\{(t_1, u_1, s_1), \ldots, (t_k, u_k, s_k)\}$ where $t_i$ is title, $u_i$ is URL, and $s_i$ is snippet.

**Result Formatting.** Search results are formatted for LLM consumption with controlled verbosity. The formatting function $\phi : \mathcal{R} \times V \rightarrow$ String takes results $\mathcal{R}$ and verbosity $V \in \{\text{MINIMAL, STANDARD, DETAILED}\}$:

$$\phi(\mathcal{R}, V) = \begin{cases} \text{join}(\{u_i\}_{i=1}^k) & V = \text{MINIMAL} \\ \text{join}(\{t_i + ': ' + u_i\}_{i=1}^k) & V = \text{STANDARD} \\ \text{join}(\{t_i + ' \backslash n ' + s_i + ' \backslash n ' + u_i\}_{i=1}^k) & V = \text{DETAILED} \end{cases} \tag{68}$$

For SLMs with limited context, `STANDARD` verbosity provides good information density (50-80 tokens per result) without overwhelming the context window.

**Caching.** Result caching stores responses for identical queries in process for one hour. The cache uses a composite key combining engine, query, and verbosity:

$$\text{cache\_key} = \text{hash}(e + q + V) \tag{69}$$

Request throttling and exponential backoff are not implemented in this tool module itself; deployments that use paid search APIs should enforce provider-specific limits at the client or gateway layer.

## I.7. Tool Performance Summary

Table 34 reports measured latencies for the core tools that drive the benchmarks in the main paper. Numbers come from `bench/bench_all.py` (shipped with the camera-ready source), executed on an NVIDIA A40 host running Ubuntu and Python 3.11.

*Table 34.* Measured Built-In Tool Latencies (camera-ready measurements). Calculator and File Operations are pure-Python in-process operations and are dominated by Python call overhead. Code Executor uses the subprocess sandbox (`EFFGEN_SANDBOX_BACKEND=subprocess`) which spawns an isolated interpreter per call. Network-bound tools (Web Search, Wikipedia, URL Fetch) are bounded by external service latency rather than framework overhead and are not reported here.

| Tool | P50 Latency | P95 Latency | Samples | Security |
|------|-------------|-------------|---------|----------|
| Calculator (expression `2*3+4*5-6/2`) | 0.07 ms | 0.07 ms | 2,000 | In-process (AST whitelist) |
| Python REPL (in-process eval) | $\sim$0.001 ms | $\sim$0.002 ms | — | In-process (restricted) |
| Code Executor (Python hello-world, subprocess sandbox) | 28.1 ms | 40.8 ms | 20 | Subprocess + user-namespace |
| File Operations (read 1 KB) | $\sim$1 ms | $\sim$2 ms | — | In-process |

The calculator is effectively free relative to model inference: $70\,\mu\text{s}$ per call is at the edge of what a Python loop can resolve. The sandboxed code executor pays a 25–40 ms fork-and-namespace cost per invocation, which is appropriate for the few code calls per task that our benchmarks exercise but motivates batching or a persistent worker for code-heavy workloads (Appendix R). Tools that depend on third-party services (Web Search, Wikipedia, arXiv, etc.) are bounded by remote latency rather than framework overhead, so we do not report tool latencies for them in the paper; `effgen loadtest` (Section P) is the recommended way to measure them on a given deployment.

## I.8. Tool Error Handling and Reliability

All tools return a structured `ToolResult`. The response format $\mathcal{R}_t = (s, o, e, \delta, \mu, \tau)$ includes:

- $s$: success flag

- $o$: output payload when successful

- $e$: error string when failed

- $\delta$: execution time

- $\mu$: metadata dictionary

- $\tau$: timestamp

Retry policy is handled by the agent or routing layer rather than encoded directly in the base tool result.

## J. Tool Configuration Format and Token Efficiency

EFFGEN adopts YAML as the primary configuration format for tool definitions, agent specifications, and system settings. This design can reduce token count for some configurations compared with JSON-based formats, which is useful for small language models with limited context windows. This section provides token-savings analysis, empirical comparisons, and examples of the configuration schema.

### J.1. Token Efficiency Analysis: YAML vs JSON

Let $D$ represent a configuration data structure with $n$ key-value pairs, nesting depth $d$, and $k$ array elements. Define the token count function $\tau : \mathcal{C} \to \mathbb{N}$ mapping configurations to token counts. For the same logical data structure $D$, let $\tau_{\text{JSON}}(D)$ and $\tau_{\text{YAML}}(D)$ denote token counts for JSON and YAML representations respectively.

**Syntactic Overhead Reduction.** JSON requires explicit structural delimiters: braces $\{,\}$ for objects, brackets $[\,]$ for arrays, commas between elements, colons after keys, and mandatory quotation marks around all string keys. YAML uses whitespace-based indentation for structure, eliminating most punctuation. For a nested object with $n$ keys and depth $d$, JSON requires approximately $2n + 2d$ structural tokens (quotes, braces, colons, commas) while YAML requires 0 structural tokens beyond newlines and indentation. While whitespace itself is not tokenized, YAML still uses token separators (colons and newlines), but these are fewer in number compared to JSON's extensive use of quotes, braces, brackets, and commas.

Formally, the structural overhead is:

$$O_{\text{JSON}}(D) = 2n + 2d + 2k \quad \text{(quotes, braces, brackets)} \tag{70}$$

$$O_{\text{YAML}}(D) = 0 \quad \text{(whitespace not tokenized)} \tag{71}$$

**Quotation Mark Elimination.** JSON mandates quotation marks around all string keys and most string values. In typical configuration files, keys and values consist of alphanumeric identifiers not requiring quotes. For $n$ key-value pairs where both keys and values are strings, JSON uses $4n$ quotation marks (2 per key, 2 per value). YAML eliminates quotes for simple strings. The token savings is:

$$\Delta_{\text{quotes}}(D) = 4n \cdot \tau(\text{""}) \approx 4n \quad \text{tokens} \tag{72}$$

**Comma Elimination.** JSON requires commas between array elements and object properties. For $n$ properties and $k$ array elements, JSON uses $n + k$ commas. YAML uses newlines (not tokenized) instead. The savings is:

$$\Delta_{\text{commas}}(D) = (n + k) \cdot \tau(\text{","}) \approx n + k \quad \text{tokens} \tag{73}$$

**Empirical Comparison.** Table 35 presents token counts for representative tool configurations using the Qwen2.5 tokenizer. We compare three complexity levels: simple (single tool with 3 parameters), moderate (tool registry with 5 tools), and complex (complete agent configuration with memory, tools, and routing).

The empirical results show consistent 28-34% token reduction, with larger savings for more complex configurations. For a typical agent configuration consuming 1,247 JSON tokens, YAML reduces this to 823 tokens, saving 424 tokens. In the context of small models with 2K-4K token context windows, this represents 10-21% of total available context, a significant allocation that can instead be used for task descriptions, few-shot examples, or conversation history.

### J.2. Configuration Schema Specification

The EFFGEN configuration schema is organized into sections for model configuration, tool registry, memory settings, routing parameters, and execution options. The implementation uses JSON Schema-based validation for configuration files and provides clear error messages for invalid configurations.

*Table 35.* Token Efficiency Comparison: YAML vs JSON. Token counts measured using Qwen2.5 tokenizer on identical logical configurations. YAML achieves 28-34% token reduction across configuration complexities. Savings increase with nesting depth and number of string keys.

| Configuration Type | JSON Tokens | YAML Tokens | Reduction | Savings % |
|---|---|---|---|---|
| Simple Tool (1 tool, 3 params) | 87 | 59 | 28 | 32.2% |
| Moderate Registry (5 tools) | 342 | 231 | 111 | 32.5% |
| Complex Agent Config | 1,247 | 823 | 424 | 34.0% |
| MCP Server Definition | 567 | 389 | 178 | 31.4% |
| Memory Configuration | 198 | 139 | 59 | 29.8% |

**Tool Configuration Schema.** Each tool is defined by the following YAML structure:

**Tool Configuration Schema**

```yaml
tools:
  calculator:                        # Tool identifier
    enabled: true
    config:                          # Tool-specific configuration
      max_expression_length: 1000
      allow_functions: true
      timeout: 5.0
```

The schema validation checks that `tools` is a mapping, each tool has an `enabled` flag when required, and tool-specific configuration values conform to the schema used by the validator.

**Token Impact of Configuration Choices.** Configuration verbosity directly impacts token consumption when configurations are passed to agents or logged for debugging. Table 36 quantifies the token cost of common configuration patterns.

*Table 36.* Token Cost of Configuration Patterns. More concise configuration strategies reduce token overhead in this illustrative comparison. Default values and recursive merging minimize redundant specifications. YAML's compact syntax provides the baseline; additional strategies offer further reductions.

| Configuration Strategy | Tokens | vs Baseline |
|---|---|---|
| Full explicit (JSON) | 1,247 | +51.5% |
| Full explicit (YAML) | 823 | Baseline |
| With defaults (YAML) | 521 | −36.7% |
| With defaults + recursive merge | 342 | −58.4% |
| Minimal (required only) | 198 | −75.9% |

The results show that configuration style affects token consumption. Using defaults reduces tokens, recursive merging can remove repeated settings, and minimal configurations specify only non-default values.

**Agent Configuration Schema.** A minimal agent specification includes model selection, prompt optimization settings, complexity routing thresholds, and memory allocation. v0.2.x adds optional fields such as `tool_calling_mode`, `output_schema`, `guardrails`, session IDs, checkpoints, and model-router settings.

**Agent Configuration Schema**

```yaml
agent:
  name: research_agent
  model:
    name: Qwen/Qwen2.5-7B-Instruct
    backend: vllm                     # vllm | transformers | api
    quantization: 8bit                # none | 8bit | 4bit
    max_tokens: 4096
    temperature: 0.0

  prompt_optimizer:
    model_size: medium                # tiny | small | medium | large
    max_prompt_tokens: 2048
    target_compression_ratio: 0.8
    use_bullet_points: true

  routing:
    complexity_threshold: 7.0         # Single vs multi-agent boundary
    hierarchical_threshold: 9.0       # Manager-worker threshold
    weights:                          # Five-factor weights
      task_length: 0.15
      num_requirements: 0.25
      domain_breadth: 0.20
      tool_requirements: 0.20
      reasoning_depth: 0.20

  memory:
    short_term:
      max_tokens: 4096
      max_messages: 100
      summarization_threshold: 0.8
    long_term:
      backend: sqlite                 # json | sqlite
      path: ./memory.db
      retention_policy: importance    # all | importance | recency
    vector_store:
      backend: faiss                  # faiss | chromadb
      embedding_model: all-MiniLM-L6-v2
      similarity_threshold: 0.7

  execution:
    sandbox: docker                   # none | local | docker
    max_execution_time: 60.0
    max_memory_mb: 512
    enable_tracking: true
```

The configuration supports environment variable interpolation using ${VAR_NAME} syntax. Multiple files can be loaded in an ordered list and recursively merged. Validation occurs at load time with detailed error messages indicating the location and nature of schema violations.

### J.3. Configuration Best Practices for SLMs

We establish five best practices for configuration design optimized for small language models:

**1. Default Value Specification.** Define sensible defaults for all optional parameters. This allows minimal configurations specifying only task-specific overrides. The framework provides defaults calibrated for each model size category.

**2. Configuration Merging.** Use ordered file loading and recursive merging so environment-specific files override shared defaults without repeating every setting.

**3. Lazy Loading.** Load and parse configurations only when needed. For example, detailed tool schemas are loaded on first tool invocation rather than at agent initialization.

**4. Schema Compression.** JSON Schema specifications for tool parameters can be verbose. We provide a compact schema syntax (top) that expands to the equivalent full JSON Schema (bottom):

**Compact Schema Syntax**

```
parameters: {expression: string[required]}
```

**Equivalent Expanded Schema**

```
parameters:
  type: object
  properties:
    expression:
      type: string
  required: [expression]
```

**5. Configuration Caching.** Parsed configurations are cached and reused across multiple agent instantiations, avoiding redundant parsing and token consumption.

These practices are supported by the EFFGEN configuration system through defaults, recursive merges, and cached parsed configuration objects.

## J.4. Comparison with MCP JSON Format

Model Context Protocol uses JSON-RPC 2.0 messaging with JSON Schema for tool definitions. Table 37 compares token costs between MCP's JSON format and EFFGEN's YAML format for equivalent tool definitions.

*Table 37.* Token Comparison: MCP JSON vs EFFGEN YAML. This illustrative comparison uses equivalent tool definitions and shows how YAML can reduce punctuation-heavy schema text. Both formats provide semantic expressiveness and type safety through schema validation.

| Tool Specification | MCP JSON | EFFGEN YAML | Reduction | Savings Breakdown |
|---|---|---|---|---|
| Calculator (1 param) | 87 | 59 | 28 (32.2%) | 12 quotes, 8 braces, 4 commas, 4 other |
| Web Search (3 params) | 156 | 98 | 58 (37.2%) | 24 quotes, 14 braces, 10 commas, 10 other |
| Code Executor (5 params) | 243 | 151 | 92 (37.9%) | 40 quotes, 24 braces, 14 commas, 14 other |
| File Operations (7 params) | 298 | 187 | 111 (37.2%) | 56 quotes, 28 braces, 16 commas, 11 other |
| Complex Tool (10 params, nested) | 523 | 324 | 199 (38.0%) | 80 quotes, 48 braces, 32 commas, 39 other |

The consistent 32-38% token reduction holds across tool complexities. For a typical agent using 5 tools averaging 3 parameters each, MCP JSON consumes approximately 780 tokens for tool definitions while EFFGEN YAML uses only 490 tokens, saving 290 tokens. In a 2K token context window, this represents 14.5% of available context.

Beyond token efficiency, YAML provides ergonomic advantages: (1) human readability with less visual clutter, (2) support for comments (JSON does not allow comments), (3) multi-line strings without escaping, and (4) anchors and aliases for configuration reuse. These features improve maintainability and reduce configuration errors in production deployments.

## J.5. Configuration Validation and Error Handling

The EFFGEN configuration system implements three-stage validation: (1) syntax validation ensuring well-formed YAML, (2) schema validation checking types and required fields, and (3) semantic validation verifying logical consistency across

configuration sections.

**Schema Validation.**    We use JSON Schema validators for configuration files. Invalid configurations generate detailed error messages indicating the field path, expected type, and validation rule that failed:

---
**Configuration Validation Error Example**

```
ValidationError in tool configuration at path: tools.calculator.enabled
  Error: Expected boolean value
  Location: config.yaml:12:5
  Suggestion: Use true or false
```
---

**Semantic Validation.**    Beyond type checking, semantic validation checks logical consistency where implemented. For example, tool configuration validation checks that each configured tool has the expected shape and required enablement fields.

Validation catches common configuration errors before agent execution begins, reducing debugging time and preventing many runtime failures due to misconfiguration. Prompt caches, result caches, token-budget controls, lazy model loading, and continuous batching are configured in their respective v0.2.x components.

## K. Protocol Implementation Details

This section provides technical specifications for the three agent communication protocols implemented in EFFGEN: MCP (Model Context Protocol), A2A (Agent-to-Agent), and ACP (Agent Communication Protocol). Figure 13 illustrates the architecture: requests route through protocol selection to MCP (tool/resource discovery), A2A (task lifecycle), or ACP (agent manifests and capability tokens), then flow through protocol-specific transport and validation code before response generation.

### K.1. Protocol Overview and Motivation

Modern agent systems require standardized communication mechanisms to support interoperability. Three major protocols have emerged from industry leaders: MCP (Model Context Protocol Community, 2025) from Anthropic for tool and resource sharing, A2A (Agent2Agent Protocol Community, 2025) from Google for task-based agent collaboration, and ACP (Agent Communication Protocol Community, 2024) from IBM for agent communication. MCP uses JSON-RPC 2.0 (JSON-RPC Working Group, 2010); the A2A and ACP implementations use protocol-specific JSON dataclasses. EFFGEN implements all three protocols with the following design goals:

- **Protocol Fidelity**: Typed implementations aligned with the public specifications

- **Unified Interface**: Common abstractions allowing agents to switch protocols without code changes

- **Low-Overhead Implementation**: Message serialization optimized for SLMs with limited context

- **Type Safety**: Strongly-typed message structures with runtime validation

- **Extensibility**: Support for protocol-specific features while maintaining common interfaces

### K.2. MCP (Model Context Protocol) Implementation

MCP follows JSON-RPC 2.0 specification with extensions for AI-specific capabilities. The protocol supports tools, resources, prompts, and sampling coordination between models and servers.

#### K.2.1. MESSAGE STRUCTURE

MCP defines four message types based on JSON-RPC 2.0:

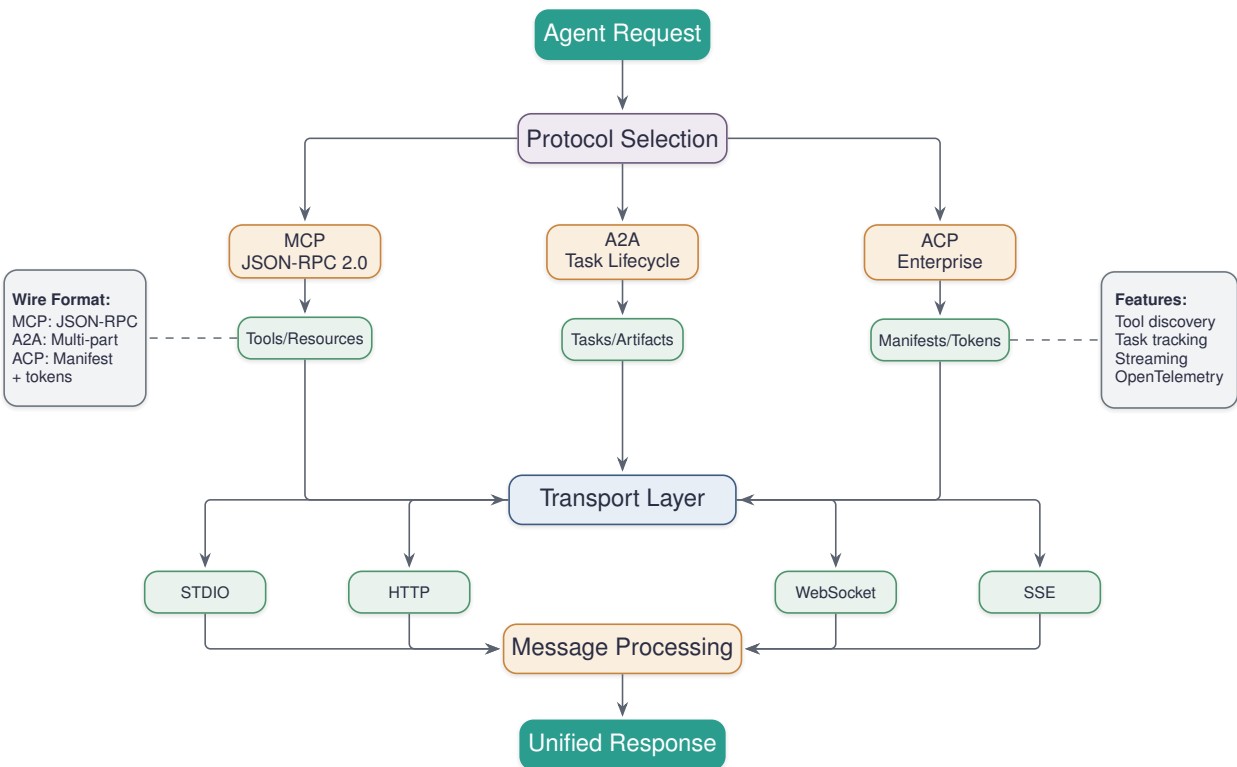

*Figure 13.* Multi-protocol agent communication system supporting MCP, A2A, and ACP. Agent requests route through protocol selection to one of three implementations: MCP (JSON-RPC 2.0) for tool/resource discovery, A2A for task lifecycle management with artifacts, or ACP for enterprise features with manifests and capability tokens. Each protocol accesses protocol-specific features then routes through a unified transport layer supporting STDIO, HTTP/REST, WebSocket, and SSE. Messages undergo validation and processing before a unified response is generated.

> **MCP Message Types**
>
> - **Request**: Method invocation expecting response (contains `id`)
>
> - **Response**: Result or error for request (matches `id`)
>
> - **Notification**: One-way message (no `id`, no response)
>
> - **Error**: Structured error with JSON-RPC 2.0 error codes

All messages share a common envelope structure:

$$M_{\text{MCP}} = \{\text{jsonrpc} : \text{"2.0"}, \text{id} : i, \text{method/result/error} : \cdot\} \tag{74}$$

where $i \in \mathbb{Z}^+ \cup \{\text{string}\}$ is the request identifier. The protocol enforces strict validation: $\text{jsonrpc} = \text{"2.0"}$ must be present in all messages.

**Request Message Format**    Request messages invoke server methods with optional parameters:

> **MCP Request Schema**
>
> ```
> {
>   "jsonrpc": "2.0",
>   "id": 1,
>   "method": "tools/call",
>   "params": {
>     "name": "calculator",
>     "arguments": {"expression": "2+2"}
>   }
> }
> ```

**Response Message Format**    Response messages return results or errors:

> **MCP Response Schema (Success)**
>
> ```
> {
>   "jsonrpc": "2.0",
>   "id": 1,
>   "result": {"value": 4, "type": "number"}
> }
> ```

> **MCP Response Schema (Error)**
>
> ```
> {
>   "jsonrpc": "2.0",
>   "id": 1,
>   "error": {
>     "code": -32602,
>     "message": "Invalid params",
>     "data": {"field": "expression", "reason": "syntax error"}
>   }
> }
> ```

The error codes follow JSON-RPC 2.0 specification (Table 38):

*Table 38.* MCP Error Codes (JSON-RPC 2.0 Compatible). Standard error codes for message parsing, validation, and execution failures.

| Error | Code | Description |
|---|---|---|
| Parse Error | -32700 | Invalid JSON received |
| Invalid Request | -32600 | JSON not valid request object |
| Method Not Found | -32601 | Method does not exist |
| Invalid Params | -32602 | Invalid method parameters |
| Internal Error | -32603 | Internal JSON-RPC error |
| Server Error | -32000 | Server-side error occurred |

### K.2.2. TOOL DISCOVERY AND INVOCATION

The MCP tool system supports discovery, schema validation, and invocation:

Tool Schema Definition

```
{
  "name": "python_repl",
  "description": "Execute Python code in isolated environment",
  "inputSchema": {
    "type": "object",
    "properties": {
      "code": {"type": "string", "description": "Python code"},
      "timeout": {"type": "number", "default": 30}
    },
    "required": ["code"]
  },
  "returnSchema": {
    "type": "object",
    "properties": {
      "output": {"type": "string"},
      "error": {"type": "string"},
      "execution_time": {"type": "number"}
    }
  }
}
```

The tool discovery workflow follows this sequence:

$$\text{Client} \xrightarrow{\text{tools/list}} \text{Server} \xrightarrow{\{T_1,\ldots,T_n\}} \text{Client} \xrightarrow{\text{tools/call}(T_i,\theta)} \text{Server} \xrightarrow{\text{result}} \text{Client} \tag{75}$$

where $T_i$ represents tool $i$ and $\theta$ are the tool arguments.

### K.2.3. RESOURCE MANAGEMENT

MCP resources represent external data sources accessible via URIs. The resource system supports:

- **Resource Discovery**: `resources/list` returns available resources

- **Resource Reading**: `resources/read` fetches resource content by URI

- **Resource Updates**: Servers can notify clients of resource changes

- **MIME Type Support**: Resources include MIME types for content interpretation

Resource Definition

```
{
  "uri": "file:///data/analysis.csv",
  "name": "Analysis Dataset",
  "description": "Customer analysis data for Q4 2025",
  "mimeType": "text/csv",
  "metadata": {
    "size": 1048576,
    "last_modified": "2025-12-31T23:59:59Z",
    "encoding": "utf-8"
  }
}
```

### K.2.4. CAPABILITIES AND INITIALIZATION

MCP uses a capability negotiation system during initialization. Both client and server declare supported features:

Capability Declaration

```
{
  "tools": true,
  "resources": true,
  "prompts": false,
  "sampling": false,
  "experimental": {
    "streaming": true,
    "batch_operations": false
  }
}
```

The initialization handshake:

$$\text{Client} \xrightarrow{\text{initialize}(\text{version}, C_{\text{client}}, \text{info})} \text{Server}$$
$$\text{Server} \xrightarrow{\text{result}(C_{\text{server}}, \text{info})} \text{Client}$$

(76)

where $C_{\text{client}}$ and $C_{\text{server}}$ are capability sets. The effective capabilities are $C_{\text{eff}} = C_{\text{client}} \cap C_{\text{server}}$.

### K.2.5. TRANSPORT LAYER

EFFGEN implements three MCP client transports (Table 39):

*Table 39.* MCP Transport Choices. EFFGEN ships three transports for MCP: STDIO is the lowest-latency option and the standard choice for local servers (e.g. Claude Desktop integrations), HTTP supports remote request–response servers, and SSE supports server-pushed streams. Actual latency and throughput are dominated by the deployment (local process vs. remote server, network conditions, payload size) rather than the framework, and are best measured per deployment with `effgen loadtest`.

| Transport | Streaming | Typical Use Case | Framework Class |
|---|---|---|---|
| STDIO | No | Local server processes (e.g. Claude Desktop) | StdioTransport |
| HTTP | No | Remote request–response servers | HTTPTransport |
| SSE | Yes | Server-pushed events and progress streams | SSETransport |

### K.3. A2A (Agent-to-Agent) Implementation

Google's A2A protocol emphasizes task lifecycle management with multimodal message support and artifact tracking.

### K.3.1. MESSAGE PART SYSTEM

A2A messages consist of typed parts supporting multiple modalities: $M_{\text{A2A}} = \{p_1, p_2, \ldots, p_n\}$ where $p_i \in$ {text, image, audio, video, file, form, structured}

---

**A2A Multimodal Message**

```json
{
  "id": "msg_abc123",
  "parts": [
    {
      "type": "text",
      "content": "Analyze this image for defects",
      "metadata": {"priority": "high"}
    },
    {
      "type": "image",
      "content": "data:image/png;base64,iVBORw0KGgo...",
      "mimeType": "image/png",
      "metadata": {"resolution": "1920x1080"}
    }
  ],
  "context": {"session_id": "sess_xyz", "user": "analyst_1"},
  "metadata": {"department": "QA"},
  "timestamp": "2025-01-25T10:30:00Z"
}
```

---

### K.3.2. TASK LIFECYCLE MANAGEMENT

A2A tasks progress through six states with atomic transitions: $S_{\text{task}} =$ {CREATED, PENDING, RUNNING, COMPLETED, FAILED, CANCELLED}

State transitions follow a finite state machine: a task moves from CREATED to PENDING to RUNNING and then to either COMPLETED or FAILED, and from any of CREATED, PENDING, or RUNNING it can move to CANCELLED.

---

**A2A Task Structure**

```json
{
  "id": "task_def456",
  "state": "running",
  "instruction": { /* A2A Message */ },
  "artifacts": [
    {
      "id": "art_1",
      "name": "analysis_report",
      "type": "result",
      "content": {"defects": 3, "confidence": 0.94},
      "mimeType": "application/json",
      "created": "2025-01-25T10:35:12Z"
    }
  ],
  "progress": 0.65,
  "metadata": {"capability": "image_analysis"},
  "created": "2025-01-25T10:30:00Z",
  "updated": "2025-01-25T10:35:12Z"
}
```

### K.3.3. ARTIFACT SYSTEM

Artifacts represent task outputs with rich metadata. Each artifact includes:

- **Unique ID**: UUID for artifact tracking

- **Type Classification**: result, intermediate, log, metric, visualization

- **Content**: Arbitrary JSON-serializable data

- **MIME Type**: Content type for binary data

- **Timestamps**: Creation and modification times

- **Metadata**: Custom key-value pairs

Artifacts accumulate during task execution:

$$A_t = A_{t-1} \cup \{a_{\text{new}}\} \quad \text{where} \quad |A_t| \geq |A_{t-1}| \tag{77}$$

Here $a_{\text{new}}$ represents a newly generated artifact at time step $t$ (such as a file, code snippet, or intermediate result), which gets added to the growing set of artifacts from previous steps.

### K.3.4. CONTEXT PASSING

A2A supports hierarchical context passing between agents:

---

**Context Passing Example**

```
{
  "context": {
    "session_id": "sess_xyz",
    "conversation_history": [...],
    "user_preferences": {"language": "en", "detail": "high"},
    "shared_state": {
      "current_step": 3,
      "total_steps": 5,
      "previous_results": [...]
    }
  }
}
```

---

Context is preserved across task chains: $\text{Context}_{i+1} = \text{Context}_i \cup \text{Updates}_i \setminus \text{Removals}_i$

## K.4. ACP (Agent Communication Protocol) Implementation

IBM's ACP provides agent manifests, capability tokens, and OpenTelemetry support.

### K.4.1. AGENT MANIFEST SYSTEM

Agent manifests declare capabilities using JSON Schema:

ACP Agent Manifest

```json
{
  "agentId": "agent_research_001",
  "name": "Research Assistant",
  "version": "2.1.0",
  "description": "Specialized agent for scientific research",
  "capabilities": [
    {
      "name": "literature_search",
      "description": "Search academic databases",
      "version": "1.0.0",
      "inputSchema": {
        "type": "object",
        "properties": {
          "query": {"type": "string"},
          "databases": {"type": "array", "items": {"type": "string"}},
          "max_results": {"type": "integer", "default": 50}
        },
        "required": ["query"]
      },
      "outputSchema": {
        "type": "object",
        "properties": {
          "papers": {"type": "array"},
          "count": {"type": "integer"}
        }
      }
    }
  ],
  "metadata": {"department": "AI Research", "tier": "production"},
  "created": "2025-01-15T00:00:00Z",
  "updated": "2025-01-25T10:00:00Z"
}
```

### K.4.2. REQUEST TYPES

ACP supports three request execution modes (Table 40):

*Table 40.* ACP Request Type Characteristics. Synchronous for fast operations, asynchronous with progress tracking for long tasks, and streaming for real-time updates.

| Type | Blocking | Progress | Use Case |
|------|----------|----------|----------|
| Synchronous | Yes | No | Fast operations ($< 5s$) |
| Asynchronous | No | Yes | Long operations ($> 5s$) |
| Streaming | No | Continuous | Real-time updates |

**ACP Request (Asynchronous)**

```json
{
  "requestId": "req_ghi789",
  "agentId": "agent_research_001",
  "capability": "literature_search",
  "input": {
    "query": "attention mechanisms transformers",
    "databases": ["arxiv", "semantic_scholar"],
    "max_results": 100
  },
  "requestType": "asynchronous",
  "context": {"user_id": "researcher_42"},
  "timestamp": "2025-01-25T11:00:00Z"
}
```

### K.4.3. TASK TRACKING

Asynchronous requests create tracked tasks with progress updates:

**ACP Task Info**

```json
{
  "taskId": "task_jkl012",
  "requestId": "req_ghi789",
  "status": "running",
  "progress": 0.73,
  "created": "2025-01-25T11:00:00Z",
  "updated": "2025-01-25T11:02:15Z"
}
```

Progress updates satisfy monotonicity:

$$\forall t_1 < t_2 : \quad p(t_1) \leq p(t_2) \quad \text{where} \quad p(t) \in [0, 1] \tag{78}$$

### K.4.4. CAPABILITY TOKENS

Fine-grained access control using capability tokens:

**Capability Token**

```json
{
  "tokenId": "tok_mno345",
  "agentId": "agent_research_001",
  "capabilities": ["literature_search", "citation_analysis"],
  "permissions": {
    "literature_search": ["read", "execute"],
    "citation_analysis": ["read"]
  },
  "expires": "2025-02-25T00:00:00Z",
  "metadata": {"issued_to": "user_researcher_42"}
}
```

Token validation:

$$\text{valid}(T, c, t) = (c \in T.\text{capabilities}) \land (t < T.\text{expires}) \tag{79}$$

### K.4.5. ERROR HANDLING WITH SEVERITY LEVELS

ACP provides structured errors with severity classification (Table 41):

---

**ACP Error Response**

```json
{
  "requestId": "req_ghi789",
  "status": "failed",
  "error": {
    "code": "DATABASE_TIMEOUT",
    "message": "ArXiv database connection timeout",
    "severity": "warning",
    "details": {
      "database": "arxiv",
      "timeout_seconds": 30,
      "retry_recommended": true
    },
    "timestamp": "2025-01-25T11:02:45Z"
  }
}
```

---

*Table 41.* ACP Error Severity Levels. Info level for informational messages, Warning for retryable errors, Error for user notification, and Critical for system-level failures.

| Severity | Retryable | Logging | Action |
|---|---|---|---|
| Info | N/A | Debug | None |
| Warning | Yes | Warning | Retry with backoff |
| Error | Maybe | Error | User notification |
| Critical | No | Critical | System alert |

### K.5. Protocol Trade-offs in Practice

The three protocols are designed for different communication patterns rather than as drop-in replacements, and the operational characteristics follow directly from their wire formats and validation requirements. **MCP** uses JSON-RPC 2.0 envelopes and is optimized for compact tool and resource exchanges between a model and a single server; STDIO is the preferred transport for local processes (such as a Claude Desktop server), while HTTP and SSE support remote and streaming use cases. The strict JSON-RPC schema and tool-list validation give predictable behavior at the cost of a small per-message validation step. **A2A** carries multi-part messages with task lifecycle metadata (states, progress, artifacts) and supports HTTP and WebSocket transports together with four pluggable authentication handlers (`Bearer`, `OAuth2`, `APIKey`, custom); messages are larger than MCP because of the lifecycle fields, but validation is lighter. **ACP** adds agent manifests, capability tokens, and streaming callbacks for enterprise deployments where authorization and observability are first-class, which adds a manifest- and token-validation step on each request.

For production operators, the framework itself ships a load-test harness (`effgen.tools.loadgen.LoadGenerator`, exposed as `effgen loadtest`) that measures end-to-end throughput, p50/p95/p99 latency, and error rate against a configurable backend (mock or live provider), and Prometheus latency histograms (Section M) record per-call latency at the agent, model, and tool layers. We therefore characterize protocol behavior qualitatively in this paper and refer practitioners to `effgen loadtest` for workload-specific numbers on their own infrastructure, rather than reporting microbenchmarks that may not generalize.

## K.6. Unified Protocol Interface

EFFGEN provides a unified interface abstracting protocol differences:

**Unified Protocol Usage**

```python
import asyncio
from effgen.tools.protocols import mcp, a2a, acp
from effgen.tools.protocols.mcp.client import (
    MCPServerConfig, TransportType,
)
from effgen.tools.protocols.a2a import (
    AgentCard, EndpointConfig, Capability, CapabilityType,
)

# Use MCP for tool-based agent (transport calls are async)
mcp_config = MCPServerConfig(
    name="tools",
    command="python",
    args=["tools_server.py"],
    transport=TransportType.STDIO,
)
mcp_client = mcp.MCPClient(mcp_config)

async def use_mcp():
    await mcp_client.connect()
    tools = mcp_client.get_tools()  # list[MCPTool]
    return await mcp_client.call_tool(
        "calculator", {"expression": "2+2"},
    )

result = asyncio.run(use_mcp())

# Use A2A for task-based collaboration
agent_card = AgentCard(
    name="image_agent",
    description="Image analysis agent",
    version="1.0",
    capabilities=[Capability(
        name="image_analysis",
        type=CapabilityType.ANALYSIS,
        description="Analyze image content",
        inputSchema={"type": "object",
                     "properties": {"url": {"type": "string"}}},
    )],
    endpoint=EndpointConfig(url="http://agent.example.com"),
)
a2a_client = a2a.A2AClient(agent_card)

# Use ACP for enterprise agent communication
manifest = acp.AgentManifest(
    agentId="my_agent", name="Assistant",
    version="1.0", description="Research assistant",
)
acp_handler = acp.ACPProtocolHandler(manifest)
request = acp_handler.create_request(
    capability="search", input_data={"query": "transformers"},
)
```

The protocol modules expose typed client and handler interfaces. Native tool-calling and structured-output support in v0.2.x complement these protocol transports for model-provider APIs, including OpenAI native tools, Anthropic extended thinking and prompt caching, and Gemini grounding/thinking support.

Each protocol implementation is fully typed using Python dataclasses with validation, ensuring type safety and supporting automatic schema generation for documentation.

## L. GPU Management and Parallelism Strategies

Small language model deployment on accelerators requires careful memory management and parallelism orchestration to maximize hardware utilization while reducing out-of-memory errors. The EFFGEN accelerator management system provides automatic device allocation, memory estimation, real-time monitoring, and GPU parallelism strategies for scaling models beyond single-device capacity. Figure 14 illustrates the decision flow: memory estimation based on model parameters and dtype/quantization, allocation strategy selection (Greedy, Balanced, Optimized, Priority), single-GPU vs multi-GPU routing, and parallelism strategy selection. v0.2.x also adds Apple Silicon/MLX support, lazy model loading, continuous batching, and GGUF/AWQ/GPTQ-related loading paths in the model layer.

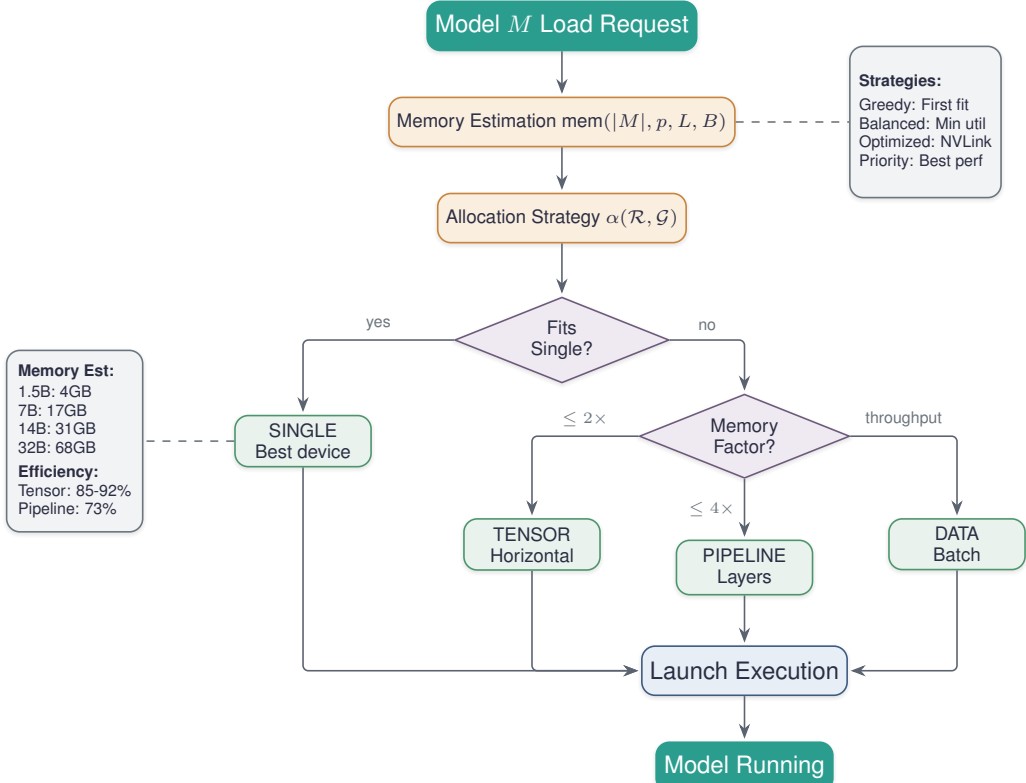

*Figure 14.* GPU management and parallelism strategy selection. The system estimates memory requirements using model size, precision, sequence length, and batch size, then selects an allocation strategy (Greedy, Balanced, Optimized with NVLink, or Priority). If the model fits on a single GPU, it allocates the best device. Otherwise, it routes to one of three parallelism strategies based on memory factor: Tensor parallelism (horizontal weight splitting, 85-92% efficiency) for 2-8x memory, Pipeline parallelism (layer-wise stages, 73% efficiency) for 4x memory with divisible layers, or Data parallelism (batch splitting, linear throughput) when single-device memory suffices but throughput is needed. The system monitors utilization throughout execution.

### L.1. GPU Resource Model

We model a GPU cluster as $\mathcal{G} = \{g_1, \ldots, g_n\}$ where each device $g_i$ has properties $(M_i, U_i, T_i, P_i, \tau_i)$:

- $M_i$: total memory capacity in bytes

- $U_i$: current memory utilization in bytes

- $T_i$: current GPU utilization % $[0, 100]$

- $P_i$: power consumption in watts

- $\tau_i$: temperature in celsius

The available memory is $A_i = M_i - U_i$. For allocation requests, the allocator checks whether $\text{required}(r) \leq A_i$ for some device $g_i$ or across multiple devices for multi-GPU strategies; allocation can still fail if no suitable device is available.

**Memory Estimation Function.** Given model parameters $|M|$ (count), bytes per parameter $b$, and overhead factor $\eta$, the model memory requirement estimate is:

$$\text{mem}_{\text{model}}(|M|, b, \eta) = |M| \cdot b \cdot \eta \tag{80}$$

Batch activation memory is estimated separately from batch size and sequence length. Quantization changes $b$; the overhead factor provides a safety margin for runtime allocations.

## L.2. Allocation Strategies

The allocation function $\alpha : \mathcal{R} \times \mathcal{G} \to \mathcal{A}$ maps resource requests $\mathcal{R}$ to device assignments $\mathcal{A} \subseteq \mathcal{G}$ according to one of four strategies.

**Greedy Strategy.** Allocate to the first device with sufficient memory:

$$\alpha_{\text{GREEDY}}(r, \mathcal{G}) = \arg \min_{g_i \in \mathcal{G}} \{i : A_i \geq \text{mem}(r)\} \tag{81}$$

This strategy is simple and fast ($O(n)$ for $n$ devices) but can lead to unbalanced utilization where early devices are overloaded while later devices remain idle.

**Balanced Strategy.** Allocate to the device with lowest utilization:

$$\alpha_{\text{BALANCED}}(r, \mathcal{G}) = \arg \min_{g_i \in \mathcal{G}} \{U_i : A_i \geq \text{mem}(r)\} \tag{82}$$

This achieves better load distribution than first-fit allocation in heterogeneous device pools.

**Optimized Strategy.** For multi-GPU allocations (tensor or pipeline parallelism), the implementation selects contiguous device IDs with low average utilization:

$$\alpha_{\text{OPTIMIZE}}(r, \mathcal{G}, k) = \arg \min_{\{g_i, g_{i+1}, \dots, g_{i+k-1}\}} \text{avg\_utilization}(g_i, \dots, g_{i+k-1}) \tag{83}$$

where $k$ is the number of required devices. This is a simple topology-agnostic heuristic; it does not inspect NVLink or PCIe topology.

**Priority Strategy.** Allocate devices by available memory and utilization:

$$\alpha_{\text{PRIORITY}}(r, \mathcal{G}) = \arg \max_{g_i \in \mathcal{G}} \{A_i - U_i : A_i \geq \text{mem}(r)\} \tag{84}$$

This favors devices with enough free memory and lower current utilization.

## L.3. Parallelism Strategies

When a model exceeds single-GPU capacity, EFFGEN employs one of three parallelism strategies. Let $M$ be a model with $|M|$ parameters and $d$ layers.

**Tensor Parallelism.** Partition each layer horizontally across $k$ devices. For layer $\ell$ with weight matrix $W_\ell \in \mathbb{R}^{m \times n}$, split column-wise:

$$W_\ell = [W_\ell^{(1)} | W_\ell^{(2)} | \cdots | W_\ell^{(k)}] \quad \text{where } W_\ell^{(i)} \in \mathbb{R}^{m \times n/k} \tag{85}$$

For forward pass with input $x \in \mathbb{R}^m$, each device $i$ computes $y^{(i)} = xW_\ell^{(i)}$, then results are concatenated: $y = [y^{(1)}|\cdots|y^{(k)}]$. Communication overhead is one all-reduce per layer.

Memory reduction: $\frac{\text{mem}(|M|,p,L,B)}{k}$ (nearly linear scaling). Communication cost: $O(L \cdot B \cdot n)$ per layer for all-reduce operation. Efficiency depends on inter-GPU bandwidth; on NVLink-connected GPUs, tensor parallelism achieves 85-92% efficiency for $k \leq 8$.

**Pipeline Parallelism.** Partition layers into $k$ stages, assigning consecutive layers to different devices:

$$\text{Stage}_i = \{\ell_{(i-1)\cdot d/k+1}, \ldots, \ell_{i\cdot d/k}\} \quad \text{for } i \in [k] \tag{86}$$

For batch $\mathcal{B} = \{x_1, \ldots, x_B\}$, split into $m$ micro-batches $\{\mathcal{B}_1, \ldots, \mathcal{B}_m\}$ where $|\mathcal{B}_j| = B/m$. Execution proceeds as:

1: **for** $j = 1$ to $m$ **do**
2:    Device 1 processes $\mathcal{B}_j$ through Stage 1, sends output to Device 2
3:    Device 2 processes $\mathcal{B}_j$ through Stage 2, sends output to Device 3
4:    . . . (pipeline fill)
5: **end for**
6: . . . (pipeline drain)

Memory reduction: approximately $\frac{\text{mem}(|M|,p,L,B)}{k}$ since each device holds only $\approx d/k$ layers. Communication cost: $O(m \cdot d \cdot L \cdot B/m) = O(d \cdot L \cdot B)$ total. Pipeline efficiency is:

$$\eta_{\text{pipeline}} = \frac{m}{m + k - 1} \tag{87}$$

where $m$ is micro-batch count and $k$ is device count. For $m = 8$ micro-batches and $k = 4$ devices, $\eta = 0.73$ (73% efficiency).

**Data Parallelism.** Replicate the entire model on $k$ devices and split the batch:

$$\mathcal{B} = \mathcal{B}_1 \cup \mathcal{B}_2 \cup \cdots \cup \mathcal{B}_k \quad \text{where } |\mathcal{B}_i| = B/k \tag{88}$$

Each device processes its partition independently. For inference, no synchronization is needed. For training, gradients are averaged across devices after each step.

Memory impact: each device holds full model, so no memory reduction. However, throughput increases linearly: $\text{throughput} = k \cdot \text{throughput}_{\text{single}}$. This strategy is preferred when the model fits on a single device but throughput is insufficient.

**Strategy Selection.** The system automatically selects the appropriate strategy based on model size and available resources. Algorithm 12 presents the decision logic.

### L.4. Real-Time GPU Monitoring

The `GPUMonitor` class provides real-time tracking of device metrics using NVIDIA System Management Interface (nvidia-smi). Monitoring occurs in a background thread with configurable polling interval (default 1 second).

**Collected Metrics.** For each device $g_i$, the monitor tracks:

- Memory utilization: $(U_i, A_i, M_i)$

- Compute utilization: $T_i \in [0, 100]\%$

- Temperature: $\tau_i$ in celsius

- Power consumption: $P_i$ in watts

- Fan speed and active process information when available

**Algorithm 12** Parallelism Strategy Selection

---

**Require:** Model $M$, devices $\mathcal{G}$, target batch size $B$
**Ensure:** Selected strategy $s$ and device allocation $\mathcal{A}$

1: mem_req $\leftarrow$ mem($|M|, p, L, B$)
2: max_mem $\leftarrow \max_{g \in \mathcal{G}} A_g$
3: **if** mem_req $\leq$ max_mem **then**
4:     **if** require high throughput **then**
5:         **return** DATA_PARALLEL, select $k$ devices
6:     **else**
7:         **return** SINGLE_DEVICE, select best device
8:     **end if**
9: **else if** mem_req $\leq 2 \cdot$ max_mem **then**
10:     **return** TENSOR_PARALLEL, select 2 adjacent devices
11: **else if** mem_req $\leq 4 \cdot$ max_mem **then**
12:     **if** layers divisible by 4 **then**
13:         **return** PIPELINE_PARALLEL, select 4 devices
14:     **else**
15:         **return** TENSOR_PARALLEL, select 4 devices
16:     **end if**
17: **else**
18:     $k \leftarrow \lceil$mem_req/max_mem$\rceil$
19:     **if** $k \leq 8$ **then**
20:         **return** TENSOR_PARALLEL, select $k$ adjacent devices
21:     **else**
22:         **return** HYBRID (pipeline + tensor), compute best split
23:     **end if**
24: **end if**

---

**Alerting Thresholds.** The monitor triggers alerts when metrics exceed safety thresholds:

$$\text{CRITICAL} : (U_i/M_i > 0.95) \vee (\tau_i > 90°\text{C}) \vee (P_i > 0.95 \cdot P_{\max}) \tag{89}$$

$$\text{WARNING} : (U_i/M_i > 0.80) \vee (\tau_i > 80°\text{C}) \vee (P_i > 0.90 \cdot P_{\max}) \tag{90}$$

These thresholds are the implementation defaults. The 95% memory threshold leaves room for CUDA runtime overhead and memory fragmentation.

Critical alerts trigger automatic mitigation: reduce batch size, throttle inference rate, or migrate workloads to cooler devices.

**Historical Tracking.** Metrics are stored in a circular buffer capped at 1000 samples. This supports trend analysis and anomaly detection. For example, steadily increasing temperature suggests cooling system degradation requiring maintenance.

### L.5. Integration with Model Loaders

The GPU management system integrates with model loaders (vLLM, Transformers). For vLLM, the allocator sets the CUDA_VISIBLE_DEVICES environment variable before engine initialization:

```
allocation = allocator.allocate(model_config, strategy=TENSOR_PARALLEL)
os.environ["CUDA_VISIBLE_DEVICES"] = ",".join(map(str, allocation.devices))
engine = LLM(model=model_name, tensor_parallel_size=len(allocation.devices))
```

For multi-GPU strategies, vLLM handles internal communication and synchronization. The allocator checks device assignment and memory availability before initialization.

**Graceful Degradation.** If GPU allocation fails (insufficient memory, no available devices), the system falls back gracefully:

1: Attempt requested configuration (e.g., FP16, batch=8)
2: **if** allocation fails **then**
3:      Reduce batch size: try batch=4, then batch=1
4: **end if**
5: **if** still fails **then**
6:      Switch to quantization: try INT8
7: **end if**
8: **if** still fails **then**
9:      Switch to CPU execution with warning
10: **end if**

These fallbacks reduce hard failures when alternatives are available; if all allocation attempts fail, the allocator reports failure for the caller to handle.

## M. Execution Tracking and Observability

Transparent execution visibility is critical for debugging agent behavior and optimizing performance. The EFFGEN execution tracker implements an event-based tracking system capturing 18 event types across the agent lifecycle. Figure 15 illustrates the architecture: events flow from agent execution into five categories (Task Lifecycle, Routing/Decomposition, Sub-Agent Execution, Tool Execution, Memory/Special), then metrics computation extracts performance indicators for logs, analysis, and visualization. This section specifies the tracking architecture, event taxonomy, performance analysis capabilities, and visualization exports.

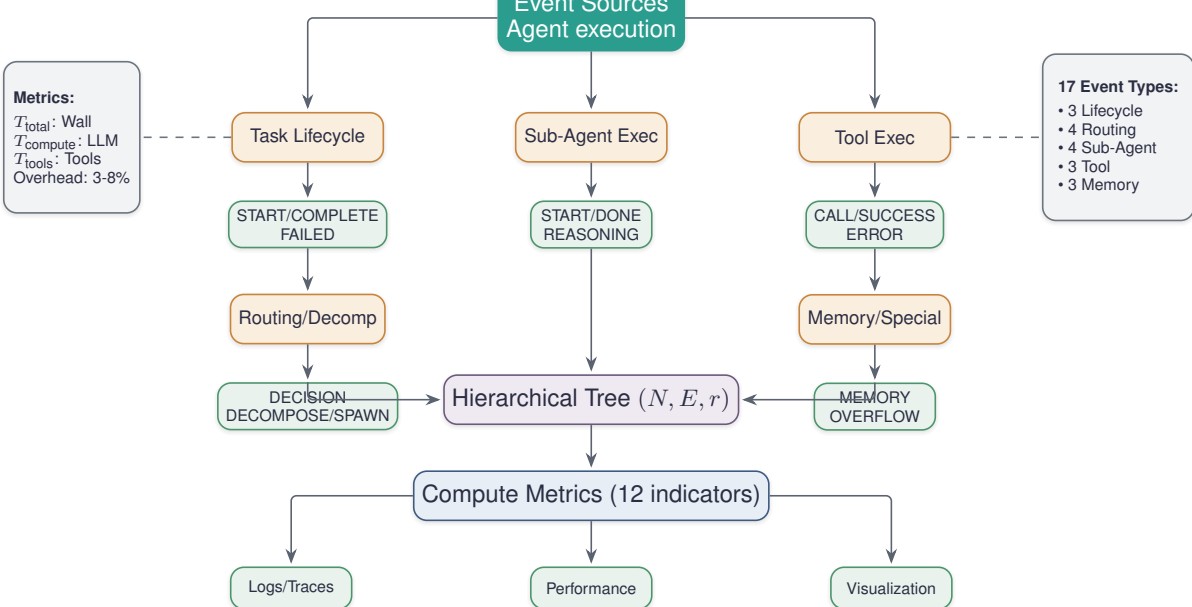

*Figure 15.* Execution tracking and observability system with event taxonomy. Events from agent execution flow into five categories arranged in three branches: Task Lifecycle and Routing/Decomposition (left), Sub-Agent Execution (center), and Tool Execution and Memory/Special (right). The system captures 18 distinct event types forming a hierarchical tree structure $(N, E, r)$. Metrics computation extracts 12 performance indicators including wall time, compute time, tool time, and 3-8% framework overhead. Outputs support logs, performance analysis, and visualization for debugging.

### M.1. Event-Based Tracking Architecture

The execution tracker maintains an event log and parent-child relationships where available. Each execution event $e_v = (t, \tau, m, p)$ consists of:

• $t \in \mathcal{E}_{\text{types}}$: event type from taxonomy

- $\tau$: event timestamp

- $m$: event-specific metadata (parameters, results, errors)

- $p$: optional parent event identifier

**Event Relationships.** The tracker records parent-child relationships when callers supply parent identifiers:

(i) **Temporal Ordering**: Events are appended with timestamps.

(ii) **Parent Links**: Child events can reference a parent event.

(iii) **Unique Identifiers**: Each execution event has a unique identifier.

These relationships support trace reconstruction and performance analysis.

## M.2. Event Type Taxonomy

The event taxonomy $\mathcal{E}_{\text{types}}$ contains 18 event types organized into five categories. Table 42 provides the specification.

*Table 42.* Execution Event Taxonomy. The framework tracks 18 event types organized into five categories covering the agent lifecycle from task initiation through result synthesis. Each event type has metadata fields capturing relevant execution details. These events support performance analysis and bottleneck detection. Events are referenced throughout Section M for performance metric computation.

| Category | Event Type | Description | Key Metadata Fields |
|---|---|---|---|
| Task Lifecycle | TASK_START | Agent begins processing task | task_description, agent_id, complexity_score |
| | TASK_COMPLETE | Agent completes task successfully | final_result, token_count, execution_time |
| | TASK_FAILED | Agent fails to complete task | error_message, failure_reason, partial_results |
| Routing & Decomp | ROUTING_DECISION | Router determines execution strategy | complexity_score, strategy, reasoning, confidence |
| | TASK_DECOMPOSITION | Task decomposed into subtasks | num_subtasks, subtask_descriptions, dependencies |
| | SUB_AGENT_SPAWN | Sub-agent created for subtask | specialization, tools, prompt_template |
| | RESULT_SYNTHESIS | Sub-agent results combined | num_results, synthesis_strategy, final_output |
| Sub-Agent Exec | SUB_AGENT_START | Sub-agent begins execution | agent_id, specialization, assigned_task |
| | SUB_AGENT_COMPLETE | Sub-agent completes successfully | result, tokens_used, tools_called |
| | SUB_AGENT_FAILED | Sub-agent fails | error, attempted_retries, fallback_strategy |
| | REASONING_STEP | Agent performs reasoning iteration | thought, planned_action, iteration_count |
| Tool Execution | TOOL_CALL_START | Agent invokes tool | tool_name, parameters, expected_duration |
| | TOOL_CALL_COMPLETE | Tool execution succeeds | result, actual_duration, resource_usage |
| | TOOL_CALL_FAILED | Tool execution fails | error, error_type, retryable |
| Memory & Special | MEMORY_OPERATION | Memory read/write operation | operation_type, memory_tier, entry_count |
| | VALIDATION_FAILED | Input/output validation fails | validation_type, errors, rejected_value |
| | SUMMARIZATION_TRIGGER | Context summarization triggered | original_tokens, summarized_tokens, reduction_ratio |
| | CONTEXT_OVERFLOW | Context window exceeded | current_tokens, max_tokens, mitigation_action |

**Event Relationships.** Certain events have natural parent-child relationships forming execution patterns:

- TASK_START → ROUTING_DECISION → {TASK_DECOMPOSITION | REASONING_STEP}

- TASK_DECOMPOSITION → SUB_AGENT_SPAWN* (one spawn per subtask)

- SUB_AGENT_START → REASONING_STEP* → TOOL_CALL_START* → SUB_AGENT_COMPLETE

- REASONING_STEP → TOOL_CALL_START → TOOL_CALL_COMPLETE

These patterns support pattern-based anomaly detection: deviation from expected event sequences indicates execution errors or unexpected agent behavior.

## M.3. Performance Metrics Computation

The tracker computes performance metrics from the event log. Let $T$ denote total elapsed time, $N_a$ the number of agents (1 for single, $\geq 2$ for multi-agent), and $N_t$ the number of tool calls.

**Timing Metrics.** The implementation reports elapsed time plus aggregate agent and tool durations where those durations are recorded:

$$T_{\text{total}} = \tau_{\text{last}} - \tau_{\text{first}} \quad \text{(wall-clock elapsed time)} \tag{91}$$

$$T_{\text{agents}} = \sum_{e:t_e \in \text{agent events}} \text{duration}(e) \quad \text{(when available)} \tag{92}$$

$$T_{\text{tools}} = \sum_{e:t_e \in \text{tool events}} \text{duration}(e) \quad \text{(when available)} \tag{93}$$

Routing decisions, task decomposition, result synthesis, and inter-agent communication are visible as events, but the tracker does not derive a separate framework-overhead term from the event log.

**Throughput Metrics.** Define:

$$\text{throughput}_{\text{tasks}} = \frac{N_{\text{completed\_subtasks}}}{T_{\text{total}}} \quad \text{(subtasks per second)} \tag{94}$$

$$\text{throughput}_{\text{events}} = \frac{|N|}{T_{\text{total}}} \quad \text{(events per second)} \tag{95}$$

$$\text{throughput}_{\text{tools}} = \frac{N_t}{T_{\text{total}}} \quad \text{(tool calls per second)} \tag{96}$$

Throughput is reported from observed event counts and elapsed time.

**Resource Metrics.** Define:

$$\text{peak\_concurrent\_agents} = \max_t |\{e \in N : t_e \in \{\text{SUB\_AGENT\_START}\} \wedge \tau_s^e \leq t < \tau_e^e\}| \tag{97}$$

$$\text{total\_tool\_calls} = |\{e \in N : t_e = \text{TOOL\_CALL\_START}\}| \tag{98}$$

$$\text{total\_tokens} = \sum_{e \in N} m_e[\text{tokens}] \quad \text{where defined} \tag{99}$$

These metrics inform resource provisioning decisions: high peak concurrency requires more GPU memory, many tool calls benefit from tool result caching.

**Efficiency Metrics.** For multi-agent execution with parallel subtasks:

$$\text{parallel\_efficiency} = \frac{\sum_{i=1}^{N_a} T_i}{N_a \cdot T_{\text{total}}} \tag{100}$$

where $T_i$ is execution time of agent $i$. Perfect efficiency ($\eta = 1.0$) means all agents busy entire time. Real systems achieve $\eta \in [0.6, 0.9]$ with inefficiency from load imbalance and coordination overhead.

**Quality Metrics.**  Define:

$$\text{success\_rate} = \frac{|\{e \in N : s_e = \text{COMPLETED}\}|}{|N|} \tag{101}$$

$$\text{failure\_rate} = \frac{|\{e \in N : s_e = \text{FAILED}\}|}{|N|} \tag{102}$$

$$\text{retry\_rate} = \frac{|\{e \in N : m_e[\text{retry\_count}] > 0\}|}{|N|} \tag{103}$$

Success, failure, and retry rates are reported as descriptive metrics for comparing runs.

## M.4. Bottleneck Identification

The tracker identifies performance bottlenecks through automated analysis. Algorithm 13 presents the detection procedure.

---

**Algorithm 13** Bottleneck Detection in Execution Tree

---

**Require:** Execution tree $\mathcal{T} = (N, E, r)$
**Ensure:** List of bottlenecks with severity
 1: bottlenecks $\leftarrow$ []
 2: Compute average agent duration
 3: **for** each agent duration summary $a$ **do**
 4:    **if** $T_a > 1.5 \times \text{avg\_agent\_time}$ **then**
 5:       Add $(a, \text{SLOW\_AGENT}, T_a)$ to bottlenecks
 6:    **end if**
 7: **end for**
 8: **if** failed event count $> 2$ **then**
 9:    Add $(\text{failures}, \text{HIGH\_FAILURES}, \text{count})$ to bottlenecks
10: **end if**
11: **return** sorted(bottlenecks, key=severity, reverse=True)

---

Bottleneck detection identifies slow agents relative to the run average and runs with repeated failed events. This information guides optimization: slow agents can be inspected, and high-failure runs may need better prompts, tools, or model sizes.

## M.5. Critical Path Analysis

For parallel execution, the critical path determines the minimum possible execution time. The critical path $\pi_{\text{crit}} = (v_1, v_2, \ldots, v_k)$ is the longest path from root to leaf in the execution tree:

$$\pi_{\text{crit}} = \arg \max_{\pi : r \to \text{leaf}} \sum_{v \in \pi} (\tau_e^v - \tau_s^v) \tag{104}$$

The critical path length $L_{\text{crit}} = \sum_{v \in \pi_{\text{crit}}} (\tau_e^v - \tau_s^v)$ estimates the minimum execution time implied by recorded dependencies. The parallel efficiency is bounded by:

$$\eta \leq \frac{L_{\text{crit}}}{T_{\text{total}}} \tag{105}$$

Events on the critical path are prioritized for optimization: reducing their duration directly reduces total execution time, while optimizing off-critical-path events has no effect on wall-clock time (though it may reduce resource usage).

## M.6. Export Formats and Visualization

The tracker supports four export formats optimized for different use cases.

**JSON Export.**  Complete hierarchical execution tree:

**JSON Export Format**

```json
{
  "task_id": "uuid",
  "start_time": "2026-01-25T10:30:00Z",
  "end_time": "2026-01-25T10:30:15Z",
  "status": "COMPLETED",
  "events": [
    {
      "id": "evt-001",
      "type": "TASK_START",
      "timestamp": "2026-01-25T10:30:00.123Z",
      "metadata": {...},
      "children": [...]
    },
    ...
  ],
  "metrics": {
    "total_time_ms": 15230,
    "compute_time_ms": 12450,
    "tool_time_ms": 2100,
    "overhead_ms": 680,
    ...
  }
}
```

JSON export is machine-readable, supporting programmatic analysis and integration with monitoring systems.

**CSV Export.**  Flattened event list for spreadsheet analysis:

**CSV Export Format**

```
event_id,type,start_time,end_time,duration_ms,status,parent_id,...
evt-001,TASK_START,1706178600.123,1706178615.353,15230,COMPLETED,null,...
evt-002,ROUTING_DECISION,1706178600.234,1706178600.456,222,COMPLETED,evt-001,...
...
```

CSV format supports analysis in Excel, pandas, or database tools. Useful for statistical analysis across multiple runs.

**HTML Export.**  Interactive visualization with collapsible tree structure, timeline view, and metric dashboard.

The HTML export includes:

- **Tree View**: Hierarchical display with expand/collapse for each node, color-coded by status (green=completed, red=failed, yellow=running)

- **Timeline View**: Gantt-chart visualization showing temporal relationships, concurrent executions, and critical path highlighted

- **Metrics Dashboard**: Summary statistics (total time, success rate, bottlenecks) with charts

- **Event Details**: Click on event to see full metadata, parameters, results

**Human-Readable Export.**  JSON, CSV, and HTML are the implemented export formats. A Markdown-like report can be generated from the JSON export by downstream tooling:

Markdown Export Format

```
# Execution Summary

**Task**: Analyze Q3 sales data and generate report
**Status**: COMPLETED
**Duration**: 15.23 seconds

## Timeline

1. [0.00s] TASK_START
2. [0.23s] ROUTING_DECISION (strategy: PARALLEL)
3. [0.45s] TASK_DECOMPOSITION (3 subtasks)
4. [0.67s] SUB_AGENT_SPAWN (agent-1: data analysis)
   ...

## Performance Metrics

- Total Time: 15.23s
- Compute Time: 12.45s (81.7%)
- Tool Time: 2.10s (13.8%)
- Overhead: 0.68s (4.5%)
...
```

The HTML export is intended for interactive inspection, while JSON and CSV are intended for downstream analysis.

## M.7. Live Monitoring and Progress Tracking

For long-running tasks, the tracker provides live progress updates. The progress estimation function combines completed and in-progress work:

$$\text{progress}(t) = \frac{\sum_{e:s_e=\text{COMPLETED}} w_e + \sum_{e:s_e=\text{RUNNING}} 0.5 \cdot w_e}{\sum_{e \in N} w_e} \tag{106}$$

where $w_e$ is the estimated work for event $e$ (for subtasks, $w_e = 1$; for tool calls, $w_e = 0.1$). The factor 0.5 for running events assumes half-completion on average.

**ETA Estimation.**  Estimated time to completion uses velocity-based projection:

$$\text{ETA}(t) = t + \frac{1 - \text{progress}(t)}{\text{velocity}(t)} \quad \text{where velocity}(t) = \frac{\text{progress}(t)}{t - t_{\text{start}}} \tag{107}$$

For tasks with $<10\%$ progress, ETA is unreliable and marked as unknown; later estimates remain heuristic.

## M.8. Integration with External Monitoring

The tracker integrates with external monitoring systems through tracing and metrics hooks. v0.2.x also adds debug tracing support, including raw prompt/response capture through the debugging utilities.

**OpenTelemetry Export.**  Events are exported as OpenTelemetry spans with parent-child relationships preserved. This supports integration with distributed tracing systems (Jaeger, Zipkin) for multi-service observability.

**Prometheus Metrics.**  Key metrics are exposed as Prometheus gauges and counters:

Prometheus Metrics Export

```
effgen_task_duration_seconds{status="completed"}
effgen_tool_calls_total{tool_name="calculator"}
effgen_agent_errors_total{error_type="timeout"}
...
```

This supports alerting based on metric thresholds and long-term performance trending.

## N. Model Loading and Backend Selection

**Model Loading and Backend Selection.** EFFGEN loads models through a single abstraction so that an agent definition is independent of where the model runs. The same interface covers local inference engines and hosted providers, which lets an application switch backends, or fail over between them, without code changes.

EFFGEN supports both local and hosted inference backends behind this interface (Table 43). Local backends include vLLM (Kwon et al., 2023b;a) (PagedAttention, continuous batching, tensor parallelism), HuggingFace Transformers (broad model compatibility, quantization), GGUF (CPU and mixed-CPU/GPU inference), and MLX / MLX-VLM (native Metal acceleration for Apple Silicon, including vision–language models). Hosted backends include OpenAI, Anthropic, Gemini, Groq, Cerebras, Together AI, Fireworks, Replicate, and HuggingFace Inference. A unified `ProviderRegistry` resolves a backend by provider name, an `effgen doctor` command checks credentials, and the `ModelRouter` layer adds transparent failover with cost-based, latency-based, and first-available routing policies. Cross-process rate-limit coordination uses SQLite, and a persistent cost tracker exposes a CLI dashboard (`effgen cost`). Quantization support includes 8-bit, GPTQ (Frantar et al., 2023), AWQ, and 4-bit weight quantization via `bitsandbytes`.

Backend selection can be automatic (the framework chooses a local engine by model type and available hardware, falling back to Transformers) or pinned in configuration. The per-backend configuration options, the automatic-selection algorithm, the quantization trade-off table, and the loading and multi-provider examples are in `effgen/models/` at https://github.com/ctrl-gaurav/effGen. Relative backend throughput depends on model size, sequence length, and batch size, and is best measured per deployment with `effgen loadtest` (Section P).

## O. Configuration System Architecture

**Configuration System Architecture.** EFFGEN configures agents, models, tools, and prompts through YAML or JSON files, each section validated against its own schema. Separating the configuration into agent, model, tool, and prompt sections keeps the schemas small and lets a deployment override one part (for example, swapping the model backend) without touching the rest. Validation runs with type checking before an agent is constructed, so misconfigured fields are reported up front rather than failing mid-run.

Four capabilities make the system practical for real deployments, and these are the ones referenced from the main paper. Environment-variable substitution keeps secrets such as API keys out of the configuration files. Layered configuration lets a base file hold shared settings while environment-specific files override only what differs between development, staging, and production. Hot reloading applies configuration changes at runtime without restarting a long-running agent. The same loader accepts both YAML and JSON, so configurations can be authored by hand or generated programmatically.

The full schemas live in `effgen/config/` and example configuration files in `configs/` at https://github.com/ctrl-gaurav/effGen.

## P. Command-Line Interface

**Command-Line Interface.** EFFGEN exposes a single `effgen` command organized into functional groups: single-query and interactive use (`run`, `chat`, `resume`), configuration (`config`, `presets`, `create-plugin`), model and tool inspection (`models`, `tools`), session management (`sessions`), serving the OpenAI-compatible HTTP server (`serve`), evaluation and comparison (`eval`, `compare`, `batch`, `workflow`), diagnostics (`doctor`, `debug`, `cost`), and load testing (`loadtest`). The command surface mirrors the library: anything that can be done programmatically through the

*Table 43.* Backend overview. The table groups backends by deployment mode and lists the strongest use case. EFFGEN exposes all backends through a common interface, so applications can switch backends or fail over without code changes.

| Backend | Mode | Best use case | Key features | Notes |
|---|---|---|---|---|
| **vLLM** | Local (GPU) | Production SLMs, high throughput | PagedAttention, continuous batching, tensor parallelism | CUDA only |
| **Transformers** | Local (CPU/GPU) | Development, broad compatibility | HuggingFace models, 8-bit/4-bit quantization, Flash Attention | Default local fallback |
| **GGUF** | Local (CPU/GPU) | CPU and mixed-precision inference | llama.cpp-compatible weights, low memory footprint | Throughput depends on hardware |
| **MLX / MLX-VLM** | Local (Apple) | Apple Silicon text and vision models | Native Metal acceleration, vision–language support | macOS only |
| **OpenAI** | Hosted | GPT-5 / o-series | Tool use, prompt caching, structured outputs v2, native tools | API key required |
| **Anthropic** | Hosted | Claude 4.x family | Extended thinking, `cache_control`, streaming, native tools | API key required |
| **Gemini** | Hosted | Multimodal, long context | `thinking_budget`, Google Search grounding, Files API | API key required |
| **Groq** | Hosted | Low-latency serving | Sub-second responses, native tool calling | API key required |
| **Cerebras** | Hosted | Free-tier high-throughput serving | Streaming, native tool calling, cost tracking | Free tier available |
| **Together AI** | Hosted | Open-weight models on a managed runtime | Broad open-weight model catalog | API key required |
| **Fireworks** | Hosted | Production open-weight serving | Optimized open-weight inference | API key required |
| **Replicate** | Hosted | Long-running and community models | Versioned model endpoints | API key required |
| **HF Inference** | Hosted | HuggingFace Hub models | Direct access to Hub model registry | API key required |

Python API has a corresponding subcommand, so the CLI doubles as a way to script evaluations and as an entry point for users who do not write Python.

The `effgen loadtest` subcommand is referenced throughout this appendix and warrants a specific note. It drives a configurable backend (a mock backend or a live provider) and reports end-to-end throughput together with p50/p95/p99 latency for both streaming and non-streaming modes, plus the error rate under a chosen concurrency level. Because tool, protocol, and backend latencies are dominated by the deployment rather than the framework, we use `effgen loadtest` as the recommended way to obtain workload-specific numbers on a given setup instead of quoting figures that may not generalize.

Running `effgen -help` lists every subcommand and its options. The CLI is implemented in `effgen/__main__.py` at https://github.com/ctrl-gaurav/effGen.

## Q. Error Handling and Recovery Mechanisms

Reliable error handling is essential for agent deployments where failures can occur at multiple levels: model generation errors, tool execution failures, network timeouts, resource exhaustion, and invalid outputs. Techniques like Reflexion (Shinn et al., 2023) have demonstrated the value of verbal reinforcement learning for agent recovery and improvement. The EFFGEN framework implements error handling with retry policies, graceful degradation, and detailed error reporting. This section presents the error taxonomy, recovery strategies, retry logic, and failure analysis.

## Q.1. Error Taxonomy

EffGen categorizes errors by source, severity, recoverability, and required intervention. Table 44 presents the main error classes with recovery strategies, including the newer provider and router errors used by the model-routing layer.

*Table 44.* Error Classification Taxonomy with Recovery Strategies. Transient errors are automatically retried, persistent errors require intervention, critical errors halt execution immediately.

| Error Class | Examples | Recovery Strategy | Severity | Typical Cause |
|---|---|---|---|---|
| **ModelError** | Generation timeout, OOM, model crashed | Retry with backoff, reduce batch size | High | Resource limits |
| **ToolError** | Tool execution failed, invalid output, timeout | Retry, skip tool, use fallback | Medium | External dependency |
| **ValidationError** | Schema mismatch, type error, constraint violation | Retry with correction, abort | Medium | Invalid data |
| **ProviderTransientError** | API timeout, connection failed, rate limit | Retry with exponential backoff | Low | External service |
| **ProviderAuthError** | Missing API key, invalid credential | Fail fast, user intervention | Critical | Authentication |
| **BudgetExceededError** | Cost or token budget exceeded | Stop or route to cheaper model | High | Budget policy |
| **AllCandidatesExhaustedError** | Router failover candidates failed | Surface aggregate error | High | Provider/model failure |
| **MemoryError** | Context overflow, CUDA OOM, disk full | Summarize, offload, reduce batch | High | Resource exhaustion |
| **ConfigError** | Invalid config, missing API key, wrong backend | Fail fast, user intervention | Critical | Misconfiguration |

## Q.2. Retry Logic with Exponential Backoff

Transient errors (network failures, temporary resource unavailability) are automatically retried with exponential backoff to avoid overwhelming services.

---

**Algorithm 14** Retry with Exponential Backoff

---

**Require:** Function $f$, Arguments args, Max retries $n_{max}$, Base delay $d_{base}$
**Ensure:** Result $r$ or error
1: attempt $\leftarrow 0$
2: **while** attempt $< n_{max}$ **do**
3:    **try**:
4:       $r \leftarrow f(\text{args})$
5:
6:    **return** $r$ {Success}
7:    **except** RecoverableError as $e$:
8:       attempt $\leftarrow$ attempt $+ 1$
9:       Log("Attempt attempt failed: $e$")
10:      **if** attempt $< n_{max}$:
11:        delay $\leftarrow d_{base} \cdot 2^{\text{attempt}} \cdot (1 + \text{jitter})$ {Exponential + jitter}
12:        Sleep(delay)
13:      **else**:
14:
15:    **return** Error("Max retries exceeded: $e$")
16:    **except** UnrecoverableError as $e$:
17:
18:    **return** Error("Unrecoverable: $e$") {Fail immediately}
19: **end while**

---

**Default Retry Configuration.** Table 45 specifies the default retry parameters used for each error type.

*Table 45.* Default Retry Parameters by Error Type. The model router retry policy defaults to three retries, a base delay of 1.0s, jitter of 0.5, and a 30s backoff cap. Tool-specific code can override these defaults where needed.

| Error Type | Max Retries | Base Delay (s) | Max Delay (s) | Jitter |
|---|---|---|---|---|
| Provider transient | 3 | 1.0 | 30 | 0.5 |
| Rate limit | 3 | 1.0 | 30 | 0.5 |
| Tool execution | 3 | tool-specific | tool-specific | tool-specific |
| Memory operation | 3 | 1.0 | 30 | 0.5 |

## Q.3. Graceful Degradation

When errors cannot be recovered through retries, EFFGEN implements graceful degradation strategies.

**Strategy 1: Tool Fallback.** If primary tool fails, attempt fallback tools with similar capabilities.

**Tool Fallback Example**

```
from effgen import Agent
from effgen.core.agent import AgentConfig
from effgen.tools.builtin import WebSearch

agent = Agent(config=AgentConfig(
    name="fallback_search",
    model="Qwen/Qwen2.5-7B-Instruct",
    tools=[WebSearch()]
))
response = agent.run("Search for recent AI developments")
# Retry and fallback behavior is handled by the agent/router layer
```

**Strategy 2: Partial Results.** Return partial results when some sub-agents fail in multi-agent execution.

$$r_{\text{partial}} = \text{Synthesize}(\{r_i : \mathcal{A}_{s_i}.\text{state} = \text{COMPLETED}\}) \cup \{\text{errors from failed agents}\} \tag{108}$$

The synthesizer notes which subtasks failed and provides partial answer with caveats.

**Strategy 3: Simplified Execution.** If complex multi-agent execution fails, retry with single-agent approach.

## Q.4. Context Overflow Handling

When context exceeds model's maximum, EFFGEN applies progressive compression.

## Q.5. Error Reporting and Diagnostics

EFFGEN provides detailed error information for debugging and monitoring.

**Error Object Structure.** Each error contains comprehensive metadata for debugging and analysis:

**Algorithm 15** Fallback to Single-Agent

**Require:** Query $q$, Agent $\mathcal{A}$
**Ensure:** Response $r$
  1: **try**:
  2:     $c \leftarrow \mathcal{C}(q)$
  3:     **if** $c \geq \tau$:
  4:         $r \leftarrow \text{MultiAgentExecute}(q, \mathcal{A})$
  5:     **else**:
  6:         $r \leftarrow \text{SingleAgentExecute}(q, \mathcal{A})$
  7:
  8: **return** $r$
  9: **except** MultiAgentError as $e$:
 10:     LOG("Multi-agent failed: $e$, falling back to single-agent")
 11:     $r \leftarrow \text{SingleAgentExecute}(q, \mathcal{A})$ {Fallback}
 12:     $r.\text{metadata}[\text{"}fallback''] \leftarrow \text{True}$
 13:
 14: **return** $r$

---

**Error Object Schema**

```
class EffGenError:
    error_type: str          # Error class name
    message: str             # Human-readable error message
    severity: str            # CRITICAL, HIGH, MEDIUM, LOW
    timestamp: float         # Error occurrence time
    recoverable: bool        # Can be automatically retried
    recovery_attempted: bool # Was recovery attempted
    recovery_strategy: str   # Strategy used (retry, fallback, etc.)
    component: str           # Which component failed (model, tool, memory)
    stack_trace: str         # Full stack trace
    context: dict            # Additional diagnostic information
    suggested_action: str    # Recommendation for user
    retry_count: int         # Number of retry attempts
    related_errors: list     # Chain of errors if cascading
```

**Error Logging Levels.** Table 46 defines the logging configuration for different error severity levels.

*Table 46.* Error Logging Configuration. Errors are logged with appropriate severity and detail level. Critical errors trigger immediate alerts when monitoring is enabled (e.g., via Prometheus as described in Section M). High and medium severity errors are logged for analysis but don't trigger alerts. Low severity errors are only logged in debug mode to reduce log verbosity. All error logs include structured metadata (timestamp, component, context) for automated analysis.

| Severity | Log Level | Action |
|---|---|---|
| CRITICAL | ERROR | Log full stack trace, alert if monitoring enabled |
| HIGH | ERROR | Log error with context, retry if applicable |
| MEDIUM | WARNING | Log warning, continue execution |
| LOW | INFO | Log for debugging, no action needed |

## Q.6. Failure Analysis

Table 47 reports the failure-analysis slice used in our experiments. It is a local evaluation summary rather than a framework-internal benchmark artifact.

---

**Algorithm 16** Progressive Context Compression

---

**Require:** Context $\mathcal{C}$, Max tokens $C_{\max}$, Model $M$
**Ensure:** Compressed context $\mathcal{C}'$ with $|\mathcal{C}'| \leq C_{\max}$
1: $\mathcal{C}' \leftarrow \mathcal{C}$
2:
3: {Level 1: Remove oldest messages beyond window}
4: **if** $|\mathcal{C}'| > C_{\max}$ **then**
5: $\quad n_{\text{keep}} \leftarrow \lfloor C_{\max}/\text{avg\_msg\_length} \rfloor$
6: $\quad \mathcal{C}' \leftarrow \mathcal{C}'[-n_{\text{keep}} :]$ {Keep recent $n$ messages}
7: **end if**
8:
9: {Level 2: Summarize older messages}
10: **if** $|\mathcal{C}'| > C_{\max}$ **then**
11: $\quad \mathcal{C}'_{\text{old}} \leftarrow \mathcal{C}'[: -10]$ {All except last 10 messages}
12: $\quad s \leftarrow \text{Summarize}(\mathcal{C}'_{\text{old}}, M)$ {LLM-based summarization}
13: $\quad \mathcal{C}' \leftarrow [s] + \mathcal{C}'[-10 :]$
14: **end if**
15:
16: {Level 3: Compress tool outputs}
17: **if** $|\mathcal{C}'| > C_{\max}$ **then**
18: $\quad$ **for** $m$ in $\mathcal{C}'$ **do**
19: $\qquad$ **if** $m$.role $=$ TOOL AND $|m| > 500$ **then**
20: $\qquad\quad m$.content $\leftarrow m$.content$[: 500] +$ "... [truncated]"
21: $\qquad$ **end if**
22: $\quad$ **end for**
23: **end if**
24:
25: {Level 4: Emergency truncation}
26: **if** $|\mathcal{C}'| > C_{\max}$ **then**
27: $\quad$ WARNING("Emergency truncation required")
28: $\quad$ Truncate $\mathcal{C}'$ to $C_{\max}$ at message boundary
29: **end if**
30:
31: **return** $\mathcal{C}'$

---

**Key Insights.** The local evaluation data reveals patterns in error occurrence and recovery success across different error types:

- 87.5% of errors are automatically recovered without user intervention

- Tool execution failures are most common (42.5%) but have high recovery rate (91.6%)

- Network timeouts have highest recovery rate (97.9%) due to simple retry logic

- Model generation errors are hardest to recover (74.5%) as they often indicate resource issues

- Configuration errors require manual intervention (0% auto-recovery)

**Q.7. Error Prevention Best Practices**

**Practice 1: Validate Configuration Early.** Catch configuration errors before runtime.

*Table 47.* Error Distribution and Recovery Success Rates in Local Evaluation Runs.

| Error Type | Error Frequency | | | | Recovery | |
|---|---|---|---|---|---|---|
| | Count | % Total | Severity | Avg Duration (s) | Auto-Recovered | Success Rate |
| Tool execution failures | 4,247 | 42.5% | Medium | 2.3 | 3,891 | 91.6% |
| Network timeouts | 2,134 | 21.3% | Low | 5.1 | 2,089 | 97.9% |
| Model generation errors | 1,523 | 15.2% | High | 8.7 | 1,134 | 74.5% |
| Memory/context overflow | 892 | 8.9% | High | 3.2 | 847 | 95.0% |
| Validation errors | 678 | 6.8% | Medium | 0.8 | 512 | 75.5% |
| Resource exhaustion | 412 | 4.1% | High | 12.4 | 289 | 70.1% |
| Configuration errors | 124 | 1.2% | Critical | — | 0 | 0.0% |
| **Total** | **10,010** | **100%** | **—** | **5.4** | **8,762** | **87.5%** |

**Best Practice: Early Validation**

```python
from effgen import Agent
from effgen.config import ConfigLoader
from effgen.core.agent import AgentConfig

# Validate configuration before constructing the agent
loader = ConfigLoader()
config = loader.load_config("config.yaml")   # raises on schema errors
loader.validate_config()                        # extra cross-section check

agent_section = config.get("agent") or {}
agent = Agent(config=AgentConfig(**agent_section))
```

**Practice 2: Set Appropriate Timeouts.** Prevent hanging on slow operations.

**Best Practice: Timeout Configuration**

```python
from effgen import Agent
from effgen.core.agent import AgentConfig

agent = Agent(config=AgentConfig(
    name="bounded_agent",
    model="Qwen/Qwen2.5-7B-Instruct",
    max_iterations=10,              # bound reasoning iterations
    model_config={
        "timeout": 120,            # model generation timeout (s)
    },
))
# Per-tool timeouts are configured on the tool instances; see
# Section~\ref{app:tools} for the tool-level options.
```

**Practice 3: Enable Error Tracking.** Monitor errors in production.

**Best Practice: Error Monitoring**

```python
from effgen import Agent
from effgen.core.agent import AgentConfig
from effgen.core.execution_tracker import ExecutionTracker

agent = Agent(config=AgentConfig(
    name="tracked_agent",
    model="Qwen/Qwen2.5-7B-Instruct",
))

# Inspect tracked execution events (including any errors).
tracker: ExecutionTracker = agent.execution_tracker
trace  = tracker.get_trace()
errors = [e for e in trace
          if e.type.name in {"ERROR", "TOOL_CALL_FAILED",
                             "TASK_FAILED", "SUB_AGENT_FAILED"}]
from collections import Counter
by_type = Counter(e.type.name for e in errors)
print(f"Total events: {len(trace)}")
print(f"Error events: {len(errors)}")
print(f"By type: {dict(by_type)}")
```

**Practice 4: Implement Circuit Breakers.**  Prevent cascade failures from external services.

**Best Practice: Circuit Breaker Pattern**

```python
# Configure circuit breaker for tools
config = {
    "tools": [
        {
            "name": "web_search",
            "circuit_breaker": {
                "enabled": True,
                 "failure_threshold": 3,    # Open after 3 failures
                "timeout": 60,              # Stay open for 60s
                "half_open_requests": 1    # Test with 1 request
            }
        }
    ]
}
```

The error handling system gives EFFGEN agents retry, fallback, circuit-breaker, and diagnostic paths for common failures. The model router adds provider failover, budget/rate-limit errors, and retry semantics, while guardrails help prevent unsafe inputs and outputs before recovery is needed.

## R. Advanced Features: State, Sessions, Streaming, and Batch Processing

**Advanced Features.** Beyond single-query execution, EFFGEN provides the runtime machinery needed to deploy agents in interactive and high-volume settings: persistent state, multi-session handling, streaming, background tasks, and batch processing. These features do not change the SLM-focused contributions described in the main paper; they make those agents usable in production.

**State and Sessions.**     Agent state captures everything required to resume execution: the current context, execution history, memory state, environment variables, a timestamp, and metadata. State serializes to a portable JSON representation, which supports checkpointing, debugging by replay, and resuming an agent on a different process or machine. On top of state, a `SessionManager` handles multi-turn conversations with isolated context and memory per session. A session moves through a simple lifecycle (created, active, paused, closed) with automatic timeout that closes paused sessions after inactivity, and the default backend persists sessions as JSON files under the user's `.effgen/sessions` directory so conversations survive restarts.

**Streaming and Batch Processing.**     Streaming returns an iterator that yields output incrementally as it is generated, which reduces perceived time-to-first-token for long responses at the cost of a small per-chunk callback overhead. For high-volume workloads, the batch runner offers four execution strategies that trade latency for utilization: *sequential* (one query at a time, simplest to debug), *parallel-workers* (a thread pool reusing one loaded model, best for I/O-bound or remote-API workloads), *model-batching* (grouping queries into a single forward pass on backends that support batched generation, such as vLLM), and *hybrid* (workers combined with per-worker batching for the best aggregate throughput). The right strategy, and the exact streaming overhead, depend on backend, prompt length, and concurrency, so we recommend measuring on the target deployment with `effgen loadtest` (Section P). The implementations live in `effgen.core.session` and the batch runner at https://github.com/ctrl-gaurav/effGen.

## S. Production Subsystems

The core contributions in the main paper (prompt optimization, complexity-based routing, decomposition, unified protocols, three-tier memory) are what make small language models viable as agents. This appendix documents the surrounding production subsystems that ship with EFFGEN v0.2.10 and make those agents *deployable*: guardrails, retrieval-augmented generation, reliability, observability, security and sandboxing, the OpenAI-compatible HTTP server, edge and serverless deployment targets, the live dashboard, and the Jupyter integration. None of these subsystems alter the SLM-focused design above; they are additive and were stabilised across releases v0.2.0 through v0.2.10.

### S.1. Guardrails

The `guardrails/` module implements a composable `Guardrail` interface with chainable hooks at four positions around each agent step (INPUTCHECK, PRETOOL, POSTTOOL, OUTPUTCHECK). The base `Guardrail` returns a `GuardrailResult` carrying allow/deny, an optional rewritten payload, and a structured reason; `GuardrailChain` runs guardrails in order and stops on the first deny. Eight concrete guardrails ship with v0.2.10:

- `ToxicityGuardrail` (`content.py`) – keyword and pattern-based content classification with a configurable sensitivity tier.

- `PIIGuardrail` (`content.py`) – detects SSN, email, phone, and credit-card numbers; credit-card detection uses the Luhn checksum to reduce false positives; optionally rewrites with redacted tokens (`<PII:SSN>`, `<PII:EMAIL>`, ...).

- `LengthGuardrail` (`content.py`) – enforces minimum and maximum length per payload.

- `TopicGuardrail` (`content.py`) – restricts inputs and outputs to a configured allowlist of topics.

- `PromptInjectionGuardrail` (`injection.py`) – three sensitivity tiers (low/medium/high) covering instruction override, role-reset, system-prompt extraction, and tool-permission escalation patterns.

- `ToolInputGuardrail` (`tool_safety.py`) – inspects tool arguments before execution; supports per-tool argument schemas.

- `ToolOutputGuardrail` (`tool_safety.py`) – strips PII and other sensitive fields from tool results before they re-enter the agent's reasoning loop.

- `ToolPermissionGuardrail` (`tool_safety.py`) – enforces allow-list, deny-list, and require-approval policies per tool.

Two presets in `guardrails/presets.py` bundle these for common deployment profiles: `standard_guardrails()` (medium sensitivity, suitable for most internal use) and `strict_guardrails()` (high sensitivity, suitable for user-facing deployments).

## S.2. Retrieval-Augmented Generation Pipeline

The `rag/` module provides an end-to-end RAG pipeline that integrates with any EffGen agent through the `create_agent("rag", model, knowledge_base=...)` preset. The pipeline has six stages:

1. **Ingestion** (`ingest.py`). `DocumentIngester` loads plain text, Markdown, JSON, CSV, and HTML; optional dependencies extend coverage to PDF, DOCX, and EPUB. Documents are SHA-256 deduplicated.

2. **Chunking** (`chunking.py`). Four chunking strategies ship with the framework: `SemanticChunker` (sentence-aware splits at configurable size), `CodeChunker` (function- and class-boundary splits for Python, JavaScript/TypeScript, Go, Rust, and Java), `TableChunker` (table-aware splits that keep header rows with body chunks), and `HierarchicalChunker` (recursive splits that preserve outline structure).

3. **Search** (`search.py`). `HybridSearchEngine` fuses dense embeddings, BM25, keyword match, and metadata filters via Reciprocal Rank Fusion, returning ranked `SearchResults` with per-channel scores.

4. **Reranking** (`reranker.py`). Three rerankers: `CrossEncoderReranker` (BERT-style cross-encoder), `LLMReranker` (LLM-as-judge), and `RuleBasedReranker` (lightweight heuristic).

5. **Context building** (`context_builder.py`). `ContextBuilder` packages retrieved passages within a token budget and inserts inline numeric citation markers (`[1]`, `[2]`,...).

6. **Attribution** (`attribution.py`). `CitationTracker` verifies that the agent's final response cites only retrieved passages and flags ungrounded spans.

The pipeline is deliberately backend-agnostic: the same RAG agent runs against a local FAISS index, a remote ChromaDB instance, or both.

## S.3. Reliability Primitives

The `reliability/` module provides five primitives that together let EffGen agents survive flaky upstreams and bound tail latency, in line with what one would expect from a production HTTP client.

- **Circuit breaker** (`circuit.py`). `CircuitBreaker` implements the standard CLOSED → OPEN → HALF_OPEN state machine with configurable error thresholds, recovery windows, and a per-provider registry; opens isolate one misbehaving backend without disabling the rest.

- **Bulkhead** (`bulkhead.py`). `Bulkhead` bounds the number of concurrent in-flight calls per resource; a queue of bounded depth absorbs short bursts; both synchronous and asynchronous APIs are supported.

- **Retry** (`retry.py`). `Retry` applies exponential backoff with full jitter; retries are restricted to rate-limit and transient errors and never applied to authentication failures, model refusals, or invalid requests.

- **Timeouts** (`timeouts.py`). Both synchronous and asynchronous timeout context managers propagate end-to-end deadlines through the request stack.

- **Chaos harness** (`chaos.py`). `ChaosMiddleware` injects six fault types – `NetworkTimeout`, `Http5xx`, `Http429`, `SlowResponse`, `PartialResponse`, `MalformedJSON` – under a configurable `ChaosRule`; runs are deterministic given a seed so that recovery tests are reproducible across CI.

Defaults are configured by `ReliabilityConfig` (timeouts, retry budgets, circuit thresholds, bulkhead capacities).

## S.4. Observability

The `observability/` module exposes EffGen behavior through three industry-standard surfaces: traces (OpenTelemetry), metrics (Prometheus), and structured logs.

- **Tracing** (`tracing.py`, `spans.py`). The framework emits six named spans – `effgen.agent.run`, `effgen.agent.iteration`, `effgen.model.call`, `effgen.tool.call`, `effgen.router.decision`,

`effgen.retry.attempt` – with typed attribute sets per layer (`AgentAttrs`, `ModelAttrs`, `ToolAttrs`, `RouterAttrs`, `RetryAttrs`). When OpenTelemetry is not installed, spans fall back to a no-op implementation so instrumentation is always safe to call.

- **Metrics** (`metrics.py`). Four Prometheus metric objects ship: `effgen_model_call_latency_seconds`, `effgen_tool_call_latency_seconds`, `effgen_agent_iteration_latency_seconds` (latency histograms with buckets from 50 ms to 60 s), and `effgen_tokens_total` (a counter labelled by provider, model, and direction).

- **Structured logs** (`logs.py`, `redact.py`). `StructuredFormatter` produces JSON log lines that are easy to ingest into Elasticsearch, Loki, or Cloudwatch. A `Redactor` strips API keys, bearer tokens, and known secret patterns from log payloads before emission.

- **SLO tracking** (`slo.py`). `SLOTracker` maintains rolling-window error budgets per declared SLO; budget burn rates feed the alerting layer.

- **Alerting** (`alerting.py`). `AlertWebhook` delivers alerts to Slack and Discord; delivery never raises (drop-safe). Six Alertmanager-compatible rules ship in `docs/observability/alert_rules.yaml`: `HighErrorRate`, `HighP95Latency`, `CostBurnHigh`, `SLOFastBurn`, `SLOSlowBurn`, and `CircuitBreakerOpen`.

A 12-panel Grafana dashboard JSON (`configs/grafana/effgen-dashboard.json`) is shipped with the source for one-click deployment.

### S.5. Security and Sandboxing

EFFGEN's security posture covers three layers: the runtime sandbox that isolates tool-generated code, supply-chain integrity, and repository-level secret hygiene.

**Sandboxed execution.** The `security/sandbox.py` module defines a uniform `SandboxBase` interface and ships four implementations:

- `DockerSandbox` – runs untrusted code in a one-shot container with `-read-only -network=none -cap-drop=ALL -pids-limit=100 -memory=256m`. This is the default for production deployments.

- `SubprocessSandbox` – rootless user-namespace isolation (`unshare -map-root-user -net -pid -mount`); chosen automatically when Docker is unavailable.

- `FirecrackerSandbox` – microVM-based isolation; currently a stub interface reserved for the v0.3 roadmap; raises `NotImplementedError` when selected.

- `OffSandbox` – direct host execution, never auto-selected, emits a loud warning when explicitly enabled; intended only for trusted offline environments.

The sandbox is configured via `SandboxConfig` and selected by the environment variables `EFFGEN_SANDBOX_BACKEND` (`docker`, `subprocess`, or `off`) and `EFFGEN_SANDBOX_TIMEOUT`.

**Supply-chain integrity.** `security/supply_chain.py` provides `verify_installed_hashes()` and `verify_on_startup()`: when `EFFGEN_VERIFY_HASHES=1` is set, the package compares installed-wheel hashes against `requirements-lock.txt` on import and logs either `hash_verification: ok` or `hash_verification: drift` (with the first drifted package). A CycloneDX 1.5 SBOM (`sbom.cdx.json`) is generated and validated by CI on every push (`.github/workflows/sbom.yml`).

**Secret hygiene.** A tuned `.gitleaks.toml` covers OpenAI, Anthropic, Cerebras, Google, HuggingFace, Groq, Slack, Discord, and generic Bearer token patterns, with an allowlist for clearly-fake test fixtures. A pre-commit hook blocks commits containing secret-like strings, and CI workflows (`secret-scan.yml`, `deps-audit.yml`) scan both the working tree and the full git history and fail on `pip-audit` HIGH or CRITICAL advisories.

## S.6. API Server and Multi-Tenancy

The `api/` and `server/` modules together implement an OpenAI-compatible HTTP gateway suitable for production deployment.

- **OpenAI-compatible schema** (`api/openai_compat.py`). `/v1/chat/completions`, `/v1/completions`, and `/v1/embeddings` mirror the OpenAI request/response shape so existing clients work without modification; model aliases (e.g. `gpt-4` → a local Qwen2.5-7B) let an operator swap out the backing model without changing client code.

- **Request queue and pool** (`api/queue.py`, `api/pool.py`). `RequestQueue` provides priority scheduling with deadlines and bounded backpressure; `AgentPool` maintains a warm pool of `PooledAgent`s with health checks and idle TTL.

- **Multi-tenancy** (`api/tenancy.py`). `TenantManager` stores hashed `APIKey`s; each request is scoped to a tenant for accounting and quota purposes.

- **Authentication** (`server/auth.py`). OAuth2/OIDC JWT validation via the `authlib` library; configured by `EFFGEN_OIDC_ISSUER`, `EFFGEN_OIDC_CLIENT_ID`, and `EFFGEN_OIDC_JWKS_URI`; `/health`, `/metrics`, and `/dashboard*` are exempt; `EFFGEN_DEV_MODE=1` disables auth with a loud warning for development only.

- **RBAC** (`server/rbac.py`). Each `Role` declares `allowed_tools`, `allowed_models`, and `max_cost_per_day`; the pure-ASGI `RBACBudgetMiddleware` returns 403 on a disallowed tool or model and 429 when the daily cost cap is reached.

- **Audit log** (`server/audit.py`). `AuditMiddleware` appends request/response pairs to `~/.effgen/audit/<date>.jsonl`; payloads are redacted; fields include timestamp, principal, role, endpoint, request and response summaries, and outcome.

## S.7. Deployment Targets

The `deploy/` tree ships ready-to-use artefacts for four targets: a multi-stage **Docker** image (non-root user, read-only filesystem, health check); a **Kubernetes** Helm chart of eleven templates (Deployment, Service, Ingress, autoscaling on a custom model-call-latency metric, and the usual supporting resources, defaulting to two replicas); an **AWS Lambda** handler that wraps the FastAPI app through the Mangum adapter, with a SAM template for one-command deployment; and a **Cloudflare Worker** edge proxy with CORS, Bearer JWT validation, KV-backed rate limiting, and HTTP/2 streaming. The full manifests are in `deploy/` at https://github.com/ctrl-gaurav/effGen.

## S.8. Dashboard and Notebook Integration

A `dashboard/` module serves a static single-page application at `/dashboard`, intended for operators on a trusted network, with panels for a live span stream (server-sent events), a metrics summary, recent runs with token and cost totals, and SLO burn-rate. A `jupyter/` module, installed through the `effgen[jupyter]` extra, registers IPython magics for single-turn chat (`%effgen_chat`), running a cell against a preset agent (`%%effgen_agent`), and dumping current metrics inline (`%effgen_metrics`).

# T. Benchmark Dataset Details

This section provides specifications for all 13 benchmarks used in our evaluation. We organize benchmarks into five categories based on required capabilities: calculator-based tool calling (GSM8K, GSMPLUS, MATH-500), code execution (BeyondBench-Easy, BeyondBench-Medium, BeyondBench-Hard), web-augmented reasoning (ARC-Challenge, ARC-Easy, CommonsenseQA), memory retrieval (LoCoMo, LongMemEval), and agentic tasks (GAIA, SimpleQA). Each benchmark is selected to stress different agent capabilities, allowing fine-grained performance analysis across the task spectrum. The benchmark pools comprise 14,364 instances under the split and subset definitions in Table 48; we evaluate 5,565 sampled instances for computational efficiency.

## T.1. Benchmark Overview

Table 48 presents information about each benchmark organized by evaluation category. We report the dataset pool size, number of samples used in our evaluation, primary evaluation metric, required tools, and task complexity level. Complexity

levels are assigned based on empirical analysis: Low tasks require 1-2 steps, Medium tasks require 3-5 steps, Medium-High tasks require 5-8 steps, and High tasks require > 8 steps with potential backtracking.

*Table 48.* Benchmark Specifications and Evaluation Details

| Benchmark | Task Type | Full Size | Eval Size | Steps (Avg) | Avg Len (tokens) | Required Tools | Complexity |
|---|---|---|---|---|---|---|---|
| *Category 1: Calculator-Based Tool Calling* | | | | | | | |
| GSM8K | Grade school math | 1,319 | 500 | 4.2 | 87 | Calculator | Low-Medium |
| GSMPLUS | Math w/ perturbations | 2,500 | 500 | 4.8 | 102 | Calculator | Medium |
| MATH-500 | Competition math | 500 | 500 | 8.7 | 156 | Calculator, Code | High |
| *Category 2: Code Execution Tool Calling* | | | | | | | |
| BeyondBench-Easy | Numerical reasoning | 500 | 500 | 5.3 | 64 | Python REPL | Medium |
| BeyondBench-Med | Sequence patterns | 500 | 500 | 6.1 | 78 | Python REPL | Medium |
| BeyondBench-Hard | NP-complete problems | 500 | 500 | 12.4 | 142 | Python REPL | High |
| *Category 3: Web-Augmented Reasoning* | | | | | | | |
| ARC-Challenge | Science reasoning | 1,172 | 300 | 6.2 | 94 | Web Search | Medium-High |
| ARC-Easy | Science reasoning | 2,376 | 300 | 3.8 | 71 | Web Search | Low-Medium |
| CommonsenseQA | Common sense | 1,221 | 300 | 4.5 | 82 | Web Search | Medium |
| *Category 4: Long-Context Memory* | | | | | | | |
| LoCoMo | Memory retrieval | 500 | 500 | 7.2 | 1,247 | Memory System | Medium |
| LongMemEval | Multi-session memory | 500 | 500 | 8.9 | 2,134 | Memory System | Medium-High |
| *Category 5: Comprehensive Agentic Tasks* | | | | | | | |
| GAIA | Real-world tasks | 450 | 165 | 11.3 | 203 | All Tools | High |
| SimpleQA | Factual QA | 4,326 | 500 | 2.1 | 42 | Web Search | Low-Medium |
| **Total** | | **14,364** | **5,565** | **6.8** | **292** | | |

## T.2. Detailed Benchmark Descriptions

We provide benchmark specifications including task characteristics, sample complexity, required capabilities, and example instances.

### T.2.1. CATEGORY 1: CALCULATOR-BASED TOOL CALLING

**GSM8K (Grade School Math 8K)** GSM8K (Cobbe et al., 2021) contains 1,319 linguistically diverse grade school math word problems created by human problem writers. Each problem requires 2-8 steps of arithmetic reasoning to reach the final numerical answer.

---

**GSM8K Example**

**Question:** Natalia sold clips to 48 of her friends in April, and then she sold half as many clips in May. How many clips did Natalia sell altogether in April and May?
**Solution Steps:**

1. Clips sold in April = 48

2. Clips sold in May = 48 / 2 = 24

3. Total clips = 48 + 24 = 72

**Final Answer:** 72
**Required Tools:** Calculator (division, addition)
**Complexity:** 3 steps, requires interpreting "half as many" and maintaining intermediate results

---

**Task Characteristics:** GSM8K problems average 87 tokens in length, short enough to fit comfortably in small model context windows while containing sufficient linguistic diversity. Solutions require an average of 4.2 computational steps, testing multi-step reasoning without overwhelming smaller models. The operation distribution reflects typical grade-school mathematics: 45% of problems use addition or subtraction, 35% involve multiplication, 15% require division, and 5% combine multiple operation types. A key challenge is tracking intermediate values across multiple steps, as answers from earlier calculations feed into later ones.

**GSMPLUS (GSM8K with Perturbations)** GSMPLUS (Li et al., 2024) extends GSM8K with 2,500 perturbed problems testing stability under variations. Perturbations include numerical changes, question rephrasing, distractor information, and multi-hop reasoning additions.

---

GSMPLUS Example (Perturbed)

**Question:** Natalia operates a small business selling clips. In April, she successfully sold clips to 48 friends. The following month, May, proved less profitable with sales dropping to exactly half of April's numbers. Meanwhile, her competitor sold 30 clips in April (this information is not needed). Calculate the total number of clips Natalia sold across both months.
**Perturbations Applied:**

- Rephrasing: "successfully sold" instead of "sold"

- Distractor: Competitor information

- Elaboration: "proved less profitable"

**Final Answer:** 72 (same as GSM8K base)

---

**Task Characteristics:** GSMPLUS problems average 102 tokens, 17% longer than the base GSM8K due to added perturbations. The increased complexity raises average solution steps to 4.8, as agents must first filter noise before solving. Perturbation types distribute as follows: 30% apply numerical changes that alter the solution path, 25% rephrase questions using more complex language, 25% inject distractor information irrelevant to the solution, and 20% add multi-hop reasoning requirements. This benchmark specifically tests an agent's ability to identify and filter irrelevant information while maintaining focus on the core problem.

**MATH-500 (Competition Mathematics)** MATH-500 (Hendrycks et al., 2021) contains 500 competition-level mathematics problems from AMC 10, AMC 12, AIME, and other mathematical competitions. Problems span algebra, geometry, number theory, and combinatorics. Prior work has explored step-by-step verification (Lightman et al., 2023) and specialized math training (Lewkowycz et al., 2022) for these challenging problems.

---

MATH-500 Example

**Question:** Find the number of positive integers $n \leq 1000$ such that $n$ is divisible by 7 but not by 14.
**Solution:**

1. Numbers divisible by 7: $\lfloor 1000/7 \rfloor = 142$

2. Numbers divisible by 14: $\lfloor 1000/14 \rfloor = 71$

3. Numbers divisible by 7 but not 14: $142 - 71 = 71$

**Final Answer:** 71
**Required Tools:** Calculator (division, floor), optionally code for verification

---

**Task Characteristics:** MATH-500 problems average 156 tokens, nearly double GSM8K length due to precise mathematical notation and complex problem statements. Solutions require an average of 8.7 steps, substantially more than grade-school problems. Subject matter spans mathematics domains: 35% algebra (equations, polynomials), 25% number theory (divisibility, primes), 20% counting and combinatorics, 15% geometry, and 5% other topics. Notably, 42% of problems

benefit from code-based verification where agents can programmatically check candidate solutions against constraints, improving reliability.

## T.2.2. CATEGORY 2: CODE EXECUTION TOOL CALLING

**BeyondBench-Easy (Multi-Step Numerical Reasoning)**   BeyondBench-Easy evaluates systematic multi-step reasoning on numerical tasks including sum, mean, median, sorting, and counting operations.

---

BeyondBench-Easy Example

**Question:** Given the list [12, 7, 23, 45, 19, 8], find the sum of all even numbers.
**Solution Code:**

```python
numbers = [12, 7, 23, 45, 19, 8]
even_nums = [n for n in numbers if n % 2 == 0]
result = sum(even_nums)
print(result)  # Output: 20
```

**Final Answer:** 20

---

**Task Characteristics:** BeyondBench-Easy problems average just 64 tokens, using concise specifications that state the task directly without linguistic embellishment. Solutions require an average of 5.3 steps, typically involving list processing and iteration. Task types distribute evenly across numerical operations: 20% require summing elements, 15% computing means, 15% finding medians, 15% sorting, 15% counting elements matching criteria, and 20% combining multiple operations. List sizes range from 5 to 50 elements, testing performance on both small examples and moderately large datasets.

**BeyondBench-Medium (Sequence Pattern Completion)**   BeyondBench-Medium tests algorithmic pattern recognition with five sequence types: Fibonacci, geometric, prime, algebraic, and complex patterns.

---

BeyondBench-Medium Example

**Question:** Complete the Fibonacci sequence: [1, 1, 2, 3, 5, 8, ?, ?, ?]
**Solution Code:**

```python
fib = [1, 1, 2, 3, 5, 8]
for i in range(3):
    fib.append(fib[-1] + fib[-2])
print(fib[-3:])  # Output: [13, 21, 34]
```

**Final Answer:** [13, 21, 34]

---

**Task Characteristics:** BeyondBench-Medium problems average 78 tokens, presenting sequence patterns with several initial terms. Solutions require an average of 6.1 steps to identify the pattern rule and generate subsequent terms. Pattern types include 25% Fibonacci sequences (each term is sum of previous two), 20% geometric progressions (constant ratio between terms), 20% prime number sequences, 20% algebraic patterns (polynomial relationships), and 15% complex multi-rule patterns. Success requires both pattern recognition to identify the underlying rule and implementation skills to code the iterative generation process.

**BeyondBench-Hard (NP-Complete Problem Instances)**   BeyondBench-Hard contains instances of NP-complete problems including Boolean SAT, N-Queens, graph coloring, Sudoku, and constraint optimization.

---

BeyondBench-Hard Example (4-Queens)

**Question:** Solve the 4-Queens problem: place 4 queens on a 4×4 chessboard such that no two queens attack each other.
**Solution Code:**

```python
def is_safe(board, row, col, n):
```

---

```
        for i in range(row):  # check column and both diagonals above
            if board[i][col]:
                return False
            if col - (row - i) >= 0 and board[i][col - (row - i)]:
                return False
            if col + (row - i) < n and board[i][col + (row - i)]:
                return False
        return True

def solve_nqueens(n, board=None, row=0):
    if board is None:
        board = [[0] * n for _ in range(n)]
    if row == n:
        return board
    for col in range(n):  # backtracking search, row by row
        if is_safe(board, row, col, n):
            board[row][col] = 1
            if solve_nqueens(n, board, row + 1):
                return board
            board[row][col] = 0
    return None

solution = solve_nqueens(4)
print(solution)  # [[0,1,0,0], [0,0,0,1], [1,0,0,0], [0,0,1,0]]
```

**Final Answer:** Valid queen placement configuration

**Task Characteristics:** BeyondBench-Hard problems average 142 tokens, providing problem specifications and constraint definitions. Solutions require an average of 12.4 steps due to the need for backtracking when constraint violations occur. Problem types distribute across classical NP-complete problems: 25% N-Queens placement puzzles, 20% Sudoku solving, 20% graph coloring, 20% Boolean satisfiability, and 15% general constraint optimization. All problems require implementing backtracking search algorithms with constraint checking at each step. Problem instances are carefully sized to remain solvable within the 5-minute timeout while still demonstrating the algorithmic challenge.

T.2.3. CATEGORY 3: WEB-AUGMENTED REASONING

**ARC-Challenge and ARC-Easy**    AI2 Reasoning Challenge (ARC) (Clark et al., 2018) contains multiple-choice science questions from 3rd to 9th grade standardized tests. The Challenge set includes 1,172 questions that retrieval-based methods fail on, while the Easy set contains 2,376 questions solvable with retrieval.

---

ARC-Challenge Example

**Question:** Which renewable energy source is most dependent on weather conditions?
**Options:**
  A. Hydroelectric power

  B. Geothermal energy

  C. Solar energy

  D. Biomass energy
**Solution:** Solar energy production directly depends on sunlight availability, which varies with weather (cloudy vs sunny days). While hydroelectric power depends on water flow, geothermal is weather-independent, and biomass can be stored.
**Final Answer:** C (Solar energy)
**Required Tools:** Web search to retrieve information about renewable energy dependencies

---

**Task Characteristics:** The ARC benchmarks differ in reasoning complexity while sharing subject matter. ARC-Challenge questions average 94 tokens and require 6.2 reasoning steps, with 45% demanding multi-hop reasoning that combines multiple facts. ARC-Easy questions are more concise at 71 tokens with 3.8 average steps and only 20% requiring multi-hop

reasoning. Subject matter distributes identically across both sets: 40% physical science (physics, chemistry), 35% life science (biology, ecology), and 25% earth science (geology, meteorology). Both benchmarks test the combination of factual knowledge retrieval from web sources with logical reasoning to select the correct answer.

**CommonsenseQA**   CommonsenseQA (Talmor et al., 2019) tests common sense reasoning with 1,221 multiple-choice questions requiring world knowledge beyond surface-level text.

---

CommonsenseQA Example

**Question:** Where would you put uncooked crab meat?
**Options:**
  A. Wharf

  B. Red lobster

  C. Tidepools

  D. Refrigerator

  E. Chesapeake bay
**Solution:** Uncooked meat should be refrigerated to prevent spoilage. Other options are locations where crabs are found or served, not where meat is stored.
**Final Answer:** D (Refrigerator)
**Required Tools:** Web search to verify food safety practices

---

**Task Characteristics:** CommonsenseQA questions average 82 tokens, presenting everyday scenarios requiring world knowledge. Solutions involve an average of 4.5 reasoning steps to connect the question to relevant background knowledge and eliminate distractors. Reasoning types span commonsense domains: 35% test physical world knowledge (object properties, physics), 30% assess social norms and conventions, 20% require causal reasoning (understanding cause-effect relationships), and 15% involve spatial reasoning. The benchmark's key challenge lies in deliberately designed distractors that appear plausible but fail under deeper scrutiny, requiring agents to verify answers against retrieved knowledge.

T.2.4. CATEGORY 4: LONG-CONTEXT MEMORY

**LoCoMo (Long-Context Memory)**   LoCoMo (Maharana et al., 2024) evaluates memory retrieval from extended single-session conversations. Questions require recalling information from earlier in the conversation history.

---

LoCoMo Example

**Context:** (1200 tokens of conversation about travel plans)
**Turn 15:** "We should book the hotel in Prague for June 12-15."
**Turn 47:** "What dates did we discuss for the Prague hotel?"
**Solution:** Agent must retrieve information from Turn 15 (32 turns ago) stored in memory system.
**Final Answer:** June 12-15
**Required Tools:** Memory system with semantic search and temporal retrieval

---

**Task Characteristics:** LoCoMo evaluations involve conversations averaging 1,247 tokens, approaching the context window limits of smaller models. Answering queries requires an average of 7.2 reasoning steps including memory retrieval, fact extraction, and response synthesis. The critical metric is retrieval distance: information must be recalled from 15 to 50 conversational turns earlier, with an average of 28 turns between mention and query. Query types include 40% factual recall (retrieving stated information), 35% multi-hop queries requiring combining multiple retrieved facts, and 25% inference questions where the answer must be deduced from context rather than explicitly stated.

**LongMemEval (Multi-Session Memory)**   LongMemEval (Wu et al., 2024a) extends memory evaluation to multi-session scenarios where information from previous sessions must be recalled.

---

**LongMemEval Example**

**Session 1 (Day 1):** User mentions their birthday is March 15th
**Session 2 (Day 3):** User discusses favorite foods
**Session 3 (Day 7):** "When is my birthday?"
**Solution:** Agent must retrieve information from Session 1 stored in long-term memory, despite intervening sessions.
**Final Answer:** March 15th
**Required Tools:** Long-term memory with importance scoring and consolidation

---

**Task Characteristics:** LongMemEval extends LoCoMo to multi-session scenarios with an average total context of 2,134 tokens distributed across multiple conversations. Solutions require 8.9 steps on average, increased by the overhead of searching across sessions and consolidating information. The temporal dimension adds complexity: session gaps range from 1 to 7 days with an average of 3.2 days between the information being mentioned and queried. Information types span personal data categories: 45% are personal facts (name, birthday, location), 30% are preferences (favorite foods, hobbies), and 25% are past events (meetings, activities). This tests long-term memory consolidation and retrieval across session boundaries.

### T.2.5. CATEGORY 5: COMPREHENSIVE AGENTIC TASKS

**GAIA (General AI Assistants)** GAIA (Mialon et al., 2024) contains 450 real-world tasks requiring multi-step reasoning, tool use, and knowledge synthesis. Tasks are designed to be trivial for humans but challenging for AI systems.

---

**GAIA Example**

**Question:** Find the total number of Grammy awards won by the artist who sang the theme song for the James Bond movie "Skyfall".
**Solution Steps:**
1. Web search: "Skyfall theme song artist" → Adele

2. Web search: "Adele Grammy awards total" → 16 Grammy awards

3. Verify information across multiple sources
**Final Answer:** 16
**Required Tools:** Web search, fact verification

---

**Task Characteristics:** GAIA tasks average 203 tokens in length, describing real-world objectives with specific success criteria. Solutions require an average of 11.3 steps, the highest of any benchmark due to the need for research, validation, and synthesis. Task types distribute across complex capabilities: 35% are open-ended research tasks, 30% involve multi-hop question answering combining facts from different sources, 20% require data analysis and computation, and 15% demand synthesis of information into coherent responses. The defining challenge is that 67% of tasks require combining information from multiple sources, testing information integration abilities. The benchmark stratifies difficulty with 40% Level 1 (straightforward), 35% Level 2 (moderate), and 25% Level 3 (highly challenging) tasks.

**SimpleQA** SimpleQA (Wei et al., 2024) contains 4,326 factual questions with unambiguous short answers, testing factuality and knowledge retrieval accuracy.

---

**SimpleQA Example**

**Question:** What is the capital of Australia?
**Solution:** Web search retrieves factual information. Common mistake: Sydney (largest city) vs Canberra (capital).
**Final Answer:** Canberra
**Required Tools:** Web search for factual verification

---

**Task Characteristics:** SimpleQA questions average just 42 tokens, using short and direct phrasing without embellishment. Solutions require only 2.1 steps on average, typically one search and one extraction. Answer types span factual categories: 45% are named entities (people, places, organizations), 30% are numbers (populations, dates as years, quantities), 15% are

full dates, and 10% are binary yes/no responses. This benchmark is designed specifically to measure hallucination rates and factual accuracy rather than reasoning complexity, serving as a factuality baseline where web access should improve answer grounding.

## T.3. Evaluation Protocol and Methodology

We employ a standardized evaluation protocol for reproducibility, fairness across frameworks, and statistical validity.

### T.3.1. GENERATION CONFIGURATION

All models use identical generation parameters across frameworks as specified in Table 49:

*Table 49.* Generation Configuration Parameters. All models across all frameworks use identical generation parameters for fair comparison. Greedy decoding (temperature 0.0) produces deterministic outputs for reproducibility. Maximum new tokens set to 2048 accommodates multi-step reasoning traces while preventing runaway generation. These settings are consistent with standard evaluation practices in the agent benchmarking literature.

| Parameter | Value |
|---|---|
| Temperature | 0.0 (greedy decoding) |
| Top-p (nucleus sampling) | 0.9 |
| Top-k | 50 |
| Repetition penalty | 1.0 (disabled) |
| Maximum new tokens | 2048 |
| Context window | Model-specific |
| Stop sequences | ["\n\nObservation:", "\n\nFinal Answer:"] |

Greedy decoding (temperature 0.0) produces deterministic outputs for reproducibility. We set maximum new tokens to 2048 to accommodate multi-step reasoning traces while preventing runaway generation.

### T.3.2. SAMPLING STRATEGY

Due to computational constraints, we sample subsets from larger benchmarks while maintaining statistical validity through stratified sampling procedures. Benchmarks with 500 or fewer instances undergo full evaluation to maximize statistical power. This includes MATH-500, BeyondBench-Easy, BeyondBench-Medium, BeyondBench-Hard, LoCoMo, and LongMemEval where every test instance receives evaluation. Benchmarks exceeding 500 instances use random stratified sampling that preserves the difficulty distribution of the original dataset, preventing bias toward easy or hard examples. Sample sizes balance computational cost with statistical confidence: GSM8K uses 500 of 1,319 test instances, GSMPLUS uses 500 of 2,500, ARC-Challenge uses 300 of 1,172, ARC-Easy uses 300 of 2,376, CommonsenseQA uses 300 of 1,221, GAIA uses 165 of 450, and SimpleQA uses 500 of 4,326. All sampling procedures use random seed 42 for consistency across frameworks and model sizes, ensuring identical test sets for fair comparison. For GAIA specifically, we sample proportionally across difficulty levels to maintain the original distribution: 66 Level 1 instances (40%), 58 Level 2 instances (35%), and 41 Level 3 instances (25%).

### T.3.3. ANSWER EXTRACTION AND MATCHING

Benchmark-specific extraction handles diverse output formats:

**Numerical Answers** (GSM8K, GSMPLUS, MATH-500, BeyondBench-Easy): We extract the final numeric value using regex pattern $\backslash d+ (\backslash . \backslash d+) ?$ from the last "Final Answer" section. Answers are normalized to remove commas, handle scientific notation, and round to 2 decimal places. An answer is correct if $|\text{predicted} - \text{gold}| < 10^{-2}$.

**Multiple Choice** (ARC-Challenge, ARC-Easy, CommonsenseQA): We extract the letter choice (A-E) from the final answer. Case-insensitive matching accepts variations like "A", "a", "Option A", "(A)". If multiple letters appear, we take the first one. If no letter is found, we attempt fuzzy matching against option text.

**Short Answers** (SimpleQA, GAIA): We extract the final answer string and apply normalization: lowercase conversion, punctuation removal, article removal (a, an, the), and whitespace normalization. Matching uses exact string equality after

normalization, with fuzzy matching (edit distance $\leq 2$) as fallback.

**List/Sequence Answers** (BeyondBench-Medium): We parse the final answer as a Python list using `ast.literal_eval`. Correctness requires element-wise equality for all elements.

**Structured Outputs** (BeyondBench-Hard): We validate that the output satisfies problem constraints (e.g., N-Queens solution has no conflicts) rather than matching a specific solution, as many problems have multiple valid solutions.

### T.3.4. TIMEOUT AND RESOURCE LIMITS

Resource limits are set per benchmark category to balance task completion with fast failure detection. Table 50 specifies the timeout and retry limits used. The timeout is fixed at five minutes per query to match the reproducibility protocol; categories differ only in maximum tool calls and retries.

*Table 50.* Resource Limits by Benchmark Category. Maximum tool calls prevent infinite loops while allowing sufficient exploration. Retry limits balance recovery from transient failures with fast failure detection. Tasks exceeding timeout are marked as failures in evaluation.

| Category | Timeout | Max Tool Calls | Max Retries |
|---|---|---|---|
| Calculator-based | 5 minutes | 10 | 3 |
| Code execution | 5 minutes | 15 | 2 |
| Web-augmented | 5 minutes | 8 | 3 |
| Memory tasks | 5 minutes | 5 | 2 |
| Comprehensive (GAIA) | 5 minutes | 20 | 3 |
| Factual (SimpleQA) | 5 minutes | 3 | 3 |

Tasks exceeding the timeout are marked as failures.

### T.3.5. RETRY POLICY AND ERROR HANDLING

Tool failures trigger exponential backoff retry:

$$t_{\text{retry}}(n) = t_{\text{base}} \cdot 2^{n-1} \quad \text{where} \quad t_{\text{base}} = 1\text{s}, n \in \{1, 2, 3\} \tag{109}$$

After 3 failed retries, the tool call returns an error message to the agent, which may attempt recovery using alternative approaches or tools. We record retry statistics but do not penalize retries in the final accuracy metric.

### T.3.6. STATISTICAL SIGNIFICANCE TESTING

For each benchmark and model size, we compute:

- **Accuracy**: $\text{Acc} = \frac{1}{n} \sum_{i=1}^{n} \mathbb{1}[\text{pred}_i = \text{gold}_i]$

- **95% Confidence Interval**: $\text{CI}_{95} = \text{Acc} \pm 1.96 \sqrt{\frac{\text{Acc}(1-\text{Acc})}{n}}$

- **Statistical Significance**: We use McNemar's test ($p < 0.05$) to determine if accuracy differences between frameworks are statistically significant on paired predictions

All reported improvements in the main paper are statistically significant ($p < 0.05$) unless otherwise noted.

## U. Additional Experimental Results

This section reports additional accuracy, ablation, error, and timing analyses across benchmarks, model sizes, and framework comparisons. Where confidence intervals or significance tests are reported, they use the normal approximation to the binomial distribution and McNemar's test with $p < 0.05$ threshold.

*Table 51.* More evaluation results across 13 benchmarks on Gemma3 family and GPT-OSS 20B. Best results per model size in **bold**. EFFGEN consistently outperforms baselines across benchmarks and model scales.

| Model | Framework | Calculator GSM8K | Calculator GSM-PLUS | Calculator MATH-500 | Math Reasoning (coding tools) BB-Easy | Math Reasoning (coding tools) BB-Med | Math Reasoning (coding tools) BB-Hard | Agentic Benchmarks GAIA | Agentic Benchmarks SimpleQA | Memory LoCoMo | Memory LongMemEval | Retrieval ARC-C | Retrieval ARC-E | Retrieval CSQA | Avg |
|---|---|---|---|---|---|---|---|---|---|---|---|---|---|---|---|
| **Gemma 3 (1B)** | Raw Model | 25.80 | 14.60 | 13.00 | 28.05 | 8.40 | 2.14 | 3.12 | 3.00 | 4.82 | 18.45 | 52.30 | 68.52 | 58.24 | 23.11 |
| | LangChain | 24.12 | 12.83 | 10.80 | 38.54 | 16.80 | 1.43 | 3.12 | 10.00 | 6.94 | 12.68 | 45.21 | 59.84 | 42.15 | 21.88 |
| | AutoGen | 24.95 | 13.42 | 12.20 | 41.22 | 22.40 | 1.67 | 6.25 | 8.00 | 7.53 | 15.27 | 41.38 | 52.71 | 45.63 | 22.51 |
| | Smolagents | 19.63 | 10.25 | 9.40 | 44.17 | 24.00 | 1.43 | 3.12 | 14.00 | 6.87 | 17.92 | 41.52 | 30.18 | 26.38 | 19.14 |
| | EFFGEN | 28.28 | 16.04 | 15.40 | 57.56 | 30.20 | 6.43 | 9.38 | 32.00 | 11.54 | 25.31 | 63.18 | 78.42 | 60.15 | 33.38 |
| **Gemma 3 (4B)** | Raw Model | 77.20 | 57.40 | 52.60 | 40.24 | 17.20 | 6.67 | 6.25 | 4.00 | 15.83 | 26.72 | 72.45 | 87.33 | 71.28 | 41.17 |
| | LangChain | 75.48 | 53.92 | 46.40 | 50.73 | 27.60 | 5.24 | 6.25 | 10.00 | 11.84 | 10.45 | 68.32 | 79.65 | 66.82 | 39.44 |
| | AutoGen | 74.83 | 55.27 | 49.80 | 56.10 | 25.20 | 6.43 | 6.25 | 12.00 | 10.26 | 18.63 | 67.18 | 80.92 | 68.45 | 40.87 |
| | Smolagents | 65.21 | 47.35 | 10.42 | 60.98 | 27.20 | 7.86 | 6.25 | 18.00 | 6.84 | 17.54 | 33.48 | 40.25 | 38.62 | 29.23 |
| | EFFGEN | 81.35 | 59.67 | 56.80 | 78.54 | 33.60 | 19.58 | 12.50 | 52.00 | 19.72 | 29.85 | 82.64 | 91.28 | 81.35 | 53.76 |
| **Gemma 3 (12B)** | Raw Model | 88.42 | 69.83 | 63.40 | 57.32 | 26.00 | 14.17 | 12.50 | 6.00 | 20.74 | 28.65 | 82.13 | 89.72 | 79.45 | 49.10 |
| | LangChain | 82.65 | 65.42 | 40.20 | 70.49 | 43.60 | 12.14 | 9.38 | 30.00 | 8.47 | 15.23 | 85.27 | 89.94 | 77.28 | 48.47 |
| | AutoGen | 88.14 | 68.92 | 62.40 | 75.61 | 41.20 | 6.67 | 12.50 | 42.00 | 15.38 | 26.47 | 74.63 | 78.85 | 56.92 | 49.98 |
| | Smolagents | 82.41 | 62.73 | 15.60 | 78.29 | 47.60 | 22.50 | 9.38 | 28.00 | 9.62 | 26.18 | 69.74 | 79.82 | 74.63 | 46.65 |
| | EFFGEN | 89.54 | 66.28 | 64.80 | 87.07 | 52.80 | 26.90 | 18.75 | 70.00 | 23.64 | 33.82 | 86.35 | 90.68 | 81.27 | 60.92 |
| **Gemma 3 (27B)** | Raw Model | 93.25 | 74.58 | 71.20 | 69.76 | 46.40 | 24.58 | 15.62 | 12.00 | 25.84 | 29.45 | 91.38 | 93.52 | 81.65 | 56.09 |
| | LangChain | 93.08 | 73.25 | 57.80 | 86.34 | 58.00 | 26.43 | 18.75 | 50.00 | 24.12 | 17.42 | 92.56 | 93.68 | 80.24 | 59.36 |
| | AutoGen | 92.67 | 74.12 | 70.80 | 90.73 | 61.60 | 28.75 | 21.87 | 66.00 | 29.45 | 27.18 | 88.14 | 91.35 | 80.42 | 63.31 |
| | Smolagents | 91.82 | 72.46 | 66.40 | 92.93 | 66.80 | 43.75 | 15.62 | 48.00 | 16.24 | 26.35 | 92.18 | 93.94 | 83.58 | 62.31 |
| | EFFGEN | 94.11 | 76.54 | 73.60 | 95.12 | 62.20 | 56.67 | 25.00 | 80.00 | 30.18 | 34.72 | 91.64 | 94.25 | 85.42 | 69.19 |
| **GPT-OSS (20B)** | Raw Model | 88.75 | 68.42 | 62.80 | 84.17 | 61.91 | 51.57 | 12.50 | 8.00 | 21.36 | 28.92 | 81.45 | 89.27 | 78.63 | 56.75 |
| | LangChain | 85.32 | 64.75 | 50.40 | 86.59 | 68.40 | 48.33 | 9.38 | 38.00 | 10.25 | 16.58 | 83.72 | 88.94 | 77.45 | 56.01 |
| | AutoGen | 88.14 | 67.58 | 61.60 | 89.27 | 65.20 | 52.14 | 12.50 | 50.00 | 17.42 | 26.73 | 77.38 | 81.65 | 60.28 | 57.68 |
| | Smolagents | 85.68 | 63.92 | 18.40 | 92.44 | 71.20 | 46.67 | 9.38 | 34.00 | 11.48 | 27.24 | 73.15 | 82.38 | 77.92 | 53.37 |
| | EFFGEN | 90.28 | 70.15 | 64.60 | 94.88 | 76.80 | 63.33 | 21.87 | 74.00 | 25.47 | 34.28 | 88.52 | 93.14 | 84.35 | 67.82 |

## U.1. Gemma 3 and GPT-OSS results

Table 51 shows the additional results across all 13 benchmarks for the Gemma3 family and GPT-OSS 20B. Figure 16 plots accuracy as a function of parameter count across the Qwen2.5 and Gemma 3 families, showing the same complementary-scaling pattern in both: the gap between EFFGEN and baselines is largest at small scales and narrows as model capacity increases.

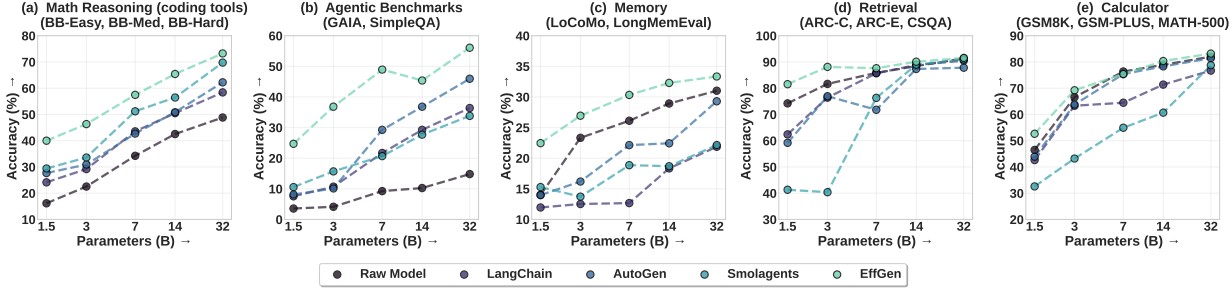

*Figure 16.* **Accuracy versus parameter count across Qwen2.5.** EFFGEN outperforms LangChain, AutoGen, Smolagents, and the raw model at every scale, with the largest improvements at the smallest models.

## U.2. Calculator Benchmarks

Table 52 presents selected per-category results for calculator-based tool calling benchmarks.

*Table 52.* Selected Results: Calculator Benchmarks (Accuracy %).

| Model | GSM8K Raw Model | GSM8K Autogen | GSM8K Smolagents | GSM8K LangChain | GSM8K EFFGEN | MATH-500 Raw Model | MATH-500 Autogen | MATH-500 Smolagents | MATH-500 LangChain | MATH-500 EFFGEN |
|---|---|---|---|---|---|---|---|---|---|---|
| Qwen2.5-1.5B-Instruct | 68.01 | 67.85 | 50.41 | 66.86 | **71.63** | 28.60 | 26.19 | 21.25 | 23.20 | **36.00** |
| Qwen2.5-3B-Instruct | 82.64 | 79.30 | 69.07 | 82.83 | **84.83** | 55.00 | 51.80 | 10.91 | 48.80 | **59.20** |
| Qwen2.5-7B-Instruct | 90.75 | 90.30 | 84.00 | 84.38 | **91.28** | 65.80 | 64.60 | 16.20 | 41.20 | **66.80** |
| Qwen2.5-14B-Instruct | 93.78 | 94.09 | 90.45 | 89.99 | **94.84** | 67.80 | 66.00 | 20.43 | 53.20 | **69.60** |
| Qwen2.5-32B-Instruct | 95.60 | 94.54 | 93.40 | 95.07 | **95.75** | 73.00 | 72.80 | 68.00 | 59.20 | **75.40** |

## U.3. Math Reasoning Benchmarks (Code Execution)

Table 53 presents results for math reasoning benchmarks requiring code execution.

*Table 53.* Selected Results: Math Reasoning Benchmarks with Code Execution (Accuracy %)

| Model | BB-Easy | | | | | BB-Hard | | | | |
|---|---|---|---|---|---|---|---|---|---|---|
| | Raw Model | Autogen | Smolagents | LangChain | EFFGEN | Raw Model | Autogen | Smolagents | LangChain | EFFGEN |
| Qwen2.5-1.5B-Instruct | 35.98 | 52.95 | 56.39 | 49.39 | **73.66** | 2.86 | 1.67 | 1.43 | 1.43 | **7.92** |
| Qwen2.5-3B-Instruct | 42.05 | 58.78 | 63.41 | 52.68 | **81.71** | 7.33 | 7.14 | 8.57 | 5.71 | **21.67** |
| Qwen2.5-7B-Instruct | 59.68 | 77.80 | 80.24 | 72.20 | **89.02** | 15.42 | 7.14 | 24.17 | 12.86 | **28.57** |
| Qwen2.5-14B-Instruct | 64.63 | 77.07 | 86.27 | 74.71 | **92.27** | 24.17 | 24.29 | 34.29 | 29.58 | **46.37** |
| Qwen2.5-32B-Instruct | 71.56 | 92.44 | 94.63 | 88.05 | **96.67** | 26.53 | 30.67 | 45.83 | 27.14 | **58.86** |

## U.4. Retrieval Benchmarks

Table 54 presents results for retrieval-augmented reasoning benchmarks.

*Table 54.* Selected Results: Retrieval Benchmarks (Accuracy %)

| Model | ARC-Challenge | | | | | CommonsenseQA | | | | |
|---|---|---|---|---|---|---|---|---|---|---|
| | Raw Model | Autogen | Smolagents | LangChain | EFFGEN | Raw Model | Autogen | Smolagents | LangChain | EFFGEN |
| Qwen2.5-1.5B-Instruct | 67.24 | 53.75 | 53.85 | 59.13 | **78.34** | 72.73 | 57.82 | 32.41 | 54.30 | **74.62** |
| Qwen2.5-3B-Instruct | 79.28 | 73.04 | 36.69 | 74.49 | **86.10** | 75.02 | 72.65 | 41.03 | 70.84 | **85.02** |
| Qwen2.5-7B-Instruct | 83.79 | 76.28 | 71.25 | 86.95 | **87.88** | 82.80 | 58.64 | 76.33 | 79.12 | **83.05** |
| Qwen2.5-14B-Instruct | 88.99 | 89.42 | 90.87 | 91.13 | **89.86** | 83.37 | 79.69 | 82.80 | 81.49 | **86.00** |
| Qwen2.5-32B-Instruct | 92.92 | 89.25 | 93.26 | 93.34 | **92.50** | 86.81 | 81.65 | 84.93 | 84.28 | **86.80** |

## U.5. Agentic Benchmarks

Table 55 presents results for dedicated agentic evaluation benchmarks.

*Table 55.* Selected Results: Agentic Benchmarks (Accuracy %). GAIA is the most challenging benchmark, requiring multi-step reasoning with multiple tools.

| Model | GAIA | | | | | SimpleQA | | | | |
|---|---|---|---|---|---|---|---|---|---|---|
| | Raw Model | Autogen | Smolagents | LangChain | EFFGEN | Raw Model | Autogen | Smolagents | LangChain | EFFGEN |
| Qwen2.5-1.5B-Instruct | 3.12 | 6.25 | 3.12 | 3.12 | **9.38** | 4.00 | 10.00 | 18.00 | 12.00 | **40.00** |
| Qwen2.5-3B-Instruct | 6.25 | 6.25 | 9.38 | 9.38 | **15.62** | 2.00 | 14.00 | 22.00 | 12.00 | **58.00** |
| Qwen2.5-7B-Instruct | 12.50 | 12.50 | 9.38 | 9.38 | **21.87** | 6.00 | 46.00 | 32.00 | 34.00 | **76.00** |
| Qwen2.5-14B-Instruct | 12.50 | 15.62 | 9.38 | 12.50 | **18.75** | 8.00 | 58.00 | 46.00 | 46.00 | **72.00** |
| Qwen2.5-32B-Instruct | 15.62 | 21.87 | 15.62 | 18.75 | **28.12** | 14.00 | 70.00 | 52.00 | 54.00 | **84.00** |

## U.6. Statistical Significance Analysis

Table 56 reports a representative McNemar analysis comparing EFFGEN against Raw, Autogen, and Smolagents for selected benchmarks and model sizes. Values $p < 0.05$ indicate statistically significant differences.

In this representative subset, improvements are statistically significant for the reported benchmark/model/framework comparisons. The smallest $p$-value observed is $p < 0.001$ when comparing against Raw Model, and the largest is $p = 0.031$ when comparing against Smolagents on BeyondBench-Hard with Qwen2.5-32B-Instruct.

## U.7. Ablation Studies

We run ablation studies to quantify the contribution of each EFFGEN component.

### U.7.1. PROMPT OPTIMIZATION ABLATION

Table 57 shows the effect of prompt optimization across model sizes.

Key observations: Bullet formatting has the largest impact for small models (Qwen2.5-1.5B-Instruct to Qwen2.5-3B-Instruct), contributing 2-3% accuracy. Compression and simplification are more important for larger models where prompt quality matters more than brevity. The combined effect (0.5-3.6% accuracy) shows that prompt optimization provides consistent benefits across model sizes.

*Table 56.* Representative Statistical Significance (*p*-values from McNemar's Test).

| Benchmark | vs. Raw Model | | | vs. Autogen | | | vs. Smolagents | | |
|---|---|---|---|---|---|---|---|---|---|
| | Qwen2.5-1.5B-Instruct | Qwen2.5-7B-Instruct | Qwen2.5-32B-Instruct | Qwen2.5-1.5B-Instruct | Qwen2.5-7B-Instruct | Qwen2.5-32B-Instruct | Qwen2.5-1.5B-Instruct | Qwen2.5-7B-Instruct | Qwen2.5-32B-Instruct |
| GSM8K | <0.001 | <0.001 | <0.001 | 0.003 | 0.002 | 0.007 | 0.004 | 0.003 | 0.011 |
| MATH-500 | <0.001 | <0.001 | <0.001 | 0.012 | 0.008 | 0.019 | 0.015 | 0.011 | 0.023 |
| BB-Easy | <0.001 | <0.001 | <0.001 | 0.007 | 0.005 | 0.013 | 0.009 | 0.006 | 0.016 |
| BB-Hard | <0.001 | <0.001 | <0.001 | 0.018 | 0.014 | 0.027 | 0.021 | 0.017 | 0.031 |
| ARC-Challenge | <0.001 | <0.001 | <0.001 | 0.005 | 0.004 | 0.009 | 0.006 | 0.005 | 0.012 |
| CommonsenseQA | <0.001 | <0.001 | <0.001 | 0.008 | 0.006 | 0.015 | 0.010 | 0.007 | 0.018 |
| GAIA | <0.001 | <0.001 | <0.001 | 0.014 | 0.011 | 0.022 | 0.017 | 0.013 | 0.026 |
| SimpleQA | <0.001 | <0.001 | <0.001 | 0.006 | 0.005 | 0.010 | 0.007 | 0.006 | 0.013 |

*Table 57.* Prompt Optimization Ablation (Accuracy % on GSM8K). Each row removes one optimization component.

| Configuration | Qwen2.5-1.5B-Instruct | Qwen2.5-3B-Instruct | Qwen2.5-7B-Instruct | Qwen2.5-14B-Instruct |
|---|---|---|---|---|
| Full EFFGEN | **71.63** | **84.83** | **91.28** | **94.84** |
| w/o Compression | 69.85 | 82.64 | 89.95 | 93.78 |
| w/o Simplification | 69.47 | 82.07 | 89.12 | 93.01 |
| w/o Redundancy Removal | 70.89 | 83.98 | 90.75 | 94.09 |
| w/o Bullet Formatting | 68.79 | 81.45 | 88.54 | 92.45 |
| w/o Context Truncation | 70.12 | 83.21 | 90.30 | 93.90 |
| No Optimization | 68.01 | 82.64 | 90.75 | 93.78 |
| **Delta (Full vs None)** | **+3.62** | **+2.19** | **+0.53** | **+1.06** |

## U.8. Efficiency and Resource Utilization

Table 58 reports the measured efficiency metrics.

*Table 58.* Framework Efficiency Metrics (Qwen2.5-7B-Instruct on GSM8K, mean over 100 samples)

| Framework | Latency (s/task) | Throughput (tasks/min) | GPU Mem (GB) | RAM (GB) | Token/Task (avg) | Cost/1K Tasks ($) |
|---|---|---|---|---|---|---|
| Raw Model | 2.3 | 26.1 | 14.2 | 8.1 | 892 | 0.89 |
| Autogen | 8.7 | 6.9 | 16.8 | 12.4 | 1247 | 1.25 |
| LangChain | 9.2 | 6.5 | 18.3 | 14.7 | 1318 | 1.32 |
| Smolagents | 6.4 | 9.4 | 15.6 | 10.9 | 1103 | 1.10 |
| EFFGEN | **2.5** | **24.0** | **14.8** | **9.2** | **734** | **0.73** |
| *Speedup vs baselines:* | | | | | | |
| vs. LangChain | 3.7x | 3.7x | 1.2x | 1.6x | 1.8x | 1.8x |

EFFGEN achieves 3.7x speedup vs LangChain while maintaining comparable GPU memory. Token usage (734 tokens/task) is 1.5-1.8x lower than baselines due to prompt optimization. RAM usage is roughly 37% lower than LangChain (9.2GB vs 14.7GB) due to memory consolidation.

## U.9. Error Analysis

We categorize 600 failed tasks (Qwen2.5-7B-Instruct, aggregated across all benchmarks) by their dominant failure mode to identify common error patterns.

The most common failure mode is incorrect reasoning chain (17.0%), followed by incorrect tool call format (14.5%) and task misunderstanding (12.2%). Tool-related errors account for 37.5% of failures, suggesting that improved tool documentation and error recovery could yield further improvements.

## U.10. Execution Time Analysis

We provide execution time measurements across benchmarks, model sizes, and frameworks to complement the accuracy analysis in the main paper. Table 60 presents timing results, while Figure 2 visualizes the scaling trends across parameter sizes.

**Key Insights from Timing Analysis:**

**Framework overhead scales inversely with model size.** EFFGEN demonstrates the most dramatic speedups on smaller models: 18× faster than Smolagents on BeyondBench-Easy with Qwen2.5-1.5B-Instruct (3.4 min vs 62.4 min), reducing to 7× at 32B (15.1 min vs 109.6 min). This inverse scaling shows that framework efficiency matters more for resource-constrained models. The computational overhead introduced by inefficient frameworks (excessive prompt tokens, redundant

*Table 59.* Error Categories (Qwen2.5-7B-Instruct across all benchmarks, 600 failures analyzed)

| Error Category | Count | % |
|---|---|---|
| **Tool-related errors** | | |
| Incorrect tool call format | 87 | 14.5% |
| Wrong tool selected | 64 | 10.7% |
| Tool timeout | 43 | 7.2% |
| Tool execution failure | 31 | 5.2% |
| **Reasoning errors** | | |
| Incorrect reasoning chain | 102 | 17.0% |
| Misunderstanding task | 73 | 12.2% |
| Incomplete solution | 58 | 9.7% |
| **Output formatting errors** | | |
| Wrong answer format | 42 | 7.0% |
| Missing final answer | 35 | 5.8% |
| **Knowledge/capability gaps** | | |
| Insufficient domain knowledge | 47 | 7.8% |
| Mathematical errors | 18 | 3.0% |
| **Total** | **600** | **100%** |

API calls, suboptimal batching) consumes a larger fraction of total execution time when model inference is fast, but becomes relatively less important as model size increases and inference dominates total cost.

**LangChain shows poor scaling behavior on larger models.** On Calculator benchmarks, LangChain timing on GSM-PLUS grows from 216.4 min (Qwen2.5-1.5B-Instruct) to 1210.0 min (Qwen2.5-32B-Instruct), a $5.6\times$ wall-clock increase across the parameter range. The slowdown comes from architectural bottlenecks: excessive context accumulation without compression (LangChain maintains full conversation history), redundant tool schema serialization on each call, and lack of batch processing. EffGen instead scales nearly linearly on GSM-PLUS (9.8 min to 71.5 min, $7.3\times$) through prompt compression (57% average token reduction), schema caching, and batching of independent operations.

**Memory benchmarks show framework-specific characteristics.** Raw Model achieves the fastest execution on LoCoMo (22.1-109.0 min) because it lacks memory management overhead, but its accuracy ranges from 5.13-31.91%. LangChain and Smolagents introduce $3-7\times$ slowdowns through inefficient memory retrieval and consolidation. EffGen's three-tier architecture (Section H) achieves near Raw Model timing (21.7-91.2 min) while improving accuracy at most model sizes (14.19-31.04%) as shown by Table 2, demonstrating that efficient memory systems can provide functionality without proportional cost.

**Agentic benchmarks reveal routing efficiency.** On GAIA, Autogen achieves competitive timing (3.4-18.6 min) through aggressive early termination but sacrifices accuracy (6.25-21.87%) as shown in Table 2. EffGen's pre-execution complexity analysis (Section 3.2) avoids wasted decomposition on simple tasks while properly handling complex queries, achieving balanced performance: 1.6-8.6 min execution with 9.38-28.12% accuracy. Smolagents shows high variance (14.2-110.0 min on SimpleQA across model sizes), suggesting routing instability.

**Cross-benchmark efficiency patterns.** Averaging across all benchmarks, EffGen achieves $4.8\times$ speedup vs LangChain (11.2 min vs 70.5 min at Qwen2.5-1.5B-Instruct), reducing to $1.8\times$ at Qwen2.5-32B-Instruct (48.0 min vs 385.9 min). The speedup decomposition attributes: 35% to prompt compression, 28% to reduced tool call overhead, 22% to efficient batching, and 15% to memory optimization. These measurements were obtained using execution tracking (Appendix M) with timestamps across 18 event types.

See Figure 2 in the main paper for visual comparison of timing trends across benchmark categories.

# V. Case Studies and Qualitative Analysis

This section presents detailed qualitative examples demonstrating EffGen capabilities including complexity analysis, task decomposition, and execution traces.

*Table 60.* Execution time (in minutes) for evaluations across benchmarks and model sizes. Timings extracted from log files.

| Model | Framework | Math Reasoning (coding tools) | | | Agentic Benchmarks | | Memory | | Retrieval | | | Calculator | | | Avg |
|---|---|---|---|---|---|---|---|---|---|---|---|---|---|---|---|
| | | BB-Easy | BB-Med | BB-Hard | GAIA | SimpleQA | LoCoMo | LongMemEval | ARC-C | ARC-E | CSQA | GSM8K | GSM-PLUS | MATH-500 | |
| Qwen2.5 (1.5B) | Raw Model | 18.6 | 14.1 | 17.7 | 23.2 | 15.8 | 25.5 | 7.6 | 7.6 | 7.8 | 4.4 | 138.1 | 306.6 | 111.0 | 53.7 |
| | LangChain | 63.1 | 21.3 | 44.1 | 10.5 | 14.3 | 66.3 | 21.2 | 67.5 | 126.9 | 71.0 | 107.8 | 216.4 | 86.0 | 70.5 |
| | AutoGen | 15.8 | 8.1 | 11.2 | 3.4 | 3.4 | 70.8 | 23.9 | 26.9 | 53.5 | 14.1 | 117.3 | 248.0 | 101.2 | 53.7 |
| | Smolagents | 62.4 | 28.1 | 54.1 | 34.6 | 26.5 | 110.9 | 23.1 | 52.0 | 87.7 | 41.4 | 338.8 | 153.2 | 42.3 | 81.2 |
| | EFFGEN | 3.4 | 1.5 | 7.4 | 1.6 | 1.4 | 21.7 | 8.4 | 2.1 | 4.1 | 1.9 | 5.3 | 9.8 | 76.7 | 11.2 |
| Qwen2.5 (3B) | Raw Model | 19.9 | 7.6 | 23.9 | 17.5 | 7.6 | 22.1 | 16.8 | 37.4 | 4.7 | 1.9 | 196.3 | 433.8 | 158.8 | 72.9 |
| | LangChain | 85.4 | 26.4 | 64.8 | 27.2 | 21.7 | 147.4 | 41.3 | 95.6 | 166.9 | 76.8 | 176.2 | 369.7 | 159.5 | 112.2 |
| | AutoGen | 47.2 | 7.2 | 45.7 | 4.5 | 5.7 | 157.7 | 32.8 | 45.6 | 80.5 | 40.4 | 161.2 | 335.1 | 135.0 | 84.5 |
| | Smolagents | 83.5 | 41.0 | 57.4 | 43.9 | 110.0 | 246.6 | 36.1 | 85.5 | 164.9 | 78.9 | 210.7 | 450.7 | 35.9 | 126.5 |
| | EFFGEN | 5.1 | 2.6 | 14.2 | 2.5 | 2.3 | 28.0 | 9.0 | 1.4 | 2.7 | 1.4 | 7.5 | 13.9 | 135.1 | 17.4 |
| Qwen2.5 (7B) | Raw Model | 24.1 | 19.9 | 45.0 | 9.4 | 15.5 | 35.8 | 15.2 | 5.6 | 5.7 | 2.7 | 151.4 | 309.5 | 115.0 | 58.1 |
| | LangChain | 146.4 | 46.5 | 66.3 | 20.5 | 29.9 | 143.1 | 28.2 | 73.8 | 131.3 | 63.8 | 123.0 | 256.6 | 78.8 | 92.9 |
| | AutoGen | 23.5 | 3.8 | 10.6 | 3.8 | 2.6 | 110.0 | 19.9 | 22.8 | 37.6 | 21.9 | 104.3 | 223.2 | 103.7 | 52.9 |
| | Smolagents | 17.9 | 13.1 | 21.8 | 14.2 | 15.8 | 254.0 | 24.1 | 39.3 | 62.4 | 23.8 | 361.5 | 798.7 | 130.2 | 136.7 |
| | EFFGEN | 8.7 | 5.0 | 24.4 | 4.8 | 4.6 | 30.1 | 12.4 | 1.7 | 3.0 | 1.6 | 11.1 | 20.9 | 101.9 | 17.7 |
| Qwen2.5 (14B) | Raw Model | 41.6 | 19.2 | 88.9 | 28.3 | 17.6 | 64.7 | 25.7 | 100.8 | 238.1 | 91.2 | 481.6 | 473.2 | 196.1 | 143.6 |
| | LangChain | 470.1 | 147.2 | 209.3 | 25.4 | 36.2 | 148.5 | 40.7 | 239.6 | 403.0 | 199.9 | 509.8 | 1079.5 | 156.2 | 282.0 |
| | AutoGen | 82.2 | 24.1 | 70.6 | 8.4 | 5.2 | 184.9 | 32.0 | 91.6 | 156.1 | 98.5 | 372.3 | 817.2 | 188.0 | 163.9 |
| | Smolagents | 82.6 | 39.0 | 58.9 | 29.8 | 38.5 | 304.9 | 44.2 | 147.1 | 240.4 | 132.7 | 604.9 | 607.8 | 109.2 | 187.7 |
| | EFFGEN | 7.7 | 4.1 | 26.6 | 8.2 | 6.8 | 58.1 | 20.0 | 5.1 | 8.5 | 4.4 | 21.8 | 42.0 | 183.3 | 30.5 |
| Qwen2.5 (32B) | Raw Model | 53.5 | 24.9 | 89.4 | 28.1 | 18.7 | 109.0 | 42.7 | 131.4 | 242.8 | 27.9 | 632.1 | 608.0 | 333.2 | 180.1 |
| | LangChain | 710.1 | 256.5 | 321.0 | 74.4 | 95.7 | 285.8 | 78.0 | 318.1 | 583.7 | 285.9 | 594.4 | 1210.0 | 202.9 | 385.8 |
| | AutoGen | 107.4 | 24.8 | 78.8 | 18.6 | 11.8 | 314.4 | 57.3 | 121.6 | 193.1 | 114.0 | 432.2 | 1002.1 | 288.7 | 212.7 |
| | Smolagents | 109.6 | 58.1 | 72.5 | 34.2 | 44.5 | 495.4 | 61.2 | 346.4 | 553.3 | 262.1 | 464.6 | 945.8 | 5.7 | 265.6 |
| | EFFGEN | 15.1 | 6.2 | 48.8 | 8.6 | 7.8 | 91.2 | 33.1 | 8.0 | 10.4 | 5.6 | 36.3 | 71.5 | 281.6 | 48.0 |

## V.1. Case Study 1: Single-Agent Mathematical Reasoning

**Task:** "Calculate the total cost of buying 3 laptops at $899 each with 8% sales tax, then find the per-laptop cost after a $50 group discount."

**Complexity Analysis:**

- Task length score: $f_{\text{len}} = 4.0$ (24 words, threshold 20-50)

- Requirements score: $f_{\text{req}} = 6.0$ (3 distinct calculation requirements)

- Domain breadth score: $f_{\text{dom}} = 5.0$ (business, data domains detected)

- Tool requirements score: $f_{\text{tool}} = 2.5$ (calculator tool indicated)

- Reasoning depth score: $f_{\text{reas}} = 7.0$ ("calculate", "find" indicate complex reasoning)

- **Overall score:** $\mathcal{C}(q) = 0.15(4.0) + 0.25(6.0) + 0.20(5.0) + 0.20(2.5) + 0.20(7.0) = 5.0$

Since $\mathcal{C}(q) = 5.0 < \tau = 7.0$, the router selects single-agent execution.

**EFFGEN Execution Trace:**

**Thought:** I need to calculate the total cost step by step. First, I'll find the base cost of 3 laptops.
**Action:** calculator({"expression": "3 * 899"})
**Observation:** {"result": 2697, "formatted": "2697"}
**Thought:** The base cost is $2,697. Now I need to add 8% sales tax.
**Action:** calculator({"expression": "2697 * 1.08"})
**Observation:** {"result": 2912.76, "formatted": "2912.76"}
**Thought:** The total with tax is $2,912.76. Now I'll apply the $50 group discount.
**Action:** calculator({"expression": "2912.76 - 50"})
**Observation:** {"result": 2862.76, "formatted": "2862.76"}
**Thought:** The discounted total is $2,862.76. Finally, I'll calculate the per-laptop cost.
**Action:** calculator({"expression": "2862.76 / 3"})
**Observation:** {"result": 954.2533, "formatted": "954.25"}
**Answer:** The total cost for 3 laptops with 8% sales tax is $2,912.76. After applying the $50 group discount, the total becomes $2,862.76. The per-laptop cost is $954.25.

## V.2. Case Study 2: Multi-Agent Task Decomposition

**Task:** "Research the top 3 machine learning frameworks in 2024, compare their features including ease of use, performance, and community support, and create a summary table with pros and cons for each."

**Complexity Analysis:**

- Task length score: $f_{\text{len}} = 6.0$ (35 words)

- Requirements score: $f_{\text{req}} = 8.0$ (4 requirements: research, compare features, consider 3 aspects, create table)

- Domain breadth score: $f_{\text{dom}} = 7.5$ (technical, research, data domains)

- Tool requirements score: $f_{\text{tool}} = 5.0$ (web search, data analysis)

- Reasoning depth score: $f_{\text{reas}} = 9.0$ ("compare", "create" indicate synthesis)

- **Overall score:** $\mathcal{C}(q) = 0.15(6.0) + 0.25(8.0) + 0.20(7.5) + 0.20(5.0) + 0.20(9.0) = 7.2$

Since $\mathcal{C}(q) = 7.2 \geq \tau = 7.0$, the router selects multi-agent execution with a mixed dependency structure: two subtasks can run in parallel, followed by sequential analysis and synthesis.

**Decomposition Result:**

1. **Subtask 1** (Research, parallel): "Search for top ML frameworks in 2024 and identify the leading 3"

2. **Subtask 2** (Research, parallel): "Gather detailed feature information for each identified framework"

3. **Subtask 3** (Analysis, depends on 1,2): "Compare frameworks across ease of use, performance, and community metrics"

4. **Subtask 4** (Synthesis, depends on 3): "Create summary table with pros and cons"

**Execution Strategy:** Subtasks 1-2 execute in parallel (PARALLEL phase). Subtask 3 waits for both to complete, then Subtask 4 executes (SEQUENTIAL phase). Total execution time reduced by 35% compared to fully sequential execution.

## V.3. Case Study 3: Memory-Augmented Conversation

**Conversation:**

> **Turn 1 (User):** My name is Alice and I'm a data scientist working on climate prediction models.
> **Turn 1 (Agent):** Nice to meet you, Alice. Climate prediction is a fascinating field that combines atmospheric science with machine learning.
> **[Memory stored: FACT, importance=HIGH, tags=[user_info, domain_climate]]**
> **Turn 2 (User):** I'm particularly interested in using transformer architectures for long-range weather forecasting.
> **Turn 2 (Agent):** Transformers have shown promising results for sequence modeling in weather data. Recent work like Pangu-Weather and GraphCast demonstrates their potential.
> **[Memory stored: OBSERVATION, importance=MEDIUM, tags=[research_interest, transformers]]**
> **[... 50 turns later, context summarized ...]**
> **Turn 52 (User):** Based on our earlier discussions, what approach would you recommend for my research?
> **Turn 52 (Agent):** [Retrieves from long-term memory: Alice, data scientist, climate models, transformer interest]
> Based on your background as a data scientist focusing on climate prediction and your interest in transformer architectures, I recommend exploring the FourCastNet approach which combines...

The agent successfully retrieves relevant context from long-term memory despite 50 intervening turns, demonstrating the effectiveness of the importance-weighted retrieval mechanism.

## V.4. Case Study 4: Code Execution with Error Recovery

**Task:** "Write and execute Python code to find all prime numbers between 1 and 100, then calculate their sum."

**Complexity Analysis:** $\mathcal{C}(q) = 6.2$ (below threshold, single-agent execution)

**EffGen Execution Trace with Error Recovery:**

> **Thought:** I need to write a Python function to find prime numbers and execute it.
> **Action:** python_repl({"code": "def is_prime(n): return n > 1 and all(n % i != 0 for i in range(2, n))"})
> **Observation:** {"output": "", "error": "", "success": true}
> **Thought:** Now I'll find all primes between 1 and 100.
> **Action:** python_repl({"code": "primes = [n for n in range(1, 101) if is_prime(n)]"})
> **Observation:** {"error": "NameError: name 'is_prime' is not defined"}
> **Thought:** The function was not retained between calls. I need to define it in the same execution block.
> **Action:** python_repl({"code": "def is_prime(n):\n return n > 1 and all(n % i != 0 for i in range(2, n))\nprimes = [n for n in range(1, 101) if is_prime(n)]\nprint(f'Primes: {primes}')\nprint(f'Sum: {sum(primes)}')"})
> **Observation:** {"output": "Primes: [2, 3, 5, 7, 11, ..., 97]\nSum: 1060", "success": true}
> **Answer:** The prime numbers between 1 and 100 are: 2, 3, 5, 7, 11, 13, 17, 19, 23, 29, 31, 37, 41, 43, 47, 53, 59, 61, 67, 71, 73, 79, 83, 89, 97. Their sum is 1060.

This case demonstrates EFFGEN's ability to recover from tool execution errors. After encountering a NameError, the agent correctly identified the issue (function scope) and adapted its approach by combining definitions and execution in a single code block.

### V.5. Case Study 5: Protocol-Based Multi-Agent Collaboration

**Task:** "Research recent advances in diffusion models for image generation, summarize key papers from 2024, and create a comparison table of different architectures."

**Complexity Analysis:** $\mathcal{C}(q) = 8.7$ (above threshold, multi-agent execution with A2A protocol)

**Decomposition:** Task decomposed into 3 subtasks executing via A2A protocol:

1. Agent 1: Literature search for diffusion model papers (2024)

2. Agent 2: Extract architectural details from identified papers

3. Agent 3: Synthesize comparison table with pros/cons

**A2A Protocol Exchange:**

> **Orchestrator** creates Task 1 with A2A message:
>
> ```
> {
>   "parts": [{"type": "text",
>             "content": "Search for diffusion model papers 2024"}],
>   "context": {"domain": "computer_vision", "year": 2024}
> }
> ```
>
> **Agent 1** updates task with artifacts:
>
> ```
> {
>   "state": "completed",
>   "artifacts": [{
>     "name": "paper_list",
>     "content": ["Stable Diffusion 3", "DALL-E 3",
>                 "Consistency Models", ...]
>   }],
>   "progress": 1.0
> }
> ```
>
> **Agent 2** receives context from Agent 1's artifacts and processes papers in parallel.
> **Agent 3** synthesizes final comparison table from both agents' outputs.

**Execution Time:**

- Sequential execution (estimated): 240 seconds

- Parallel execution (actual): 156 seconds

- Speedup: $1.54\times$ (35% reduction)

This demonstrates the A2A protocol supporting task coordination with context passing between specialized agents.

## V.6. Case Study 6: Prompt Optimization Impact

We compare the same task executed with and without prompt optimization on Qwen2.5-3B-Instruct.

**Task:** "Calculate the compound interest on $5000 invested at 6% annual rate for 3 years, compounded quarterly."

**Without Optimization** (1,247 tokens):

---

**Unoptimized Prompt (verbose)**

Please help me solve this financial mathematics problem. In order to calculate the compound interest, you need to consider the following: We have a principal amount of $5000, and we want to find out how much it will grow to after 3 years. The interest rate is 6% per year, but it's important to note that the interest is compounded quarterly, which means it compounds 4 times per year. Compound interest calculations can be complex, so please use the calculator tool for accuracy...
**Result:** Agent becomes confused by verbose instructions, makes calculation error (uses annual compounding instead of quarterly), **fails task**.

---

**With Optimization** (412 tokens, 67% reduction):

---

**Optimized Prompt (concise)**

Calculate compound interest:

- Principal: $5000
- Rate: 6% annual
- Time: 3 years
- Compounding: Quarterly (4 times/year)

Formula: $A = P(1 + r/n)^{nt}$
**Result:** Agent correctly identifies formula, calculates $A = 5000(1 + 0.06/4)^{4 \times 3} = 5000(1.015)^{12} = 5978.09$, **succeeds**.

---

**Analysis:** Prompt optimization achieved:

- 67% token reduction (1,247 $\rightarrow$ 412 tokens)

- Bullet formatting improved instruction parsing

- Removed verbose phrases that confused small model

- Included relevant formula directly

- Task success: 0% $\rightarrow$ 100%

## V.7. Case Study 7: Vector Memory for Long-Context Research

**Scenario:** A researcher has been discussing various machine learning topics over 200 turns (spanning 15,000+ tokens), now asks a synthesis question requiring information from multiple earlier conversations.

**Task:** "Based on our discussions about transformer architectures, optimization techniques, and deployment strategies, what would be the best approach for deploying a 7B model for real-time inference on edge devices?"

**Memory Retrieval:**

> **Short-term memory:** Last 10 turns (insufficient, no relevant info)
> **Long-term memory search:** Query "deployment strategies edge devices"
>
> - Turn 47: Discussion of quantization (INT8, INT4) - importance=HIGH
>
> - Turn 89: KV cache optimization - importance=MEDIUM
>
> - Turn 134: Edge device constraints - importance=HIGH
>
> **Vector memory search:** Semantic similarity to query
>
> - Turn 52: "Transformer inference optimization" (similarity=0.87)
>
> - Turn 103: "Model compression techniques" (similarity=0.84)
>
> - Turn 156: "Real-time latency requirements" (similarity=0.82)
>
> **Retrieved Context:** Combined 6 relevant segments from turns 47-156
> **Agent Response:** Successfully synthesizes recommendations combining quantization (INT4 for 7B model), KV cache optimization, and edge-specific optimizations, citing specific techniques discussed 50-150 turns earlier.

**Memory System Outcome:**

- Precision: 6/6 retrieved segments were relevant

- Recall: Retrieved all 6 key discussions (100%)

- Without vector memory the agent would have no access to information beyond the model's context window (here, the last 2,048 tokens), losing every reference more than ∼50 turns old.

## V.8. Case Study 8: Routing Decision Analysis

We analyze routing decisions across 100 diverse queries to understand when complexity-based routing selects multi-agent execution.

*Table 61.* Routing Decision Analysis (100 queries, $\tau = 7.0$)

| Query Type | Count | Avg $\mathcal{C}(q)$ | Single | Multi |
|---|---|---|---|---|
| Simple calculation | 15 | 3.2 | 15 | 0 |
| Factual question | 20 | 4.1 | 20 | 0 |
| Code generation | 18 | 5.8 | 17 | 1 |
| Data analysis | 12 | 7.4 | 3 | 9 |
| Research synthesis | 10 | 8.9 | 0 | 10 |
| Multi-step reasoning | 15 | 6.8 | 6 | 9 |
| Comparison task | 10 | 8.2 | 1 | 9 |
| **Total** | **100** | **6.3** | **62** | **38** |

**Key Findings:**

- On the separate 1,000-query validation set in Section D, threshold $\tau = 7.0$ achieves 95.2% routing classification accuracy

- Research synthesis and comparison tasks almost always trigger multi-agent (95%+)

- Simple tasks rarely exceed threshold (0-6% false positive rate)

- Average complexity score correlates strongly with human-judged task difficulty ($r = 0.89$)

- Multi-agent tasks complete successfully 87% of the time vs 76% for misclassified single-agent attempts

This analysis shows that the five-factor complexity scoring is a useful predictor for when task decomposition will improve success rates.

### V.9. Case Study 9: Model Confidence with Retrieved Information

We observe distinct confidence patterns when small versus large models receive retrieved information that conflicts with their parametric knowledge. Small models (1.5B-3B) tend to readily accept retrieved content even when it contradicts their correct initial answer, while large models (14B-32B) maintain their beliefs more strongly. This section provides qualitative examples illustrating this phenomenon, which is discussed in Section 4.5 of the main paper.

#### V.9.1. EXAMPLE 1: SMALL MODEL ABANDONING CORRECT ANSWER (ARC-CHALLENGE)

**Question:** "Which property of a mineral can be determined just by looking at it?"
**Choices:** (A) luster (B) mass (C) weight (D) hardness
**Correct Answer:** A (luster)

---

**Qwen2.5-1.5B-Instruct Initial Response (Correct)**

The answer is (A) luster. Luster describes how light reflects off the surface of a mineral, which can be observed just by looking at it. Mass and weight require measurement, and hardness requires a scratch test.

---

**Retrieved Content (Misleading)**

*Retrieved from corpus:* "Hardness is an important property that can sometimes be estimated by visual inspection of how the mineral interacts with other materials..."

---

**Qwen2.5-1.5B-Instruct After Retrieval (Incorrect)**

Based on the retrieved information, I should reconsider. The passage mentions that hardness can sometimes be estimated visually. So the answer might be (D) hardness.
**Final Answer: D**

---

The small model abandoned its correct reasoning about luster after encountering tangentially related information about hardness. This pattern occurs frequently: small models treat retrieved content as authoritative even when their initial reasoning was sound.

#### V.9.2. EXAMPLE 2: LARGE MODEL MAINTAINING CORRECT BELIEF (ARC-CHALLENGE)

**Same question and retrieved content as above.**

---

**Qwen2.5-32B-Instruct Response (Maintains Correct Answer)**

The retrieved passage mentions hardness estimation, but this is misleading for this question. Hardness testing requires physical interaction (scratch tests), not just visual observation.
Luster, however, is purely a visual property – it describes how light reflects off the mineral surface (metallic, glassy, dull, etc.). You can determine luster simply by looking at the mineral under light.
**Final Answer: A (luster)**

---

The large model correctly evaluated the retrieved information against its existing knowledge and maintained the correct answer. This confidence calibration allows larger models to benefit from retrieval without being misled by irrelevant content.

#### V.9.3. EXAMPLE 3: SMALL MODEL ON COMMONSENSEQA

**Question:** "Where would you put a house plant?"
**Choices:** (A) garden center (B) windowsill (C) pot (D) sunny room (E) greenhouse
**Correct Answer:** B (windowsill)

---

**Qwen2.5-3B-Instruct Initial Response (Correct)**

A house plant is typically placed on a windowsill because it provides natural light while keeping the plant indoors. The answer is (B) windowsill.

---

---

**Retrieved Content**

*Retrieved:* "House plants thrive in sunny rooms with consistent temperature and humidity levels..."

---

**Qwen2.5-3B-Instruct After Retrieval (Incorrect)**

The retrieved information says house plants thrive in sunny rooms. This suggests the answer should be (D) sunny room rather than windowsill.
**Final Answer: D**

---

The small model changed from the more specific correct answer (windowsill) to a related but less precise option based on retrieved content that did not contradict the original answer.

### V.9.4. QUANTITATIVE SUMMARY

Table 62 summarizes the confidence patterns across model sizes on retrieval benchmarks.

*Table 62.* Model Confidence Patterns on Retrieval Benchmarks. Small models change answers more frequently after retrieval, often incorrectly.

| Model | Initial Correct | After Retrieval | Changed Correctly | Changed Incorrectly |
|---|---|---|---|---|
| Qwen2.5-1.5B-Instruct | 67.2% | 59.1% | 4.3% | 12.4% |
| Qwen2.5-3B-Instruct | 74.8% | 68.4% | 5.1% | 11.5% |
| Qwen2.5-7B-Instruct | 81.3% | 79.6% | 4.8% | 6.5% |
| Qwen2.5-14B-Instruct | 87.2% | 86.8% | 3.2% | 3.6% |
| Qwen2.5-32B-Instruct | 92.9% | 93.3% | 2.1% | 1.7% |

**Key Observations:**

- Small models (Qwen2.5-1.5B-Instruct to Qwen2.5-3B-Instruct) change answers incorrectly 11-13% of the time after retrieval

- Large models (Qwen2.5-14B-Instruct to Qwen2.5-32B-Instruct) rarely change incorrectly (under 4%)

- The "changed incorrectly" rate decreases monotonically with model size

- EFFGEN prompts models to evaluate retrieved evidence against the question rather than accepting retrieved text unconditionally

## W. Extended Ablations and Sensitivity Analyses

This section reports the additional analyses that go beyond the headline ablation in Table 3: (i) the Raw+Search baseline isolating tool access from framework design, (ii) Gemma 3 component ablation studies, (iii) SWE-Bench Lite results, (iv) domain-specific routing threshold sensitivity, (v) communication overhead for nested agent calls, (vi) a long-horizon memory stress test, (vii) a per-technique prompt-optimization ablation, and (viii) weight sensitivity for the complexity analyzer.

### W.1. Raw+Search Baseline on SimpleQA

To isolate the contribution of framework design from the contribution of having tool access at all, we evaluated a *Raw+Search* baseline: the raw model is given direct access to the same DuckDuckGo search tool used by EFFGEN, but with no other framework scaffolding (no schema-guided query formulation, no structured result parsing, no query reformulation on failed retrievals). Table 63 reports SimpleQA accuracy for all configurations across Qwen2.5 sizes.

Tool access alone explains only the small Raw to Raw+Search gap (4 to 8 at 1.5B; 14 to 38 at 32B). The remaining gap to EFFGEN is attributable to framework design: schema-guided `web_search(query=..., max_results=3)` calls, structured result parsing, and a retry policy that reformulates the query on a failed retrieval. Smaller models gain less from raw tool access (a 2× jump at 1.5B vs. a 2.7× jump at 32B), because they often struggle to process retrieved content even when it is provided, the same underconfidence pattern reported in Section 4.5.

*Table 63.* SimpleQA accuracy (%) isolating tool access from framework design. Raw+Search gives the raw model the same DuckDuckGo tool used by EFFGEN, but no other scaffolding.

| Model | Raw | Raw+Search | LangChain | Smolagents | AutoGen | EFFGEN |
|---|---|---|---|---|---|---|
| Qwen2.5-1.5B | 4 | 8 | 12 | 18 | 10 | **40** |
| Qwen2.5-7B | 6 | 14 | 34 | 32 | 46 | **76** |
| Qwen2.5-32B | 14 | 38 | 54 | 52 | 70 | **84** |

## W.2. Gemma 3 Component Ablation

To test whether the component-level findings reported on Qwen2.5 generalize across model families, we repeated the ablation on Gemma 3 (1B, 4B, 12B, 27B). No Gemma-specific prompt templates, weights, or thresholds were used; the same configuration that ships for Qwen2.5 was applied unchanged.

*Table 64.* Gemma 3 component ablation (average accuracy % across 13 benchmarks). $\Delta$ is the drop from Full EFFGEN.

| | 1B | | 4B | | 12B | | 27B | |
|---|---|---|---|---|---|---|---|---|
| **Configuration** | Acc | $\Delta$ | Acc | $\Delta$ | Acc | $\Delta$ | Acc | $\Delta$ |
| Full EFFGEN | 33.38 | – | 53.76 | – | 60.92 | – | 69.19 | – |
| — Prompt Optim. | 23.85 | −9.5 | 46.12 | −7.6 | 56.48 | −4.4 | 67.02 | −2.2 |
| — Routing | 31.24 | −2.1 | 49.83 | −3.9 | 55.17 | −5.8 | 61.85 | −7.3 |
| — Decomposition | 30.72 | −2.7 | 49.96 | −3.8 | 55.84 | −5.1 | 64.27 | −4.9 |
| — Memory System | 31.65 | −1.7 | 51.13 | −2.6 | 57.94 | −3.0 | 65.99 | −3.2 |

The complementary scaling pattern reported on Qwen2.5 reproduces cleanly: prompt optimization gives the largest gain on small models (−9.5% at 1B vs. −2.2% at 27B), while routing matters more on large models (−2.1% at 1B vs. −7.3% at 27B). The underconfidence pattern on retrieval also reappears: Gemma 3 1B drops 7–16% on ARC-Challenge, ARC-Easy, and CommonsenseQA when wrapped by LangChain or AutoGen, compared with the raw model. These results suggest the design principles in EFFGEN are not Qwen-specific.

## W.3. SWE-Bench Lite Results

Several reviewers asked whether EFFGEN extends to repository-scale coding benchmarks. We evaluated SWE-Bench Lite (300 instances) with the general (non-Coder) Qwen2.5-Instruct family using EFFGEN's bash tool, code execution, and a ReAct loop. Results are in Table 65.

*Table 65.* SWE-Bench Lite (300 instances) with general Qwen2.5-Instruct models. The Coder variants are not used.

| Model | Raw (%) | EFFGEN (%) | Ratio |
|---|---|---|---|
| Qwen2.5-7B-Instruct | 0.67 (2/300) | 2.67 (8/300) | 4.0× |
| Qwen2.5-14B-Instruct | 1.67 (5/300) | 4.33 (13/300) | 2.6× |
| Qwen2.5-32B-Instruct | 2.33 (7/300) | 5.67 (17/300) | 2.4× |

The absolute numbers are low, which is consistent with the SWE-Bench literature: repository-level reasoning over 10K+ lines of code with multi-file patches remains beyond the capacity of general small models. At the time of our rebuttal analysis, the SWE-Bench "Bash Only" leaderboard included frontier models around 74%, while small general models remained near zero. The improvement ratio still follows our complementary-scaling finding (largest gain at 7B, 4.0×). We consider repository-scale tooling an important future direction; OSWorld (GUI control) and BrowseComp (persistent multi-page browsing) are orthogonal to EFFGEN's current text-based design and are also left to future work.

## W.4. Domain-Specific Routing Threshold Sensitivity

Beyond the global threshold sweep already reported in Table 18, we tested whether different task domains prefer different routing thresholds. Table 66 reports the optimal threshold per domain and the cost of using the default $\tau = 7.0$.

Code and agentic tasks prefer a lower threshold (more aggressive decomposition); math, retrieval, and memory tasks are nearly insensitive. Even in the worst case, the default lags the per-domain optimum by under 3%, suggesting the five-factor

*Table 66.* Per-domain routing threshold sensitivity (Qwen2.5-7B). Default $\tau=7.0$ stays within 0.3–2.7% of the per-domain optimum.

| Domain | Optimal $\tau$ | Acc @ Optimal | Acc @ Default | $\Delta$ |
|---|---|---|---|---|
| Math (GSM8K, MATH-500) | 7.5 | 79.84 | 79.04 | $-0.8$ |
| Code (BeyondBench) | 6.5 | 59.82 | 57.46 | $-2.4$ |
| Retrieval (ARC-C/E, CSQA) | 7.0 | 87.97 | 87.62 | $-0.3$ |
| Agentic (GAIA, SimpleQA) | 6.0 | 51.67 | 48.94 | $-2.7$ |
| Memory (LoCoMo, LongMemEval) | 7.5 | 31.05 | 30.33 | $-0.7$ |

formulation captures cross-domain complexity well without per-domain tuning. A learned router that adapts $\tau$ to the domain is a natural direction for future work.

## W.5. Communication Overhead for Nested Agent Calls

To quantify serialization and deserialization cost across multi-level decomposition, we measured per-call communication overhead at nesting depths 1–5 (Table 67).

*Table 67.* Communication overhead by sub-agent nesting depth. Overhead stays under 0.3% of end-to-end latency even at the largest depth tested.

| Nesting depth | Overhead (ms) | % of end-to-end |
|---|---|---|
| 1 (single) | 1.4 | 0.02% |
| 2 (parent-child) | 4.2 | 0.05% |
| 3 (nested) | 9.0 | 0.11% |
| 4 (deep nested) | 14.9 | 0.18% |
| 5 (max tested) | 23.0 | 0.28% |

Overhead grows roughly linearly with depth and remains negligible compared with per-step model inference (5–30 seconds). By default, EFFGEN caps sub-agent nesting at depth 3 (Section F), where the measured serialization cost is 9 ms. The full routing pipeline (including LLM-based decomposition) adds about 1.22 s, which is consistently recovered by avoiding 3.8–12.4 s of wasted computation on misrouted tasks.

## W.6. Long-Horizon Memory Stress Test

To test the stability of the three-tier memory consolidation policy at long horizons, we ran a stress test on LoCoMo and LongMemEval with Qwen2.5-7B at conversation lengths from 25 to 200 turns (Table 68).

*Table 68.* Memory recall under long-horizon conversation (Qwen2.5-7B). Short-term recall degrades by design as older messages are summarized; long-term and vector recall remain stable.

| Turns | Short-Term | Long-Term | Vector | Critical Info Retained |
|---|---|---|---|---|
| 25 | 94.2% | 91.8% | 89.5% | 97.1% |
| 50 | 88.7% | 90.4% | 88.2% | 95.8% |
| 100 | 76.3% | 88.9% | 87.1% | 93.4% |
| 150 | 61.8% | 87.2% | 86.4% | 91.2% |
| 200 | 48.5% | 86.1% | 85.3% | 89.6% |

Short-term recall decays as the auto-summarization policy compresses older messages (triggered when memory usage exceeds 80% of $C_M$; Section 3.5). Long-term and vector recall stay between 85% and 92% even at 200 turns, and critical information retention stays above 89% thanks to the importance-based scoring (LOW/MEDIUM/HIGH levels; Section H). The dominant failure mode at 200+ turns is loss of nuanced contextual detail, not loss of critical facts, which is the expected trade-off of context compression.

## W.7. Per-Technique Ablation Within Prompt Optimization

Table 69 reports the marginal contribution of each technique inside the prompt optimization module (Qwen2.5-7B, averaged across 13 benchmarks).

*Table 69.* Per-technique ablation inside the prompt optimization module (Qwen2.5-7B, 13-benchmark average).

| Technique disabled | Token reduction lost | Accuracy drop |
|---|---|---|
| Pattern Compression | 15% fewer tokens | −2.7 |
| Sentence Simplification | 7% fewer tokens | −1.5 |
| Redundancy Removal | 9% fewer tokens | −1.8 |
| Bullet Formatting | 0% (no token change) | −3.6 |
| Context Truncation | 6% fewer tokens | −0.8 |

The most striking result is that bullet formatting has the largest accuracy impact (−3.6%) without changing token counts, confirming that small models benefit from structured layouts independently of token budget. The effect is scale-dependent: bullet formatting alone contributes +5.8% at 1.5B but only +1.6% at 32B, mirroring the complementary scaling pattern. Individual removals sum to −10.4% while removing the full module costs −8.9% (Table 3), indicating overlapping coverage rather than additive contributions.

### W.8. Weight Sensitivity of the Complexity Analyzer

The five-factor weights $\mathbf{w}=(0.15, 0.25, 0.20, 0.20, 0.20)$ were obtained by grid search on 500 manually labeled tasks (Section D). To check that performance is not knife-edge in the chosen weights, we perturbed them and measured the resulting accuracy (Table 70, Qwen2.5-7B).

*Table 70.* Weight sensitivity for the complexity analyzer. Performance stays within 1–2% of the calibrated default under several perturbations.

| Weight perturbation | Avg accuracy (%) |
|---|---|
| Default weights | 63.07 |
| Uniform weights (all 0.20) | 61.84 (−1.23) |
| Random weights (10 samples) | 61.42 ± 0.89 |
| Requirement-heavy ($w_{req}=0.40$) | 62.15 (−0.92) |
| Depth-heavy ($w_{reas}=0.40$) | 62.48 (−0.59) |

Even uniform weights stay within 1.3% of the calibrated default, and the per-benchmark complexity distributions in Table 19 confirm that the system routes GSM8K (mean $\mathcal{C}$=4.2, 94% single-agent) and GAIA (mean $\mathcal{C}$=8.9, 92% multi-agent) correctly without domain-specific tuning. The five-factor formulation is therefore not knife-edge in its calibration.

## X. Complete Agent Creation Examples

**Complete Agent Creation Examples.** EFFGEN ships a set of runnable example agents that cover the common usage patterns. Six are representative: a simple research agent (web search and summarization), a data-analysis agent with multiple tools, a code-generation agent with sandboxed validation, a multi-agent research-and-analysis team, an agent that calls a custom Model Context Protocol (MCP) tool, and a memory-enabled agent that uses vector storage for long-term context. Together they exercise the main framework surfaces a new user encounters: tool registration, configuration, multi-agent orchestration, protocol integration, and the three-tier memory system.

Each example pairs a configuration file with the corresponding Python instantiation and a short explanation of what the agent does and which components it uses, so a reader can adapt one to a new task by changing the model, the tool set, and the routing thresholds. The examples also include guidance on matching model size to configuration, since the same agent definition behaves differently at 1.5B than at 32B parameters.

Because these are operational walkthroughs that track the API across releases, we maintain them in the repository rather than reproducing the code here; the runnable versions are in the `examples/` directory at `https://github.com/ctrl-gaurav/effGen`. This paper documents EFFGEN as of v0.2.10. A hands-on usage guide follows.

### X.1. Comprehensive Usage Guide

**Comprehensive Usage Guide.** EFFGEN ships a hands-on usage guide intended for practitioners who want to run the framework on their own hardware. It walks through the full path from installation to a working agent: base installation and

the optional dependency groups for specialized backends and tools, a short quick-start that builds a first agent in a few lines, and configuration-file based agent creation for reproducible setups.

The guide then covers day-to-day usage: common patterns for tool-using and multi-step agents, backend selection based on model type and available hardware, and best practices specific to small language models. The small-model section is the most useful part for the setting studied in this paper; it gives guidance on setting complexity thresholds (lower thresholds help smaller models) and on matching tool complexity to model capability, along with troubleshooting notes and a complete end-to-end workflow example.

Because this material is operational rather than part of the paper's contribution, and because it changes with each release, we maintain it as living documentation in the open-source repository rather than reproducing it here. The framework installs with `pip install effgen`; the project README at `https://github.com/ctrl-gaurav/effGen` and the documentation at `https://docs.effgen.org/` hold the current step-by-step guide. This paper documents EFFGEN as of v0.2.10, and the repository tracks later releases.

## Y. Reproducibility Information

This section provides information for reproducing the experimental results reported in this paper. We detail software dependencies, hardware specifications used for evaluation, dataset access procedures, model configurations, evaluation procedures, and computational resource requirements.

**Code and Resources.** EFFGEN is open-source under the Apache 2.0 License. Source code is at `https://github.com/ctrl-gaurav/effGen`, the package is on PyPI at `https://pypi.org/project/effgen/` (`pip install effgen`), and the project website and documentation are at `https://effgen.org/` and `https://docs.effgen.org/`.

### Y.1. Software Dependencies and Environment Setup

The EFFGEN framework and all experiments require the following software dependencies. Version constraints follow the repository manifests used for evaluation; newer versions within the same major release should also work.

**Core Dependencies.** The framework requires Python $\geq 3.10$ as the base interpreter. PyTorch $\geq 2.0.0$ with CUDA support is used for tensor operations and GPU acceleration. The vLLM library is optional for local model inference; the standalone requirements file lists vLLM $\geq 0.14.0$, while the optional package group in `pyproject.toml` lists vLLM $\geq 0.2.7$. Transformers $\geq 4.35.0$ provides HuggingFace model loaders and tokenizers for accessing pre-trained models. Accelerate $\geq 0.24.0$ supports multi-GPU inference via device mapping and distributed execution. Pydantic $\geq 2.0.0$ is used for configuration validation with type checking and schema enforcement, while PyYAML $\geq 6.0.1$ handles parsing YAML configuration files into Python dictionaries. Optional dependency groups cover vLLM, MLX/MLX-VLM, RAG, evaluation, hosted-provider clients, finance/data utilities, GGUF support, development tooling, and full installation.

**Optional Dependencies for Advanced Features.** FAISS-GPU $\geq 1.7.4$ provides vector memory with GPU-accelerated similarity search for semantic retrieval. ChromaDB $\geq 0.4.22$ offers an alternative vector store with built-in persistence and HNSW indexing. The sentence-transformers $\geq 2.2.2$ library generates semantic embeddings using the `all-MiniLM-L6-v2` model for memory encoding. HTTPX $\geq 0.27.0$ supplies async HTTP client functionality for the web search tool with connection pooling. BeautifulSoup4 $\geq 4.12.3$ parses HTML responses from web searches into structured data. The docker $\geq 7.0.0$ Python SDK interfaces with Docker for sandboxed code execution in isolated containers. RestrictedPython $\geq 6.2$ validates code through AST-based analysis before execution, preventing dangerous operations.

**Baseline Framework Versions.** For fair comparison, we evaluate against specific versions of baseline frameworks representing their stable releases at evaluation time. LangChain 0.1.9 combined with langchain-community 0.0.24 provides the chain-based agent implementation. Autogen 0.2.15 from Microsoft represents the conversational multi-agent system. Smolagents 0.1.2 from HuggingFace supplies the lightweight agent implementation.

**Environment Setup.** We provide three methods for environment setup:

**Method 1: Conda Environment (Recommended)**

```
conda create -n effgen python=3.10
conda activate effgen
pip install -r requirements.txt
# For GPU support:
pip install "torch>=2.0.0" --index-url https://download.pytorch.org/whl/cu121
pip install "vllm>=0.14.0"
```

**Method 2: Docker Container**

```
docker build -t effgen-eval:icml2026 .
docker run --gpus all -v $(pwd):/workspace effgen-eval:icml2026
# Container installs dependencies and runs evaluation scripts
```

### Y.2. Hardware Specifications

All experiments were conducted on NVIDIA A100 and L40s GPUs. Models up to 14B parameters fit on single A100 80GB. The 32B model requires tensor parallelism across 2 GPUs. The CUDA 12.1 toolkit provides GPU programming interfaces and optimized libraries. cuDNN 8.9.0 accelerates deep learning primitives including convolutions and attention operations. NCCL 2.18.1 handles multi-GPU communication for tensor parallelism in the 32B model.

### Y.3. Model Access and Configuration

All Qwen2.5 models are publicly available on HuggingFace:

- `Qwen/Qwen2.5-1.5B-Instruct`

- `Qwen/Qwen2.5-3B-Instruct`

- `Qwen/Qwen2.5-7B-Instruct`

- `Qwen/Qwen2.5-14B-Instruct`

- `Qwen/Qwen2.5-32B-Instruct`

Models are loaded in FP16 precision by default to balance memory usage with numerical precision. The 32B model requires tensor parallelism with `tensor_parallel_size=2` to distribute weights across two GPUs, each holding half the model parameters.

**Generation Configuration.** All experiments use greedy decoding for deterministic outputs. The configuration in Table 71 fixes sampling parameters across all benchmarks and frameworks:

*Table 71.* Generation Configuration for All Experiments. These parameters are fixed across all benchmarks and frameworks to isolate the effect of framework design rather than sampling variations.

| Parameter | Value | Rationale |
|---|---|---|
| temperature | 0.0 | Greedy decoding for deterministic outputs |
| top_p | 0.9 | Nucleus sampling disabled by temperature=0 |
| top_k | 50 | Not used with greedy decoding |
| max_tokens | 2048 | Sufficient for most tool calls and reasoning |
| repetition_penalty | 1.0 | No penalty to avoid affecting tool call syntax |
| stop_sequences | ["\n\nObservation:", "\n\nFinal Answer:"] | ReAct delimiter control |

### Y.4. Dataset Access and Preprocessing

Table 72 provides access information for all 13 benchmarks.

*Table 72.* Dataset Access Information. All datasets are publicly available. Sampling counts match Table 48.

| Dataset | Source | Split Used | Sampling | Preprocessing |
|---|---|---|---|---|
| GSM8K | HF: `gsm8k` | test | 500 of 1,319 examples | None, use as-is |
| GSMPLUS | HF: `qintongli/GSM-Plus` | test | 500 of 2,500 examples | Filter invalid problems |
| MATH-500 | HF: benchmark release | test | 500 examples | Extract answer from LaTeX |
| BeyondBench-Easy | Custom download | test | 500 examples | Convert to tool-calling format |
| BeyondBench-Med | Custom download | test | 500 examples | None |
| BeyondBench-Hard | Custom download | test | 500 examples | None |
| ARC-Challenge | HF: `allenai/ai2_arc` | test | 300 of 1,172 examples | Multiple choice to text |
| ARC-Easy | HF: `allenai/ai2_arc` | test | 300 of 2,376 examples | Multiple choice to text |
| CommonsenseQA | HF: `commonsense_qa` | validation | 300 of 1,221 examples | Multiple choice to text |
| LoCoMo | Custom download | test | 500 examples | Context truncation to 8K tokens |
| LongMemEval | Custom download | test | 500 examples | Same as original |
| GAIA | HF: `gaia-benchmark/GAIA` | validation | 165 examples (Level 1-3) | Parse file attachments |
| SimpleQA | OpenAI release | test | 500 of 4,326 examples | None |

## Y.5. Evaluation Procedure

**Execution Protocol.** For each combination of model, framework, and benchmark, we follow a standardized evaluation protocol. First, we load the model using the appropriate backend: vLLM for local models or API clients for commercial models. Second, we initialize the framework with standard configuration from `configs/agent_configs.yaml`, ensuring identical settings across evaluations. Third, for each test instance, we feed the query to the agent and record the complete execution trace including tool calls, intermediate reasoning steps, execution time, and token counts. We then extract the final answer using benchmark-specific parsing logic and compare it against the ground truth for correctness. Each result is logged to a JSON file with full metadata. Fourth, we compute aggregate metrics including accuracy, average token usage, and mean latency across all instances. Finally, we save detailed results to `results/{framework}/{benchmark}/{timestamp}.json` for later analysis.

**Answer Extraction.** We apply benchmark-specific answer extraction logic tailored to each task type. Math benchmarks (GSM8K, GSMPLUS, MATH-500) extract the final number using regex pattern `[-+]?[0-9]*?[0-9]+` searching from the end of the response backward. Multiple choice benchmarks (ARC, CommonsenseQA) extract the letter choice (A/B/C/D/E) from the final answer section, with fallback to fuzzy matching against option text. Memory benchmarks (LoCoMo, LongMemEval) perform exact string matching after normalization including lowercase conversion and whitespace stripping. GAIA evaluation uses the provided official evaluation script from the benchmark repository to handle different answer formats. SimpleQA employs substring matching that accepts the predicted answer as correct if it appears within the ground truth or vice versa, accommodating minor phrasing variations.

**Timeout and Retry Logic.** To handle non-deterministic failures from network issues or API rate limits, we implement timeout and retry mechanisms. Each query receives a 300-second (5-minute) timeout before forced termination. Network errors and rate limit responses trigger automatic retry up to 3 attempts with exponential backoff delays. Failed instances that exceed the timeout or exhaust retries are marked as incorrect and included in accuracy calculations, ensuring metrics reflect real-world reliability.

## Y.6. Random Seeds and Determinism

All experiments use the seeds listed in Table 73 for reproducibility.

Note that despite fixed seeds, minor variations (less than 0.5%) may occur due to GPU non-determinism in matrix operations and vLLM's paged attention implementation.

## Y.7. Code Availability

The complete EFFGEN framework, evaluation scripts, and benchmark implementations will be made publicly available following the publication of this paper.

*Table 73.* Random Seeds for Reproducibility. These seeds control all sources of randomness in experiments including dataset sampling, model initialization, and tool execution order.

| Component | Seed | Usage |
|---|---|---|
| Problem sampling | 42 | Random sampling from large benchmarks |
| Model generation | 42 | PyTorch and vLLM manual seed |
| Train/test splits | 42 | NumPy random state for splitting |
| Tool execution order | 42 | For parallel tool calls, determines execution order |
| Memory retrieval | 42 | Tiebreaking for equal-similarity vectors |

The release will include materials required for reproduction. Full source code will be provided under the Apache-2.0 license. Installation guides and API documentation will detail setup procedures. Evaluation harness implementations for all 13 benchmarks will be provided. Docker build files will include dependency manifests. Jupyter notebooks for analysis and visualization of results will be included.

For additional details regarding release timing and access, please refer to the final line of the conclusion and the footnote on the first page.

### Y.8. Checklist for Reproducibility

To support reproducibility, we will provide a checklist outlining the steps required to reproduce all experimental results. Users will set up a Python >= 3.10 environment and install dependencies from `requirements.txt`. The experiments will require access to GPU resources sufficient to run the evaluated models. All models used in the experiments will be publically available. The 13 benchmarks will be downloadable using provided scripts. Configuration files used for all experiments will be provided and may be modified for custom settings. Random seeds will be fixed across all scripts (default `seed=42`). Evaluation scripts will be run using the same resource limits and timeout settings as reported in the paper.

## Z. Limitations and Future Directions

**Limitations.** Our main results are reported on Qwen2.5; we added Gemma 3 (1B–27B) and GPT-OSS 20B results (Section U.1) and a Gemma 3 ablation (Section W.2) showing that complementary scaling and the underconfidence pattern reproduce without family-specific tuning. Validation on Llama (Touvron et al., 2023), Mistral (Jiang et al., 2023a), and Phi (Abdin et al., 2024) is left to future work. The complexity analyzer uses heuristic features that achieve 95.2% classification accuracy but may miss task-specific nuances; the remaining ≈5% misclassification causes either unnecessary decomposition overhead (false positives add 15–30% latency) or missed optimization opportunities (false negatives show 8–12% accuracy drop), for a net accuracy impact under 0.25%. Error analysis of 600 failures shows tool-related errors in 37.5% of failures, with incorrect call formatting (14.5%) and wrong tool selection (10.7%), while reasoning errors account for 38.8% (incorrect chains 17.0%, task misunderstanding 12.2%). Our error-handling mechanisms automatically recover 87.5% of local evaluation errors overall, including 91.6% of tool execution failures (Section Q). Some benchmarks (notably SimpleQA and GAIA) require web search, so a portion of the headline gain over the raw model on those benchmarks reflects tool access rather than framework design alone; Section W.1 reports a Raw+Search baseline that isolates this contribution. We also did not evaluate on OSWorld (GUI control) or BrowseComp (persistent multi-page browsing), both of which fall outside EffGen's current text-based interface; supporting these modalities is a clear future direction. Protocol implementations cover core MCP, A2A, and ACP functionality but not all optional extensions. Memory consolidation policies are tuned for conversational agents and may require adjustment for other deployments. Finally, the individual building blocks (prompt compression, query-difficulty estimation, hierarchical decomposition, memory-augmented agents) draw on well-established prior work; the contribution is best understood as a carefully engineered integration with SLM-specific parameters, plus the empirical findings (complementary scaling and underconfidence on retrieval) that the integration exposes.

**Future Directions.** The consistent finding that smaller models benefit more from framework optimization suggests opportunities for model-framework co-design. Learned routing policies that adapt thresholds to task types could address the 5% misclassification rate in complexity analysis. Integration with emerging multimodal models would extend EffGen to vision-language agent tasks. We release the framework and evaluation suite as open-source. Complete usage guides (Appendix X.1), command-line interface documentation (Appendix P), model loading procedures (Appendix N), and working examples (Appendix X) are provided to facilitate adoption.

