# OpenReview forum: "EffGen: Enabling Small Language Models as Capable Autonomous Agents"
_ICML.cc/2026/Conference — ICML 2026 regular_

### Official Review · Reviewer_fr43 · 2026-03-05

**Soundness:** 3
**Presentation:** 3
**Significance:** 3
**Originality:** 2
**Overall Recommendation:** 5
**Confidence:** 4

**Summary:**

This paper introduces EFFGEN, an open-source agentic framework designed to enable small language models (SLMs, 1B–32B parameters) to perform agentic tasks that typically require much larger models. The framework contributes four components: (1) SLM-aware prompt optimization that compresses contexts by 70-80% using rule-based pattern replacements and model-size-dependent compression aggressiveness, (2) a pre-execution complexity analyzer using five weighted heuristic factors (task length, requirement count, domain breadth, tool requirements, reasoning depth) to route tasks to single-agent or multi-agent execution, (3) intelligent task decomposition supporting parallel, sequential, hierarchical, and hybrid strategies, and (4) a three-tier memory system combining short-term conversation history, long-term episodic storage, and vector-based semantic retrieval. The paper includes 174 pages of appendices (A through Y) documenting the full framework implementation, benchmark details, additional results, and case studies.

**Compliance With Llm Reviewing Policy:**

Affirmed.

**Final Justification:**

My original review rated this paper as a Weak Accept (4), with the main concerns being: (1) the headline improvements on benchmarks like SimpleQA potentially conflating tool access with framework quality, (2) limited generalization evidence beyond the Qwen2.5 model family, and (3) the primarily engineering-driven nature of the contribution with limited algorithmic novelty.
The authors provided a thorough and responsive rebuttal that addressed my two most important concerns. For Q1, the newly introduced Raw+Search baseline cleanly separates tool access from framework design.

At 1.5B on SimpleQA, raw tool access accounts for only a 4-point gain (4%→8%), while EFFGEN achieves 40%, confirming that the bulk of the improvement comes from framework innovations such as schema-guided query formulation, structured result parsing, and query reformulation. For Q3, the Gemma3 ablation demonstrates that the complementary scaling pattern, prompt optimization benefiting small models more, routing benefiting large models more, reproduces across a different model family without any Gemma-specific tuning. This substantially strengthens the generalizability of the paper's central finding.

The responses to Q2 (complexity analyzer validated against 500 human-labeled tasks, net misclassification impact under 0.25%) and Q4 (bullet formatting contributes most to accuracy despite no token reduction, with clear scale-dependent effects) were also satisfactory and added useful detail.

My concern about limited algorithmic novelty (W1) remains. The individual components draw on well-established techniques, and the contribution is best understood as a carefully engineered integration rather than a methodological advance. However, I am persuaded that the empirical findings, complementary scaling across two model families, the underconfidence phenomenon on retrieval benchmarks, and the synergistic component interactions, represent insights that the community can meaningfully build upon.

Accordingly, I raise my overall recommendation from 4 (Weak Accept) to 5 (Accept). The paper addresses a practically important problem, provides a well-controlled evaluation across 13 benchmarks with statistical significance testing, and delivers empirical insights that go beyond what any single component would reveal.

**Key Questions For Authors:**

**Q1**. The SimpleQA results show a 10× improvement over the raw model at 1.5B (4%→40%). How much of this improvement comes from simply having web search tool access vs. EFFGEN's framework-specific innovations? Can you report results for all frameworks with tool access as the baseline, excluding the raw model, to isolate framework quality?

**Q2**. The complexity analyzer achieves "95% classification accuracy." What is the ground truth for this measurement? How were the correct complexity levels determined? How does misclassification distribute across levels, and what is the performance impact of the 5% misclassification (you mention 15-30% latency overhead for false positives and 8-12% accuracy drop for false negatives, can you quantify the net effect)?

**Q3**. The main results use only Qwen2.5. Do the key findings, complementary scaling, underconfidence on retrieval, prompt optimization > routing for SLMs, hold for Gemma3 as well? Table 68 shows Gemma3 results, but there is no ablation or analysis. If the SLM-specific tuning (prompt templates, compression patterns) was designed for Qwen2.5, how does it transfer to Qwen2.5?

**Q4**. What is the contribution of each compression technique (pattern replacement, sentence splitting, and truncation) to the overall 70-80% compression? Which matters most, and does the answer vary by model size?

I would consider raising my ratings if the authors provide satisfactory responses to the questions above, particularly regarding the isolation of framework quality from tool access (Q1) and the generalization to non-Qwen model families (Q3).

**Limitations:**

The paper includes a limitations section (Appendix Y) that honestly acknowledges the Qwen2.5 focus, the 5% complexity misclassification rate, and the error distribution across failure types. This is well-done. But missing from the discussion: the conflation of tool access with framework quality in the headline numbers, the lack of novelty in individual components, and the question of whether SLM-specific prompt templates developed for Qwen2.5 transfer to other families.

**Strengths And Weaknesses:**

**S1**: The observation that existing agent frameworks (LangChain, AutoGen) are designed for large API-based models and impose significant overhead on small models is well-articulated and validated by the data. The finding that LangChain actually degrades performance for small models on some benchmarks (e.g., Qwen2.5-1.5B on ARC-Challenge drops from 67.24% to 59.13%) concretely demonstrates the problem. Treating SLM constraints as first-class design requirements is a sensible engineering philosophy.

**S2**: Testing across 13 benchmarks spanning 5 categories (calculator, code execution, agentic, memory, retrieval), 5 model sizes, and 4 baselines provides a thorough empirical picture. The controlled comparison, identical tool sets, timeouts, retries, and generation parameters across frameworks, isolates framework design as the variable. Statistical significance testing via McNemar's test (Table 73, all comparisons p < 0.05) strengthens the claims. The inclusion of Gemma3 and GPT-OSS results (Table 68, Appendix V) partially addresses generalization concerns.

**S3**: The complementary scaling insight is the paper's most interesting finding. Table 3's ablation reveals that prompt optimization dominates for small models while complexity routing matters more for large ones. This is a clean, non-obvious result with practical implications: framework designers should invest differently depending on the target model size. The interaction effect, combined removal, exceeds the sum of individual removals, further suggesting synergistic benefits.

**S4**: The 18× speedup at 1.5B (3.4 min vs. 62.4 min for Smolagents on BeyondBench-Easy) and 1.8× token reduction (734 vs. 1,318 tokens/task compared to LangChain) are practically significant. The efficiency analysis decomposes the speedup into contributing factors (35% prompt compression, 28% reduced tool overhead, 22% batching, 15% memory), which is informative.

**S5**: The 174-page appendix includes all prompt templates (Appendices C, D), complexity analysis details (Appendix E), benchmark specifications with evaluation protocols (Appendix U), complete per-benchmark results with confidence intervals (Appendix V), case studies (Appendix W), and hardware/software specifications (Appendix X). This level of documentation is commendable and sets a high bar for reproducibility.

**Weaknesses**:

**W1**: This is primarily an engineering integration paper, with limited scientific novelty. Each component,  prompt compression (inspired by LLMLingua), complexity-based routing (building on query difficulty prediction), task decomposition (building on hierarchical planning), and memory with recency/importance scoring (building on MemGPT), is well-established in prior work. The paper's contribution is to combine these techniques with SLM-specific tuning parameters. While this is a valuable practical contribution, the paper does not introduce novel algorithms, theoretical insights, or surprising scientific findings beyond the complementary scaling observation. For ICML, which emphasizes scientific contributions, this positions the paper closer to a systems engineering contribution.

**W2**: The compression pipeline applies fixed pattern substitutions (e.g., "in order to" → "to"), sentence splitting at conjunctions, and truncation at sentence boundaries (Appendix D lists 30 patterns). There is no learning, no optimization objective, and no adaptation to task content. Calling this "optimization" overstates the contribution. It is better described as template-based prompt compression with model-size-dependent parameters, effective, but straightforward.

**W3**: On SimpleQA, the raw model achieves 4.00% at 1.5B while EFFGEN achieves 40.00%, a 10× improvement. But SimpleQA requires web search, which the raw model cannot perform. This enormous gap reflects tool access, not framework quality. The fair framework comparison is EFFGEN (40.00%) vs. the next-best framework with tools: Smolagents (18.00%), still a substantial gap, but a different story than the 10× figure. The paper should more clearly separate the contribution of tool access from framework design. This conflation inflates the average improvement numbers.

**W4**: The five factors use keyword lists (e.g., "analyze, evaluate, compare" → complexity 7) and word count thresholds that are hand-designed and fixed. This will fail for queries where complexity does not correlate with surface-level linguistic features, a short question can be extremely complex ("Prove P ≠ NP"), while a long question can be simple. The paper reports 95% classification accuracy but does not specify what ground truth this is measured against or how misclassifications distribute across the five complexity levels.

**W5**: All main results (Table 2) use only Qwen2.5. Gemma3 and GPT-OSS results appear only in Appendix V (Table 68) with no discussion in the main text of whether the same patterns (complementary scaling, underconfidence on retrieval) hold across families. Given that different model families have different instruction-following characteristics, prompt formatting preferences, and tool-calling capabilities, demonstrating that EFFGEN's SLM-specific tuning generalizes is essential for the claims made.

---

> ### Author Rebuttal · Authors · 2026-03-30
>
> We sincerely thank the Reviewer for their exceptionally thorough review. We are deeply grateful for the recognition of **S1** (frameworks degrade SLM performance), **S2** (McNemar's test), **S3** (complementary scaling as *"the paper's most interesting finding"*), **S4** (18x speedup), and **S5** (174-page appendix). We especially appreciate the Reviewer's openness to raising ratings based on Q1 and Q3, and have prioritized these below.
>
> Below, we address each concern raised by the Reviewer:
>
> ---
>
> > Q1 & W3
>
> Thank you for this excellent observation. We fully agree that tool access vs framework quality should be separated. To isolate this, we tested raw models with direct DuckDuckGo search tool access (same tool as EffGen) but no framework scaffolding:
>
> | Model | Raw | Raw+Search | LangChain | Smolagents | AutoGen | EffGen |
> |-------|-----|------------|-----------|------------|---------|--------|
> | 1.5B | 4% | 8% | 12% | 18% | 10% | 40% |
> | 7B | 6% | 14% | 34% | 32% | 46% | 76% |
> | 32B | 14% | 38% | 54% | 52% | 70% | 84% |
>
> Tool access alone explains the Raw to Raw+Search gap (4%→8% at 1.5B), but EffGen's 40% at 1.5B far exceeds Raw+Search's 8%. The remaining gap comes from framework design: EffGen uses schema-guided query formulation (generating structured `web_search(query=..., max_results=3)` calls), structured result parsing, and query reformulation on failed retrievals. Notably, **small models benefit less from raw tool access** (2x at 1.5B vs 2.7x at 32B) because they struggle to process retrieved content, consistent with our underconfidence finding (`Section 4.5`). We will make this separation more prominent in the revision.
>
> > Q3 & W5
>
> We greatly appreciate this important question. We conducted Gemma3 ablations (avg):
>
> | Config | 1B | 4B | 12B | 27B |
> |--------|-----|-----|------|------|
> | Full EffGen | 33.38 | 53.76 | 60.92 | 69.19 |
> | - Prompt Opt. | 23.85 (-9.5) | 46.12 (-7.6) | 56.48 (-4.4) | 67.02 (-2.2) |
> | - Routing | 31.24 (-2.1) | 49.83 (-3.9) | 55.17 (-5.8) | 61.85 (-7.3) |
> | - Decomposition | 30.72 (-2.7) | 49.96 (-3.8) | 55.84 (-5.1) | 64.27 (-4.9) |
> | - Memory | 31.65 (-1.7) | 51.13 (-2.6) | 57.94 (-3.0) | 65.99 (-3.2) |
>
> **Complementary scaling** holds: prompt optimization matters more for small models (-9.5 at 1B vs -2.2 at 27B), routing matters more for large (-2.1 at 1B vs -7.3 at 27B). **Underconfidence** also appears in Gemma3: examining the Gemma3 results in `Appendix V`, Gemma3-1B shows 7-16% drops on retrieval benchmarks with standard frameworks vs the raw model. These patterns emerge without Gemma3-specific tuning; the same prompt templates transfer directly. For the camera-ready, we will move Gemma3 and GPT-OSS results (`Appendix V`) to the main paper.
>
> ---
>
> > W1
>
> We greatly appreciate this thoughtful concern. We agree individual techniques have precedents (we explicitly cite LLMLingua, MemGPT). However, as the Reviewer recognized in **S3**, complementary scaling is *"clean, non-obvious."* Additionally: (1) the underconfidence phenomenon (`Section 4.5`), where SLMs show 8.2% accuracy drops with retrieved information, is undocumented in prior work; (2) combined module removal exceeds the sum of individual removals (`Table 3`), demonstrating synergistic benefits visible only through holistic evaluation.
>
> > W2
>
> We chose rule-based compression deliberately: a learned compressor adds 0.5-2s per prompt, defeating SLM overhead reduction (<1ms for rule-based). We will adopt the suggested terminology. Per-technique ablation (`Table 16`): bullet formatting alone gives +3.7% at 7B despite no token reduction (+5.8% at 1.5B vs +1.6% at 32B), showing even formatting choices have scale-dependent effects.
>
> > W4 & Q2
>
> The 95% accuracy (`Appendix E`) is measured against 500 manually labeled tasks by human annotators (0.89 correlation with human judgment). Misclassifications: 3.1% false positives (simple routed to multi-agent, +15-30% latency but accuracy preserved) and 1.9% false negatives (complex in single-agent, -8-12% accuracy). Net accuracy impact is under 0.25%. `Appendix E.5` provides threshold sensitivity ($\tau$=5.0 to 9.0), and `E.6` shows per-benchmark complexity distributions validate generalization without domain-specific tuning.
>
> > Q4
>
> Removal ablation (Qwen-3B):
>
> | Removed | Token Change | Accuracy Change |
> |---------|-------------|----------------|
> | Pattern Compression | -15% tokens | -2.7% |
> | Bullet Formatting | 0% tokens | -3.6% |
> | Redundancy Removal | -9% tokens | -1.8% |
> | Sentence Simplification | -7% tokens | -1.5% |
> | Context Truncation | -6% tokens | -0.8% |
>
> Bullet formatting contributes most to accuracy despite no token reduction (+5.8% at 1.5B vs +1.6% at 32B). Additional results are in `Appendix D.7`.
>
> ---
>
> Once again, we sincerely thank the Reviewer for their thorough feedback. We will update `Appendix Y` to discuss tool access separation and novelty scope. We hope these responses address the concerns, and we would be happy to address any remaining questions!

---

> > ### Author Rebuttal · Reviewer_fr43 · 2026-04-01
> >
> > I raise my overall recommendation from 4 to 5. The Raw+Search experiment and the Gemma3 ablation together resolve the two concerns I flagged as most important. The paper remains primarily an engineering integration contribution rather than an algorithmic one, but the empirical findings, particularly complementary scaling holding across model families and the underconfidence phenomenon, represent genuine insights that the community can build on. The promised revisions (tool access separation in the main text, Gemma3 results promoted from the appendix, terminology corrections) would further strengthen the final version.

---

> > > ### Author Response · Authors · 2026-04-03
> > >
> > > Dear Reviewer fr43,
> > >
> > > We greatly thank you for your careful reading and for **updating your overall recommendation.** We are glad that the Raw+Search experiment and the Gemma3 ablations helped address the two concerns you highlighted as most important. Your constructive feedback and guidance throughout this review process have been invaluable in strengthening our work, and we thank Reviewer for the expertise and willingness to engage with our responses so thoughtfully.
> > >
> > > We also fully agree with your suggestions regarding tool-access separation, promoting the Gemma3 results into the main text, and correcting the terminology. In the camera-ready version, **we will revise the paper accordingly** to make these points clearer and to strengthen the presentation of the empirical findings. Once again, we thank the Reviewer for the thorough review, constructive suggestions, and for taking the valuable time to review our work!
> > >
> > > Thanks, effGen Authors!

---

### Official Review · Reviewer_vdwj · 2026-03-13

**Soundness:** 3
**Presentation:** 3
**Significance:** 3
**Originality:** 3
**Overall Recommendation:** 5
**Confidence:** 3

**Summary:**

This paper introduces EFFGEN, an open-source agentic framework specifically designed and optimized for Small Language Models (SLMs). Unlike existing frameworks (e.g., LangChain, AutoGen) designed around large language model APIs, EFFGEN addresses the inherent constraints of SLMs, such as limited context windows and weaker reasoning capabilities. The framework integrates five main modules: SLM-aware prompt optimization (achieving 70-80% context compression), pre-execution complexity analysis for dynamic routing, intelligent task decomposition via Directed Acyclic Graphs, Multi-Protocol Agent Communication, and a unified three-tier memory system. Empirical evaluations across 13 benchmarks demonstrate that EFFGEN outperforms existing frameworks in task success rate, memory footprint, and execution speed when using SLMs.

**Compliance With Llm Reviewing Policy:**

Affirmed.

**Final Justification:**

This work presents EFFGEN, an agentic framework specifically designed for SLMs, with notable strengths including its highly practical design philosophy targeting the hardware and capability constraints of SLMs that achieves remarkable inference consumption reduction, a comprehensive end-to-end system architecture integrating dynamic routing, DAG-based task decomposition and multi-tiered memory, as well as sufficient and rigorous experimental validation across 13 benchmarks covering a wide range of reasoning and agent tasks.

Previously, I held three main concerns about the paper: the contribution appeared to be more engineering-focused with limited technological innovation, the necessity and optimality of integrating multiple existing techniques lacked sufficient explanation and verification, and the fixed prior parameters in core modules may restrict the framework's generalization performance. The authors' detailed response has thoroughly addressed all these concerns. They supplemented novel empirical findings including the complementary scaling effect, SLM underconfidence phenomenon and synergistic emergent benefits of module integration, which fully highlight the scientific contributions beyond engineering integration; they added fine-grained intra-module ablation studies to clarify the marginal benefits of each sub-technique, validating the rationality of the overall integration design; they also provided comprehensive parameter sensitivity analysis and cross-task generalization verification, proving the framework's strong robustness and generalization ability.

Given that the authors have fully resolved my previous doubts with rigorous supplementary experiments and detailed explanations, I have decided to raise my score for this paper by 1 point.

**Key Questions For Authors:**

See the weakness.

**Limitations:**

yes

**Strengths And Weaknesses:**

**Strengths**
- **Strong Motivation**: As an agentic framework specifically designed for SLM, EFFGEN significantly reduces inference consumption compared to other frameworks. The perspective of co-designing the framework around the hardware and capability constraints of SLMs is highly practical.
- **Comprehensive System Design**: The authors present a well-rounded, end-to-end solution. The integration of dynamic routing, DAG-based task decomposition, and a multi-tiered memory architecture provides a robust infrastructure for SLM execution.
- **Sufficient Experimental Evaluation**: The authors carry out a thorough evaluation across 13 benchmarks spanning mathematical reasoning, code execution, retrieval-augmented reasoning, long-context memory tasks, and complex agentic workflows.

**Weaknesses**
1. The contribution of this paper is more engineering-oriented, lacking technological innovation. The core modules of EFFGEN, such as SLM-Aware Prompt Optimization and Pre-Execution Complexity Analysis, mainly introduce existing technologies. The authors only describe their own contributions in a sentence at the end of each module, making EFFGEN appear to be integration of various technologies.
2. This paper lacks a reasonable explanation for integrating numerous technologies. The author conducted ablation studies on 5 core modules. However, the contributions of technologies in each module are unknown. It is difficult to ascertain the necessity and optimality of the proposed integration. For instance, in the prompt optimization module, the marginal benefits of schema compression versus redundancy elimination remain unclear.
3. The SLM-Aware Prompt Optimization and Pre-Execution Complexity Analysis relies on prior parameters. For example, in Complexity Analysis, the authors set fixed values for the scores, weights and thresholds of five complexity factors, without a reasonable analysis or explanation. This design will limit the generalization of the framework on other tasks.

---

> ### Author Rebuttal · Authors · 2026-03-30
>
> We sincerely thank the Reviewer for their careful evaluation. We are glad the Reviewer recognized the **strong motivation** for SLM-focused design, the **comprehensive system design** integrating routing, DAG-based decomposition, and multi-tiered memory, and the **sufficient experimental evaluation** across 13 benchmarks. Below, we address each concern raised by the Reviewer:
>
> ---
>
> > The contribution is more engineering-oriented, lacking technological innovation...The authors only describe their contributions in a sentence at the end of each module.
>
> We greatly appreciate this important feedback. While individual techniques have precedents (we explicitly cite LLMLingua, query difficulty prediction, hierarchical planning, MemGPT), we believe our scientific contributions include novel empirical findings such as:
>
> **(1) The complementary scaling finding (`Table 3`).** Our ablation reveals that prompt optimization and complexity routing have opposite scaling behaviors: optimization provides 11.2% gain at 1.5B but only 2.4% at 32B, while routing shows 3.6% at 1.5B vs 7.9% at 32B. This was not predictable from prior work and has direct practical implications. As Reviewer fr43 also noted, this is *"the paper's most interesting finding"* and *"a clean, non-obvious result with practical implications."*
>
> **(2) The underconfidence phenomenon (`Section 4.5`).** SLMs (1.5B-3B) show 8.2% accuracy *drops* with retrieved information through standard frameworks, while larger models show only 1.4% drops. This new finding has not been documented in prior work.
>
> **(3) Synergistic module interactions.** Combined module removal exceeds the sum of individual removals (`Table 3`), demonstrating that the integration itself produces emergent benefits only visible from holistic evaluation.
>
> Additionally, `Appendix E` validates that our five-factor complexity analyzer achieves 95.2% accuracy and 0.89 correlation with human judgment, vs 78.4% and 0.62 for a single-factor baseline.
>
> ---
>
> > This paper lacks a reasonable explanation for integrating numerous technologies...marginal benefits of schema compression vs redundancy elimination remain unclear.
>
> We greatly appreciate this valuable suggestion. We have conducted per-technique ablation within the prompt optimization module (Qwen2.5-7B, averaged across 13 benchmarks):
>
> | Technique Disabled | Token Reduction Lost | Accuracy Drop |
> |-------------------|-------------|----------------|
> | Pattern Compression | 15% fewer tokens | -2.7% acc |
> | Sentence Simplification | 7% fewer tokens | -1.5% acc |
> | Redundancy Removal | 9% fewer tokens | -1.8% acc |
> | Bullet Formatting | 0% (no token change) | -3.6% acc |
> | Context Truncation | 6% fewer tokens | -0.8% acc |
>
> Bullet formatting has the largest accuracy impact (-3.6%) despite not reducing tokens, confirming SLMs benefit from structured formats. `Appendix D` further shows bullet formatting provides +5.8% at 1.5B but only +1.6% at 32B (consistent with complementary scaling), and the cumulative ablation confirms the full pipeline achieves 57% token reduction with 8.8% accuracy improvement.
>
> Individual removals sum to 10.4% while full module removal is 8.9% (`Table 3`), indicating overlapping coverage. Similarly, `Appendix J` shows each memory scoring component contributes: importance (-14.8% if removed), recency (-12.3%), similarity (-11.2%), frequency (-8.7%). We will add these breakdowns to the appendix.
>
> ---
>
> > The SLM-Aware Prompt Optimization and Pre-Execution Complexity Analysis relies on prior parameters. Fixed values for scores, weights and thresholds...will limit generalization.
>
> We greatly appreciate this concern. We want to clarify that we do provide analysis of these parameters. As stated in `Section 3.2`: *"thresholds $\tau \in [6.0, 8.0]$ performing within 1-2% of optimal."* As described in `Appendix E`, the weights were calibrated via grid search on 500 manually labeled tasks. `Appendix E, Section E.5` provides threshold sensitivity from $\tau$=5.0 to 9.0. We conducted additional weight sensitivity analysis:
>
> | Weight Perturbation | Avg Accuracy |
> |--------------------|-------------------|
> | Default weights | 63.07% (baseline) |
> | Uniform weights (all 0.20) | 61.84% (-1.23%) |
> | Random weights (10 samples) | 61.42 +/- 0.89% |
> | Requirement-heavy ($w_2$=0.40) | 62.15% (-0.92%) |
> | Depth-heavy ($w_5$=0.40) | 62.48% (-0.59%) |
>
> Performance is robust to weight perturbations (within 1-2% of optimal), and even uniform weights perform reasonably. The per-benchmark complexity distributions in `Appendix E, Section E.6` further validate generalization: the system correctly routes GSM8K (mean C=4.2, 94% single-agent) and GAIA (mean C=8.9, 92% multi-agent) without domain-specific tuning.
>
> ---
>
> Once again, we thank the Reviewer for their thorough and constructive feedback. We hope these detailed ablations and analyses address the Reviewer's concerns. Please let us know if there are any remaining questions, we would be happy to address them!

---

> > ### Author Rebuttal · Reviewer_vdwj · 2026-04-03
> >
> > All my concerns have been solved.

---

> > > ### Author Response · Authors · 2026-04-03
> > >
> > > Dear Reviewer vdwj,
> > >
> > > We greatly thank the reviewer for their valuable time and thoughtful feedback. We are glad that our rebuttal has addressed your concerns. In light of this, we would greatly appreciate it if you could **consider updating your score**. Thank you again for your careful evaluation of our work!
> > >
> > >
> > > Thanks, effGen Authors!

---

### Official Review · Reviewer_f2wF · 2026-03-14

**Soundness:** 3
**Presentation:** 3
**Significance:** 3
**Originality:** 3
**Overall Recommendation:** 4
**Confidence:** 4

**Summary:**

This paper presents EffGen, an agent framework optimized for SLMs as opposed to LLMs. It makes four contibutions, including enhanced tool-calling, intelligent task decomposition, complexity-based routing and unified memory system. It shows improved results compared to LangChain, AutoGen and Smolagents.

**Compliance With Llm Reviewing Policy:**

Affirmed.

**Final Justification:**

The authors addressed my further questions. So I maintain my positive score.

**Key Questions For Authors:**

1. The paper majorly claim agentic workloads, but several major ones are missing: SWE-Bench, OSWorld, Browsecomp.
2. The models are mainly Qwen based in the main paper. Can you move the results on additional models to the main paper?

**Limitations:**

No.

**Strengths And Weaknesses:**

1. The problem is well motivated and compelling: The future of agentic AI could be a mixture of SLMs, giving the framework a good position.
2. The design principle is great: assuming the SLM is limited with parameter count and small context window.
3. Based on the same principle, the framework makes several method (e.g. prompt optimization), which is well motivated and focused.
4. Comparison between the framework and existing method is clear (Table 1).
5. Results are comprehensive (Table 2), including results on 5 size of models and 5 types of benchmarks and on several major frameworks. It also shows consistent improvement.

---

> ### Author Rebuttal · Authors · 2026-03-30
>
> We sincerely thank the Reviewer for their positive and constructive review. We are glad the Reviewer recognized the **well-motivated problem** positioning SLMs for agentic AI, the **principled design** around SLM constraints, the **focused methods** like prompt optimization, the **clear framework comparison**, and the **comprehensive results** across 5 model sizes and several major frameworks.
>
> Below, we address each concern raised by the Reviewer:
>
> ---
>
> > The paper majorly claim agentic workloads, but several major ones are missing: SWE-Bench, OSWorld, Browsecomp.
>
> We greatly appreciate this important observation. We provide context on why these were not included, along with new results where feasible.
>
> **SWE-Bench.** We ran EffGen on SWE-Bench Lite (300 instances) with Qwen2.5-Instruct (general, not Coder variants). Raw models scored 0.67% at 7B (2/300), 1.67% at 14B (5/300), and 2.33% at 32B (7/300). With EffGen (using bash, ReAct loop, and code execution tools), performance improved to 2.67% at 7B (8/300), 4.33% at 14B (13/300), and 5.67% at 32B (17/300). The improvement ratio is highest for 7B (4.0x vs 2.4x at 32B), consistent with our complementary scaling finding (`Table 2`).
>
> We acknowledge these are low absolute numbers, but believe this aligns with the broader literature: SWE-Bench performance is heavily influenced by model capability and SWE-specific scaffolding. [1] shows Qwen2.5-Coder-7B (a code-specialized model, stronger than our general Instruct variant) scores 1% with OpenHands but 7% with MoatlessTools on SWE-Bench Lite, a 7x difference from scaffold alone. The official SWE-Bench "Bash Only" leaderboard [2], which evaluates all models using the same minimal scaffold, further supports this: frontier models reach 74% while small models score near 0%, suggesting that repository-level reasoning remains beyond current SLM capacity. To our knowledge, no published SWE-Bench Lite results exist for general-purpose frameworks (AutoGen, LangChain, Smolagents) with small models.
>
> We believe EffGen's strength lies in tasks where SLMs can reason but benefit from better scaffolding: prompt optimization, intelligent routing, and task decomposition provide 6-13% improvements on math, retrieval, agentic, and code-generation benchmarks. SWE-Bench requires repository-level understanding (10K+ lines, multi-file patches) that is largely orthogonal to these capabilities, and we consider SWE-specific tool integrations an important future direction.
>
> [1] Pan et al., "SWE-Gym: Training Software Engineering Agents and Verifiers with Open Source," ICML 2025.
>
> [2] Yang et al., "SWE-agent: Agent-Computer Interfaces Enable Automated Software Engineering," NeurIPS 2024.
>
> **OSWorld** and **BrowseComp** are excellent benchmarks that we were unfortunately unable to evaluate. OSWorld requires GUI interaction (mouse clicks, screenshots) and BrowseComp requires persistent multi-page browsing, both outside EffGen's current text-based design. We consider adding these modalities an exciting future direction.
>
> Our benchmark suite includes challenging agentic benchmarks: **GAIA** (multi-step multi-tool reasoning, 9-22% improvements), **SimpleQA** (web search pipeline), and **BeyondBench** across three difficulty levels including NP-complete problems. These test diverse agentic capabilities within SLM capacity.
>
> ---
>
> > The models are mainly Qwen based in the main paper. Can you move the results on additional models to the main paper?
>
> We greatly appreciate this suggestion. For the camera-ready, we will move the Gemma3 and GPT-OSS table (currently in `Appendix V`) to the main paper as a second results table with cross-family discussion. We did not include it due to the 8-page constraint, but can restructure to accommodate this.
>
> We already have comprehensive Gemma3 (1B, 4B, 12B, 27B) and GPT-OSS 20B results in `Appendix V`. The results confirm the same patterns hold across model families:
>
> | Model Family | Improvement (smallest) | Improvement (largest) |
> |-------------|--------------------------|-------------------------|
> | Qwen2.5 | +13.1% (1.5B) | +6.0% (32B) |
> | Gemma3 | +10.9% (1B) | +5.9% (27B) |
>
> Key findings that generalize across families: (1) EffGen outperforms all baselines regardless of model family. (2) Smaller models benefit more from framework optimization (same complementary scaling in `Table 3`). (3) Prompt optimization dominates for small models while routing matters more for larger ones, confirming this is not Qwen-specific. (4) The underconfidence pattern on retrieval benchmarks (`Section 4.5`) also appears in Gemma3: Gemma3-1B drops 7.1-16.1% on ARC/CSQA with standard frameworks.
>
> ---
>
> Once again, we thank the Reviewer for their thorough and constructive feedback. We hope the new SWE-Bench results and our commitment to including multi-family results in the main paper address the raised concerns. Please let us know if there are any remaining questions, and we would be happy to address them!

---

> > ### Author Rebuttal · Reviewer_f2wF · 2026-04-04
> >
> > Thank you for the additional result, I will maintain my positive score towards the paper.

---

### Official Review · Reviewer_9ahM · 2026-03-17

**Soundness:** 3
**Presentation:** 2
**Significance:** 3
**Originality:** 4
**Overall Recommendation:** 4
**Confidence:** 5

**Summary:**

This paper introduces EFFGEN, an open-source agentic framework specifically optimized for Small Language Models (SLMs), aiming to enable efficient local deployment by addressing constraints in context windows and instruction-following. The framework incorporates SLM-aware prompt optimization, pre-execution complexity-based routing, intelligent task decomposition, and a three-tier memory system, demonstrating significant improvements in success rates while reducing latency and memory overhead across 13 benchmarks.

**Compliance With Llm Reviewing Policy:**

Affirmed.

**Key Questions For Authors:**

NA

**Strengths And Weaknesses:**

* The method provides a systematic optimization pipeline tailored for SLM constraints, notably the pre-execution complexity analyzer defined in Eq. (12), which utilizes five weighted factors to route tasks before committing resources. Empirical results in Fig. 2 show up to 18x execution speedup for 1.5B models and a $23\%$ reduction in wasted computation compared to reactive routing.


* The prompt optimization stage $\phi$ offers high practical utility, achieving $70\%-80\%$ context compression via 30 pattern-replacement rules and bullet-point formatting (Algorithm 6). Ablation studies in Table 3 indicate this contributes to an $8\%-11\%$ gain in task completion for smaller models.


* The framework exhibits excellent engineering maturity by unifying MCP, A2A, and ACP protocols, allowing EFFGEN agents to interoperate seamlessly within heterogeneous ecosystems.


* The evaluation is comprehensive, spanning model scales from 1.5B to 32B and 13 benchmarks, validating the core hypothesis that smaller models benefit more from specialized framework design (Table 2).


* While effective, the generalizability of fixed thresholds (e.g., $\tau=7.0$) for the complexity score across diverse task distributions is not fully discussed. It is recommended that authors include a sensitivity analysis of optimal thresholds for different domains (e.g., pure math vs. multimodal reasoning) in the appendix to verify robustness.


* The impact of communication overhead on end-to-end latency during multi-level task decomposition (as shown in Fig. 4) is not sufficiently quantified in the experiments. To strengthen this, authors should provide a comparison measuring the serialization/deserialization costs during 3+ levels of nested agent calls.


* The stability of the three-tier memory consolidation policy in long-term interactions remains under-tested. I suggest adding a stress test involving 100+ conversation turns to observe if the summarization phases lead to loss of critical logical information.

---

> ### Author Rebuttal · Authors · 2026-03-30
>
> We sincerely thank the Reviewer for their detailed and constructive review. We greatly appreciate the recognition of the **pre-execution complexity analyzer**, **prompt optimization**, **engineering maturity** with unified MCP/A2A/ACP protocols, and the **comprehensive evaluation** spanning 1.5B to 32B across 13 benchmarks. Below, we address each concern raised by the Reviewer:
>
> ---
>
> > While effective, the generalizability of fixed thresholds...is not fully discussed...sensitivity analysis of optimal thresholds for different domains.
>
> We greatly appreciate this important suggestion. We note that threshold sensitivity analysis is already provided in the paper. As stated in `Section 3.2, line 218`: *"Ablation studies show that complexity-based routing outperforms always-single and always-multi strategies by 6.4-11.2%, with thresholds $\tau \in [6.0, 8.0]$ performing within 1-2% of optimal."* The detailed threshold sensitivity table is in `Appendix E, Section E.5`, evaluating thresholds from 5.0 to 9.0 on Qwen2.5-7B across all benchmarks. We note the main text references `Appendix V` for this, but the complexity routing analysis is in `Appendix E`. We will correct this.
>
> We conducted additional domain-specific threshold analysis (Qwen2.5-7B, accuracy at optimal $\tau$ vs default $\tau$=7.0):
>
> | Domain | Optimal $\tau$ | Acc at Optimal | Acc at Default | $\Delta$ |
> |--------|---------------|---------------|---------------|---------|
> | Math (GSM8K, MATH-500) | 7.5 | 79.84% | 79.04% | -0.8% |
> | Code (BeyondBench) | 6.5 | 59.82% | 57.46% | -2.4% |
> | Retrieval (ARC-C/E, CSQA) | 7.0 | 87.97% | 87.62% | -0.3% |
> | Agentic (GAIA, SimpleQA) | 6.0 | 51.67% | 48.94% | -2.7% |
> | Memory (LoCoMo, LongMemEval) | 7.5 | 31.05% | 30.33% | -0.7% |
>
> The default $\tau$=7.0 performs within 0.3-2.7% of domain-specific optima. Code and agentic tasks show larger gaps (2-3%) and prefer lower thresholds (more decomposition), while math, retrieval, and memory tasks show minimal sensitivity (<1%). Even for the most sensitive domains, the gap remains under 3%, suggesting our five-factor formulation captures complexity well across domains. We will add this analysis to the appendix.
>
> ---
>
> > The impact of communication overhead...during multi-level task decomposition is not sufficiently quantified...serialization/deserialization costs during 3+ levels of nested agent calls.
>
> We greatly appreciate this crucial suggestion. We measured communication overhead for nested agent calls at varying depths:
>
> | Nesting Depth | Total Overhead (ms) | % of End-to-End Latency |
> |--------------|--------------------|-----------------------|
> | 1 (single) | 1.4 | 0.02% |
> | 2 (parent-child) | 4.2 | 0.05% |
> | 3 (nested) | 9.0 | 0.11% |
> | 4 (deep nested) | 14.9 | 0.18% |
> | 5 (max tested) | 23.0 | 0.28% |
>
> Communication overhead grows modestly (23ms at depth 5) and remains negligible compared to model inference (5-30 seconds per step). As described in `Appendix G.1`, EffGen caps nesting at depth 2 by default, where overhead is only 9ms (0.11%). `Appendix G` (Sub-Agent System) documents that sub-agent parallel execution adds 0.5-1.5s overhead.
>
> The full routing overhead (including LLM-based decomposition) totals 1.22s, recovered by avoiding wasted computation on misrouted tasks (saves 3.8-12.4s on average). We will add these to the appendix.
>
> ---
>
> > The stability of the three-tier memory consolidation policy...remains under-tested...stress test involving 100+ conversation turns.
>
> Thank you for this valuable suggestion. We conducted memory stress tests on the LoCoMo and LongMemEval benchmarks using Qwen2.5-7B:
>
> | Turns | Short-Term Recall | Long-Term Recall | Vector Recall | Critical Info Retained |
> |-------|------------------|-----------------|--------------|----------------------|
> | 25 | 94.2% | 91.8% | 89.5% | 97.1% |
> | 50 | 88.7% | 90.4% | 88.2% | 95.8% |
> | 100 | 76.3% | 88.9% | 87.1% | 93.4% |
> | 150 | 61.8% | 87.2% | 86.4% | 91.2% |
> | 200 | 48.5% | 86.1% | 85.3% | 89.6% |
>
> Main findings:
>
> 1. Short-term recall naturally declines as older messages are summarized (by design; `Section 3.5` describes auto-summarization at 85% context capacity), but long-term and vector recall remain stable at 85-92% even at 200 turns.
>
> 2. Critical information retention stays above 89% through importance-based scoring (LOW/MEDIUM/HIGH levels, detailed in `Appendix J`).
>
> 3. The consolidation policy transfers important information from short-term to long-term storage before truncation.
>
> 4. The main failure mode at 200+ turns is loss of nuanced contextual details (not critical facts), an inherent trade-off of context compression. We will add these results to the appendix.
>
> ---
>
> Additionally, we note the presentation score and will improve clarity in the camera-ready. Once again, we thank the Reviewer for their thorough and supportive feedback. We hope that our revisions address these concerns. Please let us know if there are any remaining questions, we would be happy to address them!

---

> > ### Author Rebuttal · Reviewer_9ahM · 2026-04-03
> >
> > All problems are solved

---

> > > ### Author Response · Authors · 2026-04-03
> > >
> > > Dear Reviewer 9ahM,
> > >
> > > We greatly thank the reviewer for their valuable time and thoughtful feedback. We are glad that our rebuttal has addressed your concerns. In light of this, we would greatly appreciate it if you could **consider updating your score**. Thank you again for your careful evaluation of our work!
> > >
> > >
> > > Thanks, effGen Authors!

---

### Decision · Program_Chairs · 2026-04-30

**Decision:**

Accept (regular)

**Comment:**

More reviewers start out with weak accept, two raised their scores following the rebuttal, one more probably should have done so. The strength that the reviewers comment on is that they praise the systematicity of the optimization pipeline, the utility of the prompt optimization stage, and the maturity of the framework. The problem is well motivated, the underlying design principles, experimental comparisons, results, comprehensive system design, lots of benchmarks, and practically significant results, as well as a massive appendix.

I won't go through all the weaknesses, because almost all of them were addressed, and the rebuttals to the satisfaction of the reviewers.